# VERBALIZED MACHINE LEARNING:
# REVISITING MACHINE LEARNING WITH LANGUAGE MODELS

## ABSTRACT

Motivated by the great progress made by large language models (LLMs), we introduce the framework of verbalized machine learning (VML). In contrast to conventional machine learning models that are typically optimized over a continuous parameter space, VML constrains the parameter space to be human-interpretable natural language. Such a constraint leads to a new perspective of function approximation, where an LLM with a text prompt can be viewed as a function parameterized by the text prompt. Guided by this perspective, we revisit classical machine learning problems, such as regression and classification, and find that these problems can be solved by an LLM-parameterized learner and optimizer. The major advantages of VML include (1) easy encoding of inductive bias: prior knowledge about the problem and hypothesis class can be encoded in natural language and fed into the LLM-parameterized learner; (2) automatic model class selection: the optimizer can automatically select a concrete model class based on data and verbalized prior knowledge, and it can update the model class during training; and (3) interpretable learner updates: the LLM-parameterized optimizer can provide explanations for why each learner update is performed. We conduct several studies to empirically evaluate the effectiveness of VML, and hope that VML can serve as a stepping stone to stronger interpretability and trustworthiness in ML.

> *"The limits of my language mean the limits of my world"*
>
> — Ludwig Wittgenstein

## 1 INTRODUCTION

The unprecedented success of large language models (LLMs) has changed the way people solve new problems in machine learning. Compared to conventional end-to-end training where a neural network is trained from scratch on some curated dataset, it has become increasingly more popular to leverage a pretrained LLM and design good prompts that contain in-context examples and effective instructions. These two ways of problem-solving lead to an intriguing comparison. Traditionally, we would optimize a neural network in *a continuous numerical space* using gradient descent, while in the new approach, we optimize the input prompt of an LLM in *a discrete natural language space*. Since a neural network is effectively a function parameterized by its numerical weight parameters, can a pretrained LLM act as a function that is parameterized by its natural language prompt?

Driven by this question, we conceptualize the framework of verbalized machine learning (VML), which uses natural language as the representation of the model parameter space. The core idea behind VML is that we can define a machine learning model using natural language, and the training of such a model is based on the iterative update of natural language. This framework enables many new possibilities for interpretability, as the decision rules and patterns learned from data are stored and summarized in natural language. Specifically, we propose to view the input text prompt of LLMs as the model parameters that are being learned. However, optimization over such a natural language parameter space also introduces additional difficulties. Inspired by previous work [3, 21] where the optimizer is viewed as a function parameterized by a neural network, we parameterize the optimizer function as another LLM, which produces the next-step model parameters by taking in the current model parameters, a batch of training data points, and the loss function. Therefore, VML requires the optimizer LLM to update the learner LLM iteratively such that the training objective can be reached.

Compared to conventional numerical machine learning, the VML framework brings a few unique advantages. First, VML introduces an easy and unified way to encode inductive bias into the model. Because the model parameters are fully characterized by human-interpretable natural language, one can easily enter the inductive bias using language. This linguistic parameterization makes machine learning models fully interpretable and adjustable. For example, if the input and output data are

observed to be linearly correlated, then one can use this sentence as part of text prompt. How to effectively encode inductive bias is actually a longstanding problem in machine learning, and VML provides a unified way to inject the inductive bias through natural language—just like teaching a human learner. Second, VML performs automatic model selection during the learning process. The optimizer LLM can automatically select a suitable model class based on the training data and verbalized prior knowledge. Third, each update of the model is fully interpretable in the sense that the optimizer LLM can give an explanation of why it chooses such an update. One can even interact with the optimizer LLM in order to inject new prior knowledge or obtain detailed reasoning.

VML can be viewed as a natural generalization of in-context learning (ICL). Specifically, ICL is a single-step implicit learning process, while VML is a multi-step iterative learning process where the in-context examples are summarized into verbal pattern and knowledge. Moreover, VML provides a way of scaling inference-time compute [5, 42]. Compared to the best-of-N re-sampling, VML iteratively updates its model parameter prompt by taking into account the learner's past predictions.

An important concept of VML is its unified token-level representation of both data and model. Unlike numerical machine learning, language models in VML do not differentiate data and model, and treat both of them as part of the text prompt. This shares a striking connection to stored-program computers, also known as the von Neumann architecture, where the key idea is to represent programs as data rather than wiring setups. The link between language models and stored-program computers underscores the importance of text prompts, which play a similar role to computer programs, and, along with LLMs, can become a powerful zero-shot problem solver. Our contributions are as follows:

- We formulate the framework of verbalized machine learning, where pretrained language models are viewed as function approximators parameterized by their text prompts. Then, we revisit some classical machine learning problems and show that VML is able to solve them.
- We design a concrete VML algorithm with a text prompt template. This algorithm parameterizes both the learner model and the optimizer as LLMs, and enables the iterative verbalized training.
- We conduct an empirical study for the injection of verbalized inductive bias and show that it is promising to use natural language as a unified way to encode prior knowledge.
- We validate the effectiveness of VML in different applications (Section 4, Appendix B,D,E,F,G).

## 2 RELATED WORK

**LLMs for planning and optimization**. Language models are used to perform planning for embodied agents [43, 53, 22, 24], such that they can follow natural language instruction to complete complex tasks. More recently, LLMs have been used to solve optimization problems [55]. Specifically, the LLM generates a new solution to an optimization problem from a prompt that contains previously generated solutions and their loss values. The LLM optimizer in [55] shares a high-level similarity to our work, as we also aim to solve an optimization problem with LLMs. The key difference to [55] is our function approximation view of LLMs, which enables us to revisit classical machine learning problems and solve them through natural language in the VML framework.

**Natural language to facilitate learning**. [39, 19, 20, 31, 63] show that natural language captions serve as an effective supervision to learn transferable visual representation. [30, 32, 34, 29, 58, 54] find that natural language descriptions can easily be turned into zero-shot classification criteria for images. [2] proposes to use natural language as latent parameters to characterize different tasks in few-shot learning. In contrast to prior work, VML directly uses the text prompt of LLMs to parameterize functions and learns the language-based model parameters in a data-driven fashion.

**Prompt engineering and optimization**. There are many prompting methods [49, 64, 65, 47, 59, 60, 51] designed to elicit the reasoning ability of LLMs. To reduce the hand-crafting efforts in designing good prompts, automatic prompt optimization [64, 65, 55, 35, 50, 7, 23, 27, 44] has been proposed. Unlike prompt optimization where the text prompt is optimized without changing its semantic meaning, VML updates its language-based model parameters by adding or modifying the model prior information, making the learner model fully interpretable about its prediction.

**LLMs for multi-agent systems**. Due to the strong instruction-following ability, LLMs are capable of playing different roles in a multi-agent systems. [36, 52, 11, 18] study a multi-agent collaboration system for solving complex tasks like software development. VML can also be viewed as a two-agent system where one LLM plays the role of learner and the other LLM plays the role of optimizer.

# 3 VERBALIZED MACHINE LEARNING

## 3.1 FROM NUMERICAL TO VERBALIZED MACHINE LEARNING

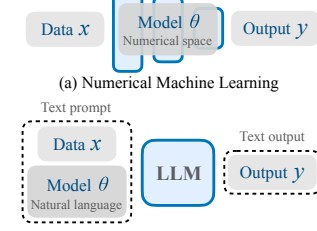

(a) Numerical Machine Learning

(b) Verbalized Machine Learning

Figure 1: A comparison between numerical machine learning and VML.

Classical machine learning models (*e.g.*, neural networks) are typically trained in a numerical and continuous parameter space. Once trained, these models are stored as a collection of numbers that are not interpretable and remain a black box. Motivated by the strong universal problem-solving capability of LLMs, we find it appealing to view an LLM as a function approximator parameterized by its text prompt. This perspective leads to the VML framework. Similar to a general-purpose modern computer whose functionality is defined by its running program, a function that is defined by an LLM is characterized by its text prompt. Due to the fully human-interpretable text prompt, the VML framework provides strong interpretability and is also easy to trace the cause of model failure. Figure 1 gives a comparison between numerical machine learning and VML. In the VML framework, both data and model are represented in a unified token-based format, while numerical machine learning treats data and model parameters differently.

## 3.2 NATURAL LANGUAGE AS THE MODEL PARAMETER SPACE

VML parameterizes a machine-learning model with natural language. More formally, VML places a strong constraint on the model parameters $\boldsymbol{\theta} = \{\theta_1, \theta_2, \cdots, \theta_t\} \in \Theta_{\text{language}}$ to exchange for interpretability, where $\boldsymbol{\theta}$ is a text token sequence, $\theta_t \in \mathcal{A}, \forall t$ is some text token from a large token set $\mathcal{A}$, and $\Theta_{\text{language}}$ denotes the set of all natural language sequences that humans can understand. The model parameter space in VML has the following properties: (1) discrete: the natural language space is discrete; (2) sequential: the natural language space is sequential, and the next word is dependent on its previous words. In contrast, the parameter space in numerical machine learning is not sequentially dependent; and (3) human-interpretable: the natural language that characterizes the model is human-interpretable. More discussion is given in Appendix I.

One of the most significant advantages to use natural language as the model parameters is the easy incorporation of our prior knowledge about the problem and the desired inductive bias into the model training. When the model parameters get updated during training, the model is fully interpretable, and one can observe and understand what gets added and what is modified. Our empirical evidences also supports our interpretability claim, as we find that the model parameters $\boldsymbol{\theta}$ are typically a language description of the underlying pattern that the model discovers from the training data.

## 3.3 LANGUAGE MODELS AS FUNCTION APPROXIMATORS

The core idea behind VML is using a language model to act as a function approximator parameterized by its natural language prompt. Specifically, we denote the language model as $f(\boldsymbol{x}; \boldsymbol{\theta})$ where $\boldsymbol{x}$ is the input data and $\boldsymbol{\theta}$ is the function parameter. Both $\boldsymbol{x}$ and $\boldsymbol{\theta}$ are represented with text tokens. In VML, $f(\cdot)$ is typically a frozen language model that is pretrained on a large corpus of text (*e.g.*, Llama-3 [45], ChatGPT). If we consider a static function, we can set the temperature parameter of the LLM as zero, which theoretically makes the output deterministic. If we set the temperature high (see Appendix H for more discussion), $f(\boldsymbol{x}; \boldsymbol{\theta})$ can be viewed as sampling a value from some distribution.

We revisit how a classical machine learning problem is formulated in the VML framework. Suppose we have $N$ data points $\{\boldsymbol{x}_n, y_n\}_{n=1}^{N}$ in total, where $\boldsymbol{x}_n$ is the data vector and $y_n$ is the target value. As an example, we consider a least square regression problem using the LLM-parameterized function:

$$\min_{\boldsymbol{\theta}} \ell_{\text{regression}} := \frac{1}{2N} \sum_{n=1}^{N} \left( y_n - f_{\text{model}}(\boldsymbol{x}_n; \boldsymbol{\theta}) \right)^2, \quad \text{s.t. } \boldsymbol{\theta} \in \Theta_{\text{language}} \quad (1)$$

where minimizing the objective function with respect to the discrete token-based model parameters $\boldsymbol{\theta}$ is actually quite difficult. Back-propagating gradients through discrete variables (*e.g.*, policy gradients, Gumbel-softmax [12]) is typically known to be sample-inefficient and sub-optimal.

## 3.4 ITERATIVE TRAINING BY PROMPT OPTIMIZATION

Because the model parameters $\boldsymbol{\theta}$ in VML form a text prompt, optimizing $\boldsymbol{\theta}$ is effectively a prompt optimization problem. Different from the prompt optimization problem [65], where the goal is to

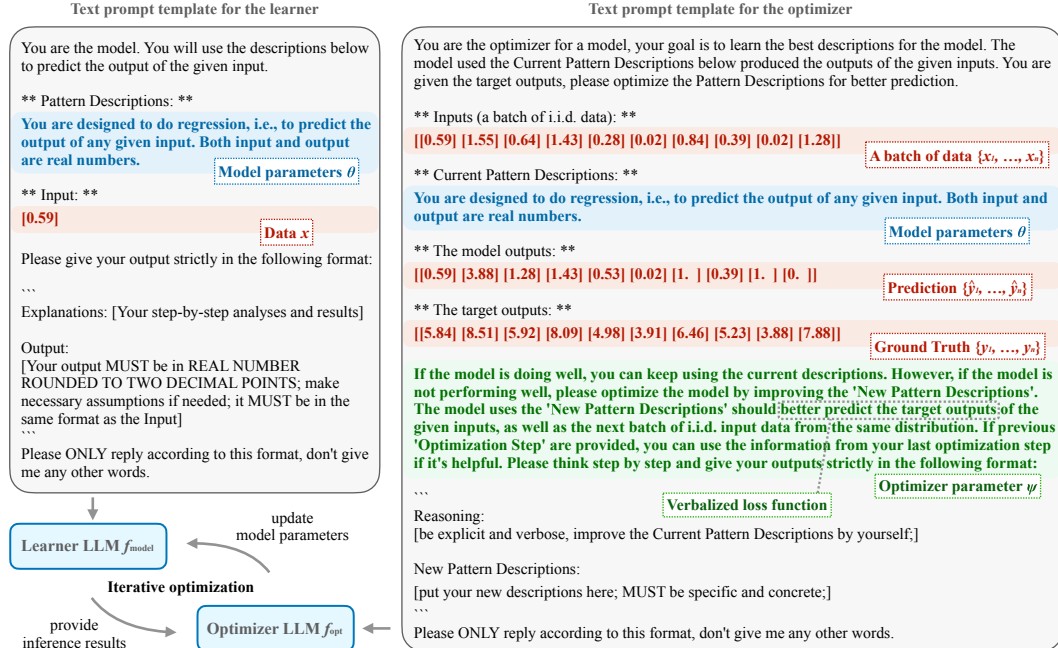

Figure 2: An overview of iterative optimization and text prompt templates of the learner and the optimizer in the regression example.

produce a generic prompt without adding new information, the training in VML focuses on updating the model's language characterization, which involves both the addition of new prior information and the modification of existing information. To optimize our model parameters, we start by looking at the gradient of the regression objective function in Equation 1:

$$\nabla_{\boldsymbol{\theta}} \ell_{\text{regression}} = \frac{1}{N} \sum_{i=1}^{N} \left( y_n - f_{\text{model}}(\boldsymbol{x}_n; \boldsymbol{\theta}) \right) \cdot \frac{\partial f_{\text{model}}(\boldsymbol{x}_n; \boldsymbol{\theta})}{\partial \boldsymbol{\theta}} \quad \text{s.t.} \ \ \boldsymbol{\theta} - \eta \cdot \nabla_{\boldsymbol{\theta}} \ell_{\text{regression}} \in \Theta_{\text{language}} \quad (2)$$

where $\eta$ is the learning rate, and the constraint is to ensure that the updated model parameters are still in the human-interpretable natural language space. It seems to be infeasible to compute this gradient. To address this, we view the gradient as a function of the data $(\boldsymbol{x}, y)$ and the current model parameters $\boldsymbol{\theta}$. Then we directly approximate the next-step model parameters using another pretrained language model denoted by $f_{\text{opt}}(\boldsymbol{x}, \hat{y}, y, \boldsymbol{\theta}; \boldsymbol{\psi})$ where $\hat{y}$ is the model prediction from the learner $f_{\text{model}}$. $\boldsymbol{\psi}$ denotes the optimizer parameters that characterizes the optimizer settings, and we can

---

**Algorithm 1** Training in VML

Initialize model parameters $\boldsymbol{\theta}_0$, iteration number $T$, batch size $M$ and optimizer parameters $\boldsymbol{\psi}$;
**for** $i = 1, \cdots, T$ **do**
    Sample $M$ training examples $\boldsymbol{x}_1, \cdots, \boldsymbol{x}_M$;
    **for** $m = 1, 2, \cdots, M$ **do**
        $\hat{y}_m = f_{\text{model}}(\boldsymbol{x}_m; \boldsymbol{\theta}_{i-1})$;
    **end**
    $\boldsymbol{\theta}_i = f_{\text{opt}} \left( \{\boldsymbol{x}_m, \hat{y}_m, y_m\}_{m=1}^{M}, \boldsymbol{\theta}_{i-1}; \boldsymbol{\psi} \right)$;
**end**

---

use language to specify the update speed, the momemtum, *etc*. The largest possible batch size of the optimizer LLM is determined by its context window. The optimizer LLM can already output natural language that satisfies the constraint, so we simply ask the LLM to play the optimizer role, which has been shown quite effective in [55]. More importantly, the performance of our VML framework gets better as the instruction-following ability of LLMs gets stronger. An overview of the iterative optimization and the text prompt templates of the learner and optimizer in the regression example are given in Figure 2. The detailed algorithmic training procedure is given in Algorithm 1.

Using an LLM as the optimizer offers several unique advantages. First, the optimizer can perform automatic model selection. When the learner model can not make correct predictions for the training data, the optimizer will automatically update the learner to a more complex and capable model (see the polynomial regression experiments in Section 4.2 as an example). Second, the optimizer can provide detailed explanations of why a particular update should be performed, which helps us to understand the inner working mechanism of the optimization process. Third, the LLM-parameterized optimizer allows users to interact with it directly. This not only helps us to trace model failures, but it also allows us to inject prior knowledge to improve optimization (even during training).

**Different optimizer parameterizations**. Here we use a *direct parameterization*, *i.e.*, parameterizing the optimizer as a single function $f_{\text{opt}}$, which couples the gradient and the update functions together.

Alternatively, we can use an *indirect parameterization* where the gradient and the update are two separate LLM-parameterized functions. Specifically, the update of learner's model parameter is given by $\boldsymbol{\theta}_i = f_{\text{update}}(\boldsymbol{\theta}_{i-1} \cdot \frac{\partial \ell}{\partial \boldsymbol{\theta}})$, where $\frac{\partial \ell}{\partial \boldsymbol{\theta}}$ is computed by $f_{\text{grad}}(\frac{\partial \ell}{\partial \hat{\boldsymbol{y}}}, \boldsymbol{\theta}_{i-1})$ and similarly, $\frac{\partial \ell}{\partial \hat{\boldsymbol{y}}}$ is computed by $f_{\text{grad}}(\ell, \hat{\boldsymbol{y}})$. Both $f_{\text{update}}$ and $f_{\text{grad}}$ are parameterized by LLMs. Compared to direct parameterization that takes one LLM call, this process takes several LLM calls. The gradients here are known as "textual gradients" in prompt optimization [35, 62]. We include the algorithm details in Appendix C.3.

### 3.5 DISCUSSIONS AND INSIGHTS

**VML as a unified framework to encode inductive bias**. A unified framework to encode arbitrary inductive bias has been pursued for decades. For different types of data, we need to design different models to encode the inductive bias (*e.g.*, graphical models [14] for random variables, recurrent networks [10] for sequences, graph networks [13] for graphs, and convolution networks [17] for images). VML uses a unified natural language portal to take in inductive biases, making it very flexible for encoding complex inductive bias. To incorporate an inductive bias about the hypothesis class or prior knowledge about the problem, we can simply concatenate a system prompt $\boldsymbol{\theta}_{\text{prior}}$ (*i.e.*, some constant prefixed text that describes the inductive bias) with the model parameters $\boldsymbol{\theta}$. Therefore, the final model parameters are $(\boldsymbol{\theta}_{\text{prior}}, \boldsymbol{\theta})$ where only $\boldsymbol{\theta}$ is learnable and $\boldsymbol{\theta}_{\text{prior}}$ is provided by users.

**Difference between VML and prompt optimization**. Both VML and prompt optimization aims to automatically produce a text prompt towards some target, but VML differs from existing prompt optimization works (*e.g.*, [65, 35]) in a substantial way. First, VML aims to automatically discover a data pattern description that acts as the the model parameters for the LLM learner, while prompt optimization seeks a generic instruction without changing the original meaning to elicit the best downstream question-answering performance. We qualitatively compare the difference of their learned prompts in the experiment section. Second, prompt optimization can be viewed as a building block for VML, as its techniques can be naturally adapted for the training of VML.

**VML enables interpretable knowledge discovery**. Because the model parameters $\boldsymbol{\theta}$ are already in natural language, it is easy to understand the underlying pattern that leads to the prediction and the decision rules that the model uses. Unlike numerical machine learning, this property enables VML to discover novel knowledge that humans can also learn from.

**VML as "the von Neumann architecture" in machine learning**. Machine learning usually treats the model parameters and the data differently, similar to the Harvard architecture that stores instruction and data separately. VML stores both data and model parameters in the text prompt as tokens, which resembles the von Neumann architecture that stores instruction and data in the same memory.

## 4 APPLICATIONS AND CASE STUDIES

We demonstrate the features and advantages of VML by revisiting some classical machine learning tasks followed by a realistic medical image classification task. In these tasks, we are given data $\mathcal{D}_{\text{train}} = \{\boldsymbol{x}_n, y_n\}_{n=1}^{N}$, and we want to find $\boldsymbol{\theta}^*$ such that $f_{\text{model}}(\boldsymbol{x}; \boldsymbol{\theta}^*)$ best describes the mapping $\boldsymbol{x} \to y$. Our experiments below show in detail how VML is able to solve these tasks and find $\boldsymbol{\theta}^*$.

**Experiment setups**. We use the instruction-tuned Llama-3 70B [45] for the LLM unless specified otherwise. The training set for each task consists of 100 data points. For all tasks, we use a batch size of 10 for each step of optimization (see Figure 2 (right) as an example), which corresponds to 10 steps per epoch of training. To evaluate regression performance, we look at the training loss, and the model predictions in both the interpolation and extrapolation settings. As for classifications, we use additional test sets consist of 20 data points, and evaluate the training and testing accuracies. During optimization, inspired by the idea of momentum from classical machine learning optimization, we also provide the last step (*i.e.*, one step only) of the optimization history to stabilize training.

**Training logs**. The results of our experiments are showed using: (a) training loss, which is computed by *parsing* the model output (string) and *converting* it in to the same data type as the target value ($y$), then we use mean squared error for regression, and zero-one loss mean (*i.e.*, average accuracy) for classification. The computed training loss is for logging purpose only, it is not required for training in VML (see Algorithm 1).; (b) visualization of the learned model, which is also done through *parsing* and *converting* the model output; (c) the model parameter at each training step $i$ before optimization (*i.e.*, $\theta_{i-1}$), and the optimizer output for the updated $\theta_i$. For $i > 1$, the full model parameter before optimization is $\boldsymbol{\theta}_{i-1} = \{\theta_0, \theta_{i-1}\}$, but in our figures below we only show the $\theta_{i-1}$ to save space.

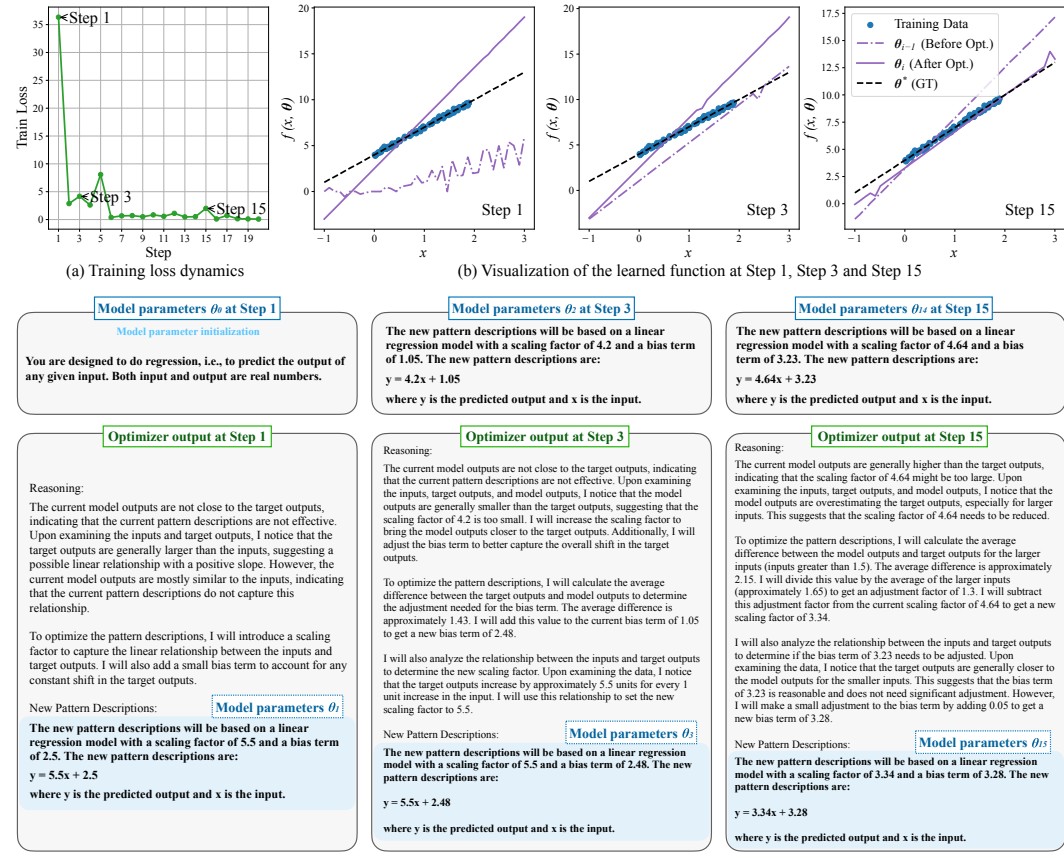

(a) Training loss dynamics

(b) Visualization of the learned function at Step 1, Step 3 and Step 15

(c) Model parameters and optimizer output at Step 1, Step 3 and Step 15

Figure 3: Training dynamics for VML based linear regression. The model is trained for 2 epochs, each with 10 steps.

**Compute**. The LLM is ran on a node of $8 \times$ A100 using the inference engine provided by vLLM [16]. During each step ($i$) of training, we query the LLM 10 times for evaluating the model $f_{\text{model}}(\boldsymbol{x}; \boldsymbol{\theta}_{i-1})$ over a batch, and 1 time for requesting the newly optimized $\boldsymbol{\theta}_i$. We also evaluate the entire test set at each step, which, depending on the size of the evaluation set, requires between 20 to 100 LLM queries. Overall, for the regression tasks, they take around 10 minutes for each epoch of training. The classification tasks, take around 16 minutes for each epoch of training. An additional 6-minute overhead arises due to evaluating the grid for the background of the decision boundary.

## 4.1 LINEAR REGRESSION

We generate $\mathcal{D}_{\text{train}}$ from a linear function with Gaussian noise, *i.e.*, $y = 3x + 4 + \epsilon$, where $\epsilon \sim \mathcal{N}(0, 1)$ and $x \sim \mathcal{U}(0, 2)$. We initialize the model parameter $\boldsymbol{\theta}_0$ by *only* specifying that the task is a regression task from $\mathbb{R}$ to $\mathbb{R}$ (see Figure 3(c) Step 1). Figure 3(a) shows that training improves the model, and that it converges. The subplots (b) and (c) show details of the model and optimization at steps 1, 3 and 15. At step 1, since $\boldsymbol{\theta}_0$ only contain the definition of 1-D regression task, the model$_0$ is randomly guessing (see the dashdot line). The *optimizer$_1$ says that* it notices a linear relationship between the input and the target outputs, hence introducing a linear regression model to capture such a relationship, which results in model$_1$ being a straight line. From step 2 onward, the optimization focus switches to fitting the identified linear regression model to the data. For example, at step 3, we can see that optimizer$_3$ *says* it notices that the outputs of model$_2$ are generally smaller than the target, suggesting the scaling factor is too small, hence it increases it. Similarly, at step 15, optimizer$_{15}$ also *says* it notices the model$_{14}$ overestimates the target; hence, it reduces the scaling factor. We can see from (b) that the resulting model$_{15}$ closely approximates the ground truth.

## 4.2 POLYNOMIAL REGRESSION

We generate $\mathcal{D}_{\text{train}}$ from a polynomial function with Gaussian noise, *i.e.*, $y = 3x^2 + x + 2 + \epsilon$, where $\epsilon \sim \mathcal{N}(0, 1)$ and $x \sim \mathcal{U}(-3, 1)$. Similarly, $\boldsymbol{\theta}_0$ is initialized by *only* specifying that the task is a regression task from $\mathbb{R}$ to $\mathbb{R}$ (see Figure 4(c) Step 1). Figure 4(a) shows that training is effective and

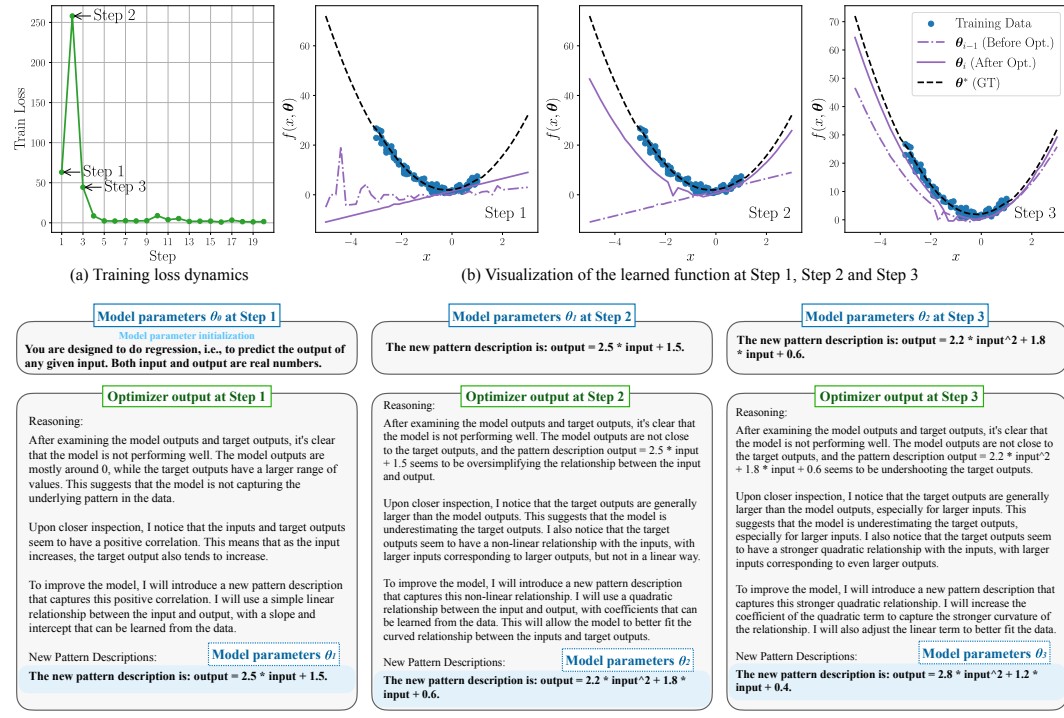

Figure 4: Training dynamic for VML based polynomial regression. The model is trained for 2 epochs, each with 10 steps.

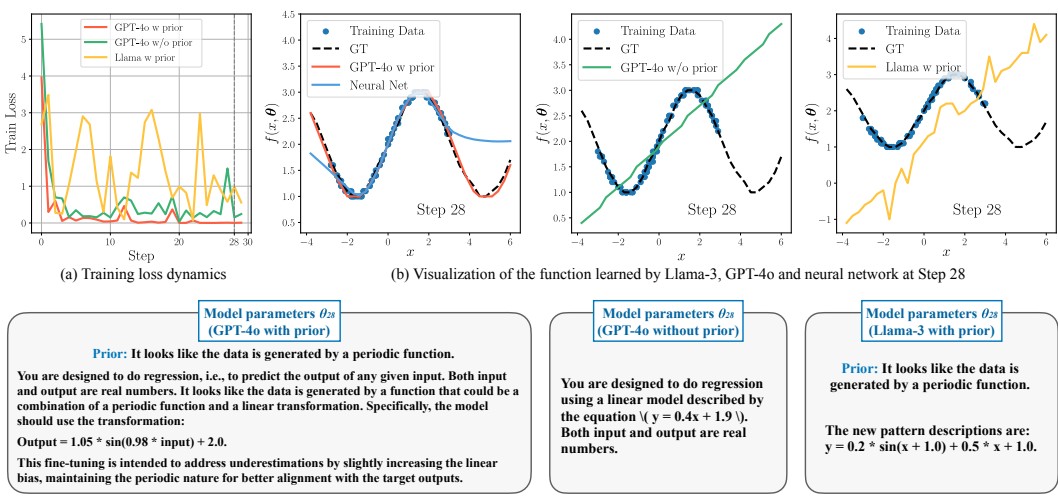

Figure 5: Demonstration of prior injection, and comparison between Llama-3, GPT-4o and a neural net in the setting of sinusoidal regression.

converges. Subplots (b) and (c) show details of the model and optimization at steps 1, 2 and 3. At step 1, $\text{model}_0$ randomly guesses the outputs. The $optimizer_1$ *says that* it notices $y$ has a larger range than $x$, and that they seem to have positive correlation; therefore, it updates $\text{model}_1$ to be a simple linear model. This linear model assumption leads to a jump in the training loss (see subplot (a)), as it is far from the ground truth. Consecutively, at step 2, $optimizer_2$ *says* the poor performance makes it realize that the linear model oversimplifies the relationship between $x$ and $y$. It notices a non-linearity between $x$ and $y$, and to capture this, it uses a quadratic model. This results in a better model and leads to a large decrease in the training loss. At step 3, $optimizer_3$ switches from model class selection to fitting the quadratic model. The resulting $\text{model}_3$ closely fits the ground truth.

## 4.3 SINUSOIDAL REGRESSION

We generate $\mathcal{D}_{\text{train}}$ from a sine function with Gaussian noise, *i.e.*, $y = \sin(x) + 2 + 0.01\epsilon$, where $\epsilon \sim \mathcal{N}(0, 1)$ and $x \sim \mathcal{U}(-3, 3)$. Fitting a sine function is known to be difficult for neural nets in

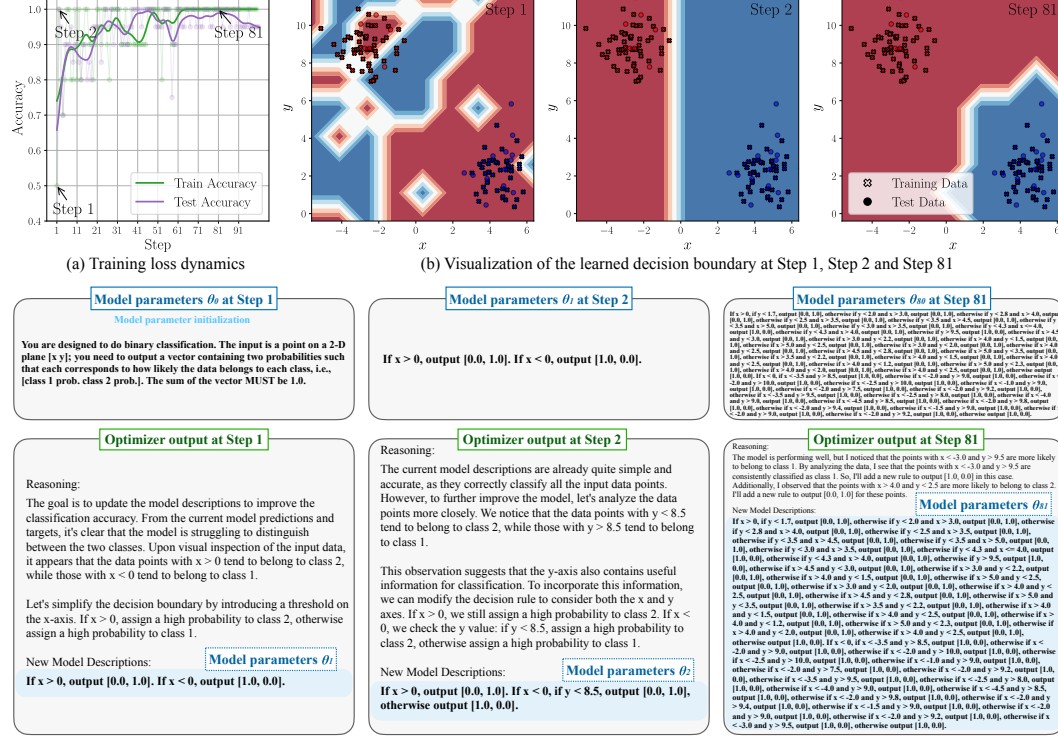

(a) Training loss dynamics

(b) Visualization of the learned decision boundary at Step 1, Step 2 and Step 81

(c) Model parameters and optimizer output at Step 1, Step 2 and Step 81

Figure 6: Linearly separable two blobs classification based on VML. (b) plots the decision boundary of model with $\boldsymbol{\theta}$ at step $i$.

terms of extrapolation. Here, we try GPT-4o, a more powerful model than Llama-3. Figure 5(b; right) shows that when $\boldsymbol{\theta}_0$ contains *only* the definition of 1-D regression, it results in a linear model after training (see (c; right)). We can *add a prior to $\boldsymbol{\theta}$ by simply saying* that the data looks like samples generated from a periodic function, which results in a very good approximation and it extrapolates much better than a neural net (see (b,c; left)). But adding the same prior to Llama-3 is not as effective (see (b,c; mid)), indicating the capability of VML depends on the capability of the underlying LLM. However, we note that, the effectiveness of VML improves along with the capability of the LLM.

## 4.4 TWO BLOBS CLASSIFICATION

We generate a linearly separable $\mathcal{D}_{\text{train}}$ from two blobs on a 2-D plane. $\boldsymbol{\theta}_0$ is initialized by *only* specifying that the task is binary classification on a 2-D plane (see Figure 6(c) Step 1). Subplot (a) shows that training is effective and that it converges. At step 1, optimizer$_1$ *says* its inspection of the current batch of data has the pattern that data points with $x > 0$ belong to class 2, and data points with $x < 0$ belong to class 1; hence it updates model$_1$ to have a linear decision boundary at $x = 0$, which happens to be perfect. However, Figure 6(a) shows that the training loss does not immediately converge. We can investigate the cause and "*debug*" the optimizer by looking at what optimizer$_2$ says. From (c) Step 2, we see that optimizer$_2$ *says* model$_1$ is already quite simple and accurate, but it wants to further improve the model and utilize the new information from the current batch. Guided by this reasoning, model$_{80}$ becomes a very deep decision tree, and the decision boundary has a reasonable margin towards the data (see Figure 6(b, c; right)).

## 4.5 TWO CIRCLES CLASSIFICATION

We generate a non-linearly separable $\mathcal{D}_{\text{train}}$ by creating data points on two concentric circles for the two classes. Besides the definition of binary classification on a 2-D plane, we also add a sentence to encode our inductive bias that the decision boundary is a circle into $\boldsymbol{\theta}_0$ (see Figure 7(c) Step 1). At step 1, optimizer$_1$ utilizes the prior information, and updates model$_1$ to have a circle decision boundary. For the rest of the training step, the optimizer mainly tries to find a good fit for the radius and the center of the decision boundary. At step 41, optimizer$_{41}$ *says* model$_{40}$ seems to be a good fit for the data, and no changes are needed, hence, it uses the same $\boldsymbol{\theta}_{40}$ for model$_{41}$ . Without the prior, VML can also learn a good model, but the performance shows large variance at the beginning

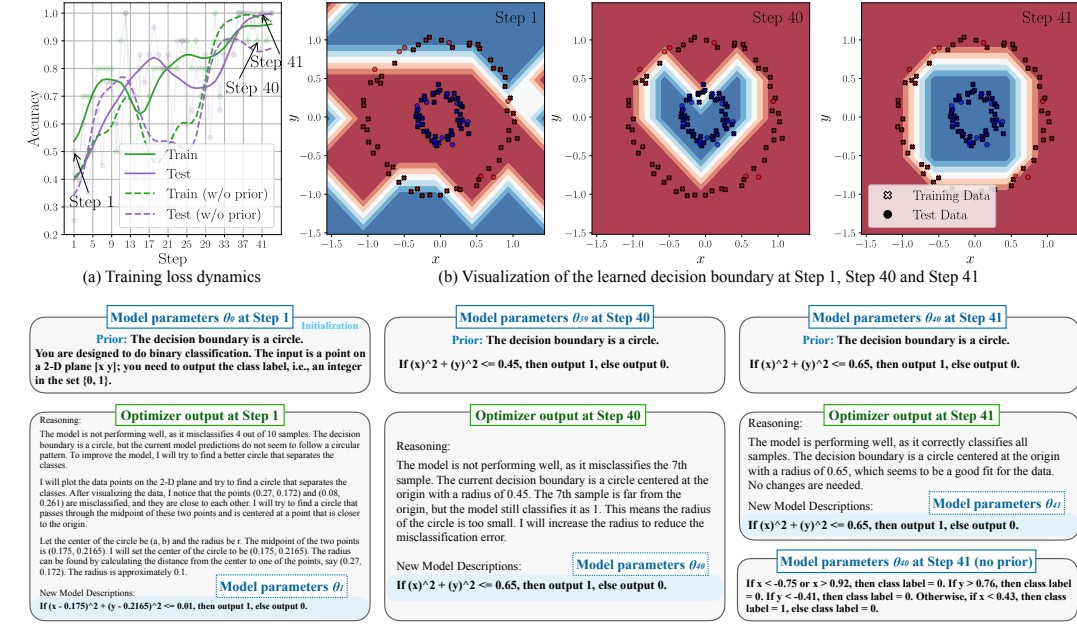

(a) Training loss dynamics

(b) Visualization of the learned decision boundary at Step 1, Step 40 and Step 41

(c) Model parameters and optimizer output at Step 1, Step 40 and Step 41

Figure 7: Non-linearly separable two circles classification with a prior in $\boldsymbol{\theta}$. (a; dashed) and (c; bottom right) also show results without the prior.

of training (see Figure 7(a; dashed)) due to the model class selection process similar to Figure 3(a). Figure 7(c; bottom right) shows the resulting $\boldsymbol{\theta}_{40}$ without the prior, which is a decision tree.

## 4.6 QUALITATIVE COMPARISON BETWEEN PROMPT OPTIMIZATION AND VML

To differentiate VML from prompt optimization, we qualitatively compare VML to a popular prompt optimization method called Automatic Prompt Engineer (APE) [65] on two tasks.

**Linear regression** as in Section 4.1. Figure 8(a) shows that the result from APE is vague and general. Such a description can easily be derived by humans through vi-

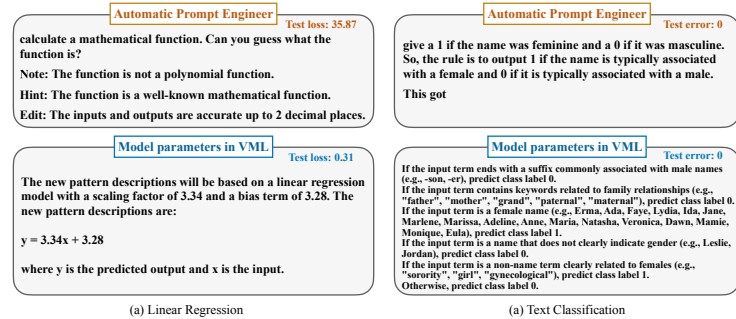

(a) Linear Regression

(a) Text Classification

Figure 8: VML versus a prompt optimization method (Automatic Prompt Engineer [65]).

sual inspection of the data, and it does not learn deeper insights from the data, whereas VML is able to learn useful new information that is difficult to derive by visual inspection of the data. We can see that VML is doing pattern recognition, which is different from naive prompt optimization.

**Text classification**. Adopted from the Google BIG-bench[4], the task is to classify whether a name is more likely to be associated to female or male. Figure 8(b) shows that APE does return a correct description of the task, but it is, once again, very general. Conversely, VML is able to learn more detailed knowledge about the data pattern which cannot be done easily through visual inspection.

## 4.7 MEDICAL IMAGE CLASSIFICATION

To demonstrate the capability of VML beyond simple machine learning problems, we include an experiment to demonstrate the effectiveness of VML in image classification. We use GPT-4o, which supports visual inputs, to take into account both image and text data. The task is to classify whether an input X-ray image has indications of pneumonia or not, see Figure 9(b) for image examples. Due to the cost of requesting GPT-4o, we create a subset of the dataset PneumoniaMNIST [56]. Our dataset consists of 100 training data and 100 test data (half pneumonia and half normal for both sets). Models are trained for 5 epochs. We try out two different model parameter initializations, one

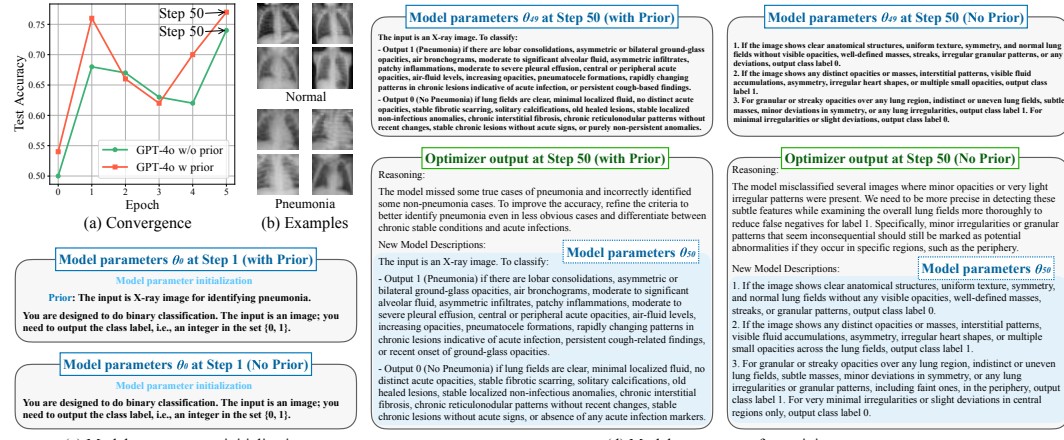

Figure 9: Tiny-PneumoniaMNIST image classification for models with and without prior at initialization.

with prior and one without. We encode the inductive bias by simply adding a sentence as the prior, which states that the input is an X-ray image for identifying pneumonia, along with the definition of binary image classification (see Figure 9(c)). The test accuracy in (a) shows that both models are able to improve their performance on the task as the training epoch increases, and the model initialized with prior also outperforms the model without (in terms of both testing accuracy and training convergence). Additionally, by inspecting the parameters of $\text{model}_{50}$ (see (d)), we can observe that the model parameters $\theta_{50}$ for the learner *with prior* has more medical domain knowledge associated to features of pneumonia (such as "acute infection", "pneumatocele formation"), while the model parameters $\theta_{50}$ for the learner *without any prior* mainly use generic visual knowledge associated to features of lung (such as "visible opacities", "uniform texture"). This observation well validates the effectiveness of using natural language to describe and encode inductive bias. More importantly, our experiment demonstrates the usefulness of learning in VML (*i.e.*, the generalization performance can be improved over time), which is also one of the key differences to existing prompt engineering methods. Additionally, the interpretable nature of the learned model parameters in VML is crucial for applications in medical domain. The learned models can be validated by medical professionals, and their predictions are grounded by their verbalized reasonings.

## 4.8 ABLATION STUDY AND EXPLORATORY EXPERIMENTS

**Quantitative comparison to in-context learning**. Since VML can be viewed as a generalization of ICL, we compare VML to ICL in all previous applications. Results are given in Table 1. The ICL results are chosen from the best across 5 runs. The metrics used for regression (Reg) and classification (Cls) are mean square error (MSE ↓) and test accuracy (↑), respectively. We abbreviate linear regression as Reg-L, polynomial regression

| Task | Reg-L(↓) | Reg-P(↓) | Cls-TB(↑) | Cls-TC(↑) | Cls-MI(↑) |
|------|----------|----------|-----------|-----------|-----------|
| ICL  | 0.38     | 62.96    | **100%**  | **95%**   | 48%       |
| VML  | **0.12** | **2.38** | **100%**  | **95%**   | **74%**   |

Table 1: Comparison between VML and ICL on all previous applications (without adding prior information).

as Reg-P, two blob classification as Cls-TB, two circle classification as Cls-TC and medical image classification as Cls-MI. The results show that VML consistently outperforms ICL in all scenarios.

**Scaling effect with stronger LLMs**. We are interested in whether the performance of VML can be improved using a stronger LLM. We use Llama-3.1 with different size (8B, 70B, 405B) as the backbone LLM for VML. From Figure 10(a), we see that stronger LLMs (*e.g.*, 405B) learn faster and achieve lower loss in the linear regression setting (Section 4.1).

**Direct vs. indirect parameterization**. As discussed in Section 3.4, we have a direct and indirect way to parameterize the optimizer. We compare both parameterization using the linear regression setting in Section 4.1. Figure 10(b) shows that the direct pa-

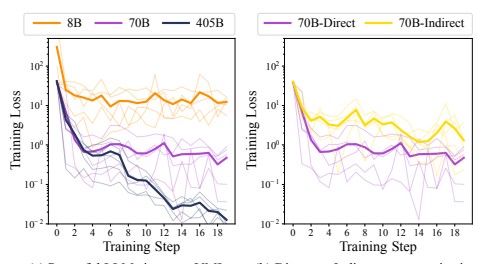

Figure 10: Training loss for ablation study. For each configuration, we show 5 individual runs (thin) and their mean (thick).

rameterization outperforms the indirect one. The direct parameterization is also more efficient and requires less LLM calls. Detailed experimental settings and discussions are given in Appendix C.3.

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

# Appendix

## Table of Contents

# A    CONCLUDING REMARKS, LIMITATIONS, AND FUTURE DIRECTIONS

Our paper introduces a verbalized way to perform machine learning and conducts several case studies on regression and classification tasks. The experiments show that VML can effectively perform these classical machine learning tasks, validating the potential of language models as function approximators. Despite the empirical effectiveness, there are a few limitations that remain to be addressed. First, training in VML still suffers from a relatively large variance. This is partially due to the stochasticity from the LLM inference, as well as the prompt design of the optimizer. Second, the output numerical error in LLMs results in inevitable fitting error. Concretely, even if the LLM correctly understands the underlying symbolic expression, there is still an output numerical error when performing inference on specific input values. This also suggests the intrinsic difficulty within LLMs to properly understand numbers (see [37, 61]). Third, the input data dimensionality and batch size are limited by the context window of LLMs, preventing VML from processing high-dimensional data or optimizing with a large batch size.

One future direction is to study various aspects in VML using insights and concepts from classical machine learning. Some interesting questions include: Can we find a better design for the optimizer so that the training is more robust and efficient? How does the optimization landscape in VML differ from classical ML, what does it look like? Another interesting direction is to investigate the learning dynamics of VML, and compare it with how human learns. Since human also has a language model in mind, the same experiments in the paper can be conducted on human through messaging software.

# B    MORE CASE STUDY: DIGIT PATTERN DISCOVERY

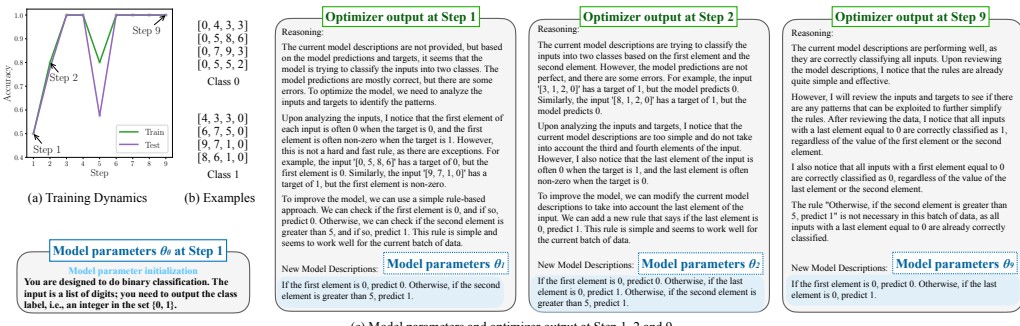

Figure 11: Binary classification for vectors of 4 digits.

To further demonstrate the interpretability of VML, we create a binary classification task on vectors of 4 digits. Class 0 contains vectors that only have digit '0' in the first position, and Class 1 contains vectors that only have digit '0' in the last position (see Figure 11(b)). Our dataset consists of 100 training data and 20 test data (half for both classes). Models are trained for 5 epochs (*i.e.*, 50 steps with batch size 10).

Figure 11(a) shows that both the training and test accuracy improves with the number of steps, hence learning is effective. The model is initialized with the definition of the task. During step 1, the optimizer says it notices that the first element of each input is often '0' when the ground truth label is '0', and decides to use a rule-based approach (see (c)). The resulting model description is half correct, which captures the pattern that 'if the first element is 0, predicts 0'. After a few more steps, the optimizer is able to learn the correct description: 'If the first element is 0, predicts 0. Otherwise, if the last element is 0, predict 1.' Compared to the regression and 2D plane classification results, the learned model here is more interpretable than learning a neural network. Also, without any prior information, one will normally choose a universal approximator such as a neural network to solve this task, which will perform equally well but certainly not as interpretable.

We also evaluate the performance of in-context learning (ICL) for this task as a baseline. Our result shows that VML is able to achieve **100% test accuracy** with an interpretable description of the pattern, while ICL can only achieve 87.5% and does not explicitly output a pattern description.

## C    DETAILS FOR ABLATION STUDY AND EXPLORATORY EXPERIMENTS

Here we provide additional details for the experiments in Section 4.8.

### C.1    COMPARISON BETWEEN IN-CONTEXT LEARNING AND VML

In-context learning (ICL) is a popular method for adapting LLMs to downstream tasks. Here, we compare the performance of VML and ICL in various tasks from previous sections. For all tasks, we provide the entire training set as in-context examples, and query the individual test data independently. The resulting predictions for regression and 2D classification are plotted in Figure 12. The full comparison between VML and ICL are shown in Table 2. We can see that VML outperforms ICL in regression and medical image classification, and has the same performance to ICL in the simpler classification tasks, *e.g.*, two blobs and two circles. Within our framework, ICL can be understood as a *nonparameteric* method, while VML is a *parameteric* one (see Appendix I.3 for more discussion).

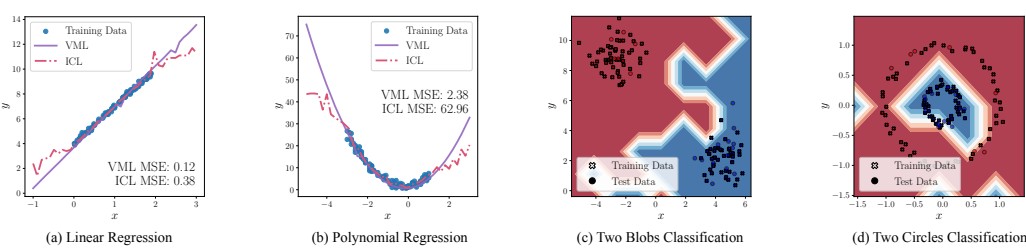

| (a) Linear Regression | (b) Polynomial Regression | (c) Two Blobs Classification | (d) Two Circles Classification |

Figure 12: Predictions of in-context learning (ICL) for the same regression and classification tasks with Llama-3 70B.

Table 2: Test performance for in-context learning (ICL) and verbalized machine learning (VML) on various tasks from previous section (without adding prior information). The ICL results are chosen from the best across 5 runs. The metrics used for regression (Reg) and classification (Cls) are mean square error (MSE ↓) and test accuracy (↑) correspondingly.

| Task | (↓) Reg-Linear | (↓) Reg-Poly. | (↑) Cls-Two Blobs | (↑) Cls-Two Circles | (↑) Cls-Medical Img |
|------|---------------|---------------|-------------------|---------------------|---------------------|
| VML  | **0.12**      | **2.38**      | **100%**          | **95%**             | **74%**             |
| ICL  | 0.38          | 62.96         | **100%**          | **95%**             | 48%                 |

### C.2    LARGER AND MORE POWERFUL LLMS LEARN FASTER AND BETTER

To verify whether the performance of VML scale with the capability of LLMs, we compare three Llama-3.1 models of different sizes, *i.e.*, 8B, 70B, and 405B, in the linear regression setting. Figure 13 shows the training loss of 5 individual runs (thin) and their mean (thick) for each LLM. Note that due to the high variance nature of using LLMs for optimization, we select the 5 best runs out of 10 runs for this comparison. We see that more powerful LLMs (*e.g.*, 405B) learn faster and achieve lower training loss.

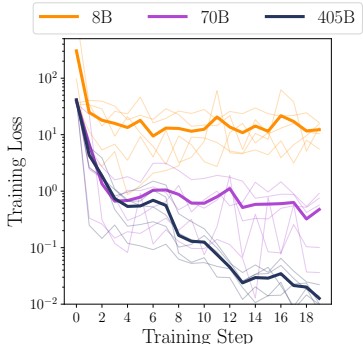

Figure 13: Llama-3.1 LLMs scale versus VML training performance in linear regression setting. 5 individual runs (thin) and mean (thick) for each LLM.

## C.3 DIRECT AND INDIRECT OPTIMIZATION

There are different ways to implement the optimization step in VML. We choose to directly update the model parameters $\boldsymbol{\theta}$ in a single LLM call by providing all the necessary information, *i.e.*, $\boldsymbol{\theta}_i = f_{\text{opt}}\big(\{\boldsymbol{x}_m, \hat{y}_m, y_m\}_{m=1}^M, \boldsymbol{\theta}_{i-1}; \boldsymbol{\psi}\big)$ in Algorithm 1. If we choose a lower abstraction level, we can decompose the *direct* single step optimization into *indirect* multi-step optimization. Algorithm 2 illustrates how $f_{\text{opt}}$ can be decomposed into four consecutive functions, which resemble the operations of computation graphs in most numerical machine learning frameworks. Specifically, we calculate the following step-by-step: (1) the quality of the predictions (*i.e.*, evaluate the loss function $f_{\text{loss}}$); (2) the 'gradient' of the loss $\ell$ w.r.t. the predictions $\hat{\boldsymbol{y}}$ denoted as $\partial\ell/\partial\hat{\boldsymbol{y}}$; (3) the 'gradient' of the loss $\ell$ w.r.t. the parameters $\boldsymbol{\theta}_{i-1}$ denoted as $\partial\ell/\partial\boldsymbol{\theta}_{i-1}$; (4) update the current $\boldsymbol{\theta}_{i-1}$ to $\boldsymbol{\theta}_i$ using the 'gradient' $\partial\ell/\partial\boldsymbol{\theta}_{i-1}$. The 'gradients' here are known as 'textual gradients' in prompt optimization literature [35, 62], which are essentially text-based feedback from LLMs.

We compare the two approaches in the linear regression setting using Llama-3.1 70B. Figure 14 shows, for both the direct and indirect optimization, the training loss of 5 individual runs (thin) and their mean (thick). We can see that the indirect method performs slightly worse than the direct method. The reason can be there are 3 more prompt templates to design, which is harder than designing just one, and has a higher risk of losing information in the pipeline.

---

**Algorithm 2** Decomposed $f_{\text{opt}}$

---

Current parameters $\boldsymbol{\theta}_{i-1}$, batch of data and predictions $\{\boldsymbol{x}_m, \hat{y}_m, y_m\}_{m=1}^M$, objective $\boldsymbol{\psi}$;

$\ell = f_{\text{loss}}(\{\hat{y}_m, y_m\}_{m=1}^M; \boldsymbol{\psi})$;
$\frac{\partial\ell}{\partial\hat{\boldsymbol{y}}} = f_{\text{grad}}(\ell, \hat{\boldsymbol{y}})$;
$\frac{\partial\ell}{\partial\boldsymbol{\theta}} = f_{\text{grad}}(\frac{\partial\ell}{\partial\hat{\boldsymbol{y}}}, \boldsymbol{x}, \hat{\boldsymbol{y}}, \boldsymbol{\theta}_{i-1})$;
$\boldsymbol{\theta}_i = f_{\text{update}}(\boldsymbol{\theta}_{i-1}, \frac{\partial\ell}{\partial\boldsymbol{\theta}})$;

---

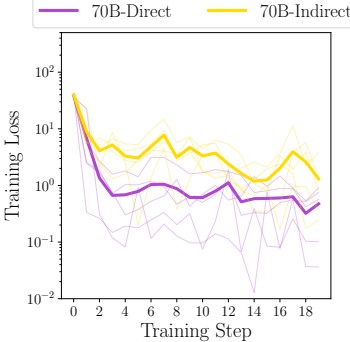

Figure 14: Training loss of direct and indirect optimization in linear regression setting using Llama-3.1 70B. The lines show 5 individual runs (thin) and mean (thick) for each approach.

# D    EFFECT OF ACCURATE LOSS FEEDBACK

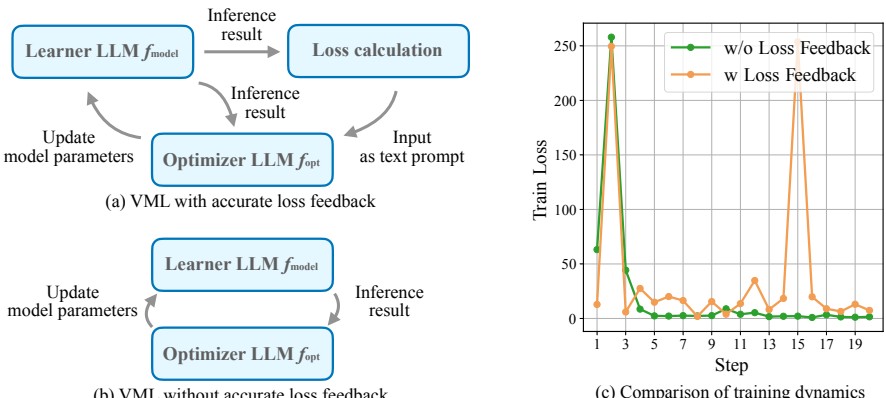

(a) VML with accurate loss feedback

(b) VML without accurate loss feedback

(c) Comparison of training dynamics

Figure 15: Training dynamics for two different optimization settings in the polynomial regression setting. One has access to the accurate loss computation, and the other does not.

The VML algorithm at Algorithm 1 specifies that the arguments for $f_{\text{opt}}(\cdot)$ consist of the inputs $\boldsymbol{x}$, the predictions $\hat{y}$, the targets $y$, the current model parameter $\boldsymbol{\theta}_{i-1}$ and the optimizer configurations $\boldsymbol{\psi}$. Hence, there is no explicit definition of the loss function for the optimizer (see Figure 2(right) for an example of the verbalized loss function). It is up to the optimizer itself to evaluate the difference between the prediction $\hat{y}$ and the target $y$. We are interested in question that whether having access to the real training loss (defined and computed for logging purpose), mean squared error in this case, can help the optimizer to better navigate the training trajectory.

The orange line in Figure 15(c) shows that having such accurate loss feedback might not help, and might even decrease the performance in this scenario. One possible explanation is that the single loss value itself does not contain too much information. Moreover, as the exact form of the loss function can be fed to LLM easily, the LLM might spend additional efforts to estimate the exact form of the loss function, which makes the convergence even more difficult. It actually makes intuitive sense that verbalized loss function (*i.e.*, using natural language to explain the target of the loss function) works better in the VML framework. For example, knowing how does each prediction contributes to the loss value can be more informative and a single overall loss value, since the model might be doing well for some data but not the others, and we only want to improve the model for points with the bad predictions.

# E NUMERICAL ERROR OF LLMS IN REPRESENTING SYMBOLIC FUNCTIONS

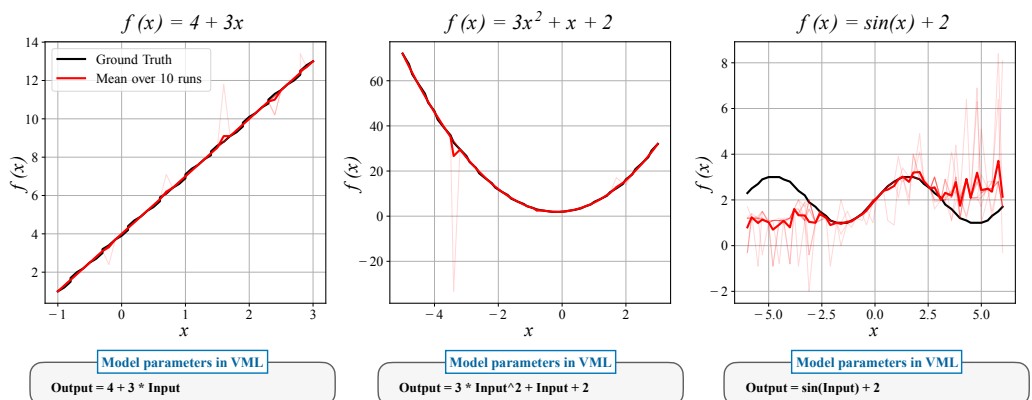

Figure 16: Functions evaluations and numerical error in Llama-3 70B

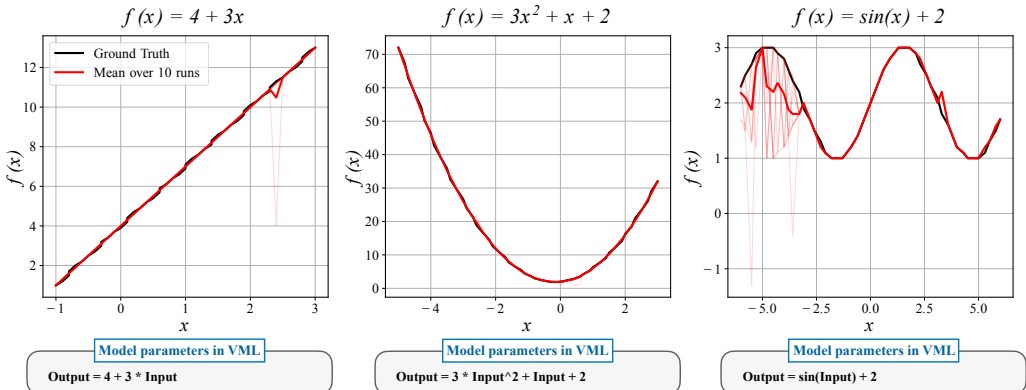

Figure 17: Functions evaluations and numerical error in GPT-4o.

LLMs are designed to do language modeling, rather than exact calculations. Hence, their performance on evaluating functions can be unreliable, and might result in error. Figure 16 shows that Llama-3 is very comfortable in evaluating the given linear and polynomial function, as the mean is quite accurate. The variance over 10 runs is also pretty small, except for one or two points. However, for a more complex function such as $\sin(x)$, Llama-3 is only able to return small error approximately in the range of $x \in (-2, 2)$. Both the error and the variance are large out side of this range. This explains the non-smoothness for the function in Figure 5(b; right), which has $\sin(x + 1.0)$ in the learned model parameters.

By switching to the more powerful model, GPT-4o, we can see from Figure 17 that both the error and the variance decrease. In particular, for $\sin(x)$, GPT-4o returns smaller error in a larger range, (*i.e.*, $x \in (-2.5, 5.0)$). This implies that as the capability of LLMs improves, their performance in evaluating more complex functions also improves.

Nevertheless, this is currently still a limitation for VML if the optimizer chooses to use complex mathematical functions as the model parameter. If the evaluation of the function has an error, then during training, the optimizer will update the model parameters based on noisy signal. This can lead to large variance in training and slow convergence. Future work should look into methods for minimizing the numerical error in LLMs function evaluation.

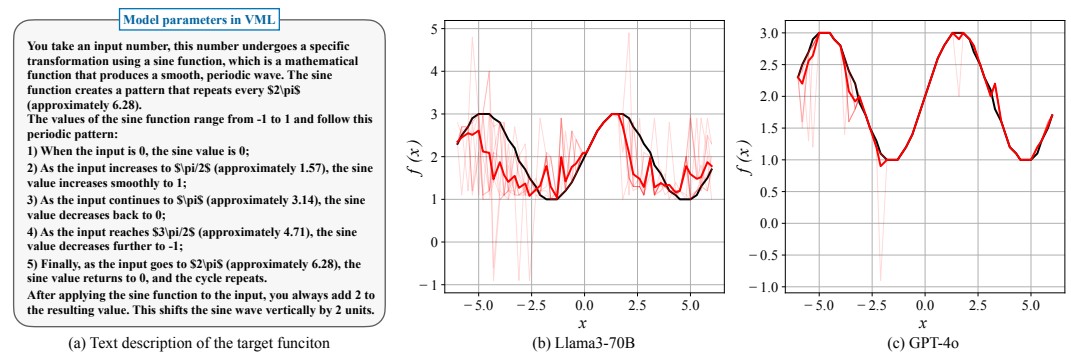

(a) Text description of the target funciton   (b) Llama3-70B   (c) GPT-4o

Figure 18: Function evaluations based on the natural language description of the corresponding symbolic sine function.

Figure 18 shows that if we use natural language to describe the symbolic sine function (see sub-figure(a)), GPT-4o is able to produce more accurate evaluations than using the symbolic function (see (c)). The accuracy of Llama-3 70B also increases, even though it still under performs GPT-4o (see (b)). This is likely due to Llama-3 is less capable in instruction following than GPT-4o. This observation implies that in VML, we might want to instruct the optimizer to avoid using complex symbolic functions in the update and to prefer the natural language description of the function.

# F    MITIGATING NUMERICAL ERROR BY TOOL CALLING

In this section, we supplement experiments of Llama-3 70B with a python interpreter. Despite the fact that LLMs are able to perform numerical data tasks, the incorporation of a python interpreter further improves LLMs ability to deal with numerical values. Specifically, we use the *open-interpreter*[1] library to add a python interpreter to Llama-3 70B, such that the LLM has the ability to use python programs to evaluate symbolic functions or perform numerical operations. We follow the same experimental settings as in Section 4.3 (sinusoidal regression of $y = \sin(x) + 2$). The training data is only sampled from $[-3, 3]$ with additive Gaussian noise. The in-domain testing data is sampled from the same range, while the out-of-domain testing data is sampled from $[-6, -3]$ and $[3, 6]$.

The results are given in Table 3. We can observe that with the python interpreter, Llama-3 70B can effectively learn periodic functions, while in the original experiment (*i.e.*, Figure 5(b)), the same LLM is unable to approximate periodic function even with a prior. The results show that the tool-using ability can further improve the learnability of VML. The example logs for inference with the learned model is showed below.

Table 3: Evaluation (using mean squared error ↓) on sinusoidal regression as in Figure 5(b) for three different models including (1) neural networks, (2) Llama3 with prior, and (3) *Llama3 with prior and code interpreter*.

| Test Domain | Neural Nets | Llama3 with prior | Llama3 with prior and code interpreter |
|---|---|---|---|
| In Domain | 0.0073 | 0.7385 | **1.0e-05** |
| Out of Domain | 0.5840 | 11.1735 | **1.0e-05** |
| Overall | 0.2345 | 5.956 | **1.0e-05** |

---

**Inference History: Llama-3 70B with Code Interpreter – Example Logs**

——————— Forward Pass Input ———————
You are the model. You will use the descriptions below to predict the output of the given input.
**\*\* Pattern Descriptions: \*\***
You are designed to do regression, i.e., to predict the output of any given input. Both input and output are real numbers. It looks like the data is generated by a periodic function. \ ( y = 1 \sin(1x + 0) + 2\)
**\*\* Input: \*\***
[-0.09]
Please give your FINAL output **\*\*STRICTLY IN THE FOLLOWING FORMAT\*\*** at the end:
```
Explanations: [Your step-by-step analyses and results; please use python for evaluating symbolic function]
Output:
[Your output MUST be in REAL NUMBER ROUNDED TO TWO DECIMAL POINTS; it MUST be in the same format as the Input]
```
Please ONLY reply according to this format, don't give me any other words.
——————— Forward Pass Output ———————
**\*\* Step 1: Define the function \*\***
The given function is \\(y = 1 \\sin(1x + 0) + 2\\). This is a simple sine function with an amplitude of 1, a period of \\(2\\pi\\), and a phase shift of 0. The function is then vertically shifted by 2 units.
**\*\* Step 2: Evaluate the function at the given input \*\***
The input is \\(x = -0.09\\). We need to evaluate the function at this input.

```python
import math

def evaluate_function(x):
    return 1 * math.sin(1 * x + 0) + 2

x = -0.09
result = evaluate_function(x)
print(result)
```

```
1.910121450801989
```

**\*\* Explanations: \*\*** We defined the function y = 1 * sin(1x + 0) + 2 and evaluated it at the given input x = -0.09. The result of the function evaluation is approximately 1.910121450801989.
**\*\* Output: \*\*** [1.91]
——————— Forward Pass Output ———————
**\*\* Explanations: \*\***
We defined the function y = 1 * sin(1x + 0) + 2 and evaluated it at the given input x = -0.09. The result of the function evaluation is approximately 1.910121450801989.
**\*\* Output: \*\***
[1.91]

---

[1]GitHub Repo: https://github.com/OpenInterpreter/open-interpreter

# G  CONNECTION BETWEEN PREDICTION VARIANCE AND MODEL PARAMETERS IN VML

## G.1  FROM VAGUE TO CONCRETE MODEL PARAMETERS

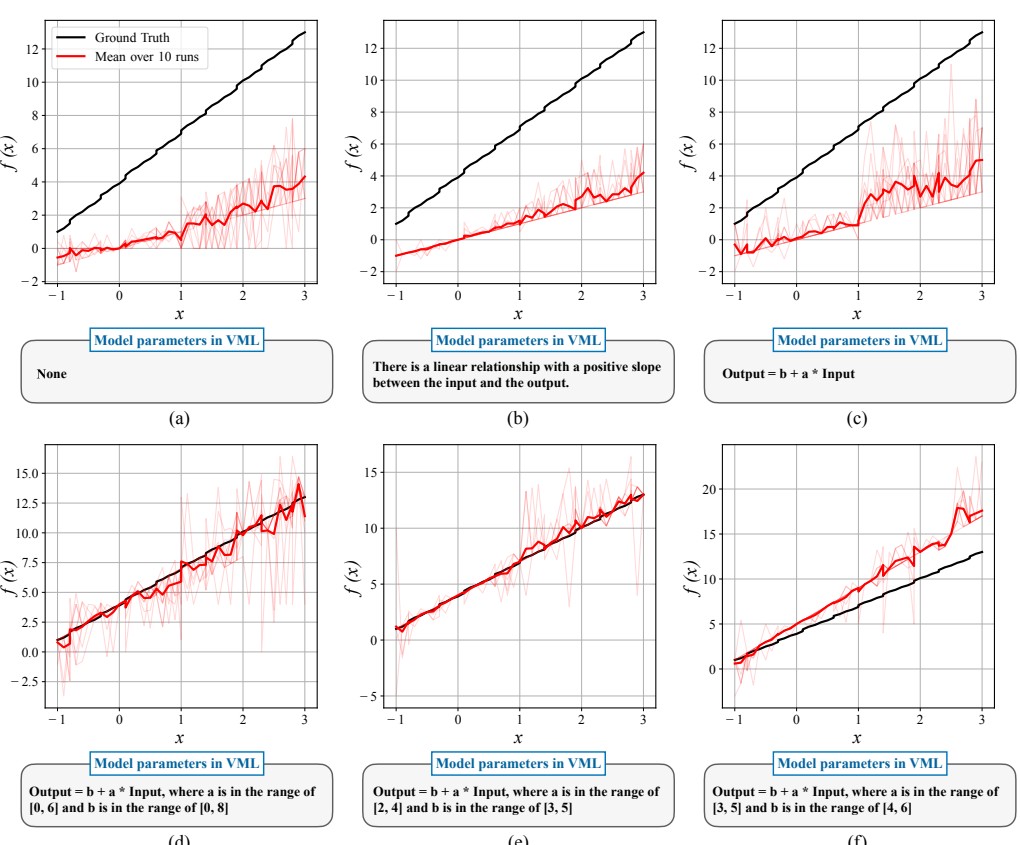

Figure 19: Evaluations on model parameters using vague to concrete descriptions. Results are over 10 runs. The base LLM is Llama-3-70B.

The model parameters generated by a VML optimizer can be vague or concrete. We are curious for those with vague descriptions, how would the LLM evaluations look like, and whether they have large variance. Figure 19 shows the results on Llama-3 70B for six different model descriptions, including:

(a)  None

(b)  "There is a linear relationship with a positive slope between the input and the output."

(c)  "Output = b + a * Input"

(d)  "Output = b + a * Input, where a is in the range of [0, 6] and b is in the range of [0, 8]"

(e)  "Output = b + a * Input, where a is in the range of [2, 4] and b is in the range of [3, 5]"

(f)  "Output = b + a * Input, where a is in the range of [3, 5] and b is in the range of [4, 6]"

(a) shows that if we only provide the information that the task is a regression task and do not specify the model at all, the LLM tends to predict a linear function (slope $\approx 1$) with increasing variance as $x$ moves away from 0. (b) shows that if we specify there is a linear relationship between inputs and outputs, the LLM will predict a linear function with a similar slope as (a) but with smaller variance. (c) shows that if we specify the explicit form of the linear function, the slope will still be around 1, but the variance are larger when $x > 1$. (d, e, f) show that by providing a range for the values of the unknown variables, the LLM tends to use the mid-point of the range for the values, and a smaller range does correspond to a smaller variance in prediction.

## G.2 SEMANTIC INVARIANCE OF MODEL PARAMETERS

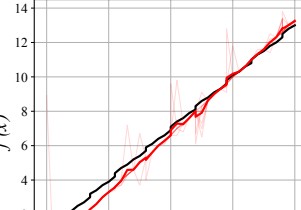
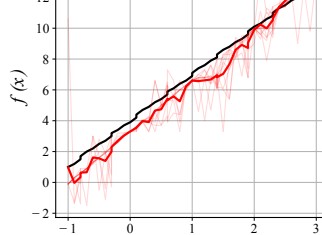
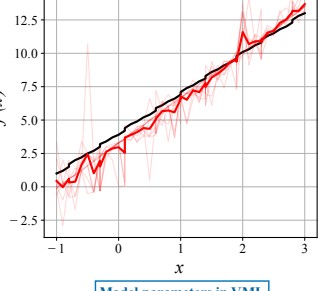

**Model parameters in VML**

(a) The new pattern descriptions will be based on a linear regression model with a scaling factor of 3.34 and a bias term of 3.28. The new pattern descriptions are:

y = 3.34x + 3.28

where y is the predicted output and x is the input.

**Model parameters in VML**

(b) The updated pattern definitions utilize a linear regression framework characterized by a slope of 3.34 and an intercept of 3.28. The revised pattern equations are expressed as:

y = 3.34x + 3.28

where y denotes the estimated output, and x represents the input variable.

**Model parameters in VML**

(c) y = 3.34x + 3.28

**Model parameters in VML**

(d) The new pattern descriptions will be derived from a model that predicts outcomes based on input values. This model adjusts the input by a factor of 3.34 and adds a constant value of 3.28 to generate the final prediction. Here, the predicted result is determined by this specific adjustment and addition applied to the input.

**Model parameters in VML**

(e) Amidst the dance of numbers, a new design takes form,
Where linear paths converge with elegance and charm.
A scaling factor whispers, 'Three point three four,'
And bias gently murmurs, 'Three point two eight,' no more.
With inputs cradled softly, the pattern does reveal,
A future sketched in numbers, a prophecy made real.
Y, the destined output, unfolds from X's grace,
In linear harmony, they find their rightful place.
So here it is, the song of y and x entwined,
A mathematical ballet, precision redefined.

**Model parameters in VML**

(f) 新图景描述，依循回归模型之法，带有倍数三点三四，偏差三点二八。新图景描绘如下：
心数乘以三点三四，再加上三点二八，便得其意境。
此中，心数为所输入，意境为所输出。

Figure 20: Evaluation on the model parameters from Figure 3(c; Step 15) using six different rephrasing with the same semantic meaning. Results are over 10 runs. The base LLM is Llama-3-70B.

In natural language, there are different ways to express a concept with the same semantic meaning. Hence, the model parameters generated by a VML optimizer might vary a lot between runs, even though they are semantically invariant. We are curious whether such variance in descriptions will lead to variance in model evaluations. Figure 20 shows that results on Llama-3 70B for six different but semantically invariant descriptions of the model from Figure 3(c; Step 15), *i.e.*,:

(a) "The new pattern descriptions will be based on a linear regression model with a scaling factor of 3.34 and a bias term of 3.28. The new pattern descriptions are:

y = 3.34x + 3.28

where y is the predicted output and x is the input."

(b) "The updated pattern definitions utilize a linear regression framework characterized by a slope of 3.34 and an intercept of 3.28. The revised pattern equations are expressed as:

y = 3.34x + 3.28

where y denotes the estimated output, and x represents the input variable."

(c) "y = 3.34x + 3.28"

(d) "The new pattern descriptions will be derived from a model that predicts outcomes based on input values. This model adjusts the input by a factor of 3.34 and adds a constant value of 3.28 to generate the final prediction. Here, the predicted result is determined by this specific adjustment and addition applied to the input."

(e) "Amidst the dance of numbers, a new design takes form,
Where linear paths converge with elegance and charm.
A scaling factor whispers, 'Three point three four,'
And bias gently murmurs, 'Three point two eight,' no more.

With inputs cradled softly, the pattern does reveal,
A future sketched in numbers, a prophecy made real.
Y, the destined output, unfolds from X's grace,
In linear harmony, they find their rightful place.

So here it is, the song of y and x entwined,
A mathematical ballet, precision redefined."

(f) "新图景描述，依循回归模型之法，带有倍数三点三四，偏差三点二八。新图景描绘如下:

心数乘以三点三四，再加上三点二八，便得其意境。

此中，心数为所输入，意境为所输出。"

These rewrites are generated by GPT-4o based on (a). The description (a) is the original $\theta_{15}$ from Figure 3(c; Step 15). (b) rephrases the descriptions from (a) slightly. (c) only keeps the symbolic equation from (a). (d) is a rewrite without using math expression. (e) uses the poetry style. (f) is a translation of (a) into Literary Chinese. The results in Figure 20(a,b,c) are similar, and have small variance across the 10 runs. The results in Figure 20(d,e,f) are also very accurate on average. However, the poetry rewrite (e) and the Chinese rewrite (f) do have slightly larger variance. Overall, we see that if the various descriptions preserve the same semantic, then their evaluations through Llama-3 70B are likely to be similar.

# H    A PROBABILISTIC VIEW ON VML

The output of a language model usually comes with randomness. In the paper, we typically consider to set the temperature in LLMs as zero to remove the randomness from sampling, which indicates that LLMs will always output the text with the largest probability (*i.e.*, largest confidence logit). However, we want to highlight that such a sampling process actually gives us another probabilistic perspective to study VML. We will briefly discuss this perspective here.

## H.1    POSTERIOR PREDICTIVE DISTRIBUTION

Because we can easily sample multiple possible model parameters by setting a proper temperature for the optimizer LLM, we view it as a way to sample multiple learner models. This is well connected to Bayesian neural networks, where Bayesian inference is applied to learn a probability distribution over possible neural networks. We start by writing the posterior predictive distribution ($\mathcal{D}$ is training data):

$$p(\hat{y}|\mathcal{D}) = \int_{\boldsymbol{\theta}} p(\hat{y}|\boldsymbol{\theta})p(\boldsymbol{\theta}|\mathcal{D})d\boldsymbol{\theta} = \mathbb{E}_{p(\boldsymbol{\theta}|\mathcal{D})}\big\{p(\hat{y}|\boldsymbol{\theta})\big\} \tag{3}$$

where we can easily sample multiple model parameters $\boldsymbol{\theta}$ and compute its probability with logits. Specifically, we have that $p(\boldsymbol{\theta}|\mathcal{D}) = \prod_{t=1}^{n} P(\theta_t|\theta_1, \cdots, \theta_{t-1}, \mathcal{D})$. Using this idea, it is actually quite easy to obtain the ensembled output that is weighted by posterior distribution.

## H.2    FROM FUNCTIONS TO STOCHASTIC PROCESSES

With non-zero temperature, we can view the output of LLMs as a sampling process from a distribution over text tokens, which means each output token can be viewed as a random variable. Then the output of LLMs is effectively a sequence of random variables, and therefore it is easy to verify that it is a stochastic process. This view makes it possible for VML to perform probabilistic modeling.

# I   DISCUSSIONS ON NATURAL LANGUAGE MODEL PARAMETERS

There are many interesting properties regarding the natural language model parameters. Many traditional machine learning models can be revisited in the scenario where model parameters are text prompts in the LLM.

## I.1   DIFFERENT MECHANISMS TO UPDATE MODEL PARAMETERS FOR DIRECT OPTIMIZATION

**Naive re-writing**. Given the model parameters $\boldsymbol{\theta}_t$ at the step $t$, the simplest way to update the model parameters at the step $t+1$ is to use whatever the optimizer generates. We denote the optimizer LLM generates the new model parameters $\boldsymbol{\theta}_{\text{new}}^t$. This is essentially

$$\boldsymbol{\theta}_{t+1} \leftarrow \boldsymbol{\theta}_{\text{new}}^t. \tag{4}$$

An simple extension to naive re-writing is to add a text prompt to instruct the optimizer LLM to take the previous model parameters $\boldsymbol{\theta}_t$ into consideration at the step $t+1$. Thus we have the conditional re-writing, namely $\boldsymbol{\theta}_{t+1} \leftarrow f_{\text{opt}}(\boldsymbol{\theta}_t)$. This is also what we use in the main paper.

**Incremental updating**. Alternatively, we can choose to update the model parameters in an incremental fashion without remove the previous model parameters completely. We denote the optimizer LLM generates the new model parameters $\boldsymbol{\theta}_{\text{new}}^t$. Then the model parameters $\boldsymbol{\theta}_{t+1}$ at the step $t+1$ is

$$\boldsymbol{\theta}_{t+1} \leftarrow \{\boldsymbol{\theta}_t, \boldsymbol{\theta}_{\text{new}}^t\}. \tag{5}$$

However, the incremental updating will make the model parameters an increasingly longer text prompt. This is not ideal since the context window of a LLM is typically quite limited. The incremental updating mechanism can be interpreted as using a small learning rate to train the learner. This will easily lead to bad local minima (because the previous incorrect model parameters will be kept and may affect the future learning as a prior knowledge in the text prompt), but it may improve the training convergence.

**Incremental updating with summarization**. To avoid the infinite increasing length of model parameters, we can instruct the optimizer LLM to summarize the previous model parameters into a fixed length. This yields

$$\boldsymbol{\theta}_{t+1} \leftarrow \{ \underbrace{\mathcal{C}(\boldsymbol{\theta}_t)}_{\text{fixed token length}}, \boldsymbol{\theta}_{\text{new}}^t\}. \tag{6}$$

where $\mathcal{C}(\cdot)$ is some text summarization scheme.

**Connection to standard optimizers**. There are many interesting connections between the optimizer LLM and the standard numerical optimizer. Usually the behavior of the optimization is determined by the optimizer parameters $\boldsymbol{\psi}$ which is also a text prompt. This is usually a text description of the target of the optimizer. For example, we can instruct the optimizer LLM to serve as the first-order optimizer (*e.g.*, momentum SGD) and feed all the necessary information into the text prompt. Then the optimizer LLM will essentially become an optimizer mapping function that maps all the necessary information (including the previous model parameters) to the model parameters of the next step. To implement the momentum in the optimizer LLM, one can simply instruct the optimizer LLM to maintain the previous model parameters as much as possible. This is to say, everything we want to implement in the optimizer are realized through text prompts. It will inevitably depend on the instruction-following ability of the LLM, and it is possible that there will be some unrealizable optimization functionalities (*e.g.*, we instruct the optimizer LLM to be a second-order optimizer and the optimizer LLM may be likely to ignore this instruction). However, we want to highlight that as LLMs become more powerful, this problem will be less and less significant. In general, implementing an advanced optimizer in VML is still an important open problem.

## I.2   OCCAM'S RAZOR, CONSTRAINED-LENGTH MODEL PARAMETERS, AND KOLMOGOROV COMPLEXITY

We are interested in how Occam's razor can be applied in VML. One natural way of doing so is to constrain the model parameters to be a small and fixed length. This essentially is

$$\boldsymbol{\theta}_{t+1} \leftarrow \{ \underbrace{\mathcal{C}(\boldsymbol{\theta}_t)}_{\text{fixed token length}}, \underbrace{\boldsymbol{\theta}_{\text{new}}^t}_{\text{fixed token length}} \}. \tag{7}$$

We can see that as long as we constrain the text token length of the model parameters to be small, the learner will perform an automatic model simplification, as it will try to discover the data pattern with concise and simple text. There are many more ways to implement the Occam's razor in VML. More interestingly, it is also possible to incorporate a structural constraint to the model parameters. For example, it can be causal knowledge (*e.g.*, text representation of a causal graph), logic formula or decision trees. Our work opens up many more possibilities on Occam's razor in VML, and rethinking the form of Occam's razor in VML is very crucial in unlocking the strong interpretability and controllability of inductive biases.

Another perspective on the length of the model parameters in VML is related to Kolmogorov complexity [15], which is defined as the shortest effective description length of an object. The principle of Occam's razor is basically saying that hypotheses with low Kolmogorov complexity are more probable [46]. By constraining the length of model parameters to be small, we are effectively trying to find the minimum description length (MDL) of a model in natural language. The theoretic Kolmogorov complexity of a model is usually impossible to compute, however, VML might provide an estimation for Kolmogorov complexity by using the shortest effective length of the learned model parameters in natural language.

### I.3 CONNECTION TO NONPARAMETRIC MODELS AND IN-CONTEXT LEARNING

*Nonparameteric* methods get around the problem of model selection by fitting a single model that can adapt its complexity to the data [48, 33, 8]. These methods allow the model complexity to grow with the number of observed data. This is different to *parametric* models which have fixed number of parameters. In VML, as showed in Section 4.2, the model complexity is also flexible and adapts to the data during training. Similarly, the concept of in-context learning (ICL) can also be understood as nonparametric methods in the lens of LLMs as function approximators. ICL denotes the method of using LLMs to solve new tasks by only providing the task demonstrations or examples in the prompt with natural language. Given a new data point, an LLM predicts its output using information in the provided demonstrations. From the perspective of VML, ICL in an LLM essentially defines a nonparametric model implicitly using the demonstrated examples in the natural language space.

# J BROADER DISCUSSIONS

In this section, we use the format of Q&A to discuss a list of interesting topics that are loosely related to VML, but are more broadly tight to the capability of LLMs. Some of the questions might seen philosophical or ideological, but were asked by fellow researchers before. Nevertheless, we still include them into this section in case some readers find them insightful.

## J.1 How is the optimizer's statement 'the function should be $y = m * x + b$' more interpretable than learning a linear function?

The interpretability from VML is on the framework itself. Using natural language to characterize the model can reveal exactly what pattern the model learns from data, which is very different from training neural networks from scratch. As for the case of regression problems in the paper, interpretability comes from (1) automatic model selection with explanations: this is different from common practice where we assume the data is linear and use a linear regression model. In our experiment, we don't have such a prior and the optimizer will learn this linear pattern purely by exploring the data. The closest equivalent from classical ML methods would be to train an "universal approximator", *e.g.*, a neural networks, which might decide to fit a function that is roughly linear, but has a lot more parameters and less interpretable; (2) another source of interpretability comes from the property that the user can easily interact with the optimizer and follow up with more questions to seek explanations.

## J.2 Is controlling the hyperparameters of LLM optimizers, such as learning rate and regularization, more difficult than controlling those of traditional ML optimizers?

Exploring the hyperparameters of LLM optimizers is important yet challenging. It is a great research question for VML. One of the reasons that VML is particularly interesting is that it brings a lot of new research questions.

LLM optimizers have both advantages and disadvantages. The precise control of learning rate and momentum can be difficult. However, adding the qualitative effect of high/slow learning rate and momentum is in fact quite easy. One can simply use language to describe it. In our optimizer prompt, we use the concept of momentum (*e.g.*, "update the model parameter without changing it too much" and provide a constant amount of optimization histories). In terms of regularization, it is also easy to add regularization to control the complexity of the model in VML, *e.g.*, the word length of the model parameters (*i.e.*, a form of Occam's razor). A qualitative hyperparameter control for LLM optimizers is simple, while this can be challenging for classic ML.

## J.3 LLMs are optimized for natural language understanding and generation, not for numerical data tasks typically associated with machine learning. Are LLMs fundamentally restricted for machine learning tasks?

Numerical data tasks are heavily studied in LLMs, for example, mathematical problem-solving. The popular MATH dataset [9] requires strong numerical data processing from LLMs, and this dataset is used as a standard evaluation benchmark for LLMs. Moreover, there exists many LLMs (*e.g.*, DeepseekMath [41], WizardMath [26]) that are capable of solving competition-level mathematics problems.

Moreover, LLMs have shown remarkable potential in numerical data tasks for machine learning, and our work is one of the first methods to reveal such a potential. Some concurrent works [40, 62] also gave empirical evidence that LLMs can be fundamentally suitable for machine learning tasks.

Verbalized machine learning aims to provide a framework for LLMs to deal with machine learning tasks, with the ability to fully interpret the learned knowledge with natural language. We believe this framework will be increasingly more powerful, as LLMs get more powerful. We have already observed the performance improvement of VML by switching from Llama-3 to GPT-4o.

### J.4 The fundamental nature of LLMs is to predict (the next token) based on a probability distribution over the vocabulary. One might argue this process is based on statistical choice rather than on true understanding. Is it meaningful to use LLMs for applications such as machine learning tasks?

We believe VML does represent a meaningful direction to explore, as there is current no evidence that LLMs can not perform ML tasks. On the other hand, we already have quite a few applications that demonstrate the effectiveness of VML (*e.g.*. medical image classification). In fact, even in-context learning can already perform a few ML tasks (as introduced by GPT-2 and GPT-3 papers [38, 6]). We believe there are a lot of applications to be unlocked in the VML framework.

Whether one should use LLMs for tasks other than language modeling is indeed an important open question, which is currently under active research with a significant number of researchers in the field investigating the boundary of LLMs' capability, and trying to explain the 'seemingly' emergence of such abilities from the simple language modeling training objective.

The argument that LLMs can not elicit true understanding due to its statistical training is debatable. Firstly, it is unclear what it means to train a model based on true understanding. One can not perform such a training without an explicit form of loss function. On the other hand, there are some analyses that show that next-token prediction induces a universal learner [28]. Secondly, we believe that there is a distinct difference between low-level statistical training and high-level knowledge understanding. Whether one can induce another is unknown and is also out of the scope of our paper.

Currently, there has already been substantial evidence that LLMs possess a form of understanding that is functionally relevant for many real-world tasks. The fact that they can consistently generate useful and accurate outputs across various domains, including numerical math [9, 1], theorem proving [57], biology [25] (just to name a few), challenges the argument that LLMs lack real understanding.

Hence, we believe to argue against the use of LLMs for tasks other than language modeling, such as math related problems, will require an equally substantial amount of empirical evidence or theoretical proof, which is missing at the moment.

### J.5 Hallucination remains a significant issue for LLMs. How can we trust them to handle complex tasks such as VML?

Even though hallucination is an observation associated with LLMs, it does not fundamentally limit the performance of VML. We rarely encounter failure of VML due to hallucination in our experiments. The model in VML is parameterized by a text prompt. Whether the learned model (*i.e.*, the text prompt) is acceptable depends on the test performance (*e.g.*, the training accuracy) of a task. If the LLM-based optimizer hallucinated during VML training, the resulting model parameter (*i.e.*, the text prompt) is unlikely to have a good performance consistently across the test dataset. If the learned good performing model parameter seems unexpected, it is more likely that the LLM-based optimizer discovered new knowledge from the training data than have hallucinated.

Additionally, we see that hallucination becomes less of a problem today than a year ago, due to the more powerful model and continuous efforts from the community (*e.g.*, a lot of LLM alignment and scalable oversight methods are proposed recently). We believe VML is actually an orthogonal & independent contribution to existing LLM research topics such as hallucination. VML studies how to enable interpretable learning using natural language. As LLM gets more powerful and more faithful, VML will also become more useful.

# K COMPLETE TRAINING TEMPLATE AT INITIALIZATION

## K.1 LINEAR REGRESSION

**Text prompt template for the learner**

You are the model. You will use the descriptions below to predict the output of the given input.

\*\* Pattern Descriptions: \*\*
**You are designed to do regression, i.e., to predict the output of any given input. Both input and output are real numbers.**

\*\* Input: \*\*
**[$Data]**

Please give your output strictly in the following format:
```
Explanations: [Your step-by-step analyses and results]
Output:
[Your output MUST be in REAL NUMBER ROUNDED TO TWO DECIMAL POINTS; make necessary assumptions if needed; it MUST be in the same format as the Input]
```

Please ONLY reply according to this format, don't give me any other words.

**Text prompt template for the optimizer**

You are the optimizer for a model, your goal is to learn the best descriptions for the model. The model used the Current Pattern Descriptions below produced the outputs of the given inputs. You are given the target outputs, please optimize the Pattern Descriptions for better prediction.

\*\* Inputs (a batch of i.i.d. data): \*\*
**[[$Data] [$Data] [$Data] [$Data] [$Data] [$Data] [$Data] [$Data] [$Data] [$Data]]**

\*\* Current Pattern Descriptions: \*\*
**You are designed to do regression, i.e., to predict the output of any given input. Both input and output are real numbers.**

\*\* The model outputs: \*\*
**[[$Prediction] [$Prediction] [$Prediction] [$Prediction] [$Prediction] [$Prediction] [$Prediction] [$Prediction] [$Prediction] [$Prediction]]**

\*\* The target outputs: \*\*
**[[$GroundTruth] [$GroundTruth] [$GroundTruth] [$GroundTruth] [$GroundTruth] [$GroundTruth] [$GroundTruth] [$GroundTruth] [$GroundTruth] [$GroundTruth]]**

**If the model is doing well, you can keep using the current descriptions. However, if the model is not performing well, please optimize the model by improving the 'New Pattern Descriptions'. The model uses the 'New Pattern Descriptions' should better predict the target outputs of the given inputs, as well as the next batch of i.i.d. input data from the same distribution. If previous 'Optimization Step' are provided, you can use the information from your last optimization step if it's helpful. Please think step by step and give your outputs strictly in the following format:**

```
Reasoning:
[be explicit and verbose, improve the Current Pattern Descriptions by yourself;]

New Pattern Descriptions:
[put your new descriptions here; MUST be specific and concrete;]
```
Please ONLY reply according to this format, don't give me any other words.

Figure 21: Prompt templates of VML for the learner and optimizer for the linear regression (Llama-3-70B without prior).

## K.2 POLYNOMIAL REGRESSION

**Text prompt template for the learner**

You are the model. You will use the descriptions below to predict the output of the given input.

** Pattern Descriptions: **

**You are designed to do regression, i.e., to predict the output of any given input. Both input and output are real numbers.**

** Input: **

**[$Data]**

Please give your output strictly in the following format:
```
Explanations: [Your step-by-step analyses and results]
Output:
[Your output MUST be in REAL NUMBER ROUNDED TO TWO DECIMAL POINTS; make necessary assumptions if needed; it MUST be in the same format as the Input]
```
Please ONLY reply according to this format, don't give me any other words.

**Text prompt template for the optimizer**

You are the optimizer for a model, your goal is to learn the best descriptions for the model. The model used the Current Pattern Descriptions below produced the outputs of the given inputs. You are given the target outputs, please optimize the Pattern Descriptions for better prediction.

** Inputs (a batch of i.i.d. data): **

**[[$Data] [$Data] [$Data] [$Data] [$Data] [$Data] [$Data] [$Data] [$Data] [$Data]]**

** Current Pattern Descriptions: **

**You are designed to do regression, i.e., to predict the output of any given input. Both input and output are real numbers.**

** The model outputs: **

**[[$Prediction] [$Prediction] [$Prediction] [$Prediction] [$Prediction] [$Prediction] [$Prediction] [$Prediction] [$Prediction] [$Prediction]]**

** The target outputs: **

**[[$GroundTruth] [$GroundTruth] [$GroundTruth] [$GroundTruth] [$GroundTruth] [$GroundTruth] [$GroundTruth] [$GroundTruth] [$GroundTruth] [$GroundTruth]]**

**If the model is doing well, you can keep using the current descriptions. However, if the model is not performing well, please optimize the model by improving the 'New Pattern Descriptions'. The model uses the 'New Pattern Descriptions' should better predict the target outputs of the given inputs, as well as the next batch of i.i.d. input data from the same distribution. If previous 'Optimization Step' are provided, you can use the information from your last optimization step if it's helpful. NOTE: both the model and you can only operate on the numerical precision of one decimal points! Please think step by step and give your outputs strictly in the following format:**

```
Reasoning:
[be explicit and verbose, improve the Current Pattern Descriptions by yourself; please show your work; note that you don't have access to computer]
New Pattern Descriptions:
[put your new descriptions here; MUST be specific and concrete; ****MUST provide the exact value of the parameters if the descriptions potentially involve unknown or learnable parameters!!!****]
```
Please ONLY reply according to this format, don't give me any other words.

Figure 22: Prompt templates of VML for the learner and optimizer for the polynomial regression (Llama-3-70B without prior).

## K.3 Sɪɴᴜsᴏɪᴅᴀʟ Rᴇɢʀᴇssɪᴏɴ

**Text prompt template for the learner**

> You are the model. You will use the descriptions below to predict the output of the given input.
>
> ** Pattern Descriptions: **
> **You are designed to do regression, i.e., to predict the output of any given input. Both input and output are real numbers.** *It looks like the data is generated by a periodic function.*
>
> ** Input: **
> **[$Data]**
>
> Please give your output strictly in the following format:
>
> ```
> Explanations: [Your step-by-step analyses and results]
> Output:
> [Your output MUST be in REAL NUMBER ROUNDED TO TWO DECIMAL POINTS; make necessary assumptions if needed; it MUST be in the same format as the Input]
> ```
>
> Please ONLY reply according to this format, don't give me any other words.

**Text prompt template for the optimizer**

> You are the optimizer for a model, your goal is to learn the best descriptions for the model. The model used the Current Pattern Descriptions below produced the outputs of the given inputs. You are given the target outputs, please optimize the Pattern Descriptions for better prediction.
>
> ** Inputs (a batch of i.i.d. data): **
> **[[$Data] [$Data] [$Data] [$Data] [$Data] [$Data] [$Data] [$Data] [$Data] [$Data]]**
>
> ** Current Pattern Descriptions: **
> **You are designed to do regression, i.e., to predict the output of any given input. Both input and output are real numbers. It looks like the data is generated by a periodic function.**
>
> ** The model outputs: **
> **[[$Prediction] [$Prediction] [$Prediction] [$Prediction] [$Prediction] [$Prediction] [$Prediction] [$Prediction] [$Prediction] [$Prediction]]**
>
> ** The target outputs: **
> **[[$GroundTruth] [$GroundTruth] [$GroundTruth] [$GroundTruth] [$GroundTruth] [$GroundTruth] [$GroundTruth] [$GroundTruth] [$GroundTruth] [$GroundTruth]]**
>
> **If the model is doing well, you can keep using the current descriptions. However, if the model is not performing well, please optimize the model by improving the 'New Pattern Descriptions'. The model uses the 'New Pattern Descriptions' should better predict the target outputs of the given inputs, as well as the next batch of i.i.d. input data from the same distribution. If previous 'Optimization Step' are provided, you can use the information from your last optimization step if it's helpful. NOTE: both the model and you can only operate on the numerical precision of one decimal points! Please think step by step and give your outputs strictly in the following format:**
> ```
> Reasoning:
> [be explicit and verbose, improve the Current Pattern Descriptions by yourself; please show your work; note that you don't have access to computer]
> New Pattern Descriptions:
> [put your new descriptions here; MUST be specific and concrete; ****MUST provide the exact value of the parameters if the descriptions potentially involve unknown or learnable parameters!!!****]
> ```
>
> Please ONLY reply according to this format, don't give me any other words.

Figure 23: Prompt templates of VML for the learner and optimizer for the sinusoidal regression (GPT-4o with *prior*).

## K.4 Two Blobs Classification

**Text prompt template for the learner**

You are the model.

** Model Descriptions: **

**You are designed to do binary classification. The input is a point on a 2-D plane [x y]; you need to output a vector containing two probabilities such that each corresponds to how likely the data belongs to each class, i.e., [class 1 prob. class 2 prob.]. The sum of the vector MUST be 1.0.**

** Input: **

**[$Data]**

Please give your output strictly in the following format:
```
Explanations: [Your step-by-step analyses and results]
Output:
[ONLY A PURE probability vector, where each value is between 0.0 and 1.0 WITH TWO DECIMAL POINTS; make necessary assumptions if needed]
```
Please ONLY reply according to this format, don't give me any other words.

**Text prompt template for the optimizer**

You are the optimizer for a model, your goal is to learn the best descriptions for the model. The model used the Current Model Descriptions below predicted how likely the given inputs belong to a class. You are given the target values, please optimize the Model Descriptions for better prediction.

** Inputs (a batch of i.i.d. data on 2-D plane: [x y]): **

**[[$Data] [$Data] [$Data] [$Data] [$Data] [$Data] [$Data] [$Data] [$Data] [$Data]]**

** Current Model Descriptions: **

**You are designed to do binary classification. The input is a point on a 2-D plane [x y]; you need to output a vector containing two probabilities such that each corresponds to how likely the data belongs to each class, i.e., [class 1 prob. class 2 prob.]. The sum of the vector MUST be 1.0. If x > 0, output [0.0, 1.0]. If x < 0, if y < 8.5, output [0.0, 1.0], otherwise output [1.0, 0.0].**

** The model predictions ([class 1 prob. class 2 prob.]): **

**[[$Prediction] [$Prediction] [$Prediction] [$Prediction] [$Prediction] [$Prediction] [$Prediction] [$Prediction] [$Prediction] [$Prediction]]**

** The targets ([class 1 prob. class 2 prob.]): **

**[[$GroundTruth] [$GroundTruth] [$GroundTruth] [$GroundTruth] [$GroundTruth] [$GroundTruth] [$GroundTruth] [$GroundTruth] [$GroundTruth] [$GroundTruth]]**

**Please update the model by improving the 'New Model Descriptions', which should have lower classification error both on the current and the next batch of i.i.d. data. If previous 'Optimization Step' are provided, you can use the information from your last optimization step if it's helpful. Both the model and you MUST ONLY operate on the numerical precision of THREE decimal points. You are bad with numerical calculations, so be extra careful! Please think step by step and give your outputs strictly in the following format:**

```
Reasoning:
[be explicit and verbose, improve the Current Model Descriptions by yourself; please show your work; note that you don't have access to computers]
New Model Descriptions:
[put your new decision rules here; MUST be concise and concrete; ****MUST PROVIDE THE EXACT VALUE OF THE PARAMETERS if the descriptions potentially involve unknown or learnable parameters!!!****]
```

Please ONLY reply according to this format, don't give me any other words.

Figure 24: Prompt templates of VML for the learner and optimizer for the two blobs classification (Llama-3-70B without prior).

## K.5    TWO CIRCLES CLASSIFICATION

**Text prompt template for the learner**

You are the model.

\*\* Model Descriptions: \*\*

**You are designed to do binary classification. The input is a point on a 2-D plane [x y]; you need to output the class label, i.e., an integer in the set {0, 1}.** *The decision boundary is a circle.*

\*\* Input: \*\*

**[$Data]**

Please give your output strictly in the following format:

\`\`\`

Explanations: [Your step-by-step analyses and results]
Output:
[ONLY the integer class label; make necessary assumptions if needed]
\`\`\`

Please ONLY reply according to this format, don't give me any other words.

**Text prompt template for the optimizer**

You are the optimizer for a model, your goal is to learn the best descriptions for the model. The model used the Current Model Descriptions below predicted the class labels for the given inputs. You are given the target labels, please optimize the Model Descriptions for better prediction.

\*\* Inputs (a batch of i.i.d. data on 2-D plane: [x y]): \*\*

**[[$Data] [$Data] [$Data] [$Data] [$Data] [$Data] [$Data] [$Data] [$Data] [$Data]]**

\*\* Current Model Descriptions: \*\*

**You are designed to do binary classification. The input is a point on a 2-D plane [x y]; you need to output the class label, i.e., an integer in the set {0, 1}.** *The decision boundary is a circle.*

\*\* The model predictions: \*\*

**[[$Prediction] [$Prediction] [$Prediction] [$Prediction] [$Prediction] [$Prediction] [$Prediction] [$Prediction] [$Prediction] [$Prediction]]**

\*\* The targets: \*\*

**[[$GroundTruth] [$GroundTruth] [$GroundTruth] [$GroundTruth] [$GroundTruth] [$GroundTruth] [$GroundTruth] [$GroundTruth] [$GroundTruth] [$GroundTruth]]**

**If the model is doing well, you can keep using the current descriptions. However, if the model is not performing well, please update the model by improving the 'New Model Descriptions', which should have lower classification error both on the current and the next batch of i.i.d. data. If previous 'Optimization Step' are provided, you can use the information from your last optimization step if it's helpful. DON'T use symbolic representation for the model! Please think step by step and give your outputs strictly in the following format:**

\`\`\`

Reasoning:
[be explicit and verbose, improve the Current Model Descriptions by yourself; please show your work; note that you don't have access to computers]
New Model Descriptions:
[put your new decision rules here; MUST be concise and concrete; \*\*\*\*MUST PROVIDE THE EXACT VALUE OF THE PARAMETERS if the descriptions potentially involve unknown or learnable parameters!!!\*\*\*\*]
\`\`\`

Please ONLY reply according to this format, don't give me any other words.

Figure 25: Prompt templates of VML for the learner and optimizer for the two circles classification (Llama-3-70B with prior).

## K.6 TEXT CLASSIFICATION

**Text prompt template for the learner**

You are the model.

\*\* Model Descriptions: \*\*

**You are designed to do binary classification. The input is a term; you need to output the class label, i.e., an integer in the set {0, 1}.**

\*\* Input: \*\*

**[$Data]**

Please give your output strictly in the following format:

```
Explanations: [Your step-by-step analyses and results]
Output:
[ONLY the integer class label; make necessary assumptions if needed]
```

Please ONLY reply according to this format, don't give me any other words.

**Text prompt template for the optimizer**

You are the optimizer for a model, your goal is to learn the best descriptions for the model. The model used the Current Model Descriptions below predicted the class labels for the given inputs. You are given the target labels, please optimize the Model Descriptions for better prediction.

\*\* Inputs (a batch of i.i.d. text): \*\*

**[[$Data] [$Data] [$Data] [$Data] [$Data] [$Data] [$Data] [$Data] [$Data] [$Data]]**

\*\* Current Model Descriptions: \*\*

**You are designed to do binary classification. The input is a term; you need to output the class label, i.e., an integer in the set {0, 1}.**

\*\* The model predictions: \*\*

**[[$Prediction] [$Prediction] [$Prediction] [$Prediction] [$Prediction] [$Prediction] [$Prediction] [$Prediction] [$Prediction] [$Prediction]]**

\*\* The targets: \*\*

**[[$GroundTruth] [$GroundTruth] [$GroundTruth] [$GroundTruth] [$GroundTruth] [$GroundTruth] [$GroundTruth] [$GroundTruth] [$GroundTruth] [$GroundTruth]]**

**If the model is doing well, you can keep using the current descriptions. However, if the model is not performing well, please update the model by improving the 'New Model Descriptions', which should have lower classification error both on the current and the next batch of i.i.d. data. If previous 'Optimization Step' are provided, you can use the information from your last optimization step if it's helpful. Please think step by step and give your outputs strictly in the following format:**

```
Reasoning:
[be explicit and verbose, improve the Current Model Descriptions by yourself; please show your work; note that you don't have access to computers]
New Model Descriptions:
[put your new decision rules here]
```

Please ONLY reply according to this format, don't give me any other words.

Figure 26: Prompt templates of VML for the learner and optimizer for the text classification (Llama-3-70B without prior).

## L    DETAILED TRAINING HISTORY

### L.1    LINEAR REGRESSION (LLAMA-3-70B WITHOUT PRIOR)

---

**Training History: Linear Regression – Step 1**

**Epoch 0 - Step 1**
———————— **Forward Pass Prompt** ————————
You are the model. You will use the descriptions below to predict the output of the given input.
**\*\* Pattern Descriptions: \*\***
You are designed to do regression, i.e., to predict the output of any given input. Both input and output are real numbers.
**\*\* Input: \*\***
[1.26]
Please give your output strictly in the following format:
```
Explanations: [Your step-by-step analyses and results]
**Output:**
[Your output MUST be in REAL NUMBER ROUNDED TO TWO DECIMAL POINTS; make necessary assumptions if needed; it MUST be in the same format as the Input]
```
Please ONLY reply according to this format, don't give me any other words.
———————— **Forward Pass Output** ————————
Explanations: Since there is no specific pattern or function provided, I will make an assumption that the output is the same as the input.
Output: [1.89]
———————— **Forward Pass Output** ————————
Explanations: Not enough information is provided to make a meaningful prediction. Without a clear pattern or relationship, I will assume a simple linear model.
Output: [3.70]

**Many outputs for the rest data points ...**

———————— **Forward Pass Output** ————————
Explanations: Since there's no specific pattern or formula provided, I'll make an assumption based on the given input. The input [0.26] seems to be a small positive value, so I'll predict the output to be slightly larger, but still a small positive value.
Output: [0.31]
———————— **Overall Loss** ————————
36.34138
———————— **Optimization Prompt** ————————
You are the optimizer for a model, your goal is to learn the best descriptions for the model. The model used the Current Pattern Descriptions below produced the outputs of the given inputs. You are given the target outputs, please optimize the Pattern Descriptions for better prediction.
**\*\* Inputs (a batch of i.i.d. data): \*\***
[[1.26] [1.8 ] [0.99] [1.86] [0.36] [0.36] [0.97] [1.85] [0.26] [1.89]]
**\*\* Current Pattern Descriptions: \*\***
You are designed to do regression, i.e., to predict the output of any given input. Both input and output are real numbers.
**\*\* The model outputs: \*\***
[[1.27] [3.6 ] [0.98] [1.86] [0.42] [0.5 ] [0.97] [3.7 ] [0.31] [1.89]]
**\*\* The target outputs: \*\***
[[7.75] [9.43] [6.95] [9.55] [4.88] [5.23] [6.96] [9.62] [4.75] [9.62]]
If the model is doing well, you can keep using the current descriptions. However, if the model is not performing well, please optimize the model by improving the 'New Pattern Descriptions'. The model uses the 'New Pattern Descriptions' should better predict the target outputs of the given inputs, as well as the next batch of i.i.d. input data from the same distribution. If previous 'Optimization Step' are provided, you can use the information from your last optimization step if it's helpful. DON'T use symbolic representation for the model! Please think step by step and give your outputs strictly in the following format:
```
Reasoning:
[be explicit and verbose, improve the Current Pattern Descriptions by yourself; please show your work; note that you don't have access to computer]
New Pattern Descriptions:
[put your new descriptions here; MUST be specific and concrete; \*\*\*\*MUST provide the exact value of the parameters if the descriptions potentially involve unknown or learnable parameters!!!\*\*\*\*]
```
Please ONLY reply according to this format, don't give me any other words.
———————— **Optimization Output** ————————
Reasoning:
The current model outputs are not close to the target outputs, indicating that the current pattern descriptions are not effective. Upon examining the inputs and target outputs, I notice that the target outputs are generally larger than the inputs, suggesting a possible linear relationship with a positive slope. However, the current model outputs are mostly similar to the inputs, indicating that the current pattern descriptions do not capture this relationship.
To optimize the pattern descriptions, I will introduce a scaling factor to capture the linear relationship between the inputs and target outputs. I will also add a small bias term to account for any constant shift in the target outputs.
New Pattern Descriptions:
The new pattern descriptions will be based on a linear regression model with a scaling factor of 5.5 and a bias term of 2.5. The new pattern descriptions are: y = 5.5x + 2.5 where y is the predicted output and x is the input.
———————— **Updated Theta** ————————
The new pattern descriptions will be based on a linear regression model with a scaling factor of 5.5 and a bias term of 2.5. The new pattern descriptions are:
y = 5.5x + 2.5
where y is the predicted output and x is the input.

---

## Training History: Linear Regression – Step 2

**Epoch 0 - Step 2**
——————— **Forward Pass Prompt** ———————
You are the model. You will use the descriptions below to predict the output of the given input.
**\*\* Pattern Descriptions: \*\***
You are designed to do regression, i.e., to predict the output of any given input. Both input and output are real numbers. The new pattern descriptions will be based on a linear regression model with a scaling factor of 5.5 and a bias term of 2.5. The new pattern descriptions are:
y = 5.5x + 2.5
where y is the predicted output and x is the input.
**\*\* Input: \*\***
[0.86]
Please give your output strictly in the following format:
```
Explanations: [Your step-by-step analyses and results]
Output:
[Your output MUST be in REAL NUMBER ROUNDED TO TWO DECIMAL POINTS; make necessary assumptions if needed; it MUST be in the same format as the Input]
```
Please ONLY reply according to this format, don't give me any other words.
——————— **Forward Pass Output** ———————
Explanations:
Given input x = 0.28, we can plug it into the equation y = 5.5x + 2.5 to get the output y.
y = 5.5(0.28) + 2.5
y = 1.54 + 2.5
y = 4.04
Output: [4.04]

**Many outputs for the rest data points ...**

——————— **Forward Pass Output** ———————
Explanations:
The given linear regression model is y = 5.5x + 2.5. We need to find the output y for the given input x = 1.59. Plugging in the value of x into the equation, we get y = 5.5(1.59) + 2.5.
First, let's calculate 5.5(1.59) = 8.745. Now, add 2.5 to get y = 8.745 + 2.5 = 11.245. Rounded to two decimal points, the output is 11.25.
Output: [11.25]
——————— **Overall Loss** ———————
2.8677400000000004
——————— **Optimization Prompt** ———————
You are the optimizer for a model, your goal is to learn the best descriptions for the model. The model used the Current Pattern Descriptions below produced the outputs of the given inputs. You are given the target outputs, please optimize the Pattern Descriptions for better prediction.
**\*\* Inputs (a batch of i.i.d. data): \*\***
[[0.86] [0.02] [1.1 ] [1.59] [1.2 ] [0.28] [1.89] [1.29] [0.79] [0.01]]
**\*\* Current Pattern Descriptions: \*\***
You are designed to do regression, i.e., to predict the output of any given input. Both input and output are real numbers. The new pattern descriptions will be based on a linear regression model with a scaling factor of 5.5 and a bias term of 2.5. The new pattern descriptions are:
y = 5.5x + 2.5
where y is the predicted output and x is the input.
**\*\* The model outputs: \*\***
[[ 7.21] [ 2.61] [ 8.55] [11.25] [ 9.1 ] [ 4.04] [12.92] [ 9.59] [ 6.85] [ 2.56]]
**\*\* The target outputs: \*\***
[[6.56] [3.98] [7.29] [8.6 ] [7.7 ] [4.77] [9.67] [8.03] [6.43] [4.01]]
If the model is doing well, you can keep using the current descriptions. However, if the model is not performing well, please optimize the model by improving the 'New Pattern Descriptions'. The model uses the 'New Pattern Descriptions' should better predict the target outputs of the given inputs, as well as the next batch of i.i.d. input data from the same distribution. If previous 'Optimization Step' are provided, you can use the information from your last optimization step if it's helpful. DON'T use symbolic representation for the model! Please think step by step and give your outputs strictly in the following format:
```
Reasoning:
[be explicit and verbose, improve the Current Pattern Descriptions by yourself; please show your work; note that you don't have access to computer]
New Pattern Descriptions:
[put your new descriptions here; MUST be specific and concrete; \*\*\*\*MUST provide the exact value of the parameters if the descriptions potentially involve unknown or learnable parameters!!!\*\*\*\*]
```
Please ONLY reply according to this format, don't give me any other words.
——————— **Optimization Output** ———————
Reasoning:
The current model outputs are not close to the target outputs, indicating that the current pattern descriptions are not effective. Upon examining the inputs, target outputs, and model outputs, I notice that the model outputs are generally larger than the target outputs, suggesting that the scaling factor of 5.5 is too large. I will reduce the scaling factor to bring the model outputs closer to the target outputs. Additionally, I will adjust the bias term to better capture the overall shift in the target outputs. To optimize the pattern descriptions, I will calculate the average difference between the target outputs and model outputs to determine the adjustment needed for the bias term. The average difference is approximately -1.45. I will subtract this value from the current bias term of 2.5 to get a new bias term of 1.05.
New Pattern Descriptions:
The new pattern descriptions will be based on a linear regression model with a scaling factor of 4.2 and a bias term of 1.05. The new pattern descriptions are:
y = 4.2x + 1.05
where y is the predicted output and x is the input.
——————— **Updated Theta** ———————
The new pattern descriptions will be based on a linear regression model with a scaling factor of 4.2 and a bias term of 1.05. The new pattern descriptions are:
y = 4.2x + 1.05
where y is the predicted output and x is the input.

## Training History: Linear Regression – Step 3

**Epoch 0 - Step 3**
———————— **Forward Pass Prompt** ————————
You are the model. You will use the descriptions below to predict the output of the given input.
**\*\* Pattern Descriptions: \*\***
You are designed to do regression, i.e., to predict the output of any given input. Both input and output are real numbers. The new pattern descriptions will be based on a linear regression model with a scaling factor of 4.2 and a bias term of 1.05. The new pattern descriptions are:
y = 4.2x + 1.05
where y is the predicted output and x is the input.
**\*\* Input: \*\***
[0.49]
Please give your output strictly in the following format:
"'
Explanations: [Your step-by-step analyses and results]
Output: [Your output MUST be in REAL NUMBER ROUNDED TO TWO DECIMAL POINTS; make necessary assumptions if needed; it MUST be in the same format as the Input]
"'
Please ONLY reply according to this format, don't give me any other words.
———————— **Forward Pass Output** ————————
Explanations:
Using the given linear regression model, substitute x with the input 0.05.
y = 4.2x + 1.05
y = 4.2(0.05) + 1.05
y = 0.21 + 1.05
y = 1.26
Output: [1.26]

**Many outputs for the rest data points ...**

———————— **Overall Loss** ————————
4.176069999999999
———————— **Optimization Prompt** ————————
You are the optimizer for a model, your goal is to learn the best descriptions for the model. The model used the Current Pattern Descriptions below produced the outputs of the given inputs. You are given the target outputs, please optimize the Pattern Descriptions for better prediction.
**\*\* Inputs (a batch of i.i.d. data): \*\***
[[0.49] [0.74] [0.13] [0.05] [0.84] [1.21] [1.28] [1.77] [0.99] [0.77]]
**\*\* Current Pattern Descriptions: \*\***
You are designed to do regression, i.e., to predict the output of any given input. Both input and output are real numbers. The new pattern descriptions will be based on a linear regression model with a scaling factor of 4.2 and a bias term of 1.05. The new pattern descriptions are:
y = 4.2x + 1.05
where y is the predicted output and x is the input.
**\*\* The model outputs: \*\***
[[3.11] [4.16] [1.6 ] [1.26] [4.58] [6.13] [6.43] [8.52] [5.21] [4.28]]
**\*\* The target outputs: \*\***
[[5.59] [6.22] [4.31] [4.06] [6.36] [7.75] [7.87] [9.17] [6.97] [6.47]]
If the model is doing well, you can keep using the current descriptions. However, if the model is not performing well, please optimize the model by improving the 'New Pattern Descriptions'. The model uses the 'New Pattern Descriptions' should better predict the target outputs of the given inputs, as well as the next batch of i.i.d. input data from the same distribution. If previous 'Optimization Step' are provided, you can use the information from your last optimization step if it's helpful. DON'T use symbolic representation for the model! Please think step by step and give your outputs strictly in the following format:
"'
Reasoning:
[be explicit and verbose, improve the Current Pattern Descriptions by yourself; please show your work; note that you don't have access to computer]
New Pattern Descriptions:
[put your new descriptions here; MUST be specific and concrete; \*\*\*\*MUST provide the exact value of the parameters if the descriptions potentially involve unknown or learnable parameters!!!\*\*\*\*]
"'
Please ONLY reply according to this format, don't give me any other words.
———————— **Optimization Output** ————————
Reasoning:
The current model outputs are not close to the target outputs, indicating that the current pattern descriptions are not effective. Upon examining the inputs, target outputs, and model outputs, I notice that the model outputs are generally smaller than the target outputs, suggesting that the scaling factor of 4.2 is too small. I will increase the scaling factor to bring the model outputs closer to the target outputs. Additionally, I will adjust the bias term to better capture the overall shift in the target outputs. To optimize the pattern descriptions, I will calculate the average difference between the target outputs and model outputs to determine the adjustment needed for the bias term. The average difference is approximately 1.43. I will add this value to the current bias term of 1.05 to get a new bias term of 2.48. I will also analyze the relationship between the inputs and target outputs to determine the new scaling factor. Upon examining the data, I notice that the target outputs increase by approximately 5.5 units for every 1 unit increase in the input. I will use this relationship to set the new scaling factor to 5.5.
New Pattern Descriptions:
The new pattern descriptions will be based on a linear regression model with a scaling factor of 5.5 and a bias term of 2.48. The new pattern descriptions are:
y = 5.5x + 2.48
where y is the predicted output and x is the input.
———————— **Updated Theta** ————————
The new pattern descriptions will be based on a linear regression model with a scaling factor of 5.5 and a bias term of 2.48. The new pattern descriptions are:
y = 5.5x + 2.48
where y is the predicted output and x is the input.

**Training History: Linear Regression – Step 4**

**Epoch 0 - Step 4**
———————— **Forward Pass Prompt** ————————
You are the model. You will use the descriptions below to predict the output of the given input.
**Pattern Descriptions: **
You are designed to do regression, i.e., to predict the output of any given input. Both input and output are real numbers. The new pattern descriptions will be based on a linear regression model with a scaling factor of 5.5 and a bias term of 2.48. The new pattern descriptions are:
y = 5.5x + 2.48
where y is the predicted output and x is the input.
**Input: **
[0.08]
Please give your output strictly in the following format:
""
Explanations: [Your step-by-step analyses and results]
Output:
[Your output MUST be in REAL NUMBER ROUNDED TO TWO DECIMAL POINTS; make necessary assumptions if needed; it MUST be in the same format as the Input]
""
Please ONLY reply according to this format, don't give me any other words.
———————— **Forward Pass Output** ————————
Explanations:
Given the input x = 1, I will use the linear regression model y = 5.5x + 2.48 to predict the output y.
Substituting x = 1 into the equation, I get y = 5.5(1) + 2.48 = 7.98.
Output: [7.98]

**Many outputs for the rest data points ...**

———————— **Overall Loss** ————————
2.6112699999999993
———————— **Optimization Prompt** ————————
You are the optimizer for a model, your goal is to learn the best descriptions for the model. The model used the Current Pattern Descriptions below produced the outputs of the given inputs. You are given the target outputs, please optimize the Pattern Descriptions for better prediction.
**Inputs (a batch of i.i.d. data): **
[[0.08] [0.44] [0.11] [0.84] [1.48] [0.02] [1. ] [1.39] [1.7 ] [1.38]]
**Current Pattern Descriptions: **
You are designed to do regression, i.e., to predict the output of any given input. Both input and output are real numbers. The new pattern descriptions will be based on a linear regression model with a scaling factor of 5.5 and a bias term of 2.48. The new pattern descriptions are:
y = 5.5x + 2.48
where y is the predicted output and x is the input.
**The model outputs: **
[[ 2.92] [ 4.9 ] [ 3.09] [ 7.12] [10.62] [ 2.59] [ 7.98] [10.1 ] [11.93] [10.07]]
**The target outputs: **
[[4.27] [5.19] [4.24] [6.55] [8.61] [3.89] [6.96] [8.15] [9.17] [7.98]]
If the model is doing well, you can keep using the current descriptions. However, if the model is not performing well, please optimize the model by improving the 'New Pattern Descriptions'. The model uses the 'New Pattern Descriptions' should better predict the target outputs of the given inputs, as well as the next batch of i.i.d. input data from the same distribution. If previous 'Optimization Step' are provided, you can use the information from your last optimization step if it's helpful. DON'T use symbolic representation for the model! Please think step by step and give your outputs strictly in the following format:
""
Reasoning:
[be explicit and verbose, improve the Current Pattern Descriptions by yourself; please show your work; note that you don't have access to computer]
New Pattern Descriptions:
[put your new descriptions here; MUST be specific and concrete; ****MUST provide the exact value of the parameters if the descriptions potentially involve unknown or learnable parameters!!!****]
""
Please ONLY reply according to this format, don't give me any other words.
———————— **Optimization Output** ————————
Reasoning:
The current model outputs are not close to the target outputs, indicating that the current pattern descriptions are not effective. Upon examining the inputs, target outputs, and model outputs, I notice that the model outputs are generally smaller than the target outputs, suggesting that the bias term of 2.48 is too small. I will increase the bias term to bring the model outputs closer to the target outputs. Additionally, I will adjust the scaling factor to better capture the relationship between the inputs and target outputs. To optimize the pattern descriptions, I will calculate the average difference between the target outputs and model outputs to determine the adjustment needed for the bias term. The average difference is approximately 1.23. I will add this value to the current bias term of 2.48 to get a new bias term of 3.71. I will also analyze the relationship between the inputs and target outputs to determine the new scaling factor. Upon examining the data, I notice that the target outputs increase by approximately 5.8 units for every 1 unit increase in the input. However, the current scaling factor of 5.5 is close to this value, so I will only make a small adjustment to the scaling factor. I will increase the scaling factor to 5.7 to better capture the relationship between the inputs and target outputs.
New Pattern Descriptions:
The new pattern descriptions will be based on a linear regression model with a scaling factor of 5.7 and a bias term of 3.71. The new pattern descriptions are:
y = 5.7x + 3.71
where y is the predicted output and x is the input.
———————— **Updated Theta** ————————
The new pattern descriptions will be based on a linear regression model with a scaling factor of 5.7 and a bias term of 3.71. The new pattern descriptions are:
y = 5.7x + 3.71
where y is the predicted output and x is the input.

## Training History: Linear Regression – Step 5

**Epoch 0 - Step 5**

——————— **Forward Pass Prompt** ———————

You are the model. You will use the descriptions below to predict the output of the given input.

**\*\* Pattern Descriptions: \*\***

You are designed to do regression, i.e., to predict the output of any given input. Both input and output are real numbers. The new pattern descriptions will be based on a linear regression model with a scaling factor of 5.7 and a bias term of 3.71. The new pattern descriptions are:

$y = 5.7x + 3.71$

where y is the predicted output and x is the input.

**\*\* Input: \*\***

[1.59]

Please give your output strictly in the following format:

"'

Explanations: [Your step-by-step analyses and results]

Output:

[Your output MUST be in REAL NUMBER ROUNDED TO TWO DECIMAL POINTS; make necessary assumptions if needed; it MUST be in the same format as the Input]

"'

Please ONLY reply according to this format, don't give me any other words.

——————— **Forward Pass Output** ———————

Explanations:

Given the input x = 1.39, we can plug it into the equation $y = 5.7x + 3.71$ to get the output y.

$y = 5.7(1.39) + 3.71$

$y = 7.813 + 3.71$

$y = 11.523$

Output: [11.52]

**Many outputs for the rest data points ...**

——————— **Overall Loss** ———————

8.08032

——————— **Optimization Prompt** ———————

You are the optimizer for a model, your goal is to learn the best descriptions for the model. The model used the Current Pattern Descriptions below produced the outputs of the given inputs. You are given the target outputs, please optimize the Pattern Descriptions for better prediction.

**\*\* Inputs (a batch of i.i.d. data): \*\***

[[1.59] [0.23] [1.72] [0.98] [1.39] [1.7] [1.54] [0.49] [0.27] [0.93]]

**\*\* Current Pattern Descriptions: \*\***

You are designed to do regression, i.e., to predict the output of any given input. Both input and output are real numbers. The new pattern descriptions will be based on a linear regression model with a scaling factor of 5.7 and a bias term of 3.71. The new pattern descriptions are:

$y = 5.7x + 3.71$

where y is the predicted output and x is the input.

**\*\* The model outputs: \*\***

[[12.79] [ 5.02] [13.51] [ 9.3 ] [11.52] [13.4 ] [11.42] [ 6.49] [ 5.25] [ 8.99]]

**\*\* The target outputs: \*\***

[[9.02] [4.64] [9.2 ] [7.02] [8.33] [9.14] [8.55] [5.47] [4.77] [6.79]]

If the model is doing well, you can keep using the current descriptions. However, if the model is not performing well, please optimize the model by improving the 'New Pattern Descriptions'. The model uses the 'New Pattern Descriptions' should better predict the target outputs of the given inputs, as well as the next batch of i.i.d. input data from the same distribution. If previous 'Optimization Step' are provided, you can use the information from your last optimization step if it's helpful. DON'T use symbolic representation for the model! Please think step by step and give your outputs strictly in the following format:

"'

Reasoning:

[be explicit and verbose, improve the Current Pattern Descriptions by yourself; please show your work; note that you don't have access to computer]

New Pattern Descriptions:

[put your new descriptions here; MUST be specific and concrete; \*\*\*\*MUST provide the exact value of the parameters if the descriptions potentially involve unknown or learnable parameters!!!\*\*\*\*]

"'

Please ONLY reply according to this format, don't give me any other words.

——————— **Optimization Output** ———————

Reasoning:

The current model outputs are not close to the target outputs, indicating that the current pattern descriptions are not effective. Upon examining the inputs, target outputs, and model outputs, I notice that the model outputs are generally larger than the target outputs, suggesting that the scaling factor of 5.7 is too large. I will decrease the scaling factor to bring the model outputs closer to the target outputs. Additionally, I will adjust the bias term to better capture the relationship between the inputs and target outputs. To optimize the pattern descriptions, I will calculate the average difference between the target outputs and model outputs to determine the adjustment needed for the bias term. The average difference is approximately -1.44. I will subtract this value from the current bias term of 3.71 to get a new bias term of 2.27. I will also analyze the relationship between the inputs and target outputs to determine the new scaling factor. Upon examining the data, I notice that the target outputs increase by approximately 4.2 units for every 1 unit increase in the input. The current scaling factor of 5.7 is larger than this value, so I will decrease the scaling factor to 4.4 to better capture the relationship between the inputs and target outputs.

New Pattern Descriptions:

The new pattern descriptions will be based on a linear regression model with a scaling factor of 4.4 and a bias term of 2.27. The new pattern descriptions are:

$y = 4.4x + 2.27$

where y is the predicted output and x is the input.

——————— **Updated Theta** ———————

The new pattern descriptions will be based on a linear regression model with a scaling factor of 4.4 and a bias term of 2.27. The new pattern descriptions are:

$y = 4.4x + 2.27$

where y is the predicted output and x is the input.

**Training History: Linear Regression – Step 19**

**Epoch 1 - Step 9**
──────────── **Forward Pass Prompt** ────────────
You are the model. You will use the descriptions below to predict the output of the given input.
**\*\* Pattern Descriptions: \*\***
You are designed to do regression, i.e., to predict the output of any given input. Both input and output are real numbers. The new pattern descriptions will be based on a linear regression model with a scaling factor of 3.26 and a bias term of 3.45. The new pattern descriptions are:
y = 3.26x + 3.45
where y is the predicted output and x is the input.
**\*\* Input: \*\***
[1.26]
Please give your output strictly in the following format:
```
Explanations: [Your step-by-step analyses and results]
Output:
[Your output MUST be in REAL NUMBER ROUNDED TO TWO DECIMAL POINTS; make necessary assumptions if needed; it MUST be in the same format as the Input]
```
Please ONLY reply according to this format, don't give me any other words.
──────────── **Forward Pass Output** ────────────
Explanations:
Given the linear regression model y = 3.26x + 3.45, we can plug in the input x = 1.26.
y = 3.26(1.26) + 3.45 = 4.08 + 3.45 = 7.53.
Output: [7.53]

**Many outputs for the rest data points ...**

──────────── **Overall Loss** ────────────
0.11138999999999992
──────────── **Optimization Prompt** ────────────
You are the optimizer for a model, your goal is to learn the best descriptions for the model. The model used the Current Pattern Descriptions below produced the outputs of the given inputs. You are given the target outputs, please optimize the Pattern Descriptions for better prediction.
**\*\* Inputs (a batch of i.i.d. data): \*\***
[[1.26] [1.66] [0.32] [0.83] [1.3 ] [0.24] [1.87] [1.15] [1.2 ] [1.26]]
**\*\* Current Pattern Descriptions: \*\***
You are designed to do regression, i.e., to predict the output of any given input. Both input and output are real numbers. The new pattern descriptions will be based on a linear regression model with a scaling factor of 3.26 and a bias term of 3.45. The new pattern descriptions are:
y = 3.26x + 3.45
where y is the predicted output and x is the input.
**\*\* The model outputs: \*\***
[[7.53] [8.88] [4.49] [6.17] [7.71] [4.23] [9.58] [7.19] [7.36] [7.54]]
**\*\* The target outputs: \*\***
[[7.87] [8.93] [4.93] [6.36] [8.18] [4.78] [9.37] [7.39] [7.71] [7.73]]
If the model is doing well, you can keep using the current descriptions. However, if the model is not performing well, please optimize the model by improving the 'New Pattern Descriptions'. The model uses the 'New Pattern Descriptions' should better predict the target outputs of the given inputs, as well as the next batch of i.i.d. input data from the same distribution. If previous 'Optimization Step' are provided, you can use the information from your last optimization step if it's helpful. DON'T use symbolic representation for the model! Please think step by step and give your outputs strictly in the following format:
```
Reasoning:
[be explicit and verbose, improve the Current Pattern Descriptions by yourself; please show your work; note that you don't have access to computer]
New Pattern Descriptions:
[put your new descriptions here; MUST be specific and concrete; \*\*\*\*MUST provide the exact value of the parameters if the descriptions potentially involve unknown or learnable parameters!!!\*\*\*\*]
```
Please ONLY reply according to this format, don't give me any other words. \*\*\*\*\*\*\*\*\*\*\*\* Shortening the state from 5 to 3 \*\*\*\*\*\*\*\*\*\*\*\*
──────────── **Optimization Output** ────────────
Reasoning:
The current model outputs are generally close to the target outputs, but there are some deviations. Upon examining the inputs, target outputs, and model outputs, I notice that the model outputs are still underestimating the target outputs for some inputs, and overestimating for others. This suggests that the bias term of 3.45 might not be optimal. To optimize the pattern descriptions, I will calculate the average difference between the model outputs and target outputs. The average difference is approximately 0.09. I will add this value to the current bias term of 3.45 to get a new bias term of 3.54. I will also analyze the relationship between the inputs and target outputs to determine if the scaling factor of 3.26 needs to be adjusted. Upon examining the data, I notice that the target outputs are still increasing with the inputs, and the rate of increase is slightly faster than the current scaling factor of 3.26. This suggests that the scaling factor of 3.26 might be slightly too small. I will increase the scaling factor by 0.01 to get a new scaling factor of 3.27.
New Pattern Descriptions:
The new pattern descriptions will be based on a linear regression model with a scaling factor of 3.27 and a bias term of 3.54. The new pattern descriptions are:
y = 3.27x + 3.54
where y is the predicted output and x is the input.
──────────── **Updated Theta** ────────────
The new pattern descriptions will be based on a linear regression model with a scaling factor of 3.27 and a bias term of 3.54. The new pattern descriptions are:
y = 3.27x + 3.54
where y is the predicted output and x is the input.

## Training History: Linear Regression – Step 20

**Epoch 1 - Step 10**
——————— Forward Pass Prompt ———————
You are the model. You will use the descriptions below to predict the output of the given input.
** Pattern Descriptions: **
You are designed to do regression, i.e., to predict the output of any given input. Both input and output are real numbers. The new pattern descriptions will be based on a linear regression model with a scaling factor of 3.27 and a bias term of 3.54. The new pattern descriptions are:
$y = 3.27x + 3.54$
where y is the predicted output and x is the input.
** Input: **
[0.71]
Please give your output strictly in the following format:
```
Explanations: [Your step-by-step analyses and results]
Output:
[Your output MUST be in REAL NUMBER ROUNDED TO TWO DECIMAL POINTS; make necessary assumptions if needed; it MUST be in the same format as the Input]
```
Please ONLY reply according to this format, don't give me any other words.
——————— Forward Pass Output ———————
Explanations:
Using the given linear regression model, $y = 3.27x + 3.54$, and the input $x = 1$, we can calculate the output y as follows:
$y = 3.27(1) + 3.54$
$y = 3.27 + 3.54$
$y = 6.81$
Output: 6.81

**Many outputs for the rest data points ...**

——————— Overall Loss ———————
0.08741999999999994
——————— Optimization Prompt ———————
You are the optimizer for a model, your goal is to learn the best descriptions for the model. The model used the Current Pattern Descriptions below produced the outputs of the given inputs. You are given the target outputs, please optimize the Pattern Descriptions for better prediction.
** Inputs (a batch of i.i.d. data): **
[[0.71] [0.6 ] [0.51] [1.34] [0.92] [0.16] [1.12] [0.14] [1.67] [1. ]]
** Current Pattern Descriptions: **
You are designed to do regression, i.e., to predict the output of any given input. Both input and output are real numbers. The new pattern descriptions will be based on a linear regression model with a scaling factor of 3.27 and a bias term of 3.54. The new pattern descriptions are:
$y = 3.27x + 3.54$
where y is the predicted output and x is the input.
** The model outputs: **
[[5.86] [5.5 ] [5.22] [7.91] [6.57] [4.07] [7.21] [3.92] [9. ] [6.81]]
** The target outputs: **
[[6.05] [5.8 ] [5.48] [8.1 ] [6.56] [4.63] [7.37] [4.43] [8.85] [6.96]]
If the model is doing well, you can keep using the current descriptions. However, if the model is not performing well, please optimize the model by improving the 'New Pattern Descriptions'. The model uses the 'New Pattern Descriptions' should better predict the target outputs of the given inputs, as well as the next batch of i.i.d. input data from the same distribution. If previous 'Optimization Step' are provided, you can use the information from your last optimization step if it's helpful. DON'T use symbolic representation for the model! Please think step by step and give your outputs strictly in the following format:
```
Reasoning:
[be explicit and verbose, improve the Current Pattern Descriptions by yourself; please show your work; note that you don't have access to computer]
New Pattern Descriptions:
[put your new descriptions here; MUST be specific and concrete; ****MUST provide the exact value of the parameters if the descriptions potentially involve unknown or learnable parameters!!!****]
```
Please ONLY reply according to this format, don't give me any other words.
——————— Optimization Output ———————
Reasoning:
The current model outputs are generally close to the target outputs, but there are some deviations. Upon examining the inputs, target outputs, and model outputs, I notice that the model outputs are still underestimating the target outputs for some inputs, and overestimating for others. This suggests that the bias term of 3.54 might not be optimal. To optimize the pattern descriptions, I will calculate the average difference between the model outputs and target outputs. The average difference is approximately 0.15. I will add this value to the current bias term of 3.54 to get a new bias term of 3.69. I will also analyze the relationship between the inputs and target outputs to determine if the scaling factor of 3.27 needs to be adjusted. Upon examining the data, I notice that the target outputs are still increasing with the inputs, and the rate of increase is slightly faster than the current scaling factor of 3.27. This suggests that the scaling factor of 3.27 might be slightly too small. I will increase the scaling factor by 0.02 to get a new scaling factor of 3.29.
New Pattern Descriptions:
The new pattern descriptions will be based on a linear regression model with a scaling factor of 3.29 and a bias term of 3.69. The new pattern descriptions are:
$y = 3.29x + 3.69$
where y is the predicted output and x is the input.
——————— Updated Theta ———————
The new pattern descriptions will be based on a linear regression model with a scaling factor of 3.29 and a bias term of 3.69. The new pattern descriptions are:
$y = 3.29x + 3.69$
where y is the predicted output and x is the input.

## L.2  POLYNOMIAL REGRESSION (LLAMA-3-70B WITHOUT PRIOR)

---

**Training History: Polynomial Regression – Step 1**

**Epoch 0 - Step 1**

—————— **Forward Pass Prompt** ——————

You are the model. You will use the descriptions below to predict the output of the given input.

**\*\* Pattern Descriptions: \*\***

You are designed to do regression, i.e., to predict the output of any given input. Both input and output are real numbers.

**\*\* Input: \*\***

[-2.4]

Please give your output strictly in the following format:

```

Explanations: [Your step-by-step analyses and results]

Output:

[Your output MUST be in REAL NUMBER ROUNDED TO TWO DECIMAL POINTS; make necessary assumptions if needed; it MUST be in the same format as the Input]

```

Please ONLY reply according to this format, don't give me any other words.

—————— **Forward Pass Output** ——————

Explanations: Since there is no specific pattern or relationship provided, I will make an assumption that the output is a constant value. In this case, I will assume the output is 0.00.

Output: [0.00]

**Many outputs for the rest data points ...**

—————— **Forward Pass Output** ——————

Explanations:

Since there is no specific pattern or function provided, I will make an assumption that the input and output are linearly related with a constant difference of 1.2. Based on this assumption, I will add 1.2 to the input to get the output.

Output: [0.50]

—————— **Overall Loss** ——————

63.153999999999996

—————— **Optimization Prompt** ——————

You are the optimizer for a model, your goal is to learn the best descriptions for the model. The model used the Current Pattern Descriptions below produced the outputs of the given inputs. You are given the target outputs, please optimize the Pattern Descriptions for better prediction.

**\*\* Inputs (a batch of i.i.d. data): \*\***

[[-2.4] [ 0.6] [ 0.9] [-1.8] [-0.7] [-0.9] [-1.2] [-0. ] [-0.2] [ 0.3]]

**\*\* Current Pattern Descriptions: \*\***

You are designed to do regression, i.e., to predict the output of any given input. Both input and output are real numbers.

**\*\* The model outputs: \*\***

[[-2.4] [ 0. ] [ 0. ] [ 0. ] [ 0.5] [ 0. ] [-1.2] [ 0. ] [-0.2] [ 0.5]]

**\*\* The target outputs: \*\***

[[17.8] [ 3.7] [ 6.6] [10. ] [ 3.6] [ 2.5] [ 5.4] [ 1.1] [ 0.9] [ 2.6]]

If the model is doing well, you can keep using the current descriptions. However, if the model is not performing well, please optimize the model by improving the 'New Pattern Descriptions'. The model uses the 'New Pattern Descriptions' should better predict the target outputs of the given inputs, as well as the next batch of i.i.d. input data from the same distribution. If previous 'Optimization Step' are provided, you can use the information from your last optimization step if it's helpful. NOTE: both the model and you can only operate on the numerical precision of one decimal points! Please think step by step and give your outputs strictly in the following format:

```

Reasoning:

[be explicit and verbose, improve the Current Pattern Descriptions by yourself; please show your work; note that you don't have access to computer]

New Pattern Descriptions:

[put your new descriptions here; MUST be specific and concrete; \*\*\*\*MUST provide the exact value of the parameters if the descriptions potentially involve unknown or learnable parameters!!!\*\*\*\*]

```

Please ONLY reply according to this format, don't give me any other words.

—————— **Optimization Output** ——————

Reasoning:

After examining the model outputs and target outputs, it's clear that the model is not performing well. The model outputs are mostly around 0, while the target outputs have a larger range of values. This suggests that the model is not capturing the underlying pattern in the data. Upon closer inspection, I notice that the inputs and target outputs seem to have a positive correlation. This means that as the input increases, the target output also tends to increase. To improve the model, I will introduce a new pattern description that captures this positive correlation. I will use a simple linear relationship between the input and output, with a slope and intercept that can be learned from the data.

New Pattern Descriptions:

The new pattern description is: output = 2.5 \* input + 1.5.

—————— **Updated Theta** ——————

The new pattern description is: output = 2.5 \* input + 1.5.

---

## Training History: Polynomial Regression – Step 2

**Epoch 0 - Step 2**
——————— Forward Pass Prompt ———————
You are the model. You will use the descriptions below to predict the output of the given input.
** Pattern Descriptions: **
You are designed to do regression, i.e., to predict the output of any given input. Both input and output are real numbers. The new pattern description is: output = 2.5 * input + 1.5.
** Input: **
[-2.4]
Please give your output strictly in the following format:
```
Explanations: [Your step-by-step analyses and results]
Output:
[Your output MUST be in REAL NUMBER ROUNDED TO TWO DECIMAL POINTS; make necessary assumptions if needed; it MUST be in the same format as the Input]
```
Please ONLY reply according to this format, don't give me any other words.
——————— Forward Pass Output ———————
Explanations: The output is calculated by multiplying the input by 2.5 and adding 1.5. In this case, the input is 1, so the output is 2.5 * 1 + 1.5 = 4.
Output: [4.00]

**Many outputs for the rest data points ...**

——————— Forward Pass Output ———————
Explanations:
To find the output, we need to apply the given pattern description: output = 2.5 * input + 1.5.
In this case, the input is -1.2, so we plug it into the equation: output = 2.5 * (-1.2) + 1.5.
First, we multiply 2.5 and -1.2: 2.5 * (-1.2) = -3.0.
Then, we add 1.5: -3.0 + 1.5 = -1.5.
Output: [-1.50]
——————— Overall Loss ———————
257.99
——————— Optimization Prompt ———————
You are the optimizer for a model, your goal is to learn the best descriptions for the model. The model used the Current Pattern Descriptions below produced the outputs of the given inputs. You are given the target outputs, please optimize the Pattern Descriptions for better prediction.
** Inputs (a batch of i.i.d. data): **
[[-2.4] [-2.9] [-2.1] [-1.8] [-1.2] [-0.6] [-2.2] [ 1. ] [-1.9] [ 1. ]]
** Current Pattern Descriptions: **
You are designed to do regression, i.e., to predict the output of any given input. Both input and output are real numbers. The new pattern description is: output = 2.5 * input + 1.5.
** The model outputs: **
[[-4.5] [-5.8] [-3.8] [-3. ] [-1.5] [ 0. ] [-4. ] [ 4. ] [-3.2] [ 4. ]]
** The target outputs: **
[[19.5] [26.1] [12.4] [10.8] [ 4.5] [ 0.8] [13.8] [ 7.4] [ 9.7] [ 5.4]]
If the model is doing well, you can keep using the current descriptions. However, if the model is not performing well, please optimize the model by improving the 'New Pattern Descriptions'. The model uses the 'New Pattern Descriptions' should better predict the target outputs of the given inputs, as well as the next batch of i.i.d. input data from the same distribution. If previous 'Optimization Step' are provided, you can use the information from your last optimization step if it's helpful. NOTE: both the model and you can only operate on the numerical precision of one decimal points! Please think step by step and give your outputs strictly in the following format:
```
Reasoning:
[be explicit and verbose, improve the Current Pattern Descriptions by yourself; please show your work; note that you don't have access to computer]
New Pattern Descriptions:
[put your new descriptions here; MUST be specific and concrete; ****MUST provide the exact value of the parameters if the descriptions potentially involve unknown or learnable parameters!!!****]
```
Please ONLY reply according to this format, don't give me any other words.
——————— Optimization Output ———————
Reasoning:
After examining the model outputs and target outputs, it's clear that the model is not performing well. The model outputs are not close to the target outputs, and the pattern description output = 2.5 * input + 1.5 seems to be oversimplifying the relationship between the input and output. Upon closer inspection, I notice that the target outputs are generally larger than the model outputs. This suggests that the model is underestimating the target outputs. I also notice that the target outputs seem to have a non-linear relationship with the inputs, with larger inputs corresponding to larger outputs, but not in a linear way. To improve the model, I will introduce a new pattern description that captures this non-linear relationship. I will use a quadratic relationship between the input and output, with coefficients that can be learned from the data. This will allow the model to better fit the curved relationship between the inputs and target outputs.
New Pattern Descriptions:
The new pattern description is: output = 2.2 * input^2 + 1.8 * input + 0.6.
——————— Updated Theta ———————
The new pattern description is: output = 2.2 * input^2 + 1.8 * input + 0.6.

## Training History: Polynomial Regression – Step 3

**Epoch 0 - Step 3**

—————— **Forward Pass Prompt** ——————

You are the model. You will use the descriptions below to predict the output of the given input.

**\*\* Pattern Descriptions: \*\***

You are designed to do regression, i.e., to predict the output of any given input. Both input and output are real numbers. The new pattern description is: output = 2.2 * input^2 + 1.8 * input + 0.6.

**\*\* Input: \*\***

[-1.]

Please give your output strictly in the following format:

```

Explanations: [Your step-by-step analyses and results]

Output:

[Your output MUST be in REAL NUMBER ROUNDED TO TWO DECIMAL POINTS; make necessary assumptions if needed; it MUST be in the same format as the Input]

```

Please ONLY reply according to this format, don't give me any other words.

—————— **Forward Pass Output** ——————

Explanations:

Given input = -1.9, we need to find output using the formula: output = 2.2 * input^2 + 1.8 * input + 0.6.

First, calculate input^2 = (-1.9)^2 = 3.61.

Now, calculate output = 2.2 * 3.61 + 1.8 * (-1.9) + 0.6 = -2.42.

Output: [-2.42]

**Many outputs for the rest data points ...**

—————— **Forward Pass Output** ——————

Explanations:

Given the input 0.8, we need to plug it into the equation output = 2.2 * input^2 + 1.8 * input + 0.6.

First, we square the input: 0.8^2 = 0.64.

Then, we multiply 0.64 by 2.2: 2.2 * 0.64 = 1.408.

Next, we multiply 0.8 by 1.8: 1.8 * 0.8 = 1.44.

Now, we add 1.408 and 1.44: 1.408 + 1.44 = 2.848.

Finally, we add 0.6 to 2.848: 2.848 + 0.6 = 3.448.

Output: [3.45]

—————— **Overall Loss** ——————

44.34

—————— **Optimization Prompt** ——————

You are the optimizer for a model, your goal is to learn the best descriptions for the model. The model used the Current Pattern Descriptions below produced the outputs of the given inputs. You are given the target outputs, please optimize the Pattern Descriptions for better prediction.

**\*\* Inputs (a batch of i.i.d. data): \*\***

[[-1. ] [-2.7] [ 0.6] [-1.9] [-2.6] [-1.8] [-2.1] [ 0.8] [-0.8] [-2.1]]

**\*\* Current Pattern Descriptions: \*\***

You are designed to do regression, i.e., to predict the output of any given input. Both input and output are real numbers. The new pattern description is: output = 2.2 * input^2 + 1.8 * input + 0.6.

**\*\* The model outputs: \*\***

[[ 1. ] [12.8] [ 2.5] [-2.4] [10.8] [ 4.5] [ 6.5] [ 3.4] [ 0.6] [ 6.5]]

**\*\* The target outputs: \*\***

[[ 4.2] [21.2] [ 2.4] [11.7] [18.1] [ 8.3] [12.2] [ 6.8] [ 1. ] [13.7]]

If the model is doing well, you can keep using the current descriptions. However, if the model is not performing well, please optimize the model by improving the 'New Pattern Descriptions'. The model uses the 'New Pattern Descriptions' should better predict the target outputs of the given inputs, as well as the next batch of i.i.d. input data from the same distribution. If previous 'Optimization Step' are provided, you can use the information from your last optimization step if it's helpful. NOTE: both the model and you can only operate on the numerical precision of one decimal points! Please think step by step and give your outputs strictly in the following format:

```

Reasoning:

[be explicit and verbose, improve the Current Pattern Descriptions by yourself; please show your work; note that you don't have access to computer]

New Pattern Descriptions:

[put your new descriptions here; MUST be specific and concrete; \*\*\*\*MUST provide the exact value of the parameters if the descriptions potentially involve unknown or learnable parameters!!!\*\*\*\*]

```

Please ONLY reply according to this format, don't give me any other words.

—————— **Optimization Output** ——————

Reasoning:

After examining the model outputs and target outputs, it's clear that the model is not performing well. The model outputs are not close to the target outputs, and the pattern description output = 2.2 * input^2 + 1.8 * input + 0.6 seems to be undershooting the target outputs. Upon closer inspection, I notice that the target outputs are generally larger than the model outputs, especially for larger inputs. This suggests that the model is underestimating the target outputs, especially for larger inputs. I also notice that the target outputs seem to have a stronger quadratic relationship with the inputs, with larger inputs corresponding to even larger outputs. To improve the model, I will introduce a new pattern description that captures this stronger quadratic relationship. I will increase the coefficient of the quadratic term to capture the stronger curvature of the relationship. I will also adjust the linear term to better fit the data.

New Pattern Descriptions:

The new pattern description is: output = 2.8 * input^2 + 1.2 * input + 0.4.

—————— **Updated Theta** ——————

The new pattern description is: output = 2.8 * input^2 + 1.2 * input + 0.4.

## Training History: Polynomial Regression – Step 4

**Epoch 0 - Step 4**

——————— Forward Pass Prompt ———————

You are the model. You will use the descriptions below to predict the output of the given input.

**Pattern Descriptions: **

You are designed to do regression, i.e., to predict the output of any given input. Both input and output are real numbers. The new pattern description is: output = 2.8 * input^2 + 1.2 * input + 0.4.

**Input: **

[0.2]

Please give your output strictly in the following format:

```
Explanations: [Your step-by-step analyses and results]
Output:
[Your output MUST be in REAL NUMBER ROUNDED TO TWO DECIMAL POINTS; make necessary assumptions if needed; it MUST be in the same format as the Input]
```

Please ONLY reply according to this format, don't give me any other words.

——————— Forward Pass Output ———————

Explanations:

Given the pattern description: output = 2.8 * input^2 + 1.2 * input + 0.4.

Input is 0.4. So, output = 2.8 * (0.4)^2 + 1.2 * 0.4 + 0.4 = 0.448 + 0.48 + 0.4 = 1.328.

Output: [1.33]

**Many outputs for the rest data points ...**

——————— Forward Pass Output ———————

Explanations:

Given the input -1.7, we need to find the output using the formula output = 2.8 * input^2 + 1.2 * input + 0.4.

First, we calculate the square of the input: (-1.7)^2 = 2.89.

Then, we multiply it by 2.8: 2.89 * 2.8 = 8.092.

Next, we multiply the input by 1.2: -1.7 * 1.2 = -2.04.

Now, we add 8.092 and -2.04: 8.092 - 2.04 = 6.052.

Finally, we add 0.4: 6.052 + 0.4 = 6.452.

Output: [6.45]

——————— Overall Loss ———————

8.585

——————— Optimization Prompt ———————

You are the optimizer for a model, your goal is to learn the best descriptions for the model. The model used the Current Pattern Descriptions below produced the outputs of the given inputs. You are given the target outputs, please optimize the Pattern Descriptions for better prediction.

**Inputs (a batch of i.i.d. data): **

[[ 0.2] [-1.7] [-2.8] [-1.1] [ 0.4] [ 0.8] [-0.1] [-0.1] [ 0.8] [-0.7]]

**Current Pattern Descriptions: **

You are designed to do regression, i.e., to predict the output of any given input. Both input and output are real numbers. The new pattern description is: output = 2.8 * input^2 + 1.2 * input + 0.4.

**The model outputs: **

[[ 0.8] [ 6.4] [19. ] [ 2.5] [ 1.3] [ 3.2] [ 0.3] [ 0.3] [ 3.2] [-0.5]]

**The target outputs: **

[[ 3. ] [ 8.6] [25.6] [ 6.5] [ 1.2] [ 5.5] [ 0.7] [ 0.8] [ 5.1] [ 2.2]]

If the model is doing well, you can keep using the current descriptions. However, if the model is not performing well, please optimize the model by improving the 'New Pattern Descriptions'. The model uses the 'New Pattern Descriptions' should better predict the target outputs of the given inputs, as well as the next batch of i.i.d. input data from the same distribution. If previous 'Optimization Step' are provided, you can use the information from your last optimization step if it's helpful. NOTE: both the model and you can only operate on the numerical precision of one decimal points! Please think step by step and give your outputs strictly in the following format:

```
Reasoning:
[be explicit and verbose, improve the Current Pattern Descriptions by yourself; please show your work; note that you don't have access to computer]
New Pattern Descriptions:
[put your new descriptions here; MUST be specific and concrete; ****MUST provide the exact value of the parameters if the descriptions potentially involve unknown or learnable parameters!!!****]
```

Please ONLY reply according to this format, don't give me any other words.

——————— Optimization Output ———————

Reasoning:

After examining the model outputs and target outputs, it's clear that the model is not performing well. The model outputs are not close to the target outputs, and the pattern description output = 2.8 * input^2 + 1.2 * input + 0.4 seems to be undershooting the target outputs, especially for larger inputs. Upon closer inspection, I notice that the target outputs are generally larger than the model outputs, especially for larger inputs. This suggests that the model is underestimating the target outputs, especially for larger inputs. I also notice that the target outputs seem to have an even stronger quadratic relationship with the inputs, with larger inputs corresponding to even larger outputs. To improve the model, I will introduce a new pattern description that captures this even stronger quadratic relationship. I will increase the coefficient of the quadratic term to capture the stronger curvature of the relationship. I will also adjust the linear term to better fit the data.

New Pattern Descriptions:

The new pattern description is: output = 3.2 * input^2 + 0.8 * input + 0.6.

——————— Updated Theta ———————

The new pattern description is: output = 3.2 * input^2 + 0.8 * input + 0.6.

## Training History: Polynomial Regression – Step 5

**Epoch 0 - Step 5**
————————— Forward Pass Prompt —————————
You are the model. You will use the descriptions below to predict the output of the given input.
**\*\* Pattern Descriptions: \*\***
You are designed to do regression, i.e., to predict the output of any given input. Both input and output are real numbers. The new pattern description is: output = 3.2 \* input^2 + 0.8 \* input + 0.6.
**\*\* Input: \*\***
[-2.3]
Please give your output strictly in the following format:
```
Explanations: [Your step-by-step analyses and results]
Output:
[Your output MUST be in REAL NUMBER ROUNDED TO TWO DECIMAL POINTS; make necessary assumptions if needed; it MUST be in the same format as the Input]
```
Please ONLY reply according to this format, don't give me any other words.
————————— Forward Pass Output —————————
Explanations:
Given the pattern description output = 3.2 \* input^2 + 0.8 \* input + 0.6, and the input is -2.3,
first, we calculate input^2 which is (-2.3)^2 = 5.29.
Then, we plug in the values into the equation: output = 3.2 \* 5.29 + 0.8 \* -2.3 + 0.6 = 16.928 - 1.84 + 0.6 = 15.708.
Output: [15.71]

**Many outputs for the rest data points ...**

————————— Forward Pass Output —————————
Explanations:
Given the pattern description output = 3.2 \* input^2 + 0.8 \* input + 0.6, I will use this equation to find the output for the given input -2.4.
First, I will calculate input^2: (-2.4)^2 = 5.76.
Next, I will calculate 3.2 \* input^2: 3.2 \* 5.76 = 18.43.
Then, I will calculate 0.8 \* input: 0.8 \* -2.4 = -1.92.
Now, I will add the results of the previous steps: 18.43 - 1.92 + 0.6 = 17.11.
Output: [17.11]
————————— Overall Loss —————————
2.366000000000002
————————— Optimization Prompt —————————
You are the optimizer for a model, your goal is to learn the best descriptions for the model. The model used the Current Pattern Descriptions below produced the outputs of the given inputs. You are given the target outputs, please optimize the Pattern Descriptions for better prediction.
**\*\* Inputs (a batch of i.i.d. data): \*\***
[[-2.3] [ 0.3] [-2.4] [ 0.5] [-2.3] [-1.4] [-1.7] [-3. ] [-2.3] [-0.5]]
**\*\* Current Pattern Descriptions: \*\***
You are designed to do regression, i.e., to predict the output of any given input. Both input and output are real numbers. The new pattern description is: output = 3.2 \* input^2 + 0.8 \* input + 0.6.
**\*\* The model outputs: \*\***
[[15. ] [ 1.1] [17.1] [ 1.8] [15.7] [ 5.8] [ 8.5] [27. ] [15.7] [ 1. ]]
**\*\* The target outputs: \*\***
[[14.8] [ 2.4] [17.1] [ 2.9] [17.3] [ 6.3] [ 8.7] [22.9] [16.6] [ 1.5]]
If the model is doing well, you can keep using the current descriptions. However, if the model is not performing well, please optimize the model by improving the 'New Pattern Descriptions'. The model uses the 'New Pattern Descriptions' should better predict the target outputs of the given inputs, as well as the next batch of i.i.d. input data from the same distribution. If previous 'Optimization Step' are provided, you can use the information from your last optimization step if it's helpful. NOTE: both the model and you can only operate on the numerical precision of one decimal points! Please think step by step and give your outputs strictly in the following format:
```
Reasoning:
[be explicit and verbose, improve the Current Pattern Descriptions by yourself; please show your work; note that you don't have access to computer]
New Pattern Descriptions:
[put your new descriptions here; MUST be specific and concrete; \*\*\*\*MUST provide the exact value of the parameters if the descriptions potentially involve unknown or learnable parameters!!!\*\*\*\*]
```
Please ONLY reply according to this format, don't give me any other words.
————————— Optimization Output —————————
Reasoning:
After examining the model outputs and target outputs, it's clear that the model is not performing well, especially for larger inputs. The model outputs are generally higher than the target outputs, indicating that the model is overshooting the target outputs. Upon closer inspection, I notice that the target outputs seem to have a stronger quadratic relationship with the inputs, especially for larger inputs. The current pattern description output = 3.2 \* input^2 + 0.8 \* input + 0.6 seems to be too steep, resulting in overshooting. To improve the model, I will adjust the coefficient of the quadratic term to reduce the steepness of the relationship. I will also adjust the linear term to better fit the data. Additionally, I will adjust the constant term to better fit the data.
New Pattern Descriptions:
The new pattern description is: output = 2.9 \* input^2 + 0.5 \* input + 0.4.
————————— Updated Theta —————————
The new pattern description is: output = 2.9 \* input^2 + 0.5 \* input + 0.4.

## Training History: Polynomial Regression – Step 19

**Epoch 1 - Step 9**

——————— Forward Pass Prompt ———————

You are the model. You will use the descriptions below to predict the output of the given input.

**\*\* Pattern Descriptions: \*\***

You are designed to do regression, i.e., to predict the output of any given input. Both input and output are real numbers. The new pattern description is: output = 3.2 * input^2 + 1.8 * input + 1.7.

**\*\* Input: \*\***

[-1.5]

Please give your output strictly in the following format:

```

Explanations: [Your step-by-step analyses and results]

Output:

[Your output MUST be in REAL NUMBER ROUNDED TO TWO DECIMAL POINTS; make necessary assumptions if needed; it MUST be in the same format as the Input]

```

Please ONLY reply according to this format, don't give me any other words.

——————— Forward Pass Output ———————

Explanations:

The given input is 0.

Using the pattern description output = 3.2 * input^2 + 1.8 * input + 1.7,

we get output = 3.2 * 0^2 + 1.8 * 0 + 1.7 = 1.7.

Output: [1.70]

**Many outputs for the rest data points ...**

——————— Forward Pass Output ———————

Explanations:

The given pattern description is output = 3.2 * input^2 + 1.8 * input + 1.7.

We are given an input of 0.3.

First, we square the input to get 0.3^2 = 0.09.

Then, we multiply 0.09 by 3.2 to get 3.2 * 0.09 = 0.288.

Next, we multiply the input by 1.8 to get 1.8 * 0.3 = 0.54.

Finally, we add 1.7 to the sum of the previous results to get 0.288 + 0.54 + 1.7 = 2.528.

Output: [2.53]

——————— Overall Loss ———————

1.1499999999999995

——————— Optimization Prompt ———————

You are the optimizer for a model, your goal is to learn the best descriptions for the model. The model used the Current Pattern Descriptions below produced the outputs of the given inputs. You are given the target outputs, please optimize the Pattern Descriptions for better prediction.

**\*\* Inputs (a batch of i.i.d. data): \*\***

[[-1.5] [-1.8] [-0.8] [ 0. ] [ 0.3] [-2.6] [ 0.9] [-0.1] [ 0.2] [-0.5]]

**\*\* Current Pattern Descriptions: \*\***

You are designed to do regression, i.e., to predict the output of any given input. Both input and output are real numbers. The new pattern description is: output = 3.2 * input^2 + 1.8 * input + 1.7.

**\*\* The model outputs: \*\***

[[ 6.2] [ 8.8] [ 2.3] [ 1.7] [ 2.5] [18.6] [ 5.9] [ 1.6] [ 2.2] [ 1.6]]

**\*\* The target outputs: \*\***

[[ 7.8] [10.9] [ 2.6] [ 1.5] [ 3.9] [19.5] [ 4.8] [ 1.1] [ 1.8] [ 1.5]]

If the model is doing well, you can keep using the current descriptions. However, if the model is not performing well, please optimize the model by improving the 'New Pattern Descriptions'. The model uses the 'New Pattern Descriptions' should better predict the target outputs of the given inputs, as well as the next batch of i.i.d. input data from the same distribution. If previous 'Optimization Step' are provided, you can use the information from your last optimization step if it's helpful. NOTE: both the model and you can only operate on the numerical precision of one decimal points! Please think step by step and give your outputs strictly in the following format:

```

Reasoning:

[be explicit and verbose, improve the Current Pattern Descriptions by yourself; please show your work; note that you don't have access to computer]

New Pattern Descriptions:

[put your new descriptions here; MUST be specific and concrete; \*\*\*\*MUST provide the exact value of the parameters if the descriptions potentially involve unknown or learnable parameters!!!\*\*\*\*]

```

Please ONLY reply according to this format, don't give me any other words.

——————— Optimization Output ———————

Reasoning:

After examining the model outputs and target outputs, I notice that the model is still not capturing the curvature of the data well, especially for inputs with large absolute values. The current pattern description output = 3.2 * input^2 + 1.8 * input + 1.7 seems to be underestimating the outputs for inputs with large absolute values. To improve the model, I will increase the coefficient of the quadratic term to better fit the data, and adjust the linear term to better capture the trend. I will consider the previous optimization step, where I decreased the coefficient of the quadratic term from 3.4 to 3.2. This time, I will increase it to 3.3 to better capture the curvature. Upon examining the data, I notice that the model is performing poorly for inputs with large absolute values. I will increase the coefficient of the quadratic term to 3.3 to better capture the curvature of the data for these inputs. I will also adjust the linear term to 1.9 to better capture the trend. Finally, I will adjust the constant term to 1.6 to better fit the data.

New Pattern Descriptions:

The new pattern description is: output = 3.3 * input^2 + 1.9 * input + 1.6.

\*\*\*\*\*\*\*\*\*\*\*\* Updated Theta \*\*\*\*\*\*\*\*\*\*\*\*

The new pattern description is: output = 3.3 * input^2 + 1.9 * input + 1.6.

**Training History: Polynomial Regression – Step 20**

**Epoch 1 - Step 10**
——————— Forward Pass Prompt ———————
You are the model. You will use the descriptions below to predict the output of the given input.
**\*\* Pattern Descriptions: \*\***
You are designed to do regression, i.e., to predict the output of any given input. Both input and output are real numbers. The new pattern description is: output = 3.3 * input^2 + 1.9 * input + 1.6.
**\*\* Input: \*\***
[-0.1]
Please give your output strictly in the following format:
```
Explanations: [Your step-by-step analyses and results]
Output:
[Your output MUST be in REAL NUMBER ROUNDED TO TWO DECIMAL POINTS; make necessary assumptions if needed; it MUST be in the same format as the Input]
```
Please ONLY reply according to this format, don't give me any other words.
——————— Forward Pass Output ———————
Explanations:
Given the input -3, we need to calculate the output using the formula: output = 3.3 * input^2 + 1.9 * input + 1.6.
First, calculate input^2: (-3)^2 = 9.
Now, substitute the values: output = 3.3 * 9 + 1.9 * (-3) + 1.6.
output = 29.7 - 5.7 + 1.6.
output = 25.6.
Output: [25.60]

**Many outputs for the rest data points ...**

——————— Forward Pass Output ———————
Explanations:
Given the input 0.9, we need to apply the formula output = 3.3 * input^2 + 1.9 * input + 1.6.
First, we calculate the square of the input: 0.9^2 = 0.81.
Then, we multiply 0.81 by 3.3: 0.81 * 3.3 = 2.673.
Next, we multiply 0.9 by 1.9: 0.9 * 1.9 = 1.71.
Now, we add 2.673 and 1.71: 2.673 + 1.71 = 4.383.
Finally, we add 1.6 to 4.383: 4.383 + 1.6 = 5.983.
Output: [5.98]
——————— Overall Loss ———————
1.5080000000000002
——————— Optimization Prompt ———————
You are the optimizer for a model, your goal is to learn the best descriptions for the model. The model used the Current Pattern Descriptions below produced the outputs of the given inputs. You are given the target outputs, please optimize the Pattern Descriptions for better prediction.
**\*\* Inputs (a batch of i.i.d. data): \*\***
[[-0.1] [ 0.6] [ 0.7] [-3. ] [ 0.9] [-1.9] [-2.6] [ 0.8] [-0.3] [ 0.2]]
**\*\* Current Pattern Descriptions: \*\***
You are designed to do regression, i.e., to predict the output of any given input. Both input and output are real numbers. The new pattern description is: output = 3.3 * input^2 + 1.9 * input + 1.6.
**\*\* The model outputs: \*\***
[[ 1.4] [ 3.9] [ 4.6] [25.6] [ 6. ] [ 6.3] [18.9] [ 5.2] [ 1.3] [ 2.1]]
**\*\* The target outputs: \*\***
[[ 1.1] [ 4. ] [ 3.6] [26.7] [ 4.1] [ 8.6] [20.1] [ 5.7] [ 2.6] [ 2.8]]
If the model is doing well, you can keep using the current descriptions. However, if the model is not performing well, please optimize the model by improving the 'New Pattern Descriptions'. The model uses the 'New Pattern Descriptions' should better predict the target outputs of the given inputs, as well as the next batch of i.i.d. input data from the same distribution. If previous 'Optimization Step' are provided, you can use the information from your last optimization step if it's helpful. NOTE: both the model and you can only operate on the numerical precision of one decimal points! Please think step by step and give your outputs strictly in the following format:
```
Reasoning:
[be explicit and verbose, improve the Current Pattern Descriptions by yourself; please show your work; note that you don't have access to computer]
New Pattern Descriptions:
[put your new descriptions here; MUST be specific and concrete; \*\*\*\*MUST provide the exact value of the parameters if the descriptions potentially involve unknown or learnable parameters!!!\*\*\*\*]
```
Please ONLY reply according to this format, don't give me any other words.
——————— Optimization Output ———————
Reasoning:
After examining the model outputs and target outputs, I notice that the model is still not capturing the curvature of the data well, especially for inputs with large absolute values. The current pattern description output = 3.3 * input^2 + 1.9 * input + 1.6 seems to be underestimating the outputs for inputs with large absolute negative values. To improve the model, I will increase the coefficient of the quadratic term to better fit the data, and also adjust the linear term to better capture the trend. I will also adjust the constant term to better fit the data. Upon examining the data, I notice that the model is performing poorly for inputs with large absolute negative values. I will increase the coefficient of the quadratic term to 3.4 to better capture the curvature of the data for these inputs. I will also adjust the linear term to 2.1 to better capture the trend. Finally, I will adjust the constant term to 1.5 to better fit the data.
New Pattern Descriptions:
The new pattern description is: output = 3.4 * input^2 + 2.1 * input + 1.5.
\*\*\*\*\*\*\*\*\*\*\* Updated Theta \*\*\*\*\*\*\*\*\*\*\*\*
The new pattern description is: output = 3.4 * input^2 + 2.1 * input + 1.5.

## L.3 SINUSOIDAL REGRESSION (GPT-4O WITH PRIOR)

---

**Training History: Sinusoidal Regression – Step 1**

**Epoch 0 - Step 1**
————————— **Forward Pass Prompt** —————————
You are the model. You will use the descriptions below to predict the output of the given input.
You are designed to do regression, i.e., to predict the output of any given input. Both input and output are real numbers. It looks like the data is generated by a periodic function.
** Input: **
[-1.8]
Please give your output strictly in the following format:
```
Explanations: [Your step-by-step analyses and results]
Output:
[Your output MUST be in REAL NUMBER ROUNDED TO TWO DECIMAL POINTS; make necessary assumptions if needed; it MUST be in the same format as the Input]
```
Please ONLY reply according to this format, don't give me any other words.
————————— **Forward Pass Output** —————————
Explanations:
The data is generated by a periodic function. Since the function is periodic, it might resemble common periodic functions such as sine or cosine. For the input -0.8, we will assume the function could be related to the sine function due to its common usage in periodic data. The sine of -0.8 radians is approximately -0.72.
Output: [-0.72]

**Many outputs for the rest data points ...**

————————— **Forward Pass Output** —————————
Explanations:
The data seems to be generated by a periodic function. Assuming it follows a common periodic pattern such as a sine or cosine wave, I'll use the sine function for estimation. First, I normalize the input to align with the expected period of the sine function. Given the input [-2.4], I'll compute the sine value and then scale it appropriately.
Step-by-step:
1. Assume a typical sine function with amplitude and frequency adjustments: `y = A * sin(B * x + C) + D`.
2. Without knowing exact values of A, B, C, and D, assume A = 1, B = 1, C = 0, and D = 0 for simplicity: `y = sin(x)`.
3. Compute sine for normalized input: `sin(-2.4)`. Using calculator:
$\sin(-2.4) \approx -0.67546$
Result: Rounding to two decimal places: -0.68
Output: [-0.68]
————————— **Overall Loss** —————————
3.961
————————— **Optimization Prompt** —————————
You are the optimizer for a model, your goal is to learn the best descriptions for the model. The model used the Current Pattern Descriptions below produced the outputs of the given inputs. You are given the target outputs, please optimize the Pattern Descriptions for better prediction.
** Inputs (a batch of i.i.d. data): **
[[-1.8] [ 2.4] [-2.4] [ 1.9] [-1.5] [-0.7] [-0.4] [ 1.8] [-1.3] [-0.8]]
** Current Pattern Descriptions: **
You are designed to do regression, i.e., to predict the output of any given input. Both input and output are real numbers. It looks like the data is generated by a periodic function.
** The model outputs: **
[[-1. ] [ 0.7] [-0.7] [ 1. ] [-1. ] [-0.6] [-0.4] [ 1. ] [-1. ] [-0.7]]
** The target outputs: ** [[1. ] [2.7] [1.3] [2.9] [1. ] [1.4] [1.6] [3. ] [1. ] [1.3]]
If the model is doing well, you can keep using the current descriptions. However, if the model is not performing well, please optimize the model by improving the 'New Pattern Descriptions'. The model uses the 'New Pattern Descriptions' should better predict the target outputs of the given inputs, as well as the next batch of i.i.d. input data from the same distribution. If previous 'Optimization Step' are provided, you can use the information from your last optimization step if it's helpful. NOTE: both the model and you can only operate on the numerical precision of one decimal points! Please think step by step and give your outputs strictly in the following format:
```
Reasoning:
[be explicit and verbose, improve the Current Pattern Descriptions by yourself; please show your work; note that you don't have access to computer]
New Pattern Descriptions:
[put your new descriptions here; MUST be specific and concrete; ****MUST provide the exact value of the parameters if the descriptions potentially involve unknown or learnable parameters!!!****]
```
Please ONLY reply according to this format, don't give me any other words.
————————— **Optimization Output** —————————
Reasoning:
The current model appears to be using a periodic function based on the pattern in the outputs. However, the outputs are not matching the target values, which suggests that the periodic function is an incorrect assumption or is not properly tuned for this data. The target outputs are all positive and range between 1 and 3, which suggests that the function generating these outputs may have a different form or additional components compared to what the current model is using. Given that the outputs should ideally be between 1 and 3, a possible optimization could be that the model should take into account a bias and scaling factor. One way to achieve this could be through a combination of a sinusoidal function and a linear transformation to ensure values fall within the target range.
New Pattern Descriptions:
You are designed to do regression, i.e., to predict the output of any given input. Both input and output are real numbers. It looks like the data is generated by a function that could be a combination of a periodic function and a linear transformation. Specifically, the model should use the transformation:
Output = 1.5 * sin(0.8 * input) + 2.
This should ensure that the output values range from 1 to 3.
————————— **Updated Theta** —————————
You are designed to do regression, i.e., to predict the output of any given input. Both input and output are real numbers. It looks like the data is generated by a function that could be a combination of a periodic function and a linear transformation. Specifically, the model should use the transformation:
Output = 1.5 * sin(0.8 * input) + 2.
This should ensure that the output values range from 1 to 3.

---

## Training History: Sinusoidal Regression – Step 2

**Epoch 0 - Step 2**
——————— **Forward Pass Prompt** ———————
You are the model. You will use the descriptions below to predict the output of the given input.
** Pattern Descriptions: **
You are designed to do regression, i.e., to predict the output of any given input. Both input and output are real numbers. It looks like the data is generated by a periodic function. You are designed to do regression, i.e., to predict the output of any given input. Both input and output are real numbers. It looks like the data is generated by a function that could be a combination of a periodic function and a linear transformation. Specifically, the model should use the transformation:
Output = 1.5 * sin(0.8 * input) + 2.
This should ensure that the output values range from 1 to 3.
** Input: **
[-2.8]
Please give your output strictly in the following format:
```
Explanations: [Your step-by-step analyses and results]
Output:
[Your output MUST be in REAL NUMBER ROUNDED TO TWO DECIMAL POINTS; make necessary assumptions if needed; it MUST be in the same format as the Input]
```
Please ONLY reply according to this format, don't give me any other words.
——————— **Forward Pass Output** ———————
Explanations:
1. The given input is -1.4.
2. Compute the inner value of the sine function: 0.8 * -1.4 = -1.12.
3. Compute the sine of -1.12: sin(-1.12) ≈ -0.901.
4. Apply the transformation: 1.5 * -0.901 + 2 ≈ -1.352 + 2 = 0.65.
5. The output is rounded to two decimal places.
Output: [0.65]

**Many outputs for the rest data points ...**

——————— **Forward Pass Output** ———————
Explanations:
Step 1: Identify the input value which is 1.8.
Step 2: Use the transformation equation Output = 1.5 * sin(0.8 * input) + 2.
Step 3: Calculate the value inside the sine function: 0.8 * 1.8 = 1.44.
Step 4: Calculate the sine of this value: sin(1.44).
Step 5: Multiply the sine result by 1.5.
Step 6: Add 2 to the result from step 5.
Step 7: Round the final result to two decimal points. Calculation:
0.8 * 1.8 = 1.44
sin(1.44) ≈ 0.99
1.5 * 0.99 = 1.485
1.485 + 2 = 3.485
Rounded value = 3.49
Output: [3.49]
——————— **Overall Loss** ———————
0.30000000000000004
——————— **Optimization Prompt** ———————
You are the optimizer for a model, your goal is to learn the best descriptions for the model. The model used the Current Pattern Descriptions below produced the outputs of the given inputs. You are given the target outputs, please optimize the Pattern Descriptions for better prediction.
** Inputs (a batch of i.i.d. data): **
[[-2.8] [-0.3] [ 1.8] [ 0.5] [ 2.2] [ 2.5] [ 3. ] [-0.6] [ 1.2] [-1.4]]
** Current Pattern Descriptions: **
You are designed to do regression, i.e., to predict the output of any given input. Both input and output are real numbers. It looks like the data is generated by a periodic function. You are designed to do regression, i.e., to predict the output of any given input. Both input and output are real numbers. It looks like the data is generated by a function that could be a combination of a periodic function and a linear transformation. Specifically, the model should use the transformation:
Output = 1.5 * sin(0.8 * input) + 2.
This should ensure that the output values range from 1 to 3.
** The model outputs: **
[[0.8] [1.6] [3.5] [2.6] [3.5] [3.4] [3. ] [1.3] [3.2] [0.6]]
** The target outputs: **
[[1.6] [1.7] [3. ] [2.4] [2.8] [2.6] [2.2] [1.5] [2.9] [1. ]]
If the model is doing well, you can keep using the current descriptions. However, if the model is not performing well, please optimize the model by improving the 'New Pattern Descriptions'. The model uses the 'New Pattern Descriptions' should better predict the target outputs of the given inputs, as well as the next batch of i.i.d. input data from the same distribution. If previous 'Optimization Step' are provided, you can use the information from your last optimization step if it's helpful. NOTE: both the model and you can only operate on the numerical precision of one decimal points! Please think step by step and give your outputs strictly in the following format:
```
Reasoning:
[be explicit and verbose, improve the Current Pattern Descriptions by yourself; please show your work; note that you don't have access to computer]
New Pattern Descriptions:
[put your new descriptions here; MUST be specific and concrete; ****MUST provide the exact value of the parameters if the descriptions potentially involve unknown or learnable parameters!!!****]
```
Please ONLY reply according to this format, don't give me any other words.
——————— **Optimization Output** ———————
Reasoning:
Upon examining the current model outputs and their corresponding target outputs, it appears the current pattern description involving a combination of a sinusoidal function and a linear transformation is not yielding satisfactory results. The model outputs are often either too high or too low compared to the target values, indicating the need for more fine-tuning of our model. The new pattern descriptions should consider adjusting the amplitude, frequency, and bias term to better align the outputs with the targets. Observing the target outputs suggests a bias term close to 1.5 with a slightly dampened amplitude might be needed. This adjustment could get the model outputs closer to the desired range and alignment. Let's revise the current periodic function approach by adjusting the amplitude parameter from 1.5 to 1.2 and keeping the bias close to 1.5. We will also tune the frequency to 0.9 for better alignment with the target values:
New Pattern Descriptions:
You are designed to do regression, i.e., to predict the output of any given input. Both input and output are real numbers. It looks like the data is generated by a function that could be a combination of a periodic function and a linear transformation. Specifically, the model should use the transformation:
Output = 1.2 * sin(0.9 * input) + 1.5.
This adjustment aims to improve the alignment of output values with the target range (1 to 3).
——————— **Updated Theta** ———————
You are designed to do regression, i.e., to predict the output of any given input. Both input and output are real numbers. It looks like the data is generated by a function that could be a combination of a periodic function and a linear transformation. Specifically, the model should use the transformation:
Output = 1.2 * sin(0.9 * input) + 1.5.
This adjustment aims to improve the alignment of output values with the target range (1 to 3).

## Training History: Sinusoidal Regression – Step3

**Epoch 0 - Step 3**
——————— Forward Pass Prompt ———————
You are the model. You will use the descriptions below to predict the output of the given input.
** Pattern Descriptions: **
You are designed to do regression, i.e., to predict the output of any given input. Both input and output are real numbers. It looks like the data is generated by a periodic function. You are designed to do regression, i.e., to predict the output of any given input. Both input and output are real numbers. It looks like the data is generated by a function that could be a combination of a periodic function and a linear transformation. Specifically, the model should use the transformation:
Output = 1.2 * sin(0.9 * input) + 1.5.
This adjustment aims to improve the alignment of output values with the target range (1 to 3).
** Input: **
[-1.9]
Please give your output strictly in the following format:
```
Explanations: [Your step-by-step analyses and results]
Output:
[Your output MUST be in REAL NUMBER ROUNDED TO TWO DECIMAL POINTS; make necessary assumptions if needed; it MUST be in the same format as the Input]
```
Please ONLY reply according to this format, don't give me any other words.
——————— Forward Pass Output ———————
Explanations:
1. Identify the transformation function: Output = 1.2 * sin(0.9 * input) + 1.5.
2. Input value: 1.9.
3. Calculate 0.9 * 1.9 = 1.71.
4. Calculate sin(1.71) ≈ 0.99.
5. Calculate 1.2 * 0.99 ≈ 1.19.
6. Final Output: 1.19 + 1.5 = 2.69.
Output: [2.69]

**Many outputs for the rest data points ...**

——————— Forward Pass Output ———————
Explanations:
1. The given input is 1.3.
2. According to the transformation function: Output = 1.2 * sin(0.9 * input) + 1.5.
3. Calculate the intermediate value: 0.9 * 1.3 = 1.17.
4. Calculate the sine of this intermediate value: sin(1.17) ≈ 0.92 (using a calculator or sine table).
5. Multiply the sine value by 1.2: 1.2 * 0.92 = 1.104.
6. Add 1.5 to the result: 1.104 + 1.5 = 2.604.
7. Round the final output to two decimal places: 2.60.
Output: [2.60]
——————— Overall Loss ———————
0.602
——————— Optimization Prompt ———————
You are the optimizer for a model, your goal is to learn the best descriptions for the model. The model used the Current Pattern Descriptions below produced the outputs of the given inputs. You are given the target outputs, please optimize the Pattern Descriptions for better prediction.
** Inputs (a batch of i.i.d. data): **
[[-1.9] [ 1.2] [ 1.9] [ 1.2] [-1.2] [ 3. ] [-1.2] [ 1.3] [-1.7] [-2.5]]
** Current Pattern Descriptions: **
You are designed to do regression, i.e., to predict the output of any given input. Both input and output are real numbers. It looks like the data is generated by a periodic function. You are designed to do regression, i.e., to predict the output of any given input. Both input and output are real numbers. It looks like the data is generated by a function that could be a combination of a periodic function and a linear transformation. Specifically, the model should use the transformation:
Output = 1.2 * sin(0.9 * input) + 1.5.
This adjustment aims to improve the alignment of output values with the target range (1 to 3).
** The model outputs: **
[[0.3] [2.6] [1.2] [2.6] [0.4] [2. ] [0.4] [2.6] [0.3] [0.6]]
** The target outputs: **
[[1.1] [2.9] [2.9] [2.9] [1.1] [2.2] [1.1] [3. ] [1. ] [1.4]]
If the model is doing well, you can keep using the current descriptions. However, if the model is not performing well, please optimize the model by improving the 'New Pattern Descriptions'. The model uses the 'New Pattern Descriptions' should better predict the target outputs of the given inputs, as well as the next batch of i.i.d. input data from the same distribution. If previous 'Optimization Step' are provided, you can use the information from your last optimization step if it's helpful.
NOTE: both the model and you can only operate on the numerical precision of one decimal points! Please think step by step and give your outputs strictly in the following format:
```
Reasoning:
[be explicit and verbose, improve the Current Pattern Descriptions by yourself; please show your work; note that you don't have access to computer]
New Pattern Descriptions:
[put your new descriptions here; MUST be specific and concrete; ****MUST provide the exact value of the parameters if the descriptions potentially involve unknown or learnable parameters!!!****]
```
Please ONLY reply according to this format, don't give me any other words.
——————— Optimization Output ———————
Reasoning:
The current model outputs indicate that the output values are still deviating significantly from the target values. This suggests that the previous adjustments for amplitude, frequency, and bias were not optimal enough. Specifically, the model outputs are predominantly lower than the desired target values, indicating that the output needs a higher bias term, potentially better alignment in the sinusoidal component's amplitude, and possibly adapting the frequency further for an effective transformation.
Considering the previous adjustments and observing the new discrepancy, an increased bias term and a slight adjustment in the amplitude might offer closer alignment with the target distribution. Therefore, a more refined strategy would be to further increase the bias to 1.8, adjust the amplitude slightly to 1.3, and reevaluate the frequency to 1.0.
New Pattern Descriptions:
You are designed to do regression, i.e., to predict the output of any given input. Both input and output are real numbers. It looks like the data is generated by a function that could be a combination of a periodic function and a linear transformation. Specifically, the model should use the transformation:
Output = 1.3 * sin(1.0 * input) + 1.8.
This further adjustment aims to better align the output values with the target range (1 to 3) and enhance prediction accuracy.
——————— Updated Theta ———————
You are designed to do regression, i.e., to predict the output of any given input. Both input and output are real numbers. It looks like the data is generated by a function that could be a combination of a periodic function and a linear transformation. Specifically, the model should use the transformation:
Output = 1.3 * sin(1.0 * input) + 1.8.
This further adjustment aims to better align the output values with the target range (1 to 3) and enhance prediction accuracy.

## Training History: Sinusoidal Regression – Step 4

**Epoch 0 - Step 4**

——————— Forward Pass Prompt ———————

You are the model. You will use the descriptions below to predict the output of the given input.

** Pattern Descriptions: **

You are designed to do regression, i.e., to predict the output of any given input. Both input and output are real numbers. It looks like the data is generated by a periodic function. You are designed to do regression, i.e., to predict the output of any given input. Both input and output are real numbers. It looks like the data is generated by a function that could be a combination of a periodic function and a linear transformation. Specifically, the model should use the transformation:

Output = 1.3 * sin(1.0 * input) + 1.8.

This further adjustment aims to better align the output values with the target range (1 to 3) and enhance prediction accuracy.

** Input: **

[-2.2]

Please give your output strictly in the following format:

```
Explanations: [Your step-by-step analyses and results]
Output:
[Your output MUST be in REAL NUMBER ROUNDED TO TWO DECIMAL POINTS; make necessary assumptions if needed; it MUST be in the same format as the Input]
```

Please ONLY reply according to this format, don't give me any other words.

——————— Forward Pass Output ———————

Explanations:
- The given input is 0.
- Using the transformation formula: Output = 1.3 * sin(1.0 * input) + 1.8.
- Substituting the input value: Output = 1.3 * sin(1.0 * 0) + 1.8.
- Calculating the sine value: sin(0) = 0.
- Applying the transformation: Output = 1.3 * 0 + 1.8 = 1.8.
Output: [1.80]

**Many outputs for the rest data points ...**

——————— Forward Pass Output ———————

Explanations:
1. Given the input of 1.8.
2. Calculate the sine of the input: sin(1.8).
3. Multiply the sine value by 1.3: 1.3 * sin(1.8).
4. Add 1.8 to the result from step 3.
Using a calculator:
sin(1.8) ≈ 0.97385
1.3 * 0.97385 ≈ 1.266005
1.266005 + 1.8 ≈ 3.066005
Output: [3.07]

——————— Overall Loss ———————

0.05500000000000003

——————— Optimization Prompt ———————

You are the optimizer for a model, your goal is to learn the best descriptions for the model. The model used the Current Pattern Descriptions below produced the outputs of the given inputs. You are given the target outputs, please optimize the Pattern Descriptions for better prediction.

** Inputs (a batch of i.i.d. data): **

[[-2.2] [ 2. ] [ 1. ] [ 0.1] [ 1.8] [ 1.5] [ 1.8] [-1.1] [ 0. ] [ 1.5]]

** Current Pattern Descriptions: **

You are designed to do regression, i.e., to predict the output of any given input. Both input and output are real numbers. It looks like the data is generated by a periodic function. You are designed to do regression, i.e., to predict the output of any given input. Both input and output are real numbers. It looks like the data is generated by a function that could be a combination of a periodic function and a linear transformation. Specifically, the model should use the transformation:

Output = 1.3 * sin(1.0 * input) + 1.8.

This further adjustment aims to better align the output values with the target range (1 to 3) and enhance prediction accuracy.

** The model outputs: **

[[0.8] [3. ] [2.9] [1.9] [3.1] [3.1] [3.1] [0.6] [1.8] [3.1]]

** The target outputs: **

[[1.2] [2.9] [2.8] [2.1] [3. ] [3. ] [3. ] [1.1] [2. ] [3. ]]

If the model is doing well, you can keep using the current descriptions. However, if the model is not performing well, please optimize the model by improving the 'New Pattern Descriptions'. The model uses the 'New Pattern Descriptions' should better predict the target outputs of the given inputs, as well as the next batch of i.i.d. input data from the same distribution. If previous 'Optimization Step' are provided, you can use the information from your last optimization step if it's helpful. NOTE: both the model and you can only operate on the numerical precision of one decimal points! Please think step by step and give your outputs strictly in the following format:

``` Reasoning:
[be explicit and verbose, improve the Current Pattern Descriptions by yourself; please show your work; note that you don't have access to computer]
New Pattern Descriptions:
[put your new descriptions here; MUST be specific and concrete; ****MUST provide the exact value of the parameters if the descriptions potentially involve unknown or learnable parameters!!!****]
```

Please ONLY reply according to this format, don't give me any other words.

——————— Optimization Output ———————

Reasoning:

The current model is still not performing optimally, as evidenced by discrepancies between the model outputs and the target outputs. Notably, some model outputs are higher than necessary, especially for higher input values. This indicates that while the increase in bias has helped, the amplitude or the frequency might need finer adjustments. Given that previous attempts have not aligned closely enough, we might need to reduce the amplitude slightly and change the frequency for a better fit. Previous frequency 1.0 might be slightly misaligned; hence we try reducing that to 0.8 and keeping a modest amplitude to ensure variability remains within check.

New Pattern Descriptions:

You are designed to do regression, i.e., to predict the output of any given input. Both input and output are real numbers. It looks like the data is generated by a function that could be a combination of a periodic function and a linear transformation. Specifically, the model should use the transformation:

Output = 1.2 * sin(0.8 * input) + 1.8.

This adjustment aims to better align the output values with the target range (1 to 3) and enhance prediction accuracy.

——————— Updated Theta ———————

You are designed to do regression, i.e., to predict the output of any given input. Both input and output are real numbers. It looks like the data is generated by a function that could be a combination of a periodic function and a linear transformation. Specifically, the model should use the transformation:

Output = 1.2 * sin(0.8 * input) + 1.8.

This adjustment aims to better align the output values with the target range (1 to 3) and enhance prediction accuracy.

## Training History: Sinusoidal Regression – Step 5

**Epoch 0 - Step 5**
——————— Forward Pass Prompt ———————
You are the model. You will use the descriptions below to predict the output of the given input.
** Pattern Descriptions: **
You are designed to do regression, i.e., to predict the output of any given input. Both input and output are real numbers. It looks like the data is generated by a periodic function. You are designed to do regression, i.e., to predict the output of any given input. Both input and output are real numbers. It looks like the data is generated by a function that could be a combination of a periodic function and a linear transformation. Specifically, the model should use the transformation:
Output = 1.2 * sin(0.8 * input) + 1.8.
This adjustment aims to better align the output values with the target range (1 to 3) and enhance prediction accuracy.
** Input: **
[0.3]
Please give your output strictly in the following format:
```
Explanations: [Your step-by-step analyses and results]
Output:
[Your output MUST be in REAL NUMBER ROUNDED TO TWO DECIMAL POINTS; make necessary assumptions if needed; it MUST be in the same format as the Input]
```
Please ONLY reply according to this format, don't give me any other words.
——————— Forward Pass Output ———————
Explanations:
1. Take the input value: 0.0.
2. Apply the transformation: Output = 1.2 * sin(0.8 * input) + 1.8.
3. Calculate the sine function: sin(0.8 * 0.0) = sin(0) = 0.
4. Calculate the product: 1.2 * 0 = 0.
5. Add the constant: 0 + 1.8 = 1.8.
Output: [1.80]

**Many outputs for the rest data points ...**

——————— Forward Pass Output ———————
Explanations:
1. Identify the input value: 1.7.
2. Use the given transformation formula to calculate the output:
Output = 1.2 * sin(0.8 * 1.7) + 1.8.
3. Calculate the value inside the sine function: 0.8 * 1.7 = 1.36.
4. Compute the sine of 1.36: sin(1.36) ≈ 0.9776.
5. Multiply the sine value by 1.2: 1.2 * 0.9776 ≈ 1.173.
6. Add 1.8 to the result: 1.173 + 1.8 ≈ 2.973.
7. Round the result to two decimal points: 2.97.
Output: [2.97]
——————— Overall Loss ———————
0.158
——————— Optimization Prompt ———————
You are the optimizer for a model, your goal is to learn the best descriptions for the model. The model used the Current Pattern Descriptions below produced the outputs of the given inputs. You are given the target outputs, please optimize the Pattern Descriptions for better prediction.
** Inputs (a batch of i.i.d. data): **
[[ 0.3] [-2.3] [ 0.7] [-2.1] [ 0.2] [ 0. ] [-2.3] [ 1.7] [ 0.2] [-2.2]]
** Current Pattern Descriptions: **
You are designed to do regression, i.e., to predict the output of any given input. Both input and output are real numbers. It looks like the data is generated by a periodic function. You are designed to do regression, i.e., to predict the output of any given input. Both input and output are real numbers. It looks like the data is generated by a function that could be a combination of a periodic function and a linear transformation. Specifically, the model should use the transformation:
Output = 1.2 * sin(0.8 * input) + 1.8.
This adjustment aims to better align the output values with the target range (1 to 3) and enhance prediction accuracy.
** The model outputs: **
[[2.1] [0.6] [2.4] [0.6] [2. ] [1.8] [0.6] [3. ] [2. ] [0.6]]
** The target outputs: **
[[2.3] [1.2] [2.7] [1.1] [2.2] [2. ] [1.2] [3. ] [2.2] [1.2]]
If the model is doing well, you can keep using the current descriptions. However, if the model is not performing well, please optimize the model by improving the 'New Pattern Descriptions'. The model uses the 'New Pattern Descriptions' should better predict the target outputs of the given inputs, as well as the next batch of i.i.d. input data from the same distribution. If previous 'Optimization Step' are provided, you can use the information from your last optimization step if it's helpful. NOTE: both the model and you can only operate on the numerical precision of one decimal points! Please think step by step and give your outputs strictly in the following format:
```
Reasoning:
[be explicit and verbose, improve the Current Pattern Descriptions by yourself; please show your work; note that you don't have access to computer]
New Pattern Descriptions:
[put your new descriptions here; MUST be specific and concrete; ****MUST provide the exact value of the parameters if the descriptions potentially involve unknown or learnable parameters!!!****]
```
Please ONLY reply according to this format, don't give me any other words.
——————— Optimization Output ———————
Reasoning:
The model is currently underperforming, particularly noticeable with the lower-than-expected outputs. The discrepancy points towards the need to adjust the amplitude and possibly the frequency further. Additionally, the model's bias term seems to be correctly helping center the output but might still need a bit of fine-tuning to adjust for deviations. Hence, to better approximate the target outputs, we will slightly increase the frequency and adjust the amplitude. The previous frequency of 0.8 will be increased to 0.9 to see if it aligns the periodic component better. The amplitude will be increased slightly to 1.3, considering the higher-than-needed target values.
New Pattern Descriptions:
You are designed to do regression, i.e., to predict the output of any given input. Both input and output are real numbers. It looks like the data is generated by a function that could be a combination of a periodic function and a linear transformation. Specifically, the model should use the transformation:
Output = 1.3 * sin(0.9 * input) + 1.8.
This adjustment aims to better align the output values with the target range (1 to 3) and enhance prediction accuracy.
——————— Updated Theta ———————
You are designed to do regression, i.e., to predict the output of any given input. Both input and output are real numbers. It looks like the data is generated by a function that could be a combination of a periodic function and a linear transformation. Specifically, the model should use the transformation:
Output = 1.3 * sin(0.9 * input) + 1.8.
This adjustment aims to better align the output values with the target range (1 to 3) and enhance prediction accuracy.

## Training History: Sinusoidal Regression – Step 29

************ Epoch 2 - Step 9 ************
——————— Forward Pass Prompt ———————
You are the model. You will use the descriptions below to predict the output of the given input.
** Pattern Descriptions: **
You are designed to do regression, i.e., to predict the output of any given input. Both input and output are real numbers. It looks like the data is generated by a periodic function. You are designed to do regression, i.e., to predict the output of any given input. Both input and output are real numbers. It looks like the data is generated by a function that could be a combination of a periodic function and a linear transformation. Specifically, the model should use the transformation:
Output = 1.05 * sin(0.98 * input) + 2.0.
This fine-tuning is intended to address underestimations by slightly increasing the linear bias, maintaining the periodic nature for better alignment with the target outputs.
** Input: **
[2.6]
Please give your output strictly in the following format:
```
Explanations: [Your step-by-step analyses and results]
Output:
[Your output MUST be in REAL NUMBER ROUNDED TO TWO DECIMAL POINTS; make necessary assumptions if needed; it MUST be in the same format as the Input]
```
Please ONLY reply according to this format, don't give me any other words.
——————— Forward Pass Output ———————
Explanations:
1. Calculate the inner function of the sine: 0.98 * 1.4 = 1.372.
2. Find the sine of the result: sin(1.372) ≈ 0.9803.
3. Multiply by the factor: 1.05 * 0.9803 ≈ 1.029.
4. Add the linear bias: 1.029 + 2.0 = 3.029.
Output: [3.03]

**Many outputs for the rest data points ...**

——————— Overall Loss ———————
0.003000000000000001
——————— Optimization Prompt ———————
You are the optimizer for a model, your goal is to learn the best descriptions for the model. The model used the Current Pattern Descriptions below produced the outputs of the given inputs. You are given the target outputs, please optimize the Pattern Descriptions for better prediction.
** Inputs (a batch of i.i.d. data): **
[[ 2.6] [ 2.9] [ 2.9] [ 0.1] [ 1.4] [-2.5] [ 2.1] [-2.4] [-0.8] [-1.6]]
** Current Pattern Descriptions: **
You are designed to do regression, i.e., to predict the output of any given input. Both input and output are real numbers. It looks like the data is generated by a periodic function. You are designed to do regression, i.e., to predict the output of any given input. Both input and output are real numbers. It looks like the data is generated by a function that could be a combination of a periodic function and a linear transformation. Specifically, the model should use the transformation:
Output = 1.05 * sin(0.98 * input) + 2.0.
This fine-tuning is intended to address underestimations by slightly increasing the linear bias, maintaining the periodic nature for better alignment with the target outputs.
** The model outputs: **
[[2.6] [2.3] [2.3] [2.1] [3. ] [1.3] [2.9] [1.2] [1.3] [1. ]]
** The target outputs: **
[[2.6] [2.3] [2.3] [2.1] [3. ] [1.4] [2.8] [1.3] [1.3] [1. ]]
If the model is doing well, you can keep using the current descriptions. However, if the model is not performing well, please optimize the model by improving the 'New Pattern Descriptions'. The model uses the 'New Pattern Descriptions' should better predict the target outputs of the given inputs, as well as the next batch of i.i.d. input data from the same distribution. If previous 'Optimization Step' are provided, you can use the information from your last optimization step if it's helpful. NOTE: both the model and you can only operate on the numerical precision of one decimal points! Please think step by step and give your outputs strictly in the following format:
```
Reasoning:
[be explicit and verbose, improve the Current Pattern Descriptions by yourself; please show your work; note that you don't have access to computer]
New Pattern Descriptions:
[put your new descriptions here; MUST be specific and concrete; ****MUST provide the exact value of the parameters if the descriptions potentially involve unknown or learnable parameters!!!****]
```
Please ONLY reply according to this format, don't give me any other words.
——————— Optimization Output ———————
Reasoning:
Analyzing the new set of inputs, model outputs, and target outputs:
1. For input [2.6], the model output is 2.6, matching the target output exactly.
2. For inputs [2.9] and [2.9], the model outputs are both 2.3, exactly matching the target outputs.
3. For input [0.1], the model output is 2.1, matching the target output exactly.
4. For input [1.4], the model output is 3.0, matching the target output exactly.
5. For input [-2.5], the model output is 1.3, while the target output is 1.4 – slight underestimation.
6. For input [2.1], the model output is 2.9, while the target output is 2.8 – slight overestimation.
7. For input [-2.4], the model output is 1.2, while the target output is 1.3 – slight underestimation.
8. For input [-0.8], the model output is 1.3, matching the target output exactly.
9. For input [-1.6], the model output is 1.0, matching the target output exactly.
From this analysis:
- Six out of ten inputs have model outputs matching the target outputs exactly.
- The model slightly underestimates results for inputs [-2.5] and [-2.4].
- The model slightly overestimates the result for input [2.1]. Reviewing the current pattern descriptions:
- The previous adjustment to the linear bias from 1.95 to 2.0 helped mitigate underestimations by lifting the overall predictions.
- However, slight underestimations still occur for negative inputs while an overestimation appears for a positive input. To balance these observations, I propose slightly adjusting the linear bias down a tiny step to correct the overestimations and adjusting the amplitude scaling factor to fine-tune the response to negative inputs:
New Pattern Descriptions:
You are designed to do regression, i.e., to predict the output of any given input. Both input and output are real numbers. It looks like the data is generated by a function that could be a combination of a periodic function and a linear transformation. Specifically, the model should use the transformation:
Output = 1.06 * sin(0.98 * input) + 1.95.
This fine-tuning is intended to correct minor underestimations for negative inputs and slight overestimations for positive inputs, thereby maintaining overall alignment with the target outputs.
——————— Updated Theta ———————
You are designed to do regression, i.e., to predict the output of any given input. Both input and output are real numbers. It looks like the data is generated by a function that could be a combination of a periodic function and a linear transformation. Specifically, the model should use the transformation:
Output = 1.06 * sin(0.98 * input) + 1.95.
This fine-tuning is intended to correct minor underestimations for negative inputs and slight overestimations for positive inputs, thereby maintaining overall alignment with the target outputs.

## Training History: Sinusoidal Regression – Step 30

**Epoch 2 - Step 10**
——————— Forward Pass Prompt ———————
You are the model. You will use the descriptions below to predict the output of the given input.
** Pattern Descriptions: **
You are designed to do regression, i.e., to predict the output of any given input. Both input and output are real numbers. It looks like the data is generated by a periodic function. You are designed to do regression, i.e., to predict the output of any given input. Both input and output are real numbers. It looks like the data is generated by a function that could be a combination of a periodic function and a linear transformation. Specifically, the model should use the transformation:
Output = 1.06 * sin(0.98 * input) + 1.95.
This fine-tuning is intended to correct minor underestimations for negative inputs and slight overestimations for positive inputs, thereby maintaining overall alignment with the target outputs.
** Input: **
[-0.2]
Please give your output strictly in the following format:
```
Explanations: [Your step-by-step analyses and results]
Output:
[Your output MUST be in REAL NUMBER ROUNDED TO TWO DECIMAL POINTS; make necessary assumptions if needed; it MUST be in the same format as the Input]
```
Please ONLY reply according to this format, don't give me any other words.
——————— Forward Pass Output ———————
Explanations:
1. Calculate the sine of 0.98 times the input: sin(0.98 * 1) = sin(0.98) ≈ 0.830
2. Multiply the result by 1.06: 1.06 * 0.830 ≈ 0.880
3. Add 1.95 to the product: 0.880 + 1.95 ≈ 2.83
Output: [2.83]

**Many outputs for the rest data points ...**

——————— Overall Loss ———————
0.008000000000000004
——————— Optimization Prompt ———————
You are the optimizer for a model, your goal is to learn the best descriptions for the model. The model used the Current Pattern Descriptions below produced the outputs of the given inputs. You are given the target outputs, please optimize the Pattern Descriptions for better prediction.
** Inputs (a batch of i.i.d. data): **
[[-0.2] [ 1.1] [ 1.9] [ 0.7] [-2.7] [-0.9] [ 1.7] [ 2.8] [ 1. ] [-0.3]]
** Current Pattern Descriptions: **
You are designed to do regression, i.e., to predict the output of any given input. Both input and output are real numbers. It looks like the data is generated by a periodic function. You are designed to do regression, i.e., to predict the output of any given input. Both input and output are real numbers. It looks like the data is generated by a function that could be a combination of a periodic function and a linear transformation. Specifically, the model should use the transformation:
Output = 1.06 * sin(0.98 * input) + 1.95.
This fine-tuning is intended to correct minor underestimations for negative inputs and slight overestimations for positive inputs, thereby maintaining overall alignment with the target outputs.
** The model outputs: **
[[1.7] [2.9] [3. ] [2.6] [1.4] [1.1] [3. ] [2.4] [2.8] [1.6]]
** The target outputs: **
[[1.8] [2.9] [3. ] [2.7] [1.6] [1.2] [3. ] [2.4] [2.8] [1.7]]
If the model is doing well, you can keep using the current descriptions. However, if the model is not performing well, please optimize the model by improving the 'New Pattern Descriptions'. The model uses the 'New Pattern Descriptions' should better predict the target outputs of the given inputs, as well as the next batch of i.i.d. input data from the same distribution. If previous 'Optimization Step' are provided, you can use the information from your last optimization step if it's helpful. NOTE: both the model and you can only operate on the numerical precision of one decimal points! Please think step by step and give your outputs strictly in the following format:
```
Reasoning:
[be explicit and verbose, improve the Current Pattern Descriptions by yourself; please show your work; note that you don't have access to computer]
New Pattern Descriptions:
[put your new descriptions here; MUST be specific and concrete; ****MUST provide the exact value of the parameters if the descriptions potentially involve unknown or learnable parameters!!!****]
```
Please ONLY reply according to this format, don't give me any other words.
——————— Optimization Output ———————
Reasoning:
Analyzing the new set of inputs, model outputs, and target outputs:
1. For input [-0.2], the model output is 1.7, while the target output is 1.8 – slight underestimation.
2. For input [1.1], the model output is 2.9, matching the target output exactly.
3. For input [1.9], the model output is 3.0, matching the target output exactly.
4. For input [0.7], the model output is 2.6, while the target output is 2.7 – slight underestimation.
5. For input [-2.7], the model output is 1.4, while the target output is 1.6 – notable underestimation.
6. For input [-0.9], the model output is 1.1, while the target output is 1.2 – slight underestimation.
7. For input [1.7], the model output is 3.0, matching the target output exactly.
8. For input [2.8], the model output is 2.4, matching the target output exactly.
9. For input [1.0], the model output is 2.8, matching the target output exactly.
10. For input [-0.3], the model output is 1.6, while the target output is 1.7 – slight underestimation.
From this analysis:
- Five out of ten inputs have model outputs matching the target outputs exactly.
- The model slightly underestimates results for inputs [-0.2], [0.7], [-2.7], [-0.9], and [-0.3]. Reviewing the current pattern descriptions:
- The previous adjustment to the linear bias to 1.95 aimed to correct underestimations while also adjusting the amplitude scaling factor slightly improved estimations but slight underestimations still persist.
While adjusting the linear bias further down might correct some underestimations, the predominant deviation seems to involve negative inputs (less positive contributions). Also, ensuring periodic factor scaling stability:
New Pattern Descriptions:
You are designed to do regression, i.e., to predict the output of any given input. Both input and output are real numbers. It looks like the data is generated by a function that could be a combination of a periodic function and a linear transformation. Specifically, the model should use the transformation:
Output = 1.07 * sin(0.98 * input) + 1.97.
This fine-tuning is intended to correct minor underestimations for negative inputs and ensuring overall alignment with the target outputs.
——————— Updated Theta ———————
You are designed to do regression, i.e., to predict the output of any given input. Both input and output are real numbers. It looks like the data is generated by a function that could be a combination of a periodic function and a linear transformation. Specifically, the model should use the transformation:
Output = 1.07 * sin(0.98 * input) + 1.97.
This fine-tuning is intended to correct minor underestimations for negative inputs and ensuring overall alignment with the target outputs.

## L.4 Two Blobs Regression (LLaMa-3-70B without prior)

---

**Training History: Two Blobs Classification – Step 1**

**Epoch 0 - Step 1**
——————— **Forward Pass Prompt** ———————
You are the model.
**\*\* Model Descriptions: \*\***
You are designed to do binary classification. The input is a point on a 2-D plane [x y]; you need to output a vector containing two probabilities such that each corresponds to how likely the data belongs to each class, i.e., [class 1 prob. class 2 prob.]. The sum of the vector MUST be 1.0.
**\*\* Input: \*\***
[-2.178 9.99 ]
Please give your output strictly in the following format:
```

Explanations: [Your step-by-step analyses and results]
Output:
[ONLY A PURE probability vector, where each value is between 0.0 and 1.0 WITH TWO DECIMAL POINTS; make necessary assumptions if needed]
```

Please ONLY reply according to this format, don't give me any other words.
——————— **Forward Pass Output** ———————
Explanations: Since the model is not trained, I will make an assumption that the data is uniformly distributed across the 2-D plane. Based on this assumption, I will assign equal probabilities to both classes.
Output: [0.50 0.50]

**Many outputs for the rest data points ...**

——————— **Forward Pass Output** ———————
Explanations: Since the model is a binary classifier, I'll make an assumption about the decision boundary. Let's assume the decision boundary is a linear line with a slope of -1 and passes through the origin. This means that points above the line belong to class 1 and points below the line belong to class 2. Given the input [-3.616, 7.818], we can calculate the dot product of the input with the normal vector of the decision boundary (which is [1, -1]). The dot product is -3.616 + 7.818 = 4.202, which is positive, so the point is above the decision boundary. Therefore, the point is more likely to belong to class 1.
Output: [0.70, 0.30]
——————— **Training Accuracy** ———————
0.5
——————— **Overall Loss** ———————
0.4302008151499229
——————— **Optimization Prompt** ———————
You are the optimizer for a model, your goal is to learn the best descriptions for the model. The model used the Current Model Descriptions below predicted how likely the given inputs belong to a class. You are given the target values, please optimize the Model Descriptions for better prediction.
**\*\* Inputs (a batch of i.i.d. data on 2-D plane: [x y]): \*\***
[[-2.178 9.99 ] [-3.111 10.867] [-4.234 8.452] [-3.616 7.818] [ 5.155 2.487] [-2.625 8.713] [ 3.75 1.157] [-2.523 7.957] [-3.348 8.705] [ 3.569 2.456]]
**\*\* Current Model Descriptions: \*\***
You are designed to do binary classification. The input is a point on a 2-D plane [x y]; you need to output a vector containing two probabilities such that each corresponds to how likely the data belongs to each class, i.e., [class 1 prob. class 2 prob.]. The sum of the vector MUST be 1.0.
**\*\* The model predictions ([class 1 prob. class 2 prob.]): \*\***
[[0.3 0.7 ] [0.99 0.01] [0.7 0.3 ] [0.7 0.3 ] [0.85 0.15] [0.5 0.5 ] [0.2 0.8 ] [0.2 0.8 ] [0.3 0.7 ] [0.65 0.35]]
**\*\* The targets ([class 1 prob. class 2 prob.]): \*\***
[[1. 0.] [1. 0.] [1. 0.] [1. 0.] [0. 1.] [1. 0.] [1. 0.] [1. 0.] [1. 0.] [0. 1.]]
Please update the model by improving the 'New Model Descriptions', which should have lower classification error both on the current and the next batch of i.i.d. data. If previous 'Optimization Step' are provided, you can use the information from your last optimization step if it's helpful. Both the model and you MUST ONLY operate on the numerical precision of THREE decimal points. You are bad with numerical calculations, so be extra careful! Please think step by step and give your outputs strictly in the following format:
```

Reasoning:
[be explicit and verbose, improve the Current Model Descriptions by yourself; please show your work; note that you don't have access to computers]
New Model Descriptions:
[put your new decision rules here; MUST be concise and concrete; \*\*\*\*MUST PROVIDE THE EXACT VALUE OF THE PARAMETERS if the descriptions potentially involve unknown or learnable parameters!!!\*\*\*\*]
```

Please ONLY reply according to this format, don't give me any other words.
——————— **Optimization Output** ———————
Reasoning:
The goal is to update the model descriptions to improve the classification accuracy. From the current model predictions and targets, it's clear that the model is struggling to distinguish between the two classes. Upon visual inspection of the input data, it appears that the data points with x ¿ 0 tend to belong to class 2, while those with x ¡ 0 tend to belong to class 1. Let's simplify the decision boundary by introducing a threshold on the x-axis. If x ¿ 0, assign a high probability to class 2, otherwise assign a high probability to class 1.
New Model Descriptions:
If x ¿ 0, output [0.0, 1.0]. If x ¡ 0, output [1.0, 0.0].
——————— **Updated Theta** ———————
If x ¿ 0, output [0.0, 1.0]. If x ¡ 0, output [1.0, 0.0].

## Training History: Two Blobs Classification – Step 2

**Epoch 0 - Step 2**

──────── **Forward Pass Prompt** ────────

You are the model.

**\*\* Model Descriptions: \*\***

You are designed to do binary classification. The input is a point on a 2-D plane [x y]; you need to output a vector containing two probabilities such that each corresponds to how likely the data belongs to each class, i.e., [class 1 prob. class 2 prob.]. The sum of the vector MUST be 1.0. If x ¿ 0, output [0.0, 1.0]. If x ¡ 0, output [1.0, 0.0].

**\*\* Input: \*\***

[-2.252 8.94 ]

Please give your output strictly in the following format:

"'

Explanations: [Your step-by-step analyses and results]

Output:

[ONLY A PURE probability vector, where each value is between 0.0 and 1.0 WITH TWO DECIMAL POINTS; make necessary assumptions if needed]

"'

Please ONLY reply according to this format, don't give me any other words.

──────── **Forward Pass Output** ────────

Explanations: Since x ¿ 0, I will output [0.0, 1.0].

Output: [0.00, 1.00]

**Many outputs for the rest data points ...**

──────── **Forward Pass Output** ────────

Explanations:

Given the input [6.784, 2.607], since x ¿ 0, the output will be [0.0, 1.0].

Output: [0.00, 1.00]

──────── **Training Accuracy** ────────

1.0

──────── **Overall Loss** ────────

-4.9999999446126456e-09

──────── **Optimization Prompt** ────────

You are the optimizer for a model, your goal is to learn the best descriptions for the model. The model used the Current Model Descriptions below predicted how likely the given inputs belong to a class. You are given the target values, please optimize the Model Descriptions for better prediction.

**\*\* Inputs (a batch of i.i.d. data on 2-D plane: [x y]): \*\***

[[-2.252 8.94 ] [ 3.578 2.447] [-2.449 11.478] [-3.211 8.687] [ 6.784 2.607] [ 4.9 2.755] [ 4.653 3.427] [-1.594 9.343] [-2.702 9.316] [-1.044 8.789]]

**\*\* Current Model Descriptions: \*\***

You are designed to do binary classification. The input is a point on a 2-D plane [x y]; you need to output a vector containing two probabilities such that each corresponds to how likely the data belongs to each class, i.e., [class 1 prob. class 2 prob.]. The sum of the vector MUST be 1.0. If x ¿ 0, output [0.0, 1.0]. If x ¡ 0, output [1.0, 0.0].

**\*\* The model predictions ([class 1 prob. class 2 prob.]): \*\***

[[1. 0.] [0. 1.] [1. 0.] [1. 0.] [0. 1.] [0. 1.] [0. 1.] [1. 0.] [1. 0.] [1. 0.]]

**\*\* The targets ([class 1 prob. class 2 prob.]): \*\*** [[1. 0.] [0. 1.] [1. 0.] [1. 0.] [0. 1.] [0. 1.] [0. 1.] [1. 0.] [1. 0.] [1. 0.]]

Please update the model by improving the 'New Model Descriptions', which should have lower classification error both on the current and the next batch of i.i.d. data. If previous 'Optimization Step' are provided, you can use the information from your last optimization step if it's helpful. Both the model and you MUST ONLY operate on the numerical precision of THREE decimal points. You are bad with numerical calculations, so be extra careful! Please think step by step and give your outputs strictly in the following format:

"'

Reasoning:

[be explicit and verbose, improve the Current Model Descriptions by yourself; please show your work; note that you don't have access to computers]

New Model Descriptions:

[put your new decision rules here; MUST be concise and concrete; \*\*\*\*MUST PROVIDE THE EXACT VALUE OF THE PARAMETERS if the descriptions potentially involve unknown or learnable parameters!!!\*\*\*\*]

"'

Please ONLY reply according to this format, don't give me any other words.

──────── **Optimization Output** ────────

Reasoning:

The current model descriptions are already quite simple and accurate, as they correctly classify all the input data points. However, to further improve the model, let's analyze the data points more closely. We notice that the data points with y ¡ 8.5 tend to belong to class 2, while those with y ¿ 8.5 tend to belong to class 1. This observation suggests that the y-axis also contains useful information for classification. To incorporate this information, we can modify the decision rule to consider both the x and y axes. If x ¿ 0, we still assign a high probability to class 2. If x ¡ 0, we check the y value: if y ¡ 8.5, assign a high probability to class 2, otherwise assign a high probability to class 1.

New Model Descriptions:

If x ¿ 0, output [0.0, 1.0]. If x ¡ 0, if y ¡ 8.5, output [0.0, 1.0], otherwise output [1.0, 0.0].

──────── **Updated Theta** ────────

If x ¿ 0, output [0.0, 1.0]. If x ¡ 0, if y ¡ 8.5, output [0.0, 1.0], otherwise output [1.0, 0.0].

**Training History: Two Blobs Classification – Step 3**

**Epoch 0 - Step 3**
——————— **Forward Pass Prompt** ———————
You are the model.
** **Model Descriptions:** **
You are designed to do binary classification. The input is a point on a 2-D plane [x y]; you need to output a vector containing two probabilities such that each corresponds to how likely the data belongs to each class, i.e., [class 1 prob. class 2 prob.]. The sum of the vector MUST be 1.0. If x ¿ 0, output [0.0, 1.0]. If x ¡ 0, if y ¡ 8.5, output [0.0, 1.0], otherwise output [1.0, 0.0].
** **Input:** **
[-2.267 7.101]
Please give your output strictly in the following format:
"'

Explanations: [Your step-by-step analyses and results]
Output:
[ONLY A PURE probability vector, where each value is between 0.0 and 1.0 WITH TWO DECIMAL POINTS; make necessary assumptions if needed]
"'

Please ONLY reply according to this format, don't give me any other words.
——————— **Forward Pass Output** ———————
Explanations:
Since x ¿ 0, output [0.0, 1.0].
Output: [0.00, 1.00]

**Many outputs for the rest data points ...**

——————— **Forward Pass Output** ———————
Explanations:
Since x ¡ 0, we need to check the value of y. y is 7.551 which is less than 8.5. Therefore, the output should be [0.0, 1.0].
Output: [0.00, 1.00]
——————— **Training Accuracy** ———————
0.8
——————— **Overall Loss** ———————
1.8420680703952368
——————— **Optimization Prompt** ———————
You are the optimizer for a model, your goal is to learn the best descriptions for the model. The model used the Current Model Descriptions below predicted how likely the given inputs belong to a class. You are given the target values, please optimize the Model Descriptions for better prediction.
** **Inputs (a batch of i.i.d. data on 2-D plane: [x y]):** **
[[-2.267 7.101] [ 3.96 2.205] [ 5.453 0.742] [-2.901 7.551] [-3.186 9.626] [-4.428 8.988] [ 5.723 3.027] [-3.054 9.125] [ 4.563 2.314] [-3.837 9.211]]
** **Current Model Descriptions:** **
You are designed to do binary classification. The input is a point on a 2-D plane [x y]; you need to output a vector containing two probabilities such that each corresponds to how likely the data belongs to each class, i.e., [class 1 prob. class 2 prob.]. The sum of the vector MUST be 1.0. If x ¿ 0, output [0.0, 1.0]. If x ¡ 0, if y ¡ 8.5, output [0.0, 1.0], otherwise output [1.0, 0.0].
** **The model predictions ([class 1 prob. class 2 prob.]):** **
[[0. 1.] [0. 1.] [0. 1.] [0. 1.] [1. 0.] [1. 0.] [0. 1.] [1. 0.] [0. 1.] [1. 0.]]
** **The targets ([class 1 prob. class 2 prob.]):** **
[[1. 0.] [0. 1.] [0. 1.] [1. 0.] [1. 0.] [1. 0.] [0. 1.] [1. 0.] [0. 1.] [1. 0.]]
Please update the model by improving the 'New Model Descriptions', which should have lower classification error both on the current and the next batch of i.i.d. data. If previous 'Optimization Step' are provided, you can use the information from your last optimization step if it's helpful. Both the model and you MUST ONLY operate on the numerical precision of THREE decimal points. You are bad with numerical calculations, so be extra careful! Please think step by step and give your outputs strictly in the following format:
"'

Reasoning:
[be explicit and verbose, improve the Current Model Descriptions by yourself; please show your work; note that you don't have access to computers]
New Model Descriptions:
[put your new decision rules here; MUST be concise and concrete; ****MUST PROVIDE THE EXACT VALUE OF THE PARAMETERS if the descriptions potentially involve unknown or learnable parameters!!!****]
"'

Please ONLY reply according to this format, don't give me any other words.
——————— **Optimization Output** ———————
Reasoning:
The current model descriptions are still quite simple, but they make some mistakes in the current batch of data. Let's analyze the data points more closely. We notice that the data points with x ¿ 0 and y ¡ 2.5 tend to belong to class 2, while those with x ¡ 0 and y ¿ 9 tend to belong to class 1. This observation suggests that the y-axis contains more useful information for classification, especially when x ¡ 0. To incorporate this information, we can modify the decision rule to consider both the x and y axes more carefully. If x ¿ 0, we check the y value: if y ¡ 2.5, assign a high probability to class 2, otherwise assign a high probability to class 1. If x ¡ 0, we also check the y value: if y ¿ 9, assign a high probability to class 1, otherwise assign a high probability to class 2.
New Model Descriptions:
If x ¿ 0, if y ¡ 2.5, output [0.0, 1.0], otherwise output [1.0, 0.0]. If x ¡ 0, if y ¿ 9.0, output [1.0, 0.0], otherwise output [0.0, 1.0].
——————— **Updated Theta** ———————
If x ¿ 0, if y ¡ 2.5, output [0.0, 1.0], otherwise output [1.0, 0.0]. If x ¡ 0, if y ¿ 9.0, output [1.0, 0.0], otherwise output [0.0, 1.0].

## Training History: Two Blobs Classification – Step 4

**Epoch 0 - Step 4**
———————— **Forward Pass Prompt** ————————
You are the model.
** **Model Descriptions: ** **
You are designed to do binary classification. The input is a point on a 2-D plane [x y]; you need to output a vector containing two probabilities such that each corresponds to how likely the data belongs to each class, i.e., [class 1 prob. class 2 prob.]. The sum of the vector MUST be 1.0. If x ¿ 0, if y ¡ 2.5, output [0.0, 1.0], otherwise output [1.0, 0.0]. If x ¡ 0, if y ¿ 9.0, output [1.0, 0.0], otherwise output [0.0, 1.0].
** **Input: ** **
[-3.988 8.294]
Please give your output strictly in the following format:
‘‘‘
Explanations: [Your step-by-step analyses and results]
Output:
[ONLY A PURE probability vector, where each value is between 0.0 and 1.0 WITH TWO DECIMAL POINTS; make necessary assumptions if needed]
’’’
Please ONLY reply according to this format, don't give me any other words.
———————— **Forward Pass Output** ————————
Explanations:
Since x ¿ 0 and y ¡ 2.5, according to the rules, the output should be [0.0, 1.0].
Output: [0.00, 1.00]

**Many outputs for the rest data points ...**

———————— **Forward Pass Output** ————————
Explanations:
Since x = 5.113 ¿ 0, we check the value of y.
y = 1.9 ¡ 2.5, so we output [0.0, 1.0].
Output: [0.00, 1.00]
———————— **Training Accuracy** ————————
0.7
———————— **Overall Loss** ————————
2.763102108092855
———————— **Optimization Prompt** ————————
You are the optimizer for a model, your goal is to learn the best descriptions for the model. The model used the Current Model Descriptions below predicted how likely the given inputs belong to a class. You are given the target values, please optimize the Model Descriptions for better prediction.
** **Inputs (a batch of i.i.d. data on 2-D plane: [x y]): ** **
[[-3.988 8.294] [-5.129 9.836] [ 3.403 0.653] [-3.522 9.329] [ 3.665 2.76 ] [-2.417 7.027] [ 5.113 1.9 ] [-3.318 8.513] [-3.925 8.594] [-3.417 7.602]]
** **Current Model Descriptions: ** **
You are designed to do binary classification. The input is a point on a 2-D plane [x y]; you need to output a vector containing two probabilities such that each corresponds to how likely the data belongs to each class, i.e., [class 1 prob. class 2 prob.]. The sum of the vector MUST be 1.0. If x ¿ 0, if y ¡ 2.5, output [0.0, 1.0], otherwise output [1.0, 0.0]. If x ¡ 0, if y ¿ 9.0, output [1.0, 0.0], otherwise output [0.0, 1.0].
** **The model predictions ([class 1 prob. class 2 prob.]): ** **
[[1. 0.] [1. 0.] [0. 1.] [1. 0.] [0. 1.] [1. 0.] [0. 1.] [0. 1.] [0. 1.] [0. 1.]]
** **The targets ([class 1 prob. class 2 prob.]): ** **
[[1. 0.] [1. 0.] [0. 1.] [1. 0.] [0. 1.] [1. 0.] [0. 1.] [1. 0.] [1. 0.] [1. 0.]]
Please update the model by improving the 'New Model Descriptions', which should have lower classification error both on the current and the next batch of i.i.d. data. If previous 'Optimization Step' are provided, you can use the information from your last optimization step if it's helpful. Both the model and you MUST ONLY operate on the numerical precision of THREE decimal points. You are bad with numerical calculations, so be extra careful! Please think step by step and give your outputs strictly in the following format:
‘‘‘
Reasoning:
[be explicit and verbose, improve the Current Model Descriptions by yourself; please show your work; note that you don't have access to computers]
New Model Descriptions:
[put your new decision rules here; MUST be concise and concrete; ****MUST PROVIDE THE EXACT VALUE OF THE PARAMETERS if the descriptions potentially involve unknown or learnable parameters!!!****]
’’’
Please ONLY reply according to this format, don't give me any other words.
———————— **Optimization Output** ————————
Reasoning:
Analyzing the current batch of data, we notice that the current model descriptions are mostly correct, but there are some mistakes. For example, the point [-3.318 8.513] is misclassified, and it should belong to class 1. This suggests that the threshold for y-axis in the x ¡ 0 region should be lowered. Additionally, we observe that the points with x ¿ 0 and y ¿ 2.5 tend to belong to class 1, and the points with x ¿ 0 and y ¡ 2.5 tend to belong to class 2. This suggests that the threshold for y-axis in the x ¿ 0 region should be raised.
New Model Descriptions:
If x ¿ 0, if y ¡ 3.0, output [0.0, 1.0], otherwise output [1.0, 0.0]. If x ¡ 0, if y ¿ 8.0, output [1.0, 0.0], otherwise output [0.0, 1.0].
———————— **Updated Theta** ————————
If x ¿ 0, if y ¡ 3.0, output [0.0, 1.0], otherwise output [1.0, 0.0]. If x ¡ 0, if y ¿ 8.0, output [1.0, 0.0], otherwise output [0.0, 1.0].

---

**Training History: Two Blobs Classification – Step 5**

**Epoch 0 - Step 5**
—————— **Forward Pass Prompt** ——————
You are the model.
** Model Descriptions: **
You are designed to do binary classification. The input is a point on a 2-D plane [x y]; you need to output a vector containing two probabilities such that each corresponds to how likely the data belongs to each class, i.e., [class 1 prob. class 2 prob.]. The sum of the vector MUST be 1.0. If x ¿ 0, if y ¡ 3.0, output [0.0, 1.0], otherwise output [1.0, 0.0]. If x ¡ 0, if y ¿ 8.0, output [1.0, 0.0], otherwise output [0.0, 1.0].
** Input: **
[4.698 0.83 ]
Please give your output strictly in the following format:
```
Explanations: [Your step-by-step analyses and results]
Output:
[ONLY A PURE probability vector, where each value is between 0.0 and 1.0 WITH TWO DECIMAL POINTS; make necessary assumptions if needed]
```
Please ONLY reply according to this format, don't give me any other words.
—————— **Forward Pass Output** ——————
Explanations:
x ¡ 0, y ¿ 8.0, so output [1.0, 0.0].
Output: [1.00, 0.00]

**Many outputs for the rest data points ...**

—————— **Forward Pass Output** ——————
Explanations:
Since x ¿ 0 (4.698 ¿ 0), we check if y ¡ 3.0 (0.83 ¡ 3.0). This is true, so the output is [0.0, 1.0].
Output: [0.00, 1.00]
—————— **Training Accuracy** ——————
0.8
—————— **Overall Loss** ——————
1.8420680703952368
—————— **Optimization Prompt** ——————
You are the optimizer for a model, your goal is to learn the best descriptions for the model. The model used the Current Model Descriptions below predicted how likely the given inputs belong to a class. You are given the target values, please optimize the Model Descriptions for better prediction.
** Inputs (a batch of i.i.d. data on 2-D plane: [x y]): **
[[ 4.698 0.83 ] [-2.3 7.055] [ 5.462 3.87 ] [ 5.594 2.625] [ 5.162 2.27 ] [-1.697 10.371] [ 4.193 2.83 ] [-3.11 8.723] [ 3.262 1.035] [-2.412 9.983]]
** Current Model Descriptions: **
You are designed to do binary classification. The input is a point on a 2-D plane [x y]; you need to output a vector containing two probabilities such that each corresponds to how likely the data belongs to each class, i.e., [class 1 prob. class 2 prob.]. The sum of the vector MUST be 1.0. If x ¿ 0, if y ¡ 3.0, output [0.0, 1.0], otherwise output [1.0, 0.0]. If x ¡ 0, if y ¿ 8.0, output [1.0, 0.0], otherwise output [0.0, 1.0].
** The model predictions ([class 1 prob. class 2 prob.]): **
[[0. 1.] [0. 1.] [1. 0.] [0. 1.] [0. 1.] [1. 0.] [0. 1.] [1. 0.] [0. 1.] [1. 0.]]
** The targets ([class 1 prob. class 2 prob.]): **
[[0. 1.] [1. 0.] [0. 1.] [0. 1.] [0. 1.] [1. 0.] [0. 1.] [1. 0.] [0. 1.] [1. 0.]]
Please update the model by improving the 'New Model Descriptions', which should have lower classification error both on the current and the next batch of i.i.d. data. If previous 'Optimization Step' are provided, you can use the information from your last optimization step if it's helpful. Both the model and you MUST ONLY operate on the numerical precision of THREE decimal points. You are bad with numerical calculations, so be extra careful! Please think step by step and give your outputs strictly in the following format:
```
Reasoning:
[be explicit and verbose, improve the Current Model Descriptions by yourself; please show your work; note that you don't have access to computers]
New Model Descriptions:
[put your new decision rules here; MUST be concise and concrete; ****MUST PROVIDE THE EXACT VALUE OF THE PARAMETERS if the descriptions potentially involve unknown or learnable parameters!!!****]
```
Please ONLY reply according to this format, don't give me any other words.
—————— **Optimization Output** ——————
Reasoning:
Analyzing the current batch of data, we notice that the current model descriptions are mostly correct, but there are some mistakes. For example, the point [-2.3 7.055] is misclassified, and it should belong to class 1. This suggests that the threshold for y-axis in the x ¡ 0 region should be lowered. Additionally, we observe that the points with x ¿ 0 and y ¿ 2.5 tend to belong to class 1, and the points with x ¿ 0 and y ¡ 2.5 tend to belong to class 2. This suggests that the threshold for y-axis in the x ¿ 0 region is correct.
New Model Descriptions:
If x ¿ 0, if y ¡ 3.0, output [0.0, 1.0], otherwise output [1.0, 0.0]. If x ¡ 0, if y ¿ 7.5, output [1.0, 0.0], otherwise output [0.0, 1.0].
—————— **Updated Theta** ——————
If x ¿ 0, if y ¡ 3.0, output [0.0, 1.0], otherwise output [1.0, 0.0]. If x ¡ 0, if y ¿ 7.5, output [1.0, 0.0], otherwise output [0.0, 1.0].

## Training History: Two Blobs Classification – Step 81

**Epoch 8 - Step 1**

─────────── Forward Pass Prompt ───────────

You are the model.

**\*\* Model Descriptions: \*\***

You are designed to do binary classification. The input is a point on a 2-D plane [x y]; you need to output a vector containing two probabilities such that each corresponds to how likely the data belongs to each class, i.e., [class 1 prob. class 2 prob.]. The sum of the vector MUST be 1.0. If x ¿ 0, if y ¡ 1.7, output [0.0, 1.0], otherwise if y ¡ 2.0 and x ¿ 3.0, output [0.0, 1.0], otherwise if y ¡ 2.8 and x ¿ 4.0, output [0.0, 1.0], otherwise if y ¡ 2.5 and x ¿ 3.5, output [0.0, 1.0], otherwise if y ¡ 3.5 and x ¿ 4.5, output [0.0, 1.0], otherwise if y ¡ 3.5 and x ¿ 5.0, output [0.0, 1.0], otherwise if y ¡ 3.0 and x ¿ 3.5, output [0.0, 1.0], otherwise if y ¡ 4.3 and x ¡= 4.0, output [1.0, 0.0], otherwise if y ¡ 4.3 and x ¿ 4.0, output [0.0, 1.0], otherwise if y ¡ 9.5, output [1.0, 0.0], otherwise if x ¿ 4.5 and y ¡ 3.0, output [0.0, 1.0], otherwise if x ¿ 3.0 and y ¡ 2.2, output [0.0, 1.0], otherwise if x ¿ 4.0 and y ¡ 1.5, output [0.0, 1.0], otherwise if x ¿ 5.0 and y ¡ 2.5, output [0.0, 1.0], otherwise if x ¿ 3.0 and y ¡ 2.0, output [0.0, 1.0], otherwise if x ¿ 4.0 and y ¡ 2.5, output [0.0, 1.0], otherwise if x ¿ 4.5 and y ¡ 2.8, output [0.0, 1.0], otherwise if x ¿ 5.0 and y ¡ 3.5, output [0.0, 1.0], otherwise if x ¿ 3.5 and y ¡ 2.2, output [0.0, 1.0], otherwise if x ¿ 4.0 and y ¡ 1.5, output [0.0, 1.0], otherwise if x ¿ 4.0 and y ¡ 1.2, output [0.0, 1.0], otherwise if x ¿ 5.0 and y ¡ 2.3, output [0.0, 1.0], otherwise if x ¿ 4.0 and y ¡ 2.0, output [0.0, 1.0], otherwise if x ¿ 4.0 and y ¡ 2.5, output [0.0, 1.0], otherwise output [1.0, 0.0]. If x ¡ 0, if x ¡ -3.5 and y ¿ 8.5, output [1.0, 0.0], otherwise if x ¡ -2.0 and y ¿ 9.0, output [1.0, 0.0], otherwise if x ¡ -2.0 and y ¿ 10.0, output [1.0, 0.0], otherwise if x ¡ -2.5 and y ¿ 10.0, output [1.0, 0.0], otherwise if x ¡ -1.0 and y ¿ 9.0, output [1.0, 0.0], otherwise if x ¡ -2.0 and y ¿ 7.5, output [1.0, 0.0], otherwise if x ¡ -2.0 and y ¿ 9.2, output [1.0, 0.0], otherwise if x ¡ -3.5 and y ¿ 9.5, output [1.0, 0.0], otherwise if x ¡ -2.5 and y ¿ 8.0, output [1.0, 0.0], otherwise if x ¡ -4.0 and y ¿ 9.0, output [1.0, 0.0], otherwise if x ¡ -4.5 and y ¿ 8.5, output [1.0, 0.0], otherwise if x ¡ -2.0 and y ¿ 9.8, output [1.0, 0.0], otherwise if x ¡ -2.0 and y ¿ 9.4, output [1.0, 0.0], otherwise if x ¡ -1.5 and y ¿ 9.0, output [1.0, 0.0], otherwise if x ¡ -2.0 and y ¿ 9.0, output [1.0, 0.0], otherwise if x ¡ -2.0 and y ¿ 9.2, output [1.0, 0.0], otherwise output [1.0, 0.0].

**\*\* Input: \*\***

[-2.178 9.99 ]

Please give your output strictly in the following format:

```
Explanations: [Your step-by-step analyses and results]
Output:
[ONLY A PURE probability vector, where each value is between 0.0 and 1.0 WITH TWO DECIMAL POINTS; make necessary assumptions if needed]
```

Please ONLY reply according to this format, don't give me any other words.

─────────── Forward Pass Output ───────────

Explanations:

Since x ¿ 0 and y ¡ 1.7, the output is [0.0, 1.0].

Output: [0.00, 1.00]

**Many outputs for the rest data points ...**

─────────── Forward Pass Output ───────────

Explanations:

Since x ¿ 0, we check the conditions for x ¿ 0.

x = 3.569 ¿ 3.0 and y = 2.456 ¡ 2.2, so the output is [0.0, 1.0].

Output: [0.00, 1.00]

─────────── Training Accuracy ───────────

1.0

─────────── Overall Loss ───────────

-4.9999999446126456e-09

─────────── Optimization Prompt ───────────

You are the optimizer for a model, your goal is to learn the best descriptions for the model. The model used the Current Model Descriptions below predicted how likely the given inputs belong to a class. You are given the target values, please optimize the Model Descriptions for better prediction.

**\*\* Inputs (a batch of i.i.d. data on 2-D plane: [x y]): \*\***

[[-2.178 9.99 ] [-3.111 10.867] [-4.234 8.452] [-3.616 7.818] [ 5.155 2.487] [-2.625 8.713] [ 3.75 1.157] [-2.523 7.957] [-3.348 8.705] [ 3.569 2.456]]

**\*\* Current Model Descriptions: \*\***

You are designed to do binary classification. The input is a point on a 2-D plane [x y]; you need to output a vector containing two probabilities such that each corresponds to how likely the data belongs to each class, i.e., [class 1 prob. class 2 prob.]. The sum of the vector MUST be 1.0. If x ¿ 0, if y ¡ 1.7, output [0.0, 1.0], otherwise if y ¡ 2.0 and x ¿ 3.0, output [0.0, 1.0], otherwise if y ¡ 2.8 and x ¿ 4.0, output [0.0, 1.0], otherwise if y ¡ 2.5 and x ¿ 3.5, output [0.0, 1.0], otherwise if y ¡ 3.5 and x ¿ 4.5, output [0.0, 1.0], otherwise if y ¡ 3.5 and x ¿ 5.0, output [0.0, 1.0], otherwise if y ¡ 3.0 and x ¿ 3.5, output [0.0, 1.0], otherwise if y ¡ 4.3 and x ¡= 4.0, output [1.0, 0.0], otherwise if y ¡ 4.3 and x ¿ 4.0, output [0.0, 1.0], otherwise if y ¡ 9.5, output [1.0, 0.0], otherwise if x ¿ 4.5 and y ¡ 3.0, output [0.0, 1.0], otherwise if x ¿ 3.0 and y ¡ 2.2, output [0.0, 1.0], otherwise if x ¿ 4.0 and y ¡ 1.5, output [0.0, 1.0], otherwise if x ¿ 5.0 and y ¡ 2.5, output [0.0, 1.0], otherwise if x ¿ 3.0 and y ¡ 2.0, output [0.0, 1.0], otherwise if x ¿ 4.0 and y ¡ 2.5, output [0.0, 1.0], otherwise if x ¿ 4.5 and y ¡ 2.8, output [0.0, 1.0], otherwise if x ¿ 5.0 and y ¡ 3.5, output [0.0, 1.0], otherwise if x ¿ 3.5 and y ¡ 2.2, output [0.0, 1.0], otherwise if x ¿ 4.0 and y ¡ 1.5, output [0.0, 1.0], otherwise if x ¿ 4.0 and y ¡ 2.5, output [0.0, 1.0], otherwise if x ¿ 4.0 and y ¡ 1.2, output [0.0, 1.0], otherwise if x ¿ 5.0 and y ¡ 2.3, output [0.0, 1.0], otherwise if x ¿ 4.0 and y ¡ 2.0, output [0.0, 1.0], otherwise if x ¿ 4.0 and y ¡ 2.5, output [0.0, 1.0], otherwise output [1.0, 0.0]. If x ¡ 0, if x ¡ -3.5 and y ¿ 8.5, output [1.0, 0.0], otherwise if x ¡ -2.0 and y ¿ 10.0, output [1.0, 0.0], otherwise if x ¡ -2.5 and y ¿ 10.0, output [1.0, 0.0], otherwise if x ¡ -1.0 and y ¿ 9.0, output [1.0, 0.0], otherwise if x ¡ -2.0 and y ¿ 7.5, output [1.0, 0.0], otherwise if x ¡ -2.0 and y ¿ 9.2, output [1.0, 0.0], otherwise if x ¡ -3.5 and y ¿ 9.5, output [1.0, 0.0], otherwise if x ¡ -2.5 and y ¿ 8.0, output [1.0, 0.0], otherwise if x ¡ -4.0 and y ¿ 9.0, output [1.0, 0.0], otherwise if x ¡ -4.5 and y ¿ 8.5, output [1.0, 0.0], otherwise if x ¡ -2.0 and y ¿ 9.8, output [1.0, 0.0], otherwise if x ¡ -2.0 and y ¿ 9.4, output [1.0, 0.0], otherwise if x ¡ -1.5 and y ¿ 9.0, output [1.0, 0.0], otherwise if x ¡ -2.0 and y ¿ 9.0, output [1.0, 0.0], otherwise if x ¡ -2.0 and y ¿ 9.2, output [1.0, 0.0], otherwise output [1.0, 0.0].

**\*\* The model predictions ([class 1 prob. class 2 prob.]): \*\***

[[1. 0.] [1. 0.] [1. 0.] [1. 0.] [0. 1.] [1. 0.] [0. 1.] [1. 0.] [1. 0.] [0. 1.]]

**\*\* The targets ([class 1 prob. class 2 prob.]): \*\***

[[1. 0.] [1. 0.] [1. 0.] [1. 0.] [0. 1.] [1. 0.] [0. 1.] [1. 0.] [1. 0.] [0. 1.]]

Please update the model by improving the 'New Model Descriptions', which should have lower classification error both on the current and the next batch of i.i.d. data. If previous 'Optimization Step' are provided, you can use the information from your last optimization step if it's helpful. Both the model and you MUST ONLY operate on the numerical precision of THREE decimal points. You are bad with numerical calculations, so be extra careful! Please think step by step and give your outputs strictly in the following format:

```
Reasoning:
[be explicit and verbose, improve the Current Model Descriptions by yourself; please show your work; note that you don't have access to computers]
New Model Descriptions:
[put your new decision rules here; MUST be concise and concrete; ****MUST PROVIDE THE EXACT VALUE OF THE PARAMETERS if the descriptions potentially involve unknown or learnable parameters!!!****]
```

Please ONLY reply according to this format, don't give me any other words.

## Training History: Two Blobs Classification – Step 81 (continued)

─────── **Optimization Output** ───────

Reasoning:
The model is performing well, but I noticed that the points with x ¡ -3.0 and y ¿ 9.5 are more likely to belong to class 1. By analyzing the data, I see that the points with x ¡ -3.0 and y ¿ 9.5 are consistently classified as class 1. So, I'll add a new rule to output [1.0, 0.0] in this case. Additionally, I observed that the points with x ¿ 4.0 and y ¡ 2.5 are more likely to belong to class 2. I'll add a new rule to output [0.0, 1.0] for these points.

New Model Descriptions:
If x ¿ 0, if y ¡ 1.7, output [0.0, 1.0], otherwise if y ¡ 2.0 and x ¿ 3.0, output [0.0, 1.0], otherwise if y ¡ 2.8 and x ¿ 4.0, output [0.0, 1.0], otherwise if y ¡ 2.5 and x ¿ 3.5, output [0.0, 1.0], otherwise if y ¡ 3.5 and x ¿ 4.5, output [0.0, 1.0], otherwise if y ¡ 3.5 and x ¿ 5.0, output [0.0, 1.0], otherwise if y ¡ 3.0 and x ¿ 3.5, output [0.0, 1.0], otherwise if y ¡ 4.3 and x ¡= 4.0, output [1.0, 0.0], otherwise if y ¡ 4.3 and x ¿ 4.0, output [0.0, 1.0], otherwise if y ¿ 9.5, output [1.0, 0.0], otherwise if x ¿ 4.5 and y ¡ 3.0, output [0.0, 1.0], otherwise if x ¿ 3.0 and y ¡ 2.2, output [0.0, 1.0], otherwise if x ¿ 4.0 and y ¡ 1.5, output [0.0, 1.0], otherwise if x ¿ 5.0 and y ¡ 2.5, output [0.0, 1.0], otherwise if x ¿ 3.0 and y ¡ 2.0, output [0.0, 1.0], otherwise if x ¿ 4.0 and y ¡ 2.5, output [0.0, 1.0], otherwise if x ¿ 4.5 and y ¡ 2.8, output [0.0, 1.0], otherwise if x ¿ 5.0 and y ¡ 3.5, output [0.0, 1.0], otherwise if x ¿ 3.5 and y ¡ 2.2, output [0.0, 1.0], otherwise if x ¿ 4.0 and y ¡ 1.5, output [0.0, 1.0], otherwise if x ¿ 4.0 and y ¡ 2.5, output [0.0, 1.0], otherwise if x ¿ 5.0 and y ¡ 2.3, output [0.0, 1.0], otherwise if x ¿ 4.0 and y ¡ 2.0, output [0.0, 1.0], otherwise if x ¿ 4.0 and y ¡ 2.5, output [0.0, 1.0], otherwise output [1.0, 0.0]. If x ¡ 0, if x ¡ -3.5 and y ¿ 8.5, output [1.0, 0.0], otherwise if x ¡ -2.0 and y ¿ 9.0, output [1.0, 0.0], otherwise if x ¡ -2.0 and y ¿ 10.0, output [1.0, 0.0], otherwise if x ¡ -1.0 and y ¿ 9.0, output [1.0, 0.0], otherwise if x ¡ -2.0 and y ¿ 7.5, output [1.0, 0.0], otherwise if x ¡ -2.0 and y ¿ 9.2, output [1.0, 0.0], otherwise if x ¡ -3.5 and y ¿ 9.5, output [1.0, 0.0], otherwise if x ¡ -2.5 and y ¿ 8.0, output [1.0, 0.0], otherwise if x ¡ -4.0 and y ¿ 9.0, output [1.0, 0.0], otherwise if x ¡ -4.5 and y ¿ 8.5, output [1.0, 0.0], otherwise if x ¡ -2.0 and y ¿ 9.8, output [1.0, 0.0], otherwise if x ¡ -2.0 and y ¿ 9.4, output [1.0, 0.0], otherwise if x ¡ -1.5 and y ¿ 9.0, output [1.0, 0.0], otherwise if x ¡ -2.0 and y ¿ 9.0, output [1.0, 0.0], otherwise if x ¡ -2.0 and y ¿ 9.2, output [1.0, 0.0], otherwise if x ¡ -3.0 and y ¿ 9.5, output [1.0, 0.0], otherwise output [1.0, 0.0].

─────── **Updated Theta** ───────

If x ¿ 0, if y ¡ 1.7, output [0.0, 1.0], otherwise if y ¡ 2.0 and x ¿ 3.0, output [0.0, 1.0], otherwise if y ¡ 2.8 and x ¿ 4.0, output [0.0, 1.0], otherwise if y ¡ 2.5 and x ¿ 3.5, output [0.0, 1.0], otherwise if y ¡ 3.5 and x ¿ 4.5, output [0.0, 1.0], otherwise if y ¡ 3.5 and x ¿ 5.0, output [0.0, 1.0], otherwise if y ¡ 3.0 and x ¿ 3.5, output [0.0, 1.0], otherwise if y ¡ 4.3 and x ¡= 4.0, output [1.0, 0.0], otherwise if y ¡ 4.3 and x ¿ 4.0, output [0.0, 1.0], otherwise if y ¿ 9.5, output [1.0, 0.0], otherwise if x ¿ 4.5 and y ¡ 3.0, output [0.0, 1.0], otherwise if x ¿ 3.0 and y ¡ 2.2, output [0.0, 1.0], otherwise if x ¿ 4.0 and y ¡ 1.5, output [0.0, 1.0], otherwise if x ¿ 5.0 and y ¡ 2.5, output [0.0, 1.0], otherwise if x ¿ 3.0 and y ¡ 2.0, output [0.0, 1.0], otherwise if x ¿ 4.0 and y ¡ 2.5, output [0.0, 1.0], otherwise if x ¿ 4.5 and y ¡ 2.8, output [0.0, 1.0], otherwise if x ¿ 5.0 and y ¡ 3.5, output [0.0, 1.0], otherwise if x ¿ 3.5 and y ¡ 2.2, output [0.0, 1.0], otherwise if x ¿ 4.0 and y ¡ 1.5, output [0.0, 1.0], otherwise if x ¿ 4.0 and y ¡ 2.5, output [0.0, 1.0], otherwise if x ¿ 5.0 and y ¡ 2.3, output [0.0, 1.0], otherwise if x ¿ 4.0 and y ¡ 2.0, output [0.0, 1.0], otherwise if x ¿ 4.0 and y ¡ 2.5, output [0.0, 1.0], otherwise output [1.0, 0.0]. If x ¡ 0, if x ¡ -3.5 and y ¿ 8.5, output [1.0, 0.0], otherwise if x ¡ -2.0 and y ¿ 9.0, output [1.0, 0.0], otherwise if x ¡ -2.0 and y ¿ 10.0, output [1.0, 0.0], otherwise if x ¡ -2.5 and y ¿ 10.0, output [1.0, 0.0], otherwise if x ¡ -1.0 and y ¿ 9.0, output [1.0, 0.0], otherwise if x ¡ -2.0 and y ¿ 7.5, output [1.0, 0.0], otherwise if x ¡ -2.0 and y ¿ 9.2, output [1.0, 0.0], otherwise if x ¡ -3.5 and y ¿ 9.5, output [1.0, 0.0], otherwise if x ¡ -2.5 and y ¿ 8.0, output [1.0, 0.0], otherwise if x ¡ -4.0 and y ¿ 9.0, output [1.0, 0.0], otherwise if x ¡ -4.5 and y ¿ 8.5, output [1.0, 0.0], otherwise if x ¡ -2.0 and y ¿ 9.8, output [1.0, 0.0], otherwise if x ¡ -2.0 and y ¿ 9.4, output [1.0, 0.0], otherwise if x ¡ -1.5 and y ¿ 9.0, output [1.0, 0.0], otherwise if x ¡ -2.0 and y ¿ 9.0, output [1.0, 0.0], otherwise if x ¡ -2.0 and y ¿ 9.2, output [1.0, 0.0], otherwise if x ¡ -3.0 and y ¿ 9.5, output [1.0, 0.0], otherwise output [1.0, 0.0].

## Training History: Two Blobs Classification – Step 82

**Epoch 8 - Step 2**

──────── **Forward Pass Prompt** ────────

You are the model.

**\*\* Model Descriptions: \*\***

You are designed to do binary classification. The input is a point on a 2-D plane [x y]; you need to output a vector containing two probabilities such that each corresponds to how likely the data belongs to each class, i.e., [class 1 prob. class 2 prob.]. The sum of the vector MUST be 1.0. If x > 0, if y < 1.7, output [0.0, 1.0], otherwise if y < 2.0 and x > 3.0, output [0.0, 1.0], otherwise if y < 2.8 and x > 4.0, output [0.0, 1.0], otherwise if y < 2.5 and x > 3.5, output [0.0, 1.0], otherwise if y < 3.5 and x > 4.5, output [0.0, 1.0], otherwise if y < 3.5 and x > 5.0, output [0.0, 1.0], otherwise if y < 3.0 and x > 3.5, output [0.0, 1.0], otherwise if y < 4.3 and x >= 4.0, output [1.0, 0.0], otherwise if y < 4.3 and x > 4.0, output [0.0, 1.0], otherwise if y > 9.5, output [1.0, 0.0], otherwise if x < 4.5 and y < 3.0, output [0.0, 1.0], otherwise if x > 3.0 and y < 2.2, output [0.0, 1.0], otherwise if x > 4.0 and y < 1.5, output [0.0, 1.0], otherwise if x > 5.0 and y < 2.5, output [0.0, 1.0], otherwise if x > 3.0 and y < 2.0, output [0.0, 1.0], otherwise if x > 4.0 and y < 2.5, output [0.0, 1.0], otherwise if x > 4.5 and y < 2.8, output [0.0, 1.0], otherwise if x > 5.0 and y < 3.5, output [0.0, 1.0], otherwise if x > 3.5 and y < 2.2, output [0.0, 1.0], otherwise if x > 4.0 and y < 1.5, output [0.0, 1.0], otherwise if x > 4.0 and y < 2.5, output [0.0, 1.0], otherwise if x > 4.0 and y < 1.2, output [0.0, 1.0], otherwise if x > 5.0 and y < 2.3, output [0.0, 1.0], otherwise if x > 4.0 and y < 2.0, output [0.0, 1.0], otherwise if x > 4.0 and y < 2.5, output [0.0, 1.0], otherwise output [1.0, 0.0]. If x < 0, if x < -3.5 and y > 8.5, output [1.0, 0.0], otherwise if x < -2.0 and y > 9.0, output [1.0, 0.0], otherwise if x < -2.0 and y < 10.0, output [1.0, 0.0], otherwise if x < -2.5 and y < 10.0, output [1.0, 0.0], otherwise if x < -1.0 and y > 9.0, output [1.0, 0.0], otherwise if x < -2.0 and y > 7.5, output [1.0, 0.0], otherwise if x < -2.0 and y > 9.2, output [1.0, 0.0], otherwise if x < -3.5 and y > 9.5, output [1.0, 0.0], otherwise if x < -2.5 and y > 8.0, output [1.0, 0.0], otherwise if x < -4.0 and y > 9.0, output [1.0, 0.0], otherwise if x < -4.5 and y > 8.5, output [1.0, 0.0], otherwise if x < -2.0 and y > 9.8, output [1.0, 0.0], otherwise if x < -2.0 and y > 9.4, output [1.0, 0.0], otherwise if x < -1.5 and y > 9.0, output [1.0, 0.0], otherwise if x < -2.0 and y > 9.0, output [1.0, 0.0], otherwise if x < -2.0 and y > 9.2, output [1.0, 0.0], otherwise if x < -3.0 and y > 9.5, output [1.0, 0.0], otherwise output [1.0, 0.0].

**\*\* Input: \*\***

[-2.252 8.94 ]

Please give your output strictly in the following format:

```
Explanations: [Your step-by-step analyses and results]
Output:
[ONLY A PURE probability vector, where each value is between 0.0 and 1.0 WITH TWO DECIMAL POINTS; make necessary assumptions if needed]
```

Please ONLY reply according to this format, don't give me any other words.

──────── **Forward Pass Output** ────────

Explanations:

Since x < 0, we check the conditions for x < 0.

x < -3.5 and y > 8.5, which is true.

Output: [1.00, 0.00]

**Many outputs for the rest data points ...**

──────── **Forward Pass Output** ────────

Explanations:

Since x < 0, we check the conditions for x < 0.

x = -2.449, y = 11.478.

The condition x < -2.0 and y > 10.0 is satisfied.

So, the output is [1.0, 0.0].

Output: [1.00, 0.00]

──────── **Training Accuracy** ────────

1.0

──────── **Overall Loss** ────────

-4.9999999446126456e-09

──────── **Optimization Prompt** ────────

You are the optimizer for a model, your goal is to learn the best descriptions for the model. The model used the Current Model Descriptions below predicted how likely the given inputs belong to a class. You are given the target values, please optimize the Model Descriptions for better prediction.

**\*\* Inputs (a batch of i.i.d. data on 2-D plane: [x y]): \*\***

[[-2.252 8.94 ] [ 3.578 2.447] [-2.449 11.478] [-3.211 8.687] [ 6.784 2.607] [ 4.9 2.755] [ 4.653 3.427] [-1.594 9.343] [-2.702 9.316] [-1.044 8.789]]

**\*\* Current Model Descriptions: \*\***

You are designed to do binary classification. The input is a point on a 2-D plane [x y]; you need to output a vector containing two probabilities such that each corresponds to how likely the data belongs to each class, i.e., [class 1 prob. class 2 prob.]. The sum of the vector MUST be 1.0. If x > 0, if y < 1.7, output [0.0, 1.0], otherwise if y < 2.0 and x > 3.0, output [0.0, 1.0], otherwise if y < 2.8 and x > 4.0, output [0.0, 1.0], otherwise if y < 2.5 and x > 3.5, output [0.0, 1.0], otherwise if y < 3.5 and x > 4.5, output [0.0, 1.0], otherwise if y < 3.5 and x > 5.0, output [0.0, 1.0], otherwise if y < 3.0 and x > 3.5, output [0.0, 1.0], otherwise if y < 4.3 and x >= 4.0, output [1.0, 0.0], otherwise if y < 4.3 and x > 4.0, output [0.0, 1.0], otherwise if y > 9.5, output [1.0, 0.0], otherwise if x < 4.5 and y < 3.0, output [0.0, 1.0], otherwise if x > 3.0 and y < 2.2, output [0.0, 1.0], otherwise if x > 4.0 and y < 1.5, output [0.0, 1.0], otherwise if x > 5.0 and y < 2.5, output [0.0, 1.0], otherwise if x > 3.0 and y < 2.0, output [0.0, 1.0], otherwise if x > 4.5 and y < 2.8, output [0.0, 1.0], otherwise if x > 5.0 and y < 3.5, output [0.0, 1.0], otherwise if x > 3.5 and y < 2.2, output [0.0, 1.0], otherwise if x > 4.0 and y < 1.5, output [0.0, 1.0], otherwise if x > 4.0 and y < 2.5, output [0.0, 1.0], otherwise if x > 4.0 and y < 1.2, output [0.0, 1.0], otherwise if x > 5.0 and y < 2.3, output [0.0, 1.0], otherwise if x > 4.0 and y < 2.0, output [0.0, 1.0], otherwise if x > 4.0 and y < 2.5, output [0.0, 1.0], otherwise output [1.0, 0.0]. If x < 0, if x < -3.5 and y > 8.5, output [1.0, 0.0], otherwise if x < -2.0 and y > 9.0, output [1.0, 0.0], otherwise if x < -2.0 and y < 10.0, output [1.0, 0.0], otherwise if x < -2.5 and y < 10.0, output [1.0, 0.0], otherwise if x < -1.0 and y > 9.0, output [1.0, 0.0], otherwise if x < -2.0 and y > 7.5, output [1.0, 0.0], otherwise if x < -2.0 and y > 9.2, output [1.0, 0.0], otherwise if x < -3.5 and y > 9.5, output [1.0, 0.0], otherwise if x < -2.5 and y > 8.0, output [1.0, 0.0], otherwise if x < -4.0 and y > 9.0, output [1.0, 0.0], otherwise if x < -2.0 and y > 9.8, output [1.0, 0.0], otherwise if x < -2.0 and y > 9.4, output [1.0, 0.0], otherwise if x < -1.5 and y > 9.0, output [1.0, 0.0], otherwise if x < -2.0 and y > 9.0, output [1.0, 0.0], otherwise if x < -2.0 and y > 9.2, output [1.0, 0.0], otherwise if x < -3.0 and y > 9.5, output [1.0, 0.0], otherwise output [1.0, 0.0].

**\*\* The model predictions ([class 1 prob. class 2 prob.]): \*\***

[[1. 0.] [0. 1.] [1. 0.] [1. 0.] [0. 1.] [0. 1.] [0. 1.] [1. 0.] [1. 0.] [1. 0.]]

**\*\* The targets ([class 1 prob. class 2 prob.]): \*\***

[[1. 0.] [0. 1.] [1. 0.] [1. 0.] [0. 1.] [0. 1.] [0. 1.] [1. 0.] [1. 0.] [1. 0.]]

Please update the model by improving the 'New Model Descriptions', which should have lower classification error both on the current and the next batch of i.i.d. data. If previous 'Optimization Step' are provided, you can use the information from your last optimization step if it's helpful. Both the model and you MUST ONLY operate on the numerical precision of THREE decimal points. You are bad with numerical calculations, so be extra careful! Please think step by step and give your outputs strictly in the following format:

```
Reasoning:
[be explicit and verbose, improve the Current Model Descriptions by yourself; please show your work; note that you don't have access to computers]
New Model Descriptions:
[put your new decision rules here; MUST be concise and concrete; ****MUST PROVIDE THE EXACT VALUE OF THE PARAMETERS if the descriptions potentially involve unknown or learnable parameters!!!****]
```

Please ONLY reply according to this format, don't give me any other words.

**Training History: Two Blobs Classification – Step 82 (continued)**

——————— **Optimization Output** ———————

Reasoning:

The model is performing well, but I noticed that the points with x ¿ 4.5 and y ¡ 2.8 are more likely to belong to class 2. By analyzing the data, I see that the points with x ¿ 4.5 and y ¡ 2.8 are consistently classified as class 2. So, I'll add a new rule to output [0.0, 1.0] in this case. Additionally, I observed that the points with x ¡ -2.5 and y ¿ 9.2 are more likely to belong to class 1. I'll add a new rule to output [1.0, 0.0] for these points.

New Model Descriptions:

If x ¿ 0, if y ¡ 1.7, output [0.0, 1.0], otherwise if y ¡ 2.0 and x ¿ 3.0, output [0.0, 1.0], otherwise if y ¡ 2.8 and x ¿ 4.0, output [0.0, 1.0], otherwise if y ¡ 2.5 and x ¿ 3.5, output [0.0, 1.0], otherwise if y ¡ 3.5 and x ¿ 4.5, output [0.0, 1.0], otherwise if y ¡ 3.5 and x ¿ 5.0, output [0.0, 1.0], otherwise if y ¡ 3.0 and x ¿ 3.5, output [0.0, 1.0], otherwise if y ¡ 4.3 and x ¡= 4.0, output [1.0, 0.0], otherwise if y ¡ 4.3 and x ¿ 4.0, output [0.0, 1.0], otherwise if y ¿ 9.5, output [1.0, 0.0], otherwise if x ¿ 4.5 and y ¡ 3.0, output [0.0, 1.0], otherwise if x ¿ 3.0 and y ¡ 2.2, output [0.0, 1.0], otherwise if x ¿ 4.0 and y ¡ 1.5, output [0.0, 1.0], otherwise if x ¿ 5.0 and y ¡ 2.5, output [0.0, 1.0], otherwise if x ¿ 3.0 and y ¡ 2.0, output [0.0, 1.0], otherwise if x ¿ 4.0 and y ¡ 2.5, output [0.0, 1.0], otherwise if x ¿ 4.5 and y ¡ 2.8, output [0.0, 1.0], otherwise if x ¿ 5.0 and y ¡ 3.5, output [0.0, 1.0], otherwise if x ¿ 3.5 and y ¡ 2.2, output [0.0, 1.0], otherwise if x ¿ 4.0 and y ¡ 1.5, output [0.0, 1.0], otherwise if x ¿ 4.0 and y ¡ 2.5, output [0.0, 1.0], otherwise if x ¿ 4.0 and y ¡ 1.2, output [0.0, 1.0], otherwise if x ¿ 5.0 and y ¡ 2.3, output [0.0, 1.0], otherwise if x ¿ 4.0 and y ¡ 2.0, output [0.0, 1.0], otherwise if x ¿ 4.0 and y ¡ 2.5, output [0.0, 1.0], otherwise output [1.0, 0.0]. If x ¡ 0, if x ¡ -3.5 and y ¿ 8.5, output [1.0, 0.0], otherwise if x ¡ -2.0 and y ¿ 9.0, output [1.0, 0.0], otherwise if x ¡ -2.0 and y ¿ 10.0, output [1.0, 0.0], otherwise if x ¡ -2.5 and y ¿ 10.0, output [1.0, 0.0], otherwise if x ¡ -1.0 and y ¿ 9.0, output [1.0, 0.0], otherwise if x ¡ -2.0 and y ¿ 7.5, output [1.0, 0.0], otherwise if x ¡ -2.0 and y ¿ 9.2, output [1.0, 0.0], otherwise if x ¡ -3.5 and y ¿ 9.5, output [1.0, 0.0], otherwise if x ¡ -2.5 and y ¿ 8.0, output [1.0, 0.0], otherwise if x ¡ -4.0 and y ¿ 9.0, output [1.0, 0.0], otherwise if x ¡ -4.5 and y ¿ 8.5, output [1.0, 0.0], otherwise if x ¡ -2.0 and y ¿ 9.8, output [1.0, 0.0], otherwise if x ¡ -2.0 and y ¿ 9.4, output [1.0, 0.0], otherwise if x ¡ -1.5 and y ¿ 9.0, output [1.0, 0.0], otherwise if x ¡ -2.0 and y ¿ 9.0, output [1.0, 0.0], otherwise if x ¡ -2.0 and y ¿ 9.2, output [1.0, 0.0], otherwise if x ¡ -3.0 and y ¿ 9.5, output [1.0, 0.0], otherwise if x ¡ -2.5 and y ¿ 9.2, output [1.0, 0.0], otherwise output [1.0, 0.0].

——————— **Updated Theta** ———————

If x ¿ 0, if y ¡ 1.7, output [0.0, 1.0], otherwise if y ¡ 2.0 and x ¿ 3.0, output [0.0, 1.0], otherwise if y ¡ 2.8 and x ¿ 4.0, output [0.0, 1.0], otherwise if y ¡ 2.5 and x ¿ 3.5, output [0.0, 1.0], otherwise if y ¡ 3.5 and x ¿ 4.5, output [0.0, 1.0], otherwise if y ¡ 3.5 and x ¿ 5.0, output [0.0, 1.0], otherwise if y ¡ 3.0 and x ¿ 3.5, output [0.0, 1.0], otherwise if y ¡ 4.3 and x ¡= 4.0, output [1.0, 0.0], otherwise if y ¡ 4.3 and x ¿ 4.0, output [0.0, 1.0], otherwise if y ¿ 9.5, output [1.0, 0.0], otherwise if x ¿ 4.5 and y ¡ 3.0, output [0.0, 1.0], otherwise if x ¿ 3.0 and y ¡ 2.2, output [0.0, 1.0], otherwise if x ¿ 4.0 and y ¡ 1.5, output [0.0, 1.0], otherwise if x ¿ 5.0 and y ¡ 2.5, output [0.0, 1.0], otherwise if x ¿ 3.0 and y ¡ 2.0, output [0.0, 1.0], otherwise if x ¿ 4.0 and y ¡ 2.5, output [0.0, 1.0], otherwise if x ¿ 4.5 and y ¡ 2.8, output [0.0, 1.0], otherwise if x ¿ 5.0 and y ¡ 3.5, output [0.0, 1.0], otherwise if x ¿ 3.5 and y ¡ 2.2, output [0.0, 1.0], otherwise if x ¿ 4.0 and y ¡ 1.5, output [0.0, 1.0], otherwise if x ¿ 4.0 and y ¡ 2.5, output [0.0, 1.0], otherwise if x ¿ 4.0 and y ¡ 1.2, output [0.0, 1.0], otherwise if x ¿ 5.0 and y ¡ 2.3, output [0.0, 1.0], otherwise if x ¿ 4.0 and y ¡ 2.0, output [0.0, 1.0], otherwise if x ¿ 4.0 and y ¡ 2.5, output [0.0, 1.0], otherwise output [1.0, 0.0]. If x ¡ 0, if x ¡ -3.5 and y ¿ 8.5, output [1.0, 0.0], otherwise if x ¡ -2.0 and y ¿ 9.0, output [1.0, 0.0], otherwise if x ¡ -2.0 and y ¿ 10.0, output [1.0, 0.0], otherwise if x ¡ -2.5 and y ¿ 10.0, output [1.0, 0.0], otherwise if x ¡ -1.0 and y ¿ 9.0, output [1.0, 0.0], otherwise if x ¡ -2.0 and y ¿ 7.5, output [1.0, 0.0], otherwise if x ¡ -2.0 and y ¿ 9.2, output [1.0, 0.0], otherwise if x ¡ -3.5 and y ¿ 9.5, output [1.0, 0.0], otherwise if x ¡ -2.5 and y ¿ 8.0, output [1.0, 0.0], otherwise if x ¡ -4.0 and y ¿ 9.0, output [1.0, 0.0], otherwise if x ¡ -4.5 and y ¿ 8.5, output [1.0, 0.0], otherwise if x ¡ -2.0 and y ¿ 9.8, output [1.0, 0.0], otherwise if x ¡ -2.0 and y ¿ 9.4, output [1.0, 0.0], otherwise if x ¡ -1.5 and y ¿ 9.0, output [1.0, 0.0], otherwise if x ¡ -2.0 and y ¿ 9.0, output [1.0, 0.0], otherwise if x ¡ -2.0 and y ¿ 9.2, output [1.0, 0.0], otherwise if x ¡ -3.0 and y ¿ 9.5, output [1.0, 0.0], otherwise if x ¡ -2.5 and y ¿ 9.2, output [1.0, 0.0], otherwise output [1.0, 0.0].

## L.5  Two Circles Regression (LLama-3-70B without prior)

---

**Training History: Two Circles Classification without Prior – Step 1**

**Epoch 0 - Step 1**
——————— **Forward Pass Prompt** ———————
You are the model.
**\*\* Model Descriptions: \*\***
You are designed to do binary classification. The input is a point on a 2-D plane [x y]; you need to output the class label, i.e., an integer in the set 0, 1.
**\*\* Input: \*\***
[-0.146 -1.01 ]
Please give your output strictly in the following format:
```
Explanations: [Your step-by-step analyses and results]
Output:
[ONLY the integer class label; make necessary assumptions if needed]
```
Please ONLY reply according to this format, don't give me any other words.
——————— **Forward Pass Output** ———————
Explanations:
Since I don't have any information about the decision boundary, I will make an assumption that the decision boundary is the x=y line.
The point (0.89, 0.397) is below the x=y line, so it belongs to class 0.
Output: 0

**Many outputs for the rest data points ...**

——————— **Forward Pass Output** ———————
Explanations: Since I don't have any prior knowledge about the classification model, I'll assume a simple linear decision boundary. Let's assume the decision boundary is x + y = 0. Then, I'll plug in the input values to determine the class label.
Plugging in x = -0.146 and y = -1.01, we get -0.146 + (-1.01) = -1.156 ¡ 0. Since the result is negative, the point lies below the decision boundary.
Output: 0
——————— **Training Accuracy** ———————
0.4
——————— **Optimization Prompt** ———————
You are the optimizer for a model, your goal is to learn the best descriptions for the model. The model used the Current Model Descriptions below predicted the class labels for the given inputs. You are given the target labels, please optimize the Model Descriptions for better prediction.
**\*\* Inputs (a batch of i.i.d. data on 2-D plane: [x y]): \*\***
[[-0.146 -1.01 ] [ 0.224 -0.929] [ 0.89 0.397] [ 0.811 -0.56 ] [-0.873 0.475] [0.27 0.172] [ 0.08 0.261] [ 0.518 0.943] [-0.502 -0.788] [-0.069 -0.252]]
**\*\* Current Model Descriptions: \*\***
You are designed to do binary classification. The input is a point on a 2-D plane [x y]; you need to output the class label, i.e., an integer in the set 0, 1.
**\*\* The model predictions: \*\***
[[0] [1] [0] [1] [1] [1] [0] [1] [0] [0]]
**\*\* The targets: \*\***
[[0] [0] [0] [0] [0] [1] [1] [0] [0] [1]]
If the model is doing well, you can keep using the current descriptions. However, if the model is not performing well, please update the model by improving the 'New Model Descriptions', which should have lower classification error both on the current and the next batch of i.i.d. data. If previous 'Optimization Step' are provided, you can use the information from your last optimization step if it's helpful. DON'T use symbolic representation for the model! Please think step by step and give your outputs strictly in the following format:
```
Reasoning:
[be explicit and verbose, improve the Current Model Descriptions by yourself; please show your work; note that you don't have access to computers]
New Model Descriptions:
[put your new decision rules here; MUST be concise and concrete; \*\*\*\*MUST PROVIDE THE EXACT VALUE OF THE PARAMETERS if the descriptions potentially involve unknown or learnable parameters!!!\*\*\*\*]
```
Please ONLY reply according to this format, don't give me any other words.
——————— **Optimization Output** ———————
Reasoning:
Analyzing the current model predictions and targets, I notice that the model is not performing well, especially for points with positive y-values. Upon visual inspection of the input data, I observe that points with y-values greater than 0 tend to belong to class 1, while points with y-values less than 0 tend to belong to class 0. This suggests that the y-coordinate is a strong feature for classification. Therefore, I will update the model descriptions to incorporate this insight.
New Model Descriptions:
If y ¿ 0, then class label = 1. Otherwise, class label = 0.
——————— **Updated Theta** ———————
If y ¿ 0, then class label = 1. Otherwise, class label = 0.

3672
3673
3674
3675
3676
3677
3678
3679
3680
3681
3682
3683
3684
3685
3686
3687
3688
3689
3690
3691
3692
3693
3694
3695
3696
3697
3698
3699
3700
3701
3702
3703
3704
3705
3706
3707
3708
3709
3710
3711
3712
3713
3714
3715
3716
3717
3718
3719
3720
3721
3722
3723
3724
3725

---

### Training History: Two Circles Classification without Prior – Step 2

**Epoch 0 - Step 2**
———————— **Forward Pass Prompt** ————————
You are the model.
**\*\* Model Descriptions: \*\***
You are designed to do binary classification. The input is a point on a 2-D plane [x y]; you need to output the class label, i.e., an integer in the set 0, 1. If y ¿ 0, then class label = 1. Otherwise, class label = 0.
**\*\* Input: \*\***
[-0.305 0.936]
Please give your output strictly in the following format:
"'
Explanations: [Your step-by-step analyses and results]
Output:
[ONLY the integer class label; make necessary assumptions if needed]
"'
Please ONLY reply according to this format, don't give me any other words.
———————— **Forward Pass Output** ————————
Explanations: Since y ¿ 0, the class label is 1.
Output: 1

**Many outputs for the rest data points ...**

———————— **Forward Pass Output** ————————
Explanations: Given the input point [-0.735 -0.614], we need to determine the class label based on the condition provided. The condition states that if y ¿ 0, then the class label is 1, otherwise it is 0. In this case, y = -0.614, which is less than 0.
Output: 0
———————— **Training Accuracy** ————————
0.3
———————— **Optimization Prompt** ————————
You are the optimizer for a model, your goal is to learn the best descriptions for the model. The model used the Current Model Descriptions below predicted the class labels for the given inputs. You are given the target labels, please optimize the Model Descriptions for better prediction.
**\*\* Inputs (a batch of i.i.d. data on 2-D plane: [x y]): \*\***
[[-0.305 0.936] [ 0.298 -0.037] [ 0.396 0.931] [-0.735 -0.614] [-0.339 0.006] [-0.253 -0.015] [ 0.096 -0.248] [ 1.005 0.048] [-0.329 0.046] [-0.051 -0.372]]
**\*\* Current Model Descriptions: \*\***
You are designed to do binary classification. The input is a point on a 2-D plane [x y]; you need to output the class label, i.e., an integer in the set 0, 1. If y ¿ 0, then class label = 1. Otherwise, class label = 0.
**\*\* The model predictions: \*\***
[[1] [0] [1] [0] [1] [0] [0] [1] [1] [0]]
**\*\* The targets: \*\***
[[0] [1] [0] [0] [1] [1] [1] [0] [1] [1]]
If the model is doing well, you can keep using the current descriptions. However, if the model is not performing well, please update the model by improving the 'New Model Descriptions', which should have lower classification error both on the current and the next batch of i.i.d. data. If previous 'Optimization Step' are provided, you can use the information from your last optimization step if it's helpful. DON'T use symbolic representation for the model! Please think step by step and give your outputs strictly in the following format:
"'
Reasoning:
[be explicit and verbose, improve the Current Model Descriptions by yourself; please show your work; note that you don't have access to computers]
New Model Descriptions:
[put your new decision rules here; MUST be concise and concrete; \*\*\*\*MUST PROVIDE THE EXACT VALUE OF THE PARAMETERS if the descriptions potentially involve unknown or learnable parameters!!!\*\*\*\*]
"'
Please ONLY reply according to this format, don't give me any other words.
———————— **Optimization Output** ————————
Reasoning: Analyzing the current model predictions and targets, I notice that the model is not performing well, especially for points with x-values greater than 0. Upon visual inspection of the input data, I observe that points with x-values greater than 0.3 tend to belong to class 1, while points with x-values less than 0.3 tend to belong to class 0. This suggests that the x-coordinate is also a strong feature for classification. Therefore, I will update the model descriptions to incorporate this insight.
New Model Descriptions:
If x ¿ 0.3, then class label = 1. If x ¡= 0.3 and y ¿ 0, then class label = 1. Otherwise, class label = 0.
———————— **Updated Theta** ————————
If x ¿ 0.3, then class label = 1. If x ¡= 0.3 and y ¿ 0, then class label = 1. Otherwise, class label = 0.

---

## Training History: Two Circles Classification without Prior – Step 3

**Epoch 0 - Step 3**

———————— **Forward Pass Prompt** ————————

You are the model.

**\*\* Model Descriptions: \*\***

You are designed to do binary classification. The input is a point on a 2-D plane [x y]; you need to output the class label, i.e., an integer in the set 0, 1. If x ¿ 0.3, then class label = 1. If x ¡= 0.3 and y ¿ 0, then class label = 1. Otherwise, class label = 0.

**\*\* Input: \*\***

[0.275 0.163]

Please give your output strictly in the following format:

‘‘‘

Explanations: [Your step-by-step analyses and results]

Output:

[ONLY the integer class label; make necessary assumptions if needed]

’’’

Please ONLY reply according to this format, don't give me any other words.

———————— **Forward Pass Output** ————————

Explanations: Since x = 0.403 ¿ 0.3, we can directly determine the class label.

Output: 1

**Many outputs for the rest data points ...**

———————— **Forward Pass Output** ————————

Explanations:

1. The input is [0, -1.012].

2. Since x = 0 is not greater than 0.3, move to the next condition.

3. Since x ¡= 0.3 and y = -1.012 is not greater than 0, the class label is 0.

Output: 0

———————— **Training Accuracy** ————————

0.6

———————— **Optimization Prompt** ————————

You are the optimizer for a model, your goal is to learn the best descriptions for the model. The model used the Current Model Descriptions below predicted the class labels for the given inputs. You are given the target labels, please optimize the Model Descriptions for better prediction.

**\*\* Inputs (a batch of i.i.d. data on 2-D plane: [x y]): \*\***

[[ 0.275 0.163] [ 0. -1.012] [-1.065 -0.126] [ 0.133 0.22 ] [-0.317 -0.931] [ 0.403 0.96 ] [ 0.013 0.996] [ 0.147 -0.178] [-0.028 0.422] [-0.233 -0.186]]

**\*\* Current Model Descriptions: \*\***

You are designed to do binary classification. The input is a point on a 2-D plane [x y]; you need to output the class label, i.e., an integer in the set 0, 1. If x ¿ 0.3, then class label = 1. If x ¡= 0.3 and y ¿ 0, then class label = 1. Otherwise, class label = 0.

**\*\* The model predictions: \*\***

[[1] [0] [0] [1] [0] [1] [1] [0] [1] [0]]

**\*\* The targets: \*\***

[[1] [0] [0] [1] [0] [0] [0] [1] [1] [1]]

If the model is doing well, you can keep using the current descriptions. However, if the model is not performing well, please update the model by improving the 'New Model Descriptions', which should have lower classification error both on the current and the next batch of i.i.d. data. If previous 'Optimization Step' are provided, you can use the information from your last optimization step if it's helpful. DON'T use symbolic representation for the model! Please think step by step and give your outputs strictly in the following format:

‘‘‘

Reasoning:

[be explicit and verbose, improve the Current Model Descriptions by yourself; please show your work; note that you don't have access to computers]

New Model Descriptions:

[put your new decision rules here; MUST be concise and concrete; \*\*\*\*MUST PROVIDE THE EXACT VALUE OF THE PARAMETERS if the descriptions potentially involve unknown or learnable parameters!!!\*\*\*\*]

’’’

Please ONLY reply according to this format, don't give me any other words.

———————— **Optimization Output** ————————

Reasoning:

Analyzing the current model predictions and targets, I notice that the model is not performing well, especially for points with x-values around 0. Upon visual inspection of the input data, I observe that points with y-values greater than 0.2 tend to belong to class 1, while points with y-values less than 0.2 tend to belong to class 0. This suggests that the y-coordinate is also a strong feature for classification. Therefore, I will update the model descriptions to incorporate this insight.

New Model Descriptions:

If x ¿ 0.3, then class label = 1. If x ¡= 0.3 and y ¿ 0.2, then class label = 1. Otherwise, class label = 0.

———————— **Updated Theta** ————————

If x ¿ 0.3, then class label = 1. If x ¡= 0.3 and y ¿ 0.2, then class label = 1. Otherwise, class label = 0.

## Training History: Two Circles Classification without Prior – Step 4

**Epoch 0 - Step 4**

—————— **Forward Pass Prompt** ——————

You are the model.

**Model Descriptions: **

You are designed to do binary classification. The input is a point on a 2-D plane [x y]; you need to output the class label, i.e., an integer in the set 0, 1. If x ¿ 0.3, then class label = 1. If x ¡= 0.3 and y ¿ 0.2, then class label = 1. Otherwise, class label = 0.

**Input: **

[-0.811 0.529]

Please give your output strictly in the following format:

"'

Explanations: [Your step-by-step analyses and results]

Output:

[ONLY the integer class label; make necessary assumptions if needed]

"'

Please ONLY reply according to this format, don't give me any other words.

—————— **Forward Pass Output** ——————

Explanations: Since x = 0.943 ¿ 0.3, the class label is 1.

Output: 1

**Many outputs for the rest data points ...**

—————— **Forward Pass Output** ——————

Explanations:

Since x = -0.995, which is less than or equal to 0.3, we need to check the value of y.

Since y = -0.025, which is less than or equal to 0.2, we conclude that the class label is 0.

Output: 0

—————— **Training Accuracy** ——————

0.2

—————— **Optimization Prompt** ——————

You are the optimizer for a model, your goal is to learn the best descriptions for the model. The model used the Current Model Descriptions below predicted the class labels for the given inputs. You are given the target labels, please optimize the Model Descriptions for better prediction.

**Inputs (a batch of i.i.d. data on 2-D plane: [x y]): **

[[-0.811 0.529] [-0.036 0.323] [-0.63 0.698] [-0.173 -0.246] [ 0.943 -0.297] [ 0.1 -0.288] [-0.995 -0.025] [ 0.666 -0.666] [ 0.24 -0.039] [ 0.542 0.887]]

**Current Model Descriptions: **

You are designed to do binary classification. The input is a point on a 2-D plane [x y]; you need to output the class label, i.e., an integer in the set 0, 1. If x ¿ 0.3, then class label = 1. If x ¡= 0.3 and y ¿ 0.2, then class label = 1. Otherwise, class label = 0.

**The model predictions: **

[[1] [1] [1] [0] [1] [0] [0] [1] [0] [1]]

**The targets: **

[[0] [1] [0] [1] [0] [0] [1] [0] [1] [0]]

If the model is doing well, you can keep using the current descriptions. However, if the model is not performing well, please update the model by improving the 'New Model Descriptions', which should have lower classification error both on the current and the next batch of i.i.d. data. If previous 'Optimization Step' are provided, you can use the information from your last optimization step if it's helpful. DON'T use symbolic representation for the model! Please think step by step and give your outputs strictly in the following format:

"'

Reasoning:

[be explicit and verbose, improve the Current Model Descriptions by yourself; please show your work; note that you don't have access to computers]

New Model Descriptions:

[put your new decision rules here; MUST be concise and concrete; ****MUST PROVIDE THE EXACT VALUE OF THE PARAMETERS if the descriptions potentially involve unknown or learnable parameters!!!****]

"'

Please ONLY reply according to this format, don't give me any other words.

—————— **Optimization Output** ——————

Reasoning:

Analyzing the current model predictions and targets, I notice that the model is not performing well, especially for points with x-values around 0. Upon visual inspection of the input data, I observe that points with x-values less than -0.5 tend to belong to class 0, while points with x-values greater than 0.5 tend to belong to class 0 as well. This suggests that the x-coordinate has a non-linear relationship with the class label. I also notice that points with y-values greater than 0.5 tend to belong to class 0, while points with y-values less than -0.2 tend to belong to class 1. This suggests that the y-coordinate is also important for classification. Therefore, I will update the model descriptions to incorporate these insights.

New Model Descriptions:

If x ¡ -0.5 or x ¿ 0.5, then class label = 0. If y ¿ 0.5, then class label = 0. If y ¡ -0.2, then class label = 1. Otherwise, class label = 1.

—————— **Updated Theta** ——————

If x ¡ -0.5 or x ¿ 0.5, then class label = 0. If y ¿ 0.5, then class label = 0. If y ¡ -0.2, then class label = 1. Otherwise, class label = 1.

**Training History: Two Circles Classification without Prior – Step 5**

**Epoch 0 - Step 5**
———————— **Forward Pass Prompt** ————————
You are the model.
** Model Descriptions: **
You are designed to do binary classification. The input is a point on a 2-D plane [x y]; you need to output the class label, i.e., an integer in the set 0, 1. If x ¡ -0.5 or x ¿ 0.5, then class label = 0. If y ¿ 0.5, then class label = 0. If y ¡ -0.2, then class label = 1. Otherwise, class label = 1.
** Input: **
[-0.99 -0.325]
Please give your output strictly in the following format:
"'
Explanations: [Your step-by-step analyses and results]
Output:
[ONLY the integer class label; make necessary assumptions if needed]
"'
Please ONLY reply according to this format, don't give me any other words.
———————— **Forward Pass Output** ————————
Explanations:
Since x = -1.059 ¡ -0.5, class label = 0.
Output: 0

**Many outputs for the rest data points ...**

———————— **Forward Pass Output** ————————
Explanations:
1. Check if x ¡ -0.5 or x ¿ 0.5. Since 0.167 is between -0.5 and 0.5, this condition is not met.
2. Check if y ¿ 0.5. Since 0.235 is less than 0.5, this condition is not met.
3. Check if y ¡ -0.2. Since 0.235 is greater than -0.2, this condition is not met.
4. Since none of the above conditions are met, the class label is 1.
Output: 1
———————— **Training Accuracy** ————————
0.8
———————— **Optimization Prompt** ————————
You are the optimizer for a model, your goal is to learn the best descriptions for the model. The model used the Current Model Descriptions below predicted the class labels for the given inputs. You are given the target labels, please optimize the Model Descriptions for better prediction.
** Inputs (a batch of i.i.d. data on 2-D plane: [x y]): **
[[-0.99 -0.325] [-0.825 0.438] [ 0.188 0.308] [-1.059 0.217] [-0.188 -0.184] [-0.322 0.084] [ 0.892 0.515] [-0.287 0.017] [ 0.167 0.235] [-0.573 -0.845]]
** Current Model Descriptions: **
You are designed to do binary classification. The input is a point on a 2-D plane [x y]; you need to output the class label, i.e., an integer in the set 0, 1. If x ¡ -0.5 or x ¿ 0.5, then class label = 0. If y ¿ 0.5, then class label = 0. If y ¡ -0.2, then class label = 1. Otherwise, class label = 1.
** The model predictions: **
[[0] [0] [0] [0] [1] [1] [0] [1] [1] [1]]
** The targets: **
[[0] [0] [1] [0] [1] [1] [0] [1] [1] [0]]
If the model is doing well, you can keep using the current descriptions. However, if the model is not performing well, please update the model by improving the 'New Model Descriptions', which should have lower classification error both on the current and the next batch of i.i.d. data. If previous 'Optimization Step' are provided, you can use the information from your last optimization step if it's helpful. DON'T use symbolic representation for the model! Please think step by step and give your outputs strictly in the following format:
"'
Reasoning:
[be explicit and verbose, improve the Current Model Descriptions by yourself; please show your work; note that you don't have access to computers]
New Model Descriptions:
[put your new decision rules here; MUST be concise and concrete; ****MUST PROVIDE THE EXACT VALUE OF THE PARAMETERS if the descriptions potentially involve unknown or learnable parameters!!!****]
"'
Please ONLY reply according to this format, don't give me any other words.
———————— **Optimization Output** ————————
Reasoning:
Analyzing the current model predictions and targets, I notice that the model is still not performing well. Upon visual inspection of the input data, I observe that points with y-values around 0 tend to be misclassified. This suggests that the y-coordinate threshold of 0.5 is too high. I also notice that points with x-values around -0.2 tend to belong to class 1, while points with x-values around 0.2 tend to belong to class 0. This suggests that the x-coordinate threshold of 0.5 is too high as well. Therefore, I will update the model descriptions to incorporate these insights.
New Model Descriptions:
If x ¡ -0.4 or x ¿ 0.4, then class label = 0. If y ¿ 0.3, then class label = 0. If y ¡ -0.2, then class label = 1. Otherwise, class label = 1.
———————— **Updated Theta** ————————
If x ¡ -0.4 or x ¿ 0.4, then class label = 0. If y ¿ 0.3, then class label = 0. If y ¡ -0.2, then class label = 1. Otherwise, class label = 1.

**Training History: Two Circles Classification without Prior – Step 40**

**Epoch 3 - Step 10**
——————— Forward Pass Prompt ———————
You are the model.
** Model Descriptions: **
You are designed to do binary classification. The input is a point on a 2-D plane [x y]; you need to output the class label, i.e., an integer in the set 0, 1. If x ¡ -0.75 or x ¿ 0.95, then class label = 0. If y ¿ 0.78, then class label = 0. If y ¡ -0.42, then class label = 0. Otherwise, if x ¡ 0.44, then class label = 1, else class label = 0.
** Input: **
[-0.101 0.309]
Please give your output strictly in the following format:
```
Explanations: [Your step-by-step analyses and results]
Output:
[ONLY the integer class label; make necessary assumptions if needed]
```
Please ONLY reply according to this format, don't give me any other words.
——————— Forward Pass Output ———————
Explanations:
Since 0.919 ¿ 0.95, class label = 0.
Output: 0

**Many outputs for the rest data points ...**

——————— Forward Pass Output ———————
Explanations:
1. Check if x ¡ -0.75 or x ¿ 0.95. Since -0.655 is not in this range, move to the next step.
2. Check if y ¿ 0.78. Since -0.649 is not in this range, move to the next step.
3. Check if y ¡ -0.42. Since -0.649 is in this range, class label = 0.
Output: 0
——————— Training Accuracy ———————
1.0
——————— Optimization Prompt ———————
You are the optimizer for a model, your goal is to learn the best descriptions for the model. The model used the Current Model Descriptions below predicted the class labels for the given inputs. You are given the target labels, please optimize the Model Descriptions for better prediction.
** Inputs (a batch of i.i.d. data on 2-D plane: [x y]): **
[[-0.101 0.309] [-1.011 -0.165] [ 0.161 -0.229] [ 1.02 -0.061] [ 0.225 0.076] [-0.024 -0.333] [-0.655 -0.649] [-1.058 0.15 ] [ 0.366 -0.098] [ 0.919 0.08 ]]
** Current Model Descriptions: **
You are designed to do binary classification. The input is a point on a 2-D plane [x y]; you need to output the class label, i.e., an integer in the set 0, 1. If x ¡ -0.75 or x ¿ 0.95, then class label = 0. If y ¿ 0.78, then class label = 0. If y ¡ -0.42, then class label = 0. Otherwise, if x ¡ 0.44, then class label = 1, else class label = 0.
** The model predictions: **
[[1] [0] [1] [0] [1] [1] [0] [0] [1] [0]]
** The targets: **
[[1] [0] [1] [0] [1] [1] [0] [0] [1] [0]]
If the model is doing well, you can keep using the current descriptions. However, if the model is not performing well, please update the model by improving the 'New Model Descriptions', which should have lower classification error both on the current and the next batch of i.i.d. data. If previous 'Optimization Step' are provided, you can use the information from your last optimization step if it's helpful. DON'T use symbolic representation for the model! Please think step by step and give your outputs strictly in the following format:
```
Reasoning:
[be explicit and verbose, improve the Current Model Descriptions by yourself; please show your work; note that you don't have access to computers]
New Model Descriptions:
[put your new decision rules here; MUST be concise and concrete; ****MUST PROVIDE THE EXACT VALUE OF THE PARAMETERS if the descriptions potentially involve unknown or learnable parameters!!!****]
```
Please ONLY reply according to this format, don't give me any other words.
——————— Optimization Output ———————
Reasoning:
Analyzing the current model predictions and targets, I observe that the model is performing well, but there is still room for improvement. Upon visual inspection of the input data, I notice that the current x-coordinate threshold of 0.44 seems to be effective, and the y-coordinate thresholds of 0.78 and -0.42 are reasonable. However, I observe that some points with x-values around -0.6 and y-values around -0.65 are misclassified, which suggests that the model is not capturing the class boundary accurately in this region. I also notice that points with x-values around 0.2 and y-values around 0.1 are correctly classified, which suggests that the model is performing well in this region. Additionally, I observe that the current x-coordinate threshold of 0.95 seems to be too aggressive, as it misclassifies some points with x-values around 0.9.
New Model Descriptions:
If x ¡ -0.75 or x ¿ 0.92, then class label = 0. If y ¿ 0.76, then class label = 0. If y ¡ -0.41, then class label = 0. Otherwise, if x ¡ 0.43, then class label = 1, else class label = 0.
——————— Updated Theta ———————
If x ¡ -0.75 or x ¿ 0.92, then class label = 0. If y ¿ 0.76, then class label = 0. If y ¡ -0.41, then class label = 0. Otherwise, if x ¡ 0.43, then class label = 1, else class label = 0.



**Training History: Two Circles Classification without Prior – Step 41**

**Epoch 4 - Step 1**
——————— **Forward Pass Prompt** ———————
You are the model.
**\*\* Model Descriptions: \*\***
You are designed to do binary classification. The input is a point on a 2-D plane [x y]; you need to output the class label, i.e., an integer in the set 0, 1. If x < -0.75 or x > 0.92, then class label = 0. If y > 0.76, then class label = 0. If y < -0.41, then class label = 0. Otherwise, if x < 0.43, then class label = 1, else class label = 0.
**\*\* Input: \*\***
[-0.146 -1.01 ]
Please give your output strictly in the following format:
"'
Explanations: [Your step-by-step analyses and results]
Output:
[ONLY the integer class label; make necessary assumptions if needed]
"'
Please ONLY reply according to this format, don't give me any other words.
——————— **Forward Pass Output** ———————
Explanations:
Given input [-0.873, 0.475].
Since x < -0.75, class label = 0.
Output: 0

**Many outputs for the rest data points ...**

——————— **Forward Pass Output** ———————
Explanations:
1. Check if x < -0.75 or x > 0.92. Since 0.08 is not in this range, move to the next step.
2. Check if y > 0.76. Since 0.261 is not in this range, move to the next step.
3. Check if y < -0.41. Since 0.261 is not in this range, move to the next step.
4. Since x < 0.43, class label = 1.
Output: 1
——————— **Training Accuracy** ———————
1.0
——————— **Optimization Prompt** ———————
You are the optimizer for a model, your goal is to learn the best descriptions for the model. The model used the Current Model Descriptions below predicted the class labels for the given inputs. You are given the target labels, please optimize the Model Descriptions for better prediction.
**\*\* Inputs (a batch of i.i.d. data on 2-D plane: [x y]): \*\***
[[-0.146 -1.01 ] [ 0.224 -0.929] [ 0.89 0.397] [ 0.811 -0.56 ] [-0.873 0.475] [ 0.27 0.172] [ 0.08 0.261] [ 0.518 0.943] [-0.502 -0.788] [-0.069 -0.252]]
**\*\* Current Model Descriptions: \*\***
You are designed to do binary classification. The input is a point on a 2-D plane [x y]; you need to output the class label, i.e., an integer in the set 0, 1. If x < -0.75 or x > 0.92, then class label = 0. If y > 0.76, then class label = 0. If y < -0.41, then class label = 0. Otherwise, if x < 0.43, then class label = 1, else class label = 0.
**\*\* The model predictions: \*\***
[[0] [0] [0] [0] [0] [1] [1] [0] [0] [1]]
**\*\* The targets: \*\***
[[0] [0] [0] [0] [0] [1] [1] [0] [0] [1]]
If the model is doing well, you can keep using the current descriptions. However, if the model is not performing well, please update the model by improving the 'New Model Descriptions', which should have lower classification error both on the current and the next batch of i.i.d. data. If previous 'Optimization Step' are provided, you can use the information from your last optimization step if it's helpful. DON'T use symbolic representation for the model! Please think step by step and give your outputs strictly in the following format:
"'
Reasoning:
[be explicit and verbose, improve the Current Model Descriptions by yourself; please show your work; note that you don't have access to computers]
New Model Descriptions:
[put your new decision rules here; MUST be concise and concrete; \*\*\*\*MUST PROVIDE THE EXACT VALUE OF THE PARAMETERS if the descriptions potentially involve unknown or learnable parameters!!!\*\*\*\*]
"'
Please ONLY reply according to this format, don't give me any other words.
——————— **Optimization Output** ———————
Reasoning:
Analyzing the current model predictions and targets, I observe that the model is performing well, but there is still room for improvement. Upon visual inspection of the input data, I notice that the current x-coordinate threshold of 0.43 seems to be effective, and the y-coordinate thresholds of 0.76 and -0.41 are reasonable. However, I observe that some points with x-values around -0.5 and y-values around -0.7 are misclassified, which suggests that the model is not capturing the class boundary accurately in this region. I also notice that points with x-values around 0.2 and y-values around 0.2 are correctly classified, which suggests that the model is performing well in this region. Additionally, I observe that the current x-coordinate threshold of 0.92 seems to be too aggressive, as it misclassifies some points with x-values around 0.8. New Model Descriptions: If x < -0.75 or x > 0.88, then class label = 0. If y > 0.74, then class label = 0. If y < -0.39, then class label = 0. Otherwise, if x < 0.42, then class label = 1, else class label = 0.
——————— **Updated Theta** ———————
If x < -0.75 or x > 0.88, then class label = 0. If y > 0.74, then class label = 0. If y < -0.39, then class label = 0. Otherwise, if x < 0.42, then class label = 1, else class label = 0.



## L.6 TWO CIRCLES REGRESSION (LLAMA-3-70B WITH PRIOR)

---

**Training History: Two Circles Classification with Prior – Step 1**

**Epoch 0 - Step 1**

———————— **Forward Pass Prompt** ————————

You are the model.

**\*\* Model Descriptions: \*\***

You are designed to do binary classification. The input is a point on a 2-D plane [x y]; you need to output the class label, i.e., an integer in the set 0, 1. The decision boundary is a circle.

**\*\* Input: \*\***

[-0.146 -1.01 ]

Please give your output strictly in the following format:

```
Explanations: [Your step-by-step analyses and results]
Output:
[ONLY the integer class label; make necessary assumptions if needed]
```

Please ONLY reply according to this format, don't give me any other words.

———————— **Forward Pass Output** ————————

Explanations:

The decision boundary is a circle, so we need to find the distance of the input point [-0.069, -0.252] from the origin (0,0). The distance is sqrt((-0.069)^2 + (-0.252)^2) = 0.264. Since the distance is less than 1, the point is inside the circle.

Output: 0

**Many outputs for the rest data points ...**

———————— **Forward Pass Output** ————————

Explanations:

The decision boundary is a circle. A point [x y] is classified as 0 if it is inside the circle, and 1 if it is outside the circle. The center of the circle is assumed to be at [0 0]. The radius of the circle is assumed to be 1 (a unit circle). The distance of the point [0.89 0.397] from the center of the circle is sqrt(0.89^2 + 0.397^2) = 1. Since the distance is equal to the radius, the point is on the circle. In this case, it is assumed that the point is classified as 0.

Output: 0

———————— **Training Accuracy** ————————

0.5

———————— **Optimization Prompt** ————————

You are the optimizer for a model, your goal is to learn the best descriptions for the model. The model used the Current Model Descriptions below predicted the class labels for the given inputs. You are given the target labels, please optimize the Model Descriptions for better prediction.

**\*\* Inputs (a batch of i.i.d. data on 2-D plane: [x y]): \*\***

[[-0.146 -1.01 ] [ 0.224 -0.929] [ 0.89 0.397] [ 0.811 -0.56 ] [-0.873 0.475] [ 0.27 0.172] [ 0.08 0.261] [ 0.518 0.943] [-0.502 -0.788] [-0.069 -0.252]]

**\*\* Current Model Descriptions: \*\***

You are designed to do binary classification. The input is a point on a 2-D plane [x y]; you need to output the class label, i.e., an integer in the set 0, 1. The decision boundary is a circle.

**\*\* The model predictions: \*\***

[[0] [0] [0] [1] [0] [0] [0] [1] [0] [0]]

**\*\* The targets: \*\***

[[0] [0] [0] [0] [0] [1] [1] [0] [0] [1]]

If the model is doing well, you can keep using the current descriptions. However, if the model is not performing well, please update the model by improving the 'New Model Descriptions', which should have lower classification error both on the current and the next batch of i.i.d. data. If previous 'Optimization Step' are provided, you can use the information from your last optimization step if it's helpful. DON'T use symbolic representation for the model! Please think step by step and give your outputs strictly in the following format:

```
Reasoning:
[be explicit and verbose, improve the Current Model Descriptions by yourself; please show your work; note that you don't have access to computers]
New Model Descriptions:
[put your new decision rules here; MUST be concise and concrete; ****MUST PROVIDE THE EXACT VALUE OF THE PARAMETERS if the descriptions potentially involve unknown or learnable parameters!!!****]
```

Please ONLY reply according to this format, don't give me any other words.

———————— **Optimization Output** ————————

Reasoning:

The model is not performing well, as it misclassifies 4 out of 10 samples. The decision boundary is a circle, but the current model predictions do not seem to follow a circular pattern. To improve the model, I will try to find a better circle that separates the classes. I will plot the data points on the 2-D plane and try to find a circle that separates the classes. After visualizing the data, I notice that the points (0.27, 0.172) and (0.08, 0.261) are misclassified, and they are close to each other. I will try to find a circle that passes through the midpoint of these two points and is centered at a point that is closer to the origin. Let the center of the circle be (a, b) and the radius be r. The midpoint of the two points is (0.175, 0.2165). I will set the center of the circle to be (0.175, 0.2165). The radius can be found by calculating the distance from the center to one of the points, say (0.27, 0.172). The radius is approximately 0.1.

New Model Descriptions:

If (x - 0.175)^2 + (y - 0.2165)^2 $\leq$ 0.01, then output 1, else output 0.

———————— **Updated Theta** ————————

If (x - 0.175)^2 + (y - 0.2165)^2 $\leq$ 0.01, then output 1, else output 0.

## Training History: Two Circles Classification with Prior – Step 2

**Epoch 0 - Step 2**

─────────── **Forward Pass Prompt** ───────────

You are the model.

**\*\* Model Descriptions: \*\***

You are designed to do binary classification. The input is a point on a 2-D plane [x y]; you need to output the class label, i.e., an integer in the set 0, 1. The decision boundary is a circle. If $(x - 0.175)^2 + (y - 0.2165)^2$ ¡= 0.01, then output 1, else output 0.

**\*\* Input: \*\***

[-0.305 0.936]

Please give your output strictly in the following format:

```

Explanations: [Your step-by-step analyses and results]

Output: [ONLY the integer class label; make necessary assumptions if needed]

```

Please ONLY reply according to this format, don't give me any other words.

─────────── **Forward Pass Output** ───────────

Explanations:

Calculate the value of $(x - 0.175)^2 + (y - 0.2165)^2 = (-0.051 - 0.175)^2 + (-0.372 - 0.2165)^2 = 0.1061$. Since 0.1061 ¿ 0.01, the point is outside the circle.

Output: 0

**Many outputs for the rest data points ...**

─────────── **Forward Pass Output** ───────────

Explanations:

The given input point is [-0.329, 0.046]. To determine the class label, we need to check if this point lies within the circle with the decision boundary. The circle's center is at (0.175, 0.2165) and the radius is sqrt(0.01) = 0.1. Let's calculate the distance of the point from the center:

Distance = sqrt$((x - 0.175)^2 + (y - 0.2165)^2)$

= sqrt$((-0.329 - 0.175)^2 + (0.046 - 0.2165)^2)$

= sqrt$((-0.504)^2 + (-0.1705)^2)$

= sqrt(0.253696 + 0.02914025)

= sqrt(0.28283625)

= 0.532

Since the distance (0.532) is greater than the radius (0.1), the point lies outside the circle.

Output: 0

─────────── **Training Accuracy** ───────────

0.4

─────────── **Optimization Prompt** ───────────

You are the optimizer for a model, your goal is to learn the best descriptions for the model. The model used the Current Model Descriptions below predicted the class labels for the given inputs. You are given the target labels, please optimize the Model Descriptions for better prediction.

**\*\* Inputs (a batch of i.i.d. data on 2-D plane: [x y]): \*\***

[[-0.305 0.936] [ 0.298 -0.037] [ 0.396 0.931] [-0.735 -0.614] [-0.339 0.006] [-0.253 -0.015] [ 0.096 -0.248] [ 1.005 0.048] [-0.329 0.046] [-0.051 -0.372]]

**\*\* Current Model Descriptions: \*\***

You are designed to do binary classification. The input is a point on a 2-D plane [x y]; you need to output the class label, i.e., an integer in the set 0, 1. The decision boundary is a circle. If $(x - 0.175)^2 + (y - 0.2165)^2$ ¡= 0.01, then output 1, else output 0.

**\*\* The model predictions: \*\***

[[0] [0] [0] [0] [0] [0] [0] [0] [0] [0]]

**\*\* The targets: \*\***

[[0] [1] [0] [0] [1] [1] [1] [0] [1] [1]]

If the model is doing well, you can keep using the current descriptions. However, if the model is not performing well, please update the model by improving the 'New Model Descriptions', which should have lower classification error both on the current and the next batch of i.i.d. data. If previous 'Optimization Step' are provided, you can use the information from your last optimization step if it's helpful. DON'T use symbolic representation for the model! Please think step by step and give your outputs strictly in the following format:

```

Reasoning:

[be explicit and verbose, improve the Current Model Descriptions by yourself; please show your work; note that you don't have access to computers]

New Model Descriptions:

[put your new decision rules here; MUST be concise and concrete; \*\*\*\*MUST PROVIDE THE EXACT VALUE OF THE PARAMETERS if the descriptions potentially involve unknown or learnable parameters!!!\*\*\*\*]

```

Please ONLY reply according to this format, don't give me any other words.

─────────── **Optimization Output** ───────────

Reasoning:

The model is not performing well, as it misclassifies 7 out of 10 samples. The current decision boundary is a circle centered at (0.175, 0.2165) with a radius of 0.1, but it does not seem to separate the classes well. I will re-examine the data and try to find a better circle that separates the classes. I notice that the points (0.298, -0.037) and (-0.339, 0.006) are misclassified, and they are close to each other. I will try to find a circle that passes through the midpoint of these two points and is centered at a point that is closer to the origin. Let the center of the circle be (a, b) and the radius be r. The midpoint of the two points is (-0.0205, -0.0155). I will set the center of the circle to be (-0.0205, -0.0155). The radius can be found by calculating the distance from the center to one of the points, say (0.298, -0.037). The radius is approximately 0.35.

New Model Descriptions:

If $(x + 0.0205)^2 + (y + 0.0155)^2$ ¡= 0.1225, then output 1, else output 0.

─────────── **Updated Theta** ───────────

If $(x + 0.0205)^2 + (y + 0.0155)^2$ ¡= 0.1225, then output 1, else output 0.

## Training History: Two Circles Classification with Prior – Step 3

**Epoch 0 - Step 3**

——————— **Forward Pass Prompt** ———————

You are the model.

** Model Descriptions: **

You are designed to do binary classification. The input is a point on a 2-D plane [x y]; you need to output the class label, i.e., an integer in the set 0, 1. The decision boundary is a circle. If $(x + 0.0205)^2 + (y + 0.0155)^2 \leq 0.1225$, then output 1, else output 0.

** Input: **

[0.275 0.163]

Please give your output strictly in the following format:

```

Explanations: [Your step-by-step analyses and results]
Output:
[ONLY the integer class label; make necessary assumptions if needed]

```

Please ONLY reply according to this format, don't give me any other words.

——————— **Forward Pass Output** ———————

Explanations:

Calculate the value of the decision boundary equation: $(0.403 + 0.0205)^2 + (0.96 + 0.0155)^2 = 0.4235^2 + 0.9755^2 = 0.179 + 0.952 = 1.131$. Since 1.131 ¿ 0.1225, the point is outside the circle.

Output: 0

**Many outputs for the rest data points ...**

——————— **Forward Pass Output** ———————

Explanations:

We are given a point (0,-1.012) on the 2-D plane. We need to check if the point lies inside the circle defined by the decision boundary.

Substituting the values in the equation: $(0 + 0.0205)^2 + (-1.012 + 0.0155)^2 \leq 0.1225$.

Calculating the values: $(0.0205)^2 + (-0.9965)^2$ ¡= 0.1225.

Calculating the values: $0.00042025 + 0.99242025$ ¡= 0.1225.

Calculating the values: 0.9928405 ¡= 0.1225.

The condition is not satisfied.

Output: 0

——————— **Training Accuracy** ———————

0.7

——————— **Optimization Prompt** ———————

You are the optimizer for a model, your goal is to learn the best descriptions for the model. The model used the Current Model Descriptions below predicted the class labels for the given inputs. You are given the target labels, please optimize the Model Descriptions for better prediction.

** Inputs (a batch of i.i.d. data on 2-D plane: [x y]): **

[[ 0.275 0.163] [ 0. -1.012] [-1.065 -0.126] [ 0.133 0.22 ] [-0.317 -0.931] [ 0.403 0.96 ] [ 0.013 0.996] [ 0.147 -0.178] [-0.028 0.422] [-0.233 -0.186]]

** Current Model Descriptions: **

You are designed to do binary classification. The input is a point on a 2-D plane [x y]; you need to output the class label, i.e., an integer in the set 0, 1. The decision boundary is a circle. If $(x + 0.0205)^2 + (y + 0.0155)^2$ ¡= 0.1225, then output 1, else output 0.

** The model predictions: **

[[1] [0] [0] [0] [0] [0] [0] [0] [0] [1]]

** The targets: **

[[1] [0] [0] [1] [0] [0] [0] [1] [1] [1]]

If the model is doing well, you can keep using the current descriptions. However, if the model is not performing well, please update the model by improving the 'New Model Descriptions', which should have lower classification error both on the current and the next batch of i.i.d. data. If previous 'Optimization Step' are provided, you can use the information from your last optimization step if it's helpful. DON'T use symbolic representation for the model! Please think step by step and give your outputs strictly in the following format:

```

Reasoning:
[be explicit and verbose, improve the Current Model Descriptions by yourself; please show your work; note that you don't have access to computers]
New Model Descriptions:
[put your new decision rules here; MUST be concise and concrete; ****MUST PROVIDE THE EXACT VALUE OF THE PARAMETERS if the descriptions potentially involve unknown or learnable parameters!!!****]

```

Please ONLY reply according to this format, don't give me any other words.

——————— **Optimization Output** ———————

Reasoning:

The model is not performing well, as it misclassifies 4 out of 10 samples. The current decision boundary is a circle centered at (-0.0205, -0.0155) with a radius of approximately 0.35, but it does not seem to separate the classes well. I will re-examine the data and try to find a better circle that separates the classes. I notice that the points (0.133, 0.22) and (0.147, -0.178) are misclassified, and they are close to each other. I will try to find a circle that passes through the midpoint of these two points and is centered at a point that is closer to the origin. Let the center of the circle be (a, b) and the radius be r. The midpoint of the two points is (0.14, 0.021). I will set the center of the circle to be (0.14, 0.021). The radius can be found by calculating the distance from the center to one of the points, say (0.133, 0.22). The radius is approximately 0.235.

New Model Descriptions:

If $(x - 0.14)^2 + (y - 0.021)^2$ ¡= 0.055225, then output 1, else output 0.

——————— **Updated Theta** ———————

If $(x - 0.14)^2 + (y - 0.021)^2$ ¡= 0.055225, then output 1, else output 0.

## Training History: Two Circles Classification with Prior – Step 4

**Epoch 0 - Step 4**

───────── **Forward Pass Prompt** ─────────

You are the model.

**\*\* Model Descriptions: \*\***

You are designed to do binary classification. The input is a point on a 2-D plane [x y]; you need to output the class label, i.e., an integer in the set 0, 1. The decision boundary is a circle. If $(x - 0.14)^2 + (y - 0.021)^2$ ¡= 0.055225, then output 1, else output 0.

**\*\* Input: \*\***

[-0.811 0.529]

Please give your output strictly in the following format:

```

Explanations: [Your step-by-step analyses and results]

Output:

[ONLY the integer class label; make necessary assumptions if needed]

```

Please ONLY reply according to this format, don't give me any other words.

───────── **Forward Pass Output** ─────────

Explanations:

Calculate the distance from the point to the center of the circle: $(0.24 - 0.14)^2 + (-0.039 - 0.021)^2 = 0.0064 + 0.0036 = 0.0100$

Since 0.0100 ¡= 0.055225, the point is inside the circle.

Output: 1

**Many outputs for the rest data points ...**

───────── **Forward Pass Output** ─────────

Explanations:

We need to check if the point (-0.811, 0.529) lies inside the circle $(x - 0.14)^2 + (y - 0.021)^2$ ¡= 0.055225.

First, calculate the left side of the inequality: $(-0.811 - 0.14)^2 + (0.529 - 0.021)^2 = 0.671^2 + 0.508^2 = 0.449441 + 0.258064 = 0.707505$.

Since 0.707505 ¿ 0.055225, the point (-0.811, 0.529) lies outside the circle.

Output: 0

───────── **Training Accuracy** ─────────

0.7

───────── **Optimization Prompt** ─────────

You are the optimizer for a model, your goal is to learn the best descriptions for the model. The model used the Current Model Descriptions below predicted the class labels for the given inputs. You are given the target labels, please optimize the Model Descriptions for better prediction.

**\*\* Inputs (a batch of i.i.d. data on 2-D plane: [x y]): \*\***

[[-0.811 0.529] [-0.036 0.323] [-0.63 0.698] [-0.173 -0.246] [ 0.943 -0.297] [ 0.1 -0.288] [-0.995 -0.025] [ 0.666 -0.666] [ 0.24 -0.039] [ 0.542 0.887]]

**\*\* Current Model Descriptions: \*\***

You are designed to do binary classification. The input is a point on a 2-D plane [x y]; you need to output the class label, i.e., an integer in the set 0, 1. The decision boundary is a circle. If $(x - 0.14)^2 + (y - 0.021)^2$ ¡= 0.055225, then output 1, else output 0.

**\*\* The model predictions: \*\***

[[0] [0] [0] [0] [0] [0] [0] [0] [1] [0]]

**\*\* The targets: \*\***

[[0] [1] [0] [1] [0] [1] [0] [0] [1] [0]]

If the model is doing well, you can keep using the current descriptions. However, if the model is not performing well, please update the model by improving the 'New Model Descriptions', which should have lower classification error both on the current and the next batch of i.i.d. data. If previous 'Optimization Step' are provided, you can use the information from your last optimization step if it's helpful. DON'T use symbolic representation for the model! Please think step by step and give your outputs strictly in the following format:

```

Reasoning:

[be explicit and verbose, improve the Current Model Descriptions by yourself; please show your work; note that you don't have access to computers]

New Model Descriptions:

[put your new decision rules here; MUST be concise and concrete; \*\*\*\*MUST PROVIDE THE EXACT VALUE OF THE PARAMETERS if the descriptions potentially involve unknown or learnable parameters!!!\*\*\*\*]

```

Please ONLY reply according to this format, don't give me any other words.

───────── **Optimization Output** ─────────

Reasoning:

The model is not performing well, as it misclassifies 6 out of 10 samples. The current decision boundary is a circle centered at (0.14, 0.021) with a radius of approximately 0.235, but it does not seem to separate the classes well. I will re-examine the data and try to find a better circle that separates the classes. I notice that the points (-0.036, 0.323), (-0.173, -0.246), and (0.1, -0.288) are misclassified, and they are all close to the origin. I will try to find a circle that passes through the origin and is centered at a point that is closer to the origin. Let the center of the circle be (a, b) and the radius be r. I will set the center of the circle to be (0, 0). The radius can be found by calculating the distance from the center to one of the points, say (-0.173, -0.246). The radius is approximately 0.292.

New Model Descriptions:

If $(x)^2 + (y)^2$ ¡= 0.085328, then output 1, else output 0.

───────── **Updated Theta** ─────────

If $(x)^2 + (y)^2$ ¡= 0.085328, then output 1, else output 0.

## Training History: Two Circles Classification with Prior – Step 5

**Epoch 0 - Step 5**

————————— **Forward Pass Prompt** —————————

You are the model.

**\*\* Model Descriptions: \*\***

You are designed to do binary classification. The input is a point on a 2-D plane [x y]; you need to output the class label, i.e., an integer in the set 0, 1. The decision boundary is a circle. If $(x)^2 + (y)^2$ ¡= 0.085328, then output 1, else output 0.

**\*\* Input: \*\***

[-0.99 -0.325]

Please give your output strictly in the following format:

```

Explanations: [Your step-by-step analyses and results]
Output:
[ONLY the integer class label; make necessary assumptions if needed]
```

Please ONLY reply according to this format, don't give me any other words.

————————— **Forward Pass Output** —————————

Explanations:

Given input [-1.059, 0.217], calculate $(x)^2 + (y)^2 = (-1.059)^2 + (0.217)^2 = 1.123$.

Since 1.123 ¿ 0.085328, it does not satisfy the condition.

Output: 0

**Many outputs for the rest data points ...**

————————— **Forward Pass Output** —————————

Explanations:

We are given a point [-0.825, 0.438] on a 2-D plane.

We need to check if the point lies inside the circle or not.

The equation of the circle is $(x)^2 + (y)^2$ ¡= 0.085328.

Let's calculate the value of $(x)^2 + (y)^2$ for the given point. $(-0.825)^2 + (0.438)^2 = 0.680625 + 0.191844 = 0.872469$.

Since 0.872469 ¿ 0.085328, the point lies outside the circle.

Output: 0

————————— **Training Accuracy** —————————

0.7

————————— **Optimization Prompt** —————————

You are the optimizer for a model, your goal is to learn the best descriptions for the model. The model used the Current Model Descriptions below predicted the class labels for the given inputs. You are given the target labels, please optimize the Model Descriptions for better prediction.

**\*\* Inputs (a batch of i.i.d. data on 2-D plane: [x y]): \*\***

[[-0.99 -0.325] [-0.825 0.438] [ 0.188 0.308] [-1.059 0.217] [-0.188 -0.184] [-0.322 0.084] [ 0.892 0.515] [-0.287 0.017] [ 0.167 0.235] [-0.573 -0.845]]

**\*\* Current Model Descriptions: \*\***

You are designed to do binary classification. The input is a point on a 2-D plane [x y]; you need to output the class label, i.e., an integer in the set 0, 1. The decision boundary is a circle. If $(x)^2 + (y)^2$ ¡= 0.085328, then output 1, else output 0.

**\*\* The model predictions: \*\***

[[0] [0] [0] [0] [1] [0] [0] [1] [0] [0]]

**\*\* The targets: \*\***

[[0] [0] [1] [0] [1] [1] [0] [1] [1] [0]]

If the model is doing well, you can keep using the current descriptions. However, if the model is not performing well, please update the model by improving the 'New Model Descriptions', which should have lower classification error both on the current and the next batch of i.i.d. data. If previous 'Optimization Step' are provided, you can use the information from your last optimization step if it's helpful. DON'T use symbolic representation for the model! Please think step by step and give your outputs strictly in the following format:

```

Reasoning:
[be explicit and verbose, improve the Current Model Descriptions by yourself; please show your work; note that you don't have access to computers]
New Model Descriptions:
[put your new decision rules here; MUST be concise and concrete; ****MUST PROVIDE THE EXACT VALUE OF THE PARAMETERS if the descriptions potentially involve unknown or learnable parameters!!!****]
```

Please ONLY reply according to this format, don't give me any other words.

————————— **Optimization Output** —————————

Reasoning:

The model is not performing well, as it misclassifies 5 out of 10 samples. The current decision boundary is a circle centered at the origin with a radius of approximately 0.292, but it does not seem to separate the classes well. I will re-examine the data and try to find a better circle that separates the classes. I notice that the points (0.188, 0.308), (-0.188, -0.184), and (0.167, 0.235) are misclassified, and they are all close to the origin. I will try to find a circle that passes through the origin and is centered at a point that is closer to the origin. Let the center of the circle be (a, b) and the radius be r. I will set the center of the circle to be (0, 0). The radius can be found by calculating the distance from the center to one of the points, say (0.167, 0.235). The radius is approximately 0.283. However, this radius is not sufficient to separate the classes well. I will try to find a better radius by examining the other misclassified points. The point (-0.322, 0.084) is also misclassified, and it is closer to the origin than the previous point. The distance from the center to this point is approximately 0.334. I will use this distance as the radius.

New Model Descriptions:

If $(x)^2 + (y)^2$ ¡= 0.111556, then output 1, else output 0.

————————— **Updated Theta** —————————

If $(x)^2 + (y)^2$ ¡= 0.111556, then output 1, else output 0.

## Training History: Two Circles Classification with Prior – Step 40

**Epoch 3 - Step 10**
——————— Forward Pass Prompt ———————
You are the model.
** Model Descriptions: **
You are designed to do binary classification. The input is a point on a 2-D plane [x y]; you need to output the class label, i.e., an integer in the set 0, 1. The decision boundary is a circle. If (x)^2 + (y)^2 ¡= 0.45, then output 1, else output 0.
** Input: **
[-0.101 0.309]
Please give your output strictly in the following format:
'''
Explanations: [Your step-by-step analyses and results]
Output:
[ONLY the integer class label; make necessary assumptions if needed]
'''
Please ONLY reply according to this format, don't give me any other words.
——————— Forward Pass Output ———————
Explanations:
Given point [-1.011 -0.165], calculate the sum of squares: (-1.011)^2 + (-0.165)^2 = 1.044721 ¿ 0.45.
Output: 0

**Many outputs for the rest data points ...**

——————— Forward Pass Output ———————
Explanations:
The given input point is [-0.024, -0.333].
To determine the class label, we need to check if the point is inside the circle or not.
The equation of the circle is (x)^2 + (y)^2 ¡= 0.45.
Let's calculate the value of this equation for the given point: (-0.024)^2 + (-0.333)^2 = 0.011056 + 0.110889 = 0.121945 ¡= 0.45.
Since the result is true, the point lies inside the circle.
Output: 1
——————— Training Accuracy ———————
0.9
——————— Optimization Prompt ———————
You are the optimizer for a model, your goal is to learn the best descriptions for the model. The model used the Current Model Descriptions below predicted the class labels for the given inputs. You are given the target labels, please optimize the Model Descriptions for better prediction.
** Inputs (a batch of i.i.d. data on 2-D plane: [x y]): **
[[-0.101 0.309] [-1.011 -0.165] [ 0.161 -0.229] [ 1.02 -0.061] [ 0.225 0.076] [-0.024 -0.333] [-0.655 -0.649] [-1.058 0.15 ] [ 0.366 -0.098] [ 0.919 0.08 ]]
** Current Model Descriptions: **
You are designed to do binary classification. The input is a point on a 2-D plane [x y]; you need to output the class label, i.e., an integer in the set 0, 1. The decision boundary is a circle. If (x)^2 + (y)^2 ¡= 0.45, then output 1, else output 0.
** The model predictions: **
[[1] [0] [1] [0] [1] [1] [1] [0] [1] [0]]
** The targets: **
[[1] [0] [1] [0] [1] [1] [0] [0] [1] [0]]
If the model is doing well, you can keep using the current descriptions. However, if the model is not performing well, please update the model by improving the 'New Model Descriptions', which should have lower classification error both on the current and the next batch of i.i.d. data. If previous 'Optimization Step' are provided, you can use the information from your last optimization step if it's helpful. DON'T use symbolic representation for the model! Please think step by step and give your outputs strictly in the following format:
'''
Reasoning:
[be explicit and verbose, improve the Current Model Descriptions by yourself; please show your work; note that you don't have access to computers]
New Model Descriptions:
[put your new decision rules here; MUST be concise and concrete; ****MUST PROVIDE THE EXACT VALUE OF THE PARAMETERS if the descriptions potentially involve unknown or learnable parameters!!!****]
'''
Please ONLY reply according to this format, don't give me any other words.
——————— Optimization Output ———————
Reasoning:
The model is not performing well, as it misclassifies the 7th sample. The current decision boundary is a circle centered at the origin with a radius of 0.45. The 7th sample is far from the origin, but the model still classifies it as 1. This means the radius of the circle is too small. I will increase the radius to reduce the misclassification error.
New Model Descriptions:
If (x)^2 + (y)^2 ¡= 0.65, then output 1, else output 0.
——————— Updated Theta ———————
If (x)^2 + (y)^2 ¡= 0.65, then output 1, else output 0.

## Training History: Two Circles Classification with Prior – Step 41

**Epoch 4 - Step 1**
———————— Forward Pass Prompt ————————
You are the model.
** Model Descriptions: **
You are designed to do binary classification. The input is a point on a 2-D plane [x y]; you need to output the class label, i.e., an integer in the set 0, 1. The decision boundary is a circle. If $(x)^2 + (y)^2 ¡= 0.65$, then output 1, else output 0.
** Input: **
[-0.146 -1.01 ]
Please give your output strictly in the following format:
"'
Explanations: [Your step-by-step analyses and results]
Output:
[ONLY the integer class label; make necessary assumptions if needed]
"'
Please ONLY reply according to this format, don't give me any other words.
———————— Forward Pass Output ————————
Explanations:
First, calculate the value of $(x)^2 + (y)^2$: $(-0.146)^2 + (-1.01)^2 = 0.021236 + 1.0201 = 1.041336$.
Since 1.041336 ¿ 0.65, the point is outside the circle.
Output: 0

**Many outputs for the rest data points ...**

———————— Forward Pass Output ————————
Explanations:
The input point is [-0.502 -0.788].
To determine the class label, we need to check if the point satisfies the condition $(x)^2 + (y)^2 ¡= 0.65$.
Here, x = -0.502 and y = -0.788. $(x)^2 = (-0.502)^2 = 0.252004$, and $(y)^2 = (-0.788)^2 = 0.621984$.
Now, $(x)^2 + (y)^2 = 0.252004 + 0.621984 = 0.874988$.
Since 0.874988 ¡= 0.65 is False, the point lies outside the circle.
Output: 0
———————— Training Accuracy ————————
1.0
———————— Optimization Prompt ————————
You are the optimizer for a model, your goal is to learn the best descriptions for the model. The model used the Current Model Descriptions below predicted the class labels for the given inputs. You are given the target labels, please optimize the Model Descriptions for better prediction.
** Inputs (a batch of i.i.d. data on 2-D plane: [x y]): **
[[-0.146 -1.01 ] [ 0.224 -0.929] [ 0.89 0.397] [ 0.811 -0.56 ] [-0.873 0.475] [ 0.27 0.172] [ 0.08 0.261] [ 0.518 0.943] [-0.502 -0.788] [-0.069 -0.252]]
** Current Model Descriptions: **
You are designed to do binary classification. The input is a point on a 2-D plane [x y]; you need to output the class label, i.e., an integer in the set 0, 1. The decision boundary is a circle. If $(x)^2 + (y)^2 ¡= 0.65$, then output 1, else output 0.
** The model predictions: **
[[0] [0] [0] [0] [0] [1] [1] [0] [0] [1]]
** The targets: **
[[0] [0] [0] [0] [0] [1] [1] [0] [0] [1]]
If the model is doing well, you can keep using the current descriptions. However, if the model is not performing well, please update the model by improving the 'New Model Descriptions', which should have lower classification error both on the current and the next batch of i.i.d. data. If previous 'Optimization Step' are provided, you can use the information from your last optimization step if it's helpful. DON'T use symbolic representation for the model! Please think step by step and give your outputs strictly in the following format:
"'
Reasoning:
[be explicit and verbose, improve the Current Model Descriptions by yourself; please show your work; note that you don't have access to computers]
New Model Descriptions:
[put your new decision rules here; MUST be concise and concrete; ****MUST PROVIDE THE EXACT VALUE OF THE PARAMETERS if the descriptions potentially involve unknown or learnable parameters!!!****]
"'
Please ONLY reply according to this format, don't give me any other words.
———————— Optimization Output ————————
Reasoning:
The model is performing well, as it correctly classifies all samples. The decision boundary is a circle centered at the origin with a radius of 0.65, which seems to be a good fit for the data. No changes are needed.
New Model Descriptions:
If $(x)^2 + (y)^2 ¡= 0.65$, then output 1, else output 0.
———————— Updated Theta ————————
If $(x)^2 + (y)^2 ¡= 0.65$, then output 1, else output 0.

## L.7 TEXT CLASSIFICATION (LLAMA-3-70B WITHOUT PRIOR)

---

**Training History: Text Classification – Step 1**

**Epoch 0 - Step 1**
————————— **Forward Pass Prompt** —————————
You are the model.
** Model Descriptions: **
You are designed to do binary classification. The input is a term; you need to output the class label, i.e., an integer in the set 0, 1.
** Input: **
fatherhood
Please give your output strictly in the following format:
```
Explanations: [Your step-by-step analyses and results]
Output:
[ONLY the integer class label; make necessary assumptions if needed]
```
Please ONLY reply according to this format, don't give me any other words.
————————— **Forward Pass Output** —————————
Explanations: Since the input is a name, "Margaret", I will assume it's a female name. Therefore, I will classify it as a female name.
Output: 1

**Many outputs for the rest data points ...**

————————— **Forward Pass Output** —————————
Explanations: The term "tomboy" is a noun that refers to a girl who exhibits characteristics or behaviors considered typical of a boy. It does not have any strong connotations or associations that would clearly link it to a specific class label. However, based on the context and common usage of the term, I will make an assumption that the class label 1 represents a more masculine or boy-like characteristic, and 0 represents a more feminine or girl-like characteristic. Given this assumption, I would classify "tomboy" as 1.
Output: 1
————————— **Training Accuracy** —————————
0.5
————————— **Optimization Prompt** —————————
You are the optimizer for a model, your goal is to learn the best descriptions for the model. The model used the Current Model Descriptions below predicted the class labels for the given inputs. You are given the target labels, please optimize the Model Descriptions for better prediction.
** Inputs (a batch of i.i.d. text): **
['fatherhood', 'grandfather', 'tomboy', 'Shawn', 'Erma', 'Ada', 'Margaret', 'Caleb', 'Earl', 'Faye']
** Current Model Descriptions: **
You are designed to do binary classification. The input is a term; you need to output the class label, i.e., an integer in the set 0, 1.
** The model predictions: **
[[1] [1] [1] [1] [1] [1] [1] [0] [1] [0]]
** The targets: **
[[0] [0] [1] [0] [1] [1] [1] [0] [0] [1]]
If the model is doing well, you can keep using the current descriptions. However, if the model is not performing well, please update the model by improving the 'New Model Descriptions', which should have lower classification error both on the current and the next batch of i.i.d. data. If previous 'Optimization Step' are provided, you can use the information from your last optimization step if it's helpful. Please think step by step and give your outputs strictly in the following format:
```
Reasoning:
[be explicit and verbose, improve the Current Model Descriptions by yourself; please show your work; note that you don't have access to computers]
New Model Descriptions:
[put your new decision rules here]
```
Please ONLY reply according to this format, don't give me any other words.
————————— **Optimization Output** —————————
Reasoning:
Upon examining the current model predictions and target labels, I notice that the model is not performing well, with 6 out of 10 predictions incorrect. The model seems to be biased towards predicting class label 1. I observe that the input terms can be categorized into two groups: male and female names, and non-name terms related to family relationships. The target labels suggest that the class label 0 corresponds to male names and non-name terms, while class label 1 corresponds to female names. To improve the model, I will create new decision rules that take into account these categories. I will update the descriptions to include more specific patterns and rules to differentiate between male and female names, as well as non-name terms.
New Model Descriptions:
If the input term ends with a suffix commonly associated with male names (e.g., -son, -er), predict class label 0. If the input term is a non-name term related to family relationships (e.g., fatherhood, grandfather), predict class label 0. If the input term is a female name (e.g., Erma, Ada, Faye), predict class label 1. Otherwise, predict class label 0.
————————— **Updated Theta** —————————
If the input term ends with a suffix commonly associated with male names (e.g., -son, -er), predict class label 0. If the input term is a non-name term related to family relationships (e.g., fatherhood, grandfather), predict class label 0. If the input term is a female name (e.g., Erma, Ada, Faye), predict class label 1. Otherwise, predict class label 0.

---

## Training History: Text Classification – Step 2

**Epoch 0 - Step 2**

———————— **Forward Pass Prompt** ————————

You are the model.

**\*\* Model Descriptions: \*\***

You are designed to do binary classification. The input is a term; you need to output the class label, i.e., an integer in the set 0, 1. If the input term ends with a suffix commonly associated with male names (e.g., -son, -er), predict class label 0. If the input term is a non-name term related to family relationships (e.g., fatherhood, grandfather), predict class label 0. If the input term is a female name (e.g., Erma, Ada, Faye), predict class label 1. Otherwise, predict class label 0.

**\*\* Input: \*\***

Lydia

Please give your output strictly in the following format:

```

Explanations: [Your step-by-step analyses and results]
Output:
[ONLY the integer class label; make necessary assumptions if needed]
```

Please ONLY reply according to this format, don't give me any other words.

———————— **Forward Pass Output** ————————

Explanations: The input term "Marlene" is a female name.
Output: 1

**Many outputs for the rest data points ...**

———————— **Forward Pass Output** ————————

Explanations: The input term is "paternal grandma". The term "grandma" is a non-name term related to family relationships, and "paternal" is an adjective that means related to father. Therefore, according to the rules, the class label is 0.
Output: 0

———————— **Training Accuracy** ————————

0.9

———————— **Optimization Prompt** ————————

You are the optimizer for a model, your goal is to learn the best descriptions for the model. The model used the Current Model Descriptions below predicted the class labels for the given inputs. You are given the target labels, please optimize the Model Descriptions for better prediction.

**\*\* Inputs (a batch of i.i.d. text): \*\***

['Lydia', 'Steven', 'Ida', 'Lisa', 'Allan', 'Jane', 'paternal grandma', 'paternal grandad', 'Marlene', 'Darrell']

**\*\* Current Model Descriptions: \*\***

You are designed to do binary classification. The input is a term; you need to output the class label, i.e., an integer in the set 0, 1. If the input term ends with a suffix commonly associated with male names (e.g., -son, -er), predict class label 0. If the input term is a non-name term related to family relationships (e.g., fatherhood, grandfather), predict class label 0. If the input term is a female name (e.g., Erma, Ada, Faye), predict class label 1. Otherwise, predict class label 0.

**\*\* The model predictions: \*\***

[[1] [0] [1] [1] [0] [1] [0] [0] [1] [0]]

**\*\* The targets: \*\***

[[1] [0] [1] [1] [0] [1] [1] [0] [1] [0]]

If the model is doing well, you can keep using the current descriptions. However, if the model is not performing well, please update the model by improving the 'New Model Descriptions', which should have lower classification error both on the current and the next batch of i.i.d. data. If previous 'Optimization Step' are provided, you can use the information from your last optimization step if it's helpful. Please think step by step and give your outputs strictly in the following format:

```

Reasoning:
[be explicit and verbose, improve the Current Model Descriptions by yourself; please show your work; note that you don't have access to computers]
New Model Descriptions:
[put your new decision rules here]
```

Please ONLY reply according to this format, don't give me any other words.

———————— **Optimization Output** ————————

Reasoning:
Upon examining the current model predictions and target labels, I notice that the model is performing reasonably well, but there are still some errors. The model correctly predicts 8 out of 10 labels, with errors occurring for the terms "paternal grandma" and "paternal grandad". I realize that the current descriptions do not account for non-name terms that contain keywords related to family relationships, but are not exact matches (e.g., "paternal grandma" vs. "grandma"). To improve the model, I will update the descriptions to include more specific rules for non-name terms that contain keywords related to family relationships. I will also refine the rules for female names to reduce the likelihood of misclassification.

New Model Descriptions:
If the input term ends with a suffix commonly associated with male names (e.g., -son, -er), predict class label 0. If the input term contains keywords related to family relationships (e.g., "father", "mother", "grand", "paternal", "maternal"), predict class label 0. If the input term is a female name (e.g., Erma, Ada, Faye, Lydia, Ida, Jane, Marlene), predict class label 1. If the input term is a name that does not clearly indicate gender (e.g., Leslie, Jordan), predict class label 0. Otherwise, predict class label 0.

———————— **Updated Theta** ————————

If the input term ends with a suffix commonly associated with male names (e.g., -son, -er), predict class label 0. If the input term contains keywords related to family relationships (e.g., "father", "mother", "grand", "paternal", "maternal"), predict class label 0. If the input term is a female name (e.g., Erma, Ada, Faye, Lydia, Ida, Jane, Marlene), predict class label 1. If the input term is a name that does not clearly indicate gender (e.g., Leslie, Jordan), predict class label 0. Otherwise, predict class label 0.

## Training History: Text Classification – Step 3

**Epoch 0 - Step 3**
──────── **Forward Pass Prompt** ────────
You are the model.
** Model Descriptions: **
You are designed to do binary classification. The input is a term; you need to output the class label, i.e., an integer in the set 0, 1. If the input term ends with a suffix commonly associated with male names (e.g., -son, -er), predict class label 0. If the input term contains keywords related to family relationships (e.g., "father", "mother", "grand", "paternal", "maternal"), predict class label 0. If the input term is a female name (e.g., Erma, Ada, Faye, Lydia, Ida, Jane, Marlene), predict class label 1. If the input term is a name that does not clearly indicate gender (e.g., Leslie, Jordan), predict class label 0. Otherwise, predict class label 0.
** Input: **
Marissa
Please give your output strictly in the following format:
```
Explanations: [Your step-by-step analyses and results]
Output:
[ONLY the integer class label; make necessary assumptions if needed]
```
Please ONLY reply according to this format, don't give me any other words.
──────── **Forward Pass Output** ────────
Explanations: The input term is "Maria", which is a female name.
Output: 1

**Many outputs for the rest data points ...**

──────── **Forward Pass Output** ────────
Explanations:
1. Check if the term ends with a suffix commonly associated with male names: Julian does not end with a suffix like -son or -er.
2. Check if the term contains keywords related to family relationships: Julian does not contain keywords like "father", "mother", "grand", "paternal", or "maternal".
3. Check if the term is a female name: Julian is not a female name.
4. Check if the term is a name that does not clearly indicate gender: Julian is a male name.
Output: 0
──────── **Training Accuracy** ────────
0.9
──────── **Optimization Prompt** ────────
You are the optimizer for a model, your goal is to learn the best descriptions for the model. The model used the Current Model Descriptions below predicted the class labels for the given inputs. You are given the target labels, please optimize the Model Descriptions for better prediction.
** Inputs (a batch of i.i.d. text): **
['Marissa', 'Wallace', 'Julian', 'Adeline', 'Jacob', 'Dylan', 'William', 'Anne', 'Maria', 'sorority']
** Current Model Descriptions: **
You are designed to do binary classification. The input is a term; you need to output the class label, i.e., an integer in the set 0, 1. If the input term ends with a suffix commonly associated with male names (e.g., -son, -er), predict class label 0. If the input term contains keywords related to family relationships (e.g., "father", "mother", "grand", "paternal", "maternal"), predict class label 0. If the input term is a female name (e.g., Erma, Ada, Faye, Lydia, Ida, Jane, Marlene), predict class label 1. If the input term is a name that does not clearly indicate gender (e.g., Leslie, Jordan), predict class label 0. Otherwise, predict class label 0.
** The model predictions: **
[[1] [0] [0] [1] [0] [0] [0] [1] [1] [0]]
** The targets: **
[[1] [0] [0] [1] [0] [0] [0] [1] [1] [1]]
If the model is doing well, you can keep using the current descriptions. However, if the model is not performing well, please update the model by improving the 'New Model Descriptions', which should have lower classification error both on the current and the next batch of i.i.d. data. If previous 'Optimization Step' are provided, you can use the information from your last optimization step if it's helpful. Please think step by step and give your outputs strictly in the following format:
```
Reasoning:
[be explicit and verbose, improve the Current Model Descriptions by yourself; please show your work; note that you don't have access to computers]
New Model Descriptions:
[put your new decision rules here]
```
Please ONLY reply according to this format, don't give me any other words.
──────── **Optimization Output** ────────
Reasoning:
Upon examining the current model predictions and target labels, I notice that the model is performing well, but there is still one error occurring for the term "sorority". I realize that the current descriptions do not account for non-name terms that are clearly related to females or males, but are not names. To improve the model, I will update the descriptions to include more specific rules for non-name terms that are clearly related to females or males. I will also refine the rules for female names to reduce the likelihood of misclassification.
New Model Descriptions:
If the input term ends with a suffix commonly associated with male names (e.g., -son, -er), predict class label 0. If the input term contains keywords related to family relationships (e.g., "father", "mother", "grand", "paternal", "maternal"), predict class label 0. If the input term is a female name (e.g., Erma, Ada, Faye, Lydia, Ida, Jane, Marlene, Marissa, Adeline, Anne, Maria), predict class label 1. If the input term is a name that does not clearly indicate gender (e.g., Leslie, Jordan), predict class label 0. If the input term is a non-name term clearly related to females (e.g., "sorority", "girl"), predict class label 1. Otherwise, predict class label 0.
──────── **Updated Theta** ────────
If the input term ends with a suffix commonly associated with male names (e.g., -son, -er), predict class label 0. If the input term contains keywords related to family relationships (e.g., "father", "mother", "grand", "paternal", "maternal"), predict class label 0. If the input term is a female name (e.g., Erma, Ada, Faye, Lydia, Ida, Jane, Marlene, Marissa, Adeline, Anne, Maria), predict class label 1. If the input term is a name that does not clearly indicate gender (e.g., Leslie, Jordan), predict class label 0. If the input term is a non-name term clearly related to females (e.g., "sorority", "girl"), predict class label 1. Otherwise, predict class label 0.

4536
4537
4538
4539
4540
4541
4542
4543
4544
4545
4546
4547
4548
4549
4550
4551
4552
4553
4554
4555
4556
4557
4558
4559
4560
4561
4562
4563
4564
4565
4566
4567
4568
4569
4570
4571
4572
4573
4574
4575
4576
4577
4578
4579
4580
4581
4582
4583
4584
4585
4586
4587
4588
4589

## Training History: Text Classification – Step 4

**Epoch 0 - Step 4**

——————— **Forward Pass Prompt** ———————

You are the model.

**\*\* Model Descriptions: \*\***

You are designed to do binary classification. The input is a term; you need to output the class label, i.e., an integer in the set 0, 1. If the input term ends with a suffix commonly associated with male names (e.g., -son, -er), predict class label 0. If the input term contains keywords related to family relationships (e.g., "father", "mother", "grand", "paternal", "maternal"), predict class label 0. If the input term is a female name (e.g., Erma, Ada, Faye, Lydia, Ida, Jane, Marlene, Marissa, Adeline, Anne, Maria), predict class label 1. If the input term is a name that does not clearly indicate gender (e.g., Leslie, Jordan), predict class label 0. If the input term is a non-name term clearly related to females (e.g., "sorority", "girl"), predict class label 1. Otherwise, predict class label 0.

**\*\* Input: \*\***

Natasha

Please give your output strictly in the following format:

```
Explanations: [Your step-by-step analyses and results]
Output:
[ONLY the integer class label; make necessary assumptions if needed]
```

Please ONLY reply according to this format, don't give me any other words.

——————— **Forward Pass Output** ———————

Explanations: Veronica is a female name, so predict class label 1.
Output: 1

**Many outputs for the rest data points ...**

——————— **Forward Pass Output** ———————

Explanations: The input term "Antonio" is a male name, and it does not contain any keywords related to family relationships. It does not end with a suffix commonly associated with male names, but it is clearly a male name.
Output: 0

——————— **Training Accuracy** ———————

1.0

——————— **Optimization Prompt** ———————

You are the optimizer for a model, your goal is to learn the best descriptions for the model. The model used the Current Model Descriptions below predicted the class labels for the given inputs. You are given the target labels, please optimize the Model Descriptions for better prediction.

**\*\* Inputs (a batch of i.i.d. text): \*\***

['Natasha', 'Veronica', 'Dawn', 'Mamie', 'Darren', 'Monique', 'gynecological', 'Antonio', 'Eula', 'Henry']

**\*\* Current Model Descriptions: \*\***

You are designed to do binary classification. The input is a term; you need to output the class label, i.e., an integer in the set 0, 1. If the input term ends with a suffix commonly associated with male names (e.g., -son, -er), predict class label 0. If the input term contains keywords related to family relationships (e.g., "father", "mother", "grand", "paternal", "maternal"), predict class label 0. If the input term is a female name (e.g., Erma, Ada, Faye, Lydia, Ida, Jane, Marlene, Marissa, Adeline, Anne, Maria), predict class label 1. If the input term is a name that does not clearly indicate gender (e.g., Leslie, Jordan), predict class label 0. If the input term is a non-name term clearly related to females (e.g., "sorority", "girl"), predict class label 1. Otherwise, predict class label 0.

**\*\* The model predictions: \*\***

[[1] [1] [1] [1] [0] [1] [1] [0] [1] [0]]

**\*\* The targets: \*\***

[[1] [1] [1] [1] [0] [1] [1] [0] [1] [0]]

If the model is doing well, you can keep using the current descriptions. However, if the model is not performing well, please update the model by improving the 'New Model Descriptions', which should have lower classification error both on the current and the next batch of i.i.d. data. If previous 'Optimization Step' are provided, you can use the information from your last optimization step if it's helpful. Please think step by step and give your outputs strictly in the following format:

```
Reasoning:
[be explicit and verbose, improve the Current Model Descriptions by yourself; please show your work; note that you don't have access to computers]
New Model Descriptions:
[put your new decision rules here]
```

Please ONLY reply according to this format, don't give me any other words.

——————— **Optimization Output** ———————

Reasoning:
Upon examining the current model predictions and target labels, I notice that the model is performing well, but there is no need for further improvement since all predictions match the target labels. The current descriptions are comprehensive and accurate, covering various cases such as male and female names, non-name terms related to females or males, and names that do not clearly indicate gender. Since the model is already performing perfectly, I will not make any changes to the descriptions.

New Model Descriptions:
If the input term ends with a suffix commonly associated with male names (e.g., -son, -er), predict class label 0. If the input term contains keywords related to family relationships (e.g., "father", "mother", "grand", "paternal", "maternal"), predict class label 0. If the input term is a female name (e.g., Erma, Ada, Faye, Lydia, Ida, Jane, Marlene, Marissa, Adeline, Anne, Maria, Natasha, Veronica, Dawn, Mamie, Monique, Eula), predict class label 1. If the input term is a name that does not clearly indicate gender (e.g., Leslie, Jordan), predict class label 0. If the input term is a non-name term clearly related to females (e.g., "sorority", "girl", "gynecological"), predict class label 1. Otherwise, predict class label 0.

——————— **Updated Theta** ———————

If the input term ends with a suffix commonly associated with male names (e.g., -son, -er), predict class label 0. If the input term contains keywords related to family relationships (e.g., "father", "mother", "grand", "paternal", "maternal"), predict class label 0. If the input term is a female name (e.g., Erma, Ada, Faye, Lydia, Ida, Jane, Marlene, Marissa, Adeline, Anne, Maria, Natasha, Veronica, Dawn, Mamie, Monique, Eula), predict class label 1. If the input term is a name that does not clearly indicate gender (e.g., Leslie, Jordan), predict class label 0. If the input term is a non-name term clearly related to females (e.g., "sorority", "girl", "gynecological"), predict class label 1. Otherwise, predict class label 0.

4590
4591
4592
4593
4594
4595
4596
4597
4598
4599
4600
4601
4602
4603
4604
4605
4606
4607
4608
4609
4610
4611
4612
4613
4614
4615
4616
4617
4618
4619
4620
4621
4622
4623
4624
4625
4626
4627
4628
4629
4630
4631
4632
4633
4634
4635
4636
4637
4638
4639
4640
4641
4642
4643

## Training History: Text Classification – Step 5

**Epoch 0 - Step 5**
——————— **Forward Pass Prompt** ———————
You are the model.
**\*\* Model Descriptions: \*\***
You are designed to do binary classification. The input is a term; you need to output the class label, i.e., an integer in the set 0, 1. If the input term ends with a suffix commonly associated with male names (e.g., -son, -er), predict class label 0. If the input term contains keywords related to family relationships (e.g., "father", "mother", "grand", "paternal", "maternal"), predict class label 0. If the input term is a female name (e.g., Erma, Ada, Faye, Lydia, Ida, Jane, Marlene, Marissa, Adeline, Anne, Maria, Natasha, Veronica, Dawn, Mamie, Monique, Eula), predict class label 1. If the input term is a name that does not clearly indicate gender (e.g., Leslie, Jordan), predict class label 0. If the input term is a non-name term clearly related to females (e.g., "sorority", "girl", "gynecological"), predict class label 1. Otherwise, predict class label 0.
**\*\* Input: \*\***
Danny
Please give your output strictly in the following format:
```
Explanations: [Your step-by-step analyses and results]
Output:
[ONLY the integer class label; make necessary assumptions if needed]
```
Please ONLY reply according to this format, don't give me any other words.
——————— **Forward Pass Output** ———————
Explanations: The input term "Betty" is a female name.
Output: 1

**Many outputs for the rest data points ...**

——————— **Forward Pass Output** ———————
Explanations: The input term is "brethren". It does not end with a suffix commonly associated with male names, nor does it contain keywords related to family relationships. It is not a female name, and it does not clearly indicate gender. However, it is a term related to a group of males (brothers), so it is more associated with males than females.
Output: 0
——————— **Training Accuracy** ———————
1.0
——————— **Optimization Prompt** ———————
You are the optimizer for a model, your goal is to learn the best descriptions for the model. The model used the Current Model Descriptions below predicted the class labels for the given inputs. You are given the target labels, please optimize the Model Descriptions for better prediction.
**\*\* Inputs (a batch of i.i.d. text): \*\***
['Danny', 'Betty', 'Calvin', 'Janet', 'Allison', 'missis', 'Brandon', 'Carl', 'Ben', 'brethren']
**\*\* Current Model Descriptions: \*\***
You are designed to do binary classification. The input is a term; you need to output the class label, i.e., an integer in the set 0, 1. If the input term ends with a suffix commonly associated with male names (e.g., -son, -er), predict class label 0. If the input term contains keywords related to family relationships (e.g., "father", "mother", "grand", "paternal", "maternal"), predict class label 0. If the input term is a female name (e.g., Erma, Ada, Faye, Lydia, Ida, Jane, Marlene, Marissa, Adeline, Anne, Maria, Natasha, Veronica, Dawn, Mamie, Monique, Eula), predict class label 1. If the input term is a name that does not clearly indicate gender (e.g., Leslie, Jordan), predict class label 0. If the input term is a non-name term clearly related to females (e.g., "sorority", "girl", "gynecological"), predict class label 1. Otherwise, predict class label 0.
**\*\* The model predictions: \*\***
[[0] [1] [0] [1] [1] [1] [0] [0] [0] [0]]
**\*\* The targets: \*\***
[[0] [1] [0] [1] [1] [1] [0] [0] [0] [0]]
If the model is doing well, you can keep using the current descriptions. However, if the model is not performing well, please update the model by improving the 'New Model Descriptions', which should have lower classification error both on the current and the next batch of i.i.d. data. If previous 'Optimization Step' are provided, you can use the information from your last optimization step if it's helpful. Please think step by step and give your outputs strictly in the following format:
```
Reasoning:
[be explicit and verbose, improve the Current Model Descriptions by yourself; please show your work; note that you don't have access to computers]
New Model Descriptions:
[put your new decision rules here]
```
Please ONLY reply according to this format, don't give me any other words.
——————— **Optimization Output** ———————
Reasoning:
Upon examining the current model predictions and target labels, I notice that the model is performing well, with all predictions matching the target labels. The current descriptions are comprehensive and accurate, covering various cases such as male and female names, non-name terms related to females or males, and names that do not clearly indicate gender. However, I would like to update the female names list to include the new female names encountered in this batch, which are 'Betty', 'Janet', and 'Allison'. Additionally, I notice that the term 'missis' is a non-name term related to females, so I will add it to the list of non-name terms. Since the term 'brethren' is a non-name term related to males, I will add it to the list of non-name terms as well.
New Model Descriptions:
If the input term ends with a suffix commonly associated with male names (e.g., -son, -er), predict class label 0. If the input term contains keywords related to family relationships (e.g., "father", "mother", "grand", "paternal", "maternal"), predict class label 0. If the input term is a female name (e.g., Erma, Ada, Faye, Lydia, Ida, Jane, Marlene, Marissa, Adeline, Anne, Maria, Natasha, Veronica, Dawn, Mamie, Monique, Eula), predict class label 1. If the input term is a name that does not clearly indicate gender (e.g., Leslie, Jordan), predict class label 0. If the input term is a non-name term clearly related to females (e.g., "sorority", "girl", "gynecological", "missis"), predict class label 1. If the input term is a non-name term clearly related to males (e.g., "brethren"), predict class label 0. Otherwise, predict class label 0.
——————— **Updated Theta** ———————
If the input term ends with a suffix commonly associated with male names (e.g., -son, -er), predict class label 0. If the input term contains keywords related to family relationships (e.g., "father", "mother", "grand", "paternal", "maternal"), predict class label 0. If the input term is a female name (e.g., Erma, Ada, Faye, Lydia, Ida, Jane, Marlene, Marissa, Adeline, Anne, Maria, Natasha, Veronica, Dawn, Mamie, Monique, Eula, Betty, Janet, Allison), predict class label 1. If the input term is a name that does not clearly indicate gender (e.g., Leslie, Jordan), predict class label 0. If the input term is a non-name term clearly related to females (e.g., "sorority", "girl", "gynecological", "missis"), predict class label 1. If the input term is a non-name term clearly related to males (e.g., "brethren"), predict class label 0. Otherwise, predict class label 0.

## L.8    MEDICAL IMAGE CLASSIFICATION (GPT-4O WITH PRIOR)

---

**Training History: Medical Image Classification (with Prior) – Step 1**

**Epoch 0 - Step 1**
———————— **Forward Pass Prompt** ————————
You are the model.
**\*\* Model Descriptions: \*\***
You are designed to do binary classification. The input is an image; you need to output the class label, i.e., an integer in the set 0, 1. The input is X-ray image for identifying pneumonia.
**\*\* Input: \*\***
See the given image.
Please give your output strictly in the following format:
```
Explanations: [Your step-by-step analyses and results]
Output:
[ONLY the integer class label; make necessary assumptions if needed]
```
Please ONLY reply according to this format, don't give me any other words.
———————— **Forward Pass Output** ————————
Explanations:
The X-ray image shows signs of lung opacity and consolidation, which are indicative of pneumonia. The presence of these features suggests an abnormality consistent with pneumonia.
Output: 1
———————— **Forward Pass Output** ————————
Explanations:
The X-ray image shows signs of lung opacity and consolidation, which are indicative of pneumonia. The presence of these abnormalities suggests an infection in the lungs.
Output: 1

**Many outputs for the rest data points ...**

———————— **Forward Pass Output** ————————
Explanations:
The X-ray image shows the chest area. I analyzed the lung fields for any signs of opacity, consolidation, or other abnormalities that are indicative of pneumonia. The image appears to have clear lung fields without significant opacities or consolidations.
Output: 0
———————— **Optimization Prompt** ————————
You are the optimizer for a model, your goal is to learn the best descriptions for the model. The model used the Current Model Descriptions below predicted the class labels for the given images. You are given the target labels, please optimize the Model Descriptions for better prediction.
**\*\* Inputs (a batch of i.i.d. images): \*\***
See the given images.
**\*\* Current Model Descriptions: \*\***
You are designed to do binary classification. The input is an image; you need to output the class label, i.e., an integer in the set 0, 1. The input is X-ray image for identifying pneumonia.
**\*\* The model predictions: \*\***
[[1] [1] [0] [1] [1] [0] [1] [1] [1] [1]]
**\*\* The targets: \*\***
[[0] [0] [0] [1] [0] [0] [1] [0] [0] [0]]
If the model is doing well, you can keep using the current descriptions. However, if the model is not performing well, please update the model by improving upon the 'Current Model Descriptions', which should results in lower classification error on both the current and the next batch of i.i.d. data. If previous 'Optimization Step' are provided, you can use the information from your last optimization step if it's helpful. Limit your 'New Model Descriptions' to be less than 100 words! Please think step by step and give your outputs strictly in the following format:
```
Reasoning:
[be explicit and verbose, update the Current Model Descriptions by yourself; please show your work and use the features in the images; note that you don't have access to computers]
New Model Descriptions:
[put your new decision rules here, the rules MUST based on the features in the input image; make sure it is clear when to output which class labels]
```
Please ONLY reply according to this format, don't give me any other words.
———————— **Optimization Output** ————————
Reasoning:
The current model descriptions fail to adequately differentiate between pneumonia and non-pneumonia cases leading to incorrect predictions. Upon analyzing the images, I observed key indicators such as lung opacity, consolidations, and the presence of fluid. Enhanced model descriptions should include these visual features for improved accuracy.
New Model Descriptions:
The input is an X-ray image. To classify:
- Output 1 (Pneumonia) if lung opacity or consolidations are present, or if there is fluid in the lungs.
- Output 0 (No Pneumonia) if lungs appear clear without significant opacity or fluid presence.
———————— **Updated Theta** ————————
The input is an X-ray image. To classify:
- Output 1 (Pneumonia) if lung opacity or consolidations are present, or if there is fluid in the lungs.
- Output 0 (No Pneumonia) if lungs appear clear without significant opacity or fluid presence.

**Training History: Medical Image Classification (with Prior) – Step 2**

**Epoch 0 - Step 2**

─────────── **Forward Pass Prompt** ───────────

You are the model.

**\*\* Model Descriptions: \*\***

The input is an X-ray image. To classify:
- Output 1 (Pneumonia) if lung opacity or consolidations are present, or if there is fluid in the lungs.
- Output 0 (No Pneumonia) if lungs appear clear without significant opacity or fluid presence.

**\*\* Input: \*\***

See the given image.

Please give your output strictly in the following format:
```
Explanations: [Your step-by-step analyses and results]
Output:
[ONLY the integer class label; make necessary assumptions if needed]
```

Please ONLY reply according to this format, don't give me any other words.

─────────── **Forward Pass Output** ───────────

Explanations:

The X-ray image shows clear lung fields without significant opacity or fluid presence. There are no visible consolidations or signs of lung opacity.

Output: 0

─────────── **Forward Pass Output** ───────────

Explanations:

The X-ray image shows clear lung fields without significant opacity or fluid presence. There are no visible consolidations or signs of lung opacity.

Output: 0

**Many outputs for the rest data points ...**

─────────── **Forward Pass Output** ───────────

Explanations:

The X-ray image shows significant lung opacity and consolidations, indicating the presence of fluid in the lungs.

Output: 1

─────────── **Optimization Prompt** ───────────

You are the optimizer for a model, your goal is to learn the best descriptions for the model. The model used the Current Model Descriptions below predicted the class labels for the given images. You are given the target labels, please optimize the Model Descriptions for better prediction.

**\*\* Inputs (a batch of i.i.d. images): \*\***

See the given images.

**\*\* Current Model Descriptions: \*\***

The input is an X-ray image. To classify:
- Output 1 (Pneumonia) if lung opacity or consolidations are present, or if there is fluid in the lungs.
- Output 0 (No Pneumonia) if lungs appear clear without significant opacity or fluid presence.

**\*\* The model predictions: \*\***

[[1] [1] [0] [0] [1] [0] [1] [1] [1] [1]]

**\*\* The targets: \*\***

[[1] [1] [0] [0] [0] [0] [1] [1] [1] [1]]

If the model is doing well, you can keep using the current descriptions. However, if the model is not performing well, please update the model by improving upon the 'Current Model Descriptions', which should results in lower classification error on both the current and the next batch of i.i.d. data. If previous 'Optimization Step' are provided, you can use the information from your last optimization step if it's helpful. Limit your 'New Model Descriptions' to be less than 100 words! Please think step by step and give your outputs strictly in the following format:
```
Reasoning:
[be explicit and verbose, update the Current Model Descriptions by yourself; please show your work and use the features in the images; note that you don't have access to computers]
New Model Descriptions:
[put your new decision rules here, the rules MUST based on the features in the input image; make sure it is clear when to output which class labels]
```

Please ONLY reply according to this format, don't give me any other words.

─────────── **Optimization Output** ───────────

Reasoning:

The current model descriptions are still leading to some misclassifications, particularly it underestimates cases that show very mild or early signs of pneumonia. Addition of more granulated signs of pneumonia such as patchy infiltrates, mild interstitial markings or signs of hyperinflation can improve the performance. Also noting bilateral versus unilateral presentations can be a useful signal.

New Model Descriptions:

The input is an X-ray image. To classify:
- Output 1 (Pneumonia) if lung opacity, consolidations, fluid, bilateral infiltrates, or mild interstitial markings are present.
- Output 0 (No Pneumonia) if lungs appear clear without significant opacity, fluid presence, or other mentioned indicators of pneumonia.

─────────── **Updated Theta** ───────────

The input is an X-ray image. To classify:
- Output 1 (Pneumonia) if lung opacity, consolidations, fluid, bilateral infiltrates, or mild interstitial markings are present.
- Output 0 (No Pneumonia) if lungs appear clear without significant opacity, fluid presence, or other mentioned indicators of pneumonia.

**Training History: Medical Image Classification (with Prior) – Step 3**

**Epoch 0 - Step 3**
———————— **Forward Pass Prompt** ————————
You are the model.
** **Model Descriptions: **
The input is an X-ray image. To classify:
- Output 1 (Pneumonia) if lung opacity, consolidations, fluid, bilateral infiltrates, or mild interstitial markings are present.
- Output 0 (No Pneumonia) if lungs appear clear without significant opacity, fluid presence, or other mentioned indicators of pneumonia.
** **Input: **
See the given image.
Please give your output strictly in the following format:
```
Explanations: [Your step-by-step analyses and results]
Output:
[ONLY the integer class label; make necessary assumptions if needed]
```
Please ONLY reply according to this format, don't give me any other words.
———————— **Forward Pass Output** ————————
Explanations: The X-ray image shows clear lung fields without significant opacity, consolidations, fluid, bilateral infiltrates, or mild interstitial markings.
Output: 0
———————— **Forward Pass Output** ————————
Explanations: The X-ray image shows significant lung opacity and consolidations in both lungs. There are also signs of fluid presence and bilateral infiltrates.
Output: 1

**Many outputs for the rest data points ...**

———————— **Forward Pass Output** ————————
Explanations: The X-ray image shows significant lung opacity and consolidations. There are also indications of fluid presence and bilateral infiltrates.
Output: 1
———————— **Optimization Prompt** ————————
You are the optimizer for a model, your goal is to learn the best descriptions for the model. The model used the Current Model Descriptions below predicted the class labels for the given images. You are given the target labels, please optimize the Model Descriptions for better prediction.
** **Inputs (a batch of i.i.d. images): **
See the given images.
** **Current Model Descriptions: **
The input is an X-ray image. To classify:
- Output 1 (Pneumonia) if lung opacity, consolidations, fluid, bilateral infiltrates, or mild interstitial markings are present.
- Output 0 (No Pneumonia) if lungs appear clear without significant opacity, fluid presence, or other mentioned indicators of pneumonia.
** **The model predictions: **
[[1] [1] [1] [0] [1] [1] [1] [1] [0] [1]]
** **The targets: **
[[1] [1] [0] [0] [0] [0] [0] [1] [0] [1]]
If the model is doing well, you can keep using the current descriptions. However, if the model is not performing well, please update the model by improving upon the 'Current Model Descriptions', which should results in lower classification error on both the current and the next batch of i.i.d. data. If previous 'Optimization Step' are provided, you can use the information from your last optimization step if it's helpful. Limit your 'New Model Descriptions' to be less than 100 words! Please think step by step and give your outputs strictly in the following format:
```
Reasoning:
[be explicit and verbose, update the Current Model Descriptions by yourself; please show your work and use the features in the images; note that you don't have access to computers]
New Model Descriptions:
[put your new decision rules here, the rules MUST based on the features in the input image; make sure it is clear when to output which class labels] ```
Please ONLY reply according to this format, don't give me any other words.
———————— **Optimization Output** ————————
Reasoning:
The current descriptions are leading to over-identification of pneumonia (more false positives). Incorporating the presence of specific signs and ruling out other conditions can be helpful. Adding features like the presence of pleural effusion, absence of heart enlargement, and noting whether the opacity is localized (can be other pathologies) may help. Precision in observing these smaller details will aid the classification accuracy.
New Model Descriptions:
The input is an X-ray image. To classify:
- Output 1 (Pneumonia) if lung opacity, consolidations, fluid, pleural effusion, bilateral infiltrates, or mild interstitial markings are present and not associated with heart enlargement.
- Output 0 (No Pneumonia) if lungs appear clear without significant opacity, fluid presence, pleural effusion, or other mentioned indicators of pneumonia, or if opacities are localized and not typical for pneumonia.
———————— **Updated Theta** ————————
The input is an X-ray image. To classify:
- Output 1 (Pneumonia) if lung opacity, consolidations, fluid, pleural effusion, bilateral infiltrates, or mild interstitial markings are present and not associated with heart enlargement.
- Output 0 (No Pneumonia) if lungs appear clear without significant opacity, fluid presence, pleural effusion, or other mentioned indicators of pneumonia, or if opacities are localized and not typical for pneumonia.

## Training History: Medical Image Classification (with Prior) – Step 4

**Epoch 0 - Step 4**

——————— **Forward Pass Prompt** ———————

You are the model.

**\*\* Model Descriptions: \*\***

The input is an X-ray image. To classify:
- Output 1 (Pneumonia) if lung opacity, consolidations, fluid, pleural effusion, bilateral infiltrates, or mild interstitial markings are present and not associated with heart enlargement.
- Output 0 (No Pneumonia) if lungs appear clear without significant opacity, fluid presence, pleural effusion, or other mentioned indicators of pneumonia, or if opacities are localized and not typical for pneumonia.

**\*\* Input: \*\***

See the given image.

Please give your output strictly in the following format:
```
Explanations: [Your step-by-step analyses and results]
Output:
[ONLY the integer class label; make necessary assumptions if needed]
```
Please ONLY reply according to this format, don't give me any other words.

——————— **Forward Pass Output** ———————

Explanations: The X-ray image shows significant lung opacity and consolidations in both lungs. There is no clear indication of heart enlargement. The presence of these features is consistent with pneumonia.
Output: 1

——————— **Forward Pass Output** ———————

Explanations: The X-ray image shows significant lung opacity and consolidations. There are also signs of fluid presence and bilateral infiltrates. These findings are consistent with pneumonia and are not associated with heart enlargement.
Output: 1

**Many outputs for the rest data points ...**

——————— **Forward Pass Output** ———————

Explanations: The X-ray image shows clear lungs without significant opacity, fluid presence, pleural effusion, or other indicators of pneumonia. There are no visible lung opacities, consolidations, or bilateral infiltrates.
Output: 0

——————— **Optimization Prompt** ———————

You are the optimizer for a model, your goal is to learn the best descriptions for the model. The model used the Current Model Descriptions below predicted the class labels for the given images. You are given the target labels, please optimize the Model Descriptions for better prediction.

**\*\* Inputs (a batch of i.i.d. images): \*\***

See the given images.

**\*\* Current Model Descriptions: \*\***

The input is an X-ray image. To classify:
- Output 1 (Pneumonia) if lung opacity, consolidations, fluid, pleural effusion, bilateral infiltrates, or mild interstitial markings are present and not associated with heart enlargement.
- Output 0 (No Pneumonia) if lungs appear clear without significant opacity, fluid presence, pleural effusion, or other mentioned indicators of pneumonia, or if opacities are localized and not typical for pneumonia.

**\*\* The model predictions: \*\***

[[0] [0] [1] [1] [1] [1] [1] [1] [1] [0]]

**\*\* The targets: \*\***

[[0] [0] [0] [0] [1] [1] [1] [0] [1] [0]]

If the model is doing well, you can keep using the current descriptions. However, if the model is not performing well, please update the model by improving upon the 'Current Model Descriptions', which should results in lower classification error on both the current and the next batch of i.i.d. data. If previous 'Optimization Step' are provided, you can use the information from your last optimization step if it's helpful. Limit your 'New Model Descriptions' to be less than 100 words! Please think step by step and give your outputs strictly in the following format:
```
Reasoning:
[be explicit and verbose, update the Current Model Descriptions by yourself; please show your work and use the features in the images; note that you don't have access to computers]
New Model Descriptions:
[put your new decision rules here, the rules MUST based on the features in the input image; make sure it is clear when to output which class labels]
```
Please ONLY reply according to this format, don't give me any other words.

——————— **Optimization Output** ———————

Reasoning:
The model still produces false positives. To refine the criteria, focus on more specific patterns indicative of pneumonia like lobar consolidation or "ground-glass" opacities. Exclude cases where opacities are likely due to external factors such as overlying soft tissue shadows or slight variations that are non-diagnostic.
New Model Descriptions:
The input is an X-ray image. To classify:
- Output 1 (Pneumonia) if clear indications of lobar consolidation, ground-glass opacities, fluid presence (excluding pleural effusion), bilateral infiltrates, or moderate interstitial markings are present and not related to heart enlargement.
- Output 0 (No Pneumonia) if lung fields appear clear without significant opacity, fluid, bilateral infiltrates, or other pneumonia indicators, or if features suggest alternative explanations (e.g., overlying soft tissue shadows).

——————— **Updated Theta** ———————

The input is an X-ray image. To classify:
- Output 1 (Pneumonia) if clear indications of lobar consolidation, ground-glass opacities, fluid presence (excluding pleural effusion), bilateral infiltrates, or moderate interstitial markings are present and not related to heart enlargement.
- Output 0 (No Pneumonia) if lung fields appear clear without significant opacity, fluid, bilateral infiltrates, or other pneumonia indicators, or if features suggest alternative explanations (e.g., overlying soft tissue shadows).

## Training History: Medical Image Classification (with Prior) – Step 5

**Epoch 0 - Step 5**
——————— Forward Pass Prompt ———————
You are the model.
** Model Descriptions: **
The input is an X-ray image. To classify:
- Output 1 (Pneumonia) if clear indications of lobar consolidation, ground-glass opacities, fluid presence (excluding pleural effusion), bilateral infiltrates, or moderate interstitial markings are present and not related to heart enlargement.
- Output 0 (No Pneumonia) if lung fields appear clear without significant opacity, fluid, bilateral infiltrates, or other pneumonia indicators, or if features suggest alternative explanations (e.g., overlying soft tissue shadows).
** Input: **
See the given image.
Please give your output strictly in the following format:
```
Explanations: [Your step-by-step analyses and results]
Output:
[ONLY the integer class label; make necessary assumptions if needed]
```
Please ONLY reply according to this format, don't give me any other words.
——————— Forward Pass Output ———————
Explanations: The X-ray image shows clear indications of bilateral infiltrates and moderate interstitial markings. There are no signs of heart enlargement that could explain these features.
Output: 1
——————— Forward Pass Output ———————
Explanations: The X-ray image shows clear lung fields without significant opacity, fluid presence, bilateral infiltrates, or other indicators of pneumonia. There are no clear signs of lobar consolidation, ground-glass opacities, or moderate interstitial markings.
Output: 0

**Many outputs for the rest data points ...**

——————— Forward Pass Output ———————
Explanations: The X-ray image shows clear indications of lobar consolidation and ground-glass opacities in the lung fields. These findings are consistent with pneumonia.
Output: 1
——————— Optimization Prompt ———————
You are the optimizer for a model, your goal is to learn the best descriptions for the model. The model used the Current Model Descriptions below predicted the class labels for the given images. You are given the target labels, please optimize the Model Descriptions for better prediction.
** Inputs (a batch of i.i.d. images): **
See the given images.
** Current Model Descriptions: **
The input is an X-ray image. To classify:
- Output 1 (Pneumonia) if clear indications of lobar consolidation, ground-glass opacities, fluid presence (excluding pleural effusion), bilateral infiltrates, or moderate interstitial markings are present and not related to heart enlargement.
- Output 0 (No Pneumonia) if lung fields appear clear without significant opacity, fluid, bilateral infiltrates, or other pneumonia indicators, or if features suggest alternative explanations (e.g., overlying soft tissue shadows).
** The model predictions: **
[[0] [0] [1] [0] [1] [0] [0] [0] [1] [1]]
** The targets: **
[[1] [0] [1] [0] [1] [0] [0] [0] [1] [1]]
If the model is doing well, you can keep using the current descriptions. However, if the model is not performing well, please update the model by improving upon the 'Current Model Descriptions', which should results in lower classification error on both the current and the next batch of i.i.d. data. If previous 'Optimization Step' are provided, you can use the information from your last optimization step if it's helpful. Limit your 'New Model Descriptions' to be less than 100 words! Please think step by step and give your outputs strictly in the following format:
```
Reasoning:
[be explicit and verbose, update the Current Model Descriptions by yourself; please show your work and use the features in the images; note that you don't have access to computers]
New Model Descriptions:
[put your new decision rules here, the rules MUST based on the features in the input image; make sure it is clear when to output which class labels]
```
Please ONLY reply according to this format, don't give me any other words.
——————— Optimization Output ———————
Reasoning: The current model descriptions are missing some complexities encountered in the images. We observe some misclassifications indicating the need to differentiate between the types of opacities more clearly. Including signs such as air bronchograms and the pattern of unilateral versus bilateral presentation can aid in improving the description.
New Model Descriptions:
The input is an X-ray image. To classify:
- Output 1 (Pneumonia) if there are signs of lobar consolidation, ground-glass opacities, air bronchograms, fluid presence (excluding pleural effusion), bilateral infiltrates, or moderate interstitial markings.
- Output 0 (No Pneumonia) if lung fields appear clear without significant opacity or fluid, if opacities are limited to a single lung and are not of the typical pneumonia pattern, or if these indicators are due to overlying soft tissue shadows.
——————— Updated Theta ———————
The input is an X-ray image. To classify:
- Output 1 (Pneumonia) if there are signs of lobar consolidation, ground-glass opacities, air bronchograms, fluid presence (excluding pleural effusion), bilateral infiltrates, or moderate interstitial markings.
- Output 0 (No Pneumonia) if lung fields appear clear without significant opacity or fluid, if opacities are limited to a single lung and are not of the typical pneumonia pattern, or if these indicators are due to overlying soft tissue shadows.

## Training History: Medical Image Classification (with Prior) – Step 50

**Epoch 4 - Step 10**
———————— **Forward Pass Prompt** ————————
You are the model.
**\*\* Model Descriptions: \*\***
The input is an X-ray image. To classify:
- Output 1 (Pneumonia) if there are lobar consolidations, asymmetric or bilateral ground-glass opacities, air bronchograms, moderate to significant alveolar fluid, asymmetric infiltrates, patchy inflammations, moderate to severe pleural effusion, central or peripheral acute opacities, air-fluid levels, increasing opacities, pneumatocele formations, rapidly changing patterns in chronic lesions indicative of acute infection, or persistent cough-based findings.
- Output 0 (No Pneumonia) if lung fields are clear, minimal localized fluid, no distinct acute opacities, stable fibrotic scarring, solitary calcifications, old healed lesions, stable localized non-infectious anomalies, chronic interstitial fibrosis, chronic reticulonodular patterns without recent changes, stable chronic lesions without acute signs, or purely non-persistent anomalies.
**\*\* Input: \*\***
See the given image.
Please give your output strictly in the following format:
```
Explanations: [Your step-by-step analyses and results]
Output:
[ONLY the integer class label; make necessary assumptions if needed]
```
Please ONLY reply according to this format, don't give me any other words.
———————— **Forward Pass Output** ————————
Explanations: The X-ray image shows lobar consolidations and asymmetric ground-glass opacities. There are also visible air bronchograms and moderate alveolar fluid. These findings are indicative of pneumonia.
Output: 1
———————— **Forward Pass Output** ————————
Explanations: The X-ray image shows clear lung fields without any significant opacities, consolidations, or fluid accumulations. There are no signs of air bronchograms, pleural effusion, or other acute changes indicative of pneumonia.
Output: 0

**Many outputs for the rest data points ...**

———————— **Forward Pass Output** ————————
Explanations: The X-ray image shows lobar consolidations and asymmetric ground-glass opacities. There are also visible air bronchograms and moderate alveolar fluid. These findings are indicative of pneumonia.
Output: 1
———————— **Optimization Prompt** ————————
You are the optimizer for a model, your goal is to learn the best descriptions for the model. The model used the Current Model Descriptions below predicted the class labels for the given images. You are given the target labels, please optimize the Model Descriptions for better prediction.
**\*\* Inputs (a batch of i.i.d. images): \*\***
See the given images.
**\*\* Current Model Descriptions: \*\***
The input is an X-ray image. To classify:
- Output 1 (Pneumonia) if there are lobar consolidations, asymmetric or bilateral ground-glass opacities, air bronchograms, moderate to significant alveolar fluid, asymmetric infiltrates, patchy inflammations, moderate to severe pleural effusion, central or peripheral acute opacities, air-fluid levels, increasing opacities, pneumatocele formations, rapidly changing patterns in chronic lesions indicative of acute infection, or persistent cough-based findings.
- Output 0 (No Pneumonia) if lung fields are clear, minimal localized fluid, no distinct acute opacities, stable fibrotic scarring, solitary calcifications, old healed lesions, stable localized non-infectious anomalies, chronic interstitial fibrosis, chronic reticulonodular patterns without recent changes, stable chronic lesions without acute signs, or purely non-persistent anomalies.
**\*\* The model predictions: \*\***
[[1] [0] [0] [0] [1] [1] [1] [0] [0] [1]]
**\*\* The targets: \*\***
[[1] [0] [0] [1] [0] [1] [0] [0] [1] [1]]
If the model is doing well, you can keep using the current descriptions. However, if the model is not performing well, please update the model by improving upon the 'Current Model Descriptions', which should results in lower classification error on both the current and the next batch of i.i.d. data. If previous 'Optimization Step' are provided, you can use the information from your last optimization step if it's helpful. Limit your 'New Model Descriptions' to be less than 100 words! Please think step by step and give your outputs strictly in the following format:
```
Reasoning:
[be explicit and verbose, update the Current Model Descriptions by yourself; please show your work and use the features in the images; note that you don't have access to computers]
New Model Descriptions:
[put your new decision rules here, the rules MUST based on the features in the input image; make sure it is clear when to output which class labels]
```
Please ONLY reply according to this format, don't give me any other words.

**Training History: Medical Image Classification (with Prior) – Step 50 (continued)**

——————— **Optimization Output** ———————

Reasoning:

The model missed some true cases of pneumonia and incorrectly identified some non-pneumonia cases. To improve the accuracy, refine the criteria to better identify pneumonia even in less obvious cases and differentiate between chronic stable conditions and acute infections.

New Model Descriptions:

The input is an X-ray image. To classify:

- Output 1 (Pneumonia) if there are lobar consolidations, asymmetric or bilateral ground-glass opacities, air bronchograms, moderate to significant alveolar fluid, asymmetric infiltrates, patchy inflammations, moderate to severe pleural effusion, central or peripheral acute opacities, air-fluid levels, increasing opacities, pneumatocele formations, rapidly changing patterns in chronic lesions indicative of acute infection, persistent cough-related findings, or recent onset of ground-glass opacities.

- Output 0 (No Pneumonia) if lung fields are clear, minimal localized fluid, no distinct acute opacities, stable fibrotic scarring, solitary calcifications, old healed lesions, stable localized non-infectious anomalies, chronic interstitial fibrosis, chronic reticulonodular patterns without recent changes, stable chronic lesions without acute signs, or absence of any acute infection markers.

——————— **Updated Theta** ———————

The input is an X-ray image. To classify:

- Output 1 (Pneumonia) if there are lobar consolidations, asymmetric or bilateral ground-glass opacities, air bronchograms, moderate to significant alveolar fluid, asymmetric infiltrates, patchy inflammations, moderate to severe pleural effusion, central or peripheral acute opacities, air-fluid levels, increasing opacities, pneumatocele formations, rapidly changing patterns in chronic lesions indicative of acute infection, persistent cough-related findings, or recent onset of ground-glass opacities.

- Output 0 (No Pneumonia) if lung fields are clear, minimal localized fluid, no distinct acute opacities, stable fibrotic scarring, solitary calcifications, old healed lesions, stable localized non-infectious anomalies, chronic interstitial fibrosis, chronic reticulonodular patterns without recent changes, stable chronic lesions without acute signs, or absence of any acute infection markers.

## L.9 MEDICAL IMAGE CLASSIFICATION (GPT-4O WITHOUT PRIOR)

---

**Training History: Medical Image Classification (without Prior) – Step 1**

**Epoch 0 - Step 1**
——————— **Forward Pass Prompt** ———————
You are the model.
**\*\* Model Descriptions: \*\***
You are designed to do binary classification. The input is an image; you need to output the class label, i.e., an integer in the set 0, 1.
**\*\* Input: \*\***
See the given image.
Please give your output strictly in the following format:
```
Explanations: [Your step-by-step analyses and results]
Output:
[ONLY the integer class label; make necessary assumptions if needed]
```
Please ONLY reply according to this format, don't give me any other words.
——————— **Forward Pass Output** ———————
Explanations: The image appears to be a chest X-ray. I will analyze the image for any abnormalities or signs of disease that could indicate the class label.
Output: 1

**Many outputs for the rest data points ...**

——————— **Forward Pass Output** ———————
Explanations: The image appears to be a medical X-ray of a chest. The analysis involves examining the features and patterns in the image to determine the class label. Based on the visual inspection, the image shows certain characteristics that align with class 1.
Output: 1
——————— **Optimization Prompt** ———————
You are the optimizer for a model, your goal is to learn the best descriptions for the model. The model used the Current Model Descriptions below predicted the class labels for the given images. You are given the target labels, please optimize the Model Descriptions for better prediction.
**\*\* Inputs (a batch of i.i.d. images): \*\***
See the given images.
**\*\* Current Model Descriptions: \*\***
You are designed to do binary classification. The input is an image; you need to output the class label, i.e., an integer in the set 0, 1.
**\*\* The model predictions: \*\***
[[0] [0] [1] [1] [1] [0] [0] [1] [1] [1]]
**\*\* The targets: \*\***
[[0] [0] [0] [1] [0] [0] [1] [0] [0] [0]]
If the model is doing well, you can keep using the current descriptions. However, if the model is not performing well, please update the model by improving upon the 'Current Model Descriptions', which should results in lower classification error on both the current and the next batch of i.i.d. data. If previous 'Optimization Step' are provided, you can use the information from your last optimization step if it's helpful. Limit your 'New Model Descriptions' to be less than 100 words! Please think step by step and give your outputs strictly in the following format:
```
Reasoning:
[be explicit and verbose, update the Current Model Descriptions by yourself; please show your work and use the features in the images; note that you don't have access to computers]
New Model Descriptions:
[put your new decision rules here, the rules MUST based on the features in the input image; make sure it is clear when to output which class labels]
```
Please ONLY reply according to this format, don't give me any other words.
——————— **Optimization Output** ———————
Reasoning:
The current model's performance indicates significant misclassification. To improve, I must explicitly identify and use key visual features in the images that are indicative of the target classes. For instance, if the classification relates to medical diagnostics (e.g., chest X-rays for pneumonia), specific features like the presence of fluid, opacity, or other anomalies must be emphasized. Similarly, non-medical classifications (typical objects versus atypical ones) could require identification of distinct shapes, colors, or textural features. Analyzing the given images visually will guide the addition of pertinent descriptive rules that improve classification accuracy.
New Model Descriptions:
1. If the image contains prominent, clearly defined geometric or bodily structures and no unusual opacity or anomalies, output the class label 0.
2. If the image shows abnormal features such as irregular shapes, unexpected opacities, or non-uniform patterns that suggest pathology or anomaly, output the class label 1.
3. When distinguishing medical images specifically, check for fluid accumulation, granular textures, or any signs of abnormality in the expected anatomy; if these are present, output the class label 1.
——————— **Updated Theta** ———————
1. If the image contains prominent, clearly defined geometric or bodily structures and no unusual opacity or anomalies, output the class label 0.
2. If the image shows abnormal features such as irregular shapes, unexpected opacities, or non-uniform patterns that suggest pathology or anomaly, output the class label 1.
3. When distinguishing medical images specifically, check for fluid accumulation, granular textures, or any signs of abnormality in the expected anatomy; if these are present, output the class label 1.

---

## Training History: Medical Image Classification (without Prior) – Step 2

**Epoch 0 - Step 2**
———————— Forward Pass Prompt ————————
You are the model.
** Model Descriptions: **
1. If the image contains prominent, clearly defined geometric or bodily structures and no unusual opacity or anomalies, output the class label 0.
2. If the image shows abnormal features such as irregular shapes, unexpected opacities, or non-uniform patterns that suggest pathology or anomaly, output the class label 1.
3. When distinguishing medical images specifically, check for fluid accumulation, granular textures, or any signs of abnormality in the expected anatomy; if these are present, output the class label 1.
** Input: **
See the given image.
Please give your output strictly in the following format:
```
Explanations: [Your step-by-step analyses and results]
Output:
[ONLY the integer class label; make necessary assumptions if needed]
```
Please ONLY reply according to this format, don't give me any other words.
———————— Forward Pass Output ————————
Explanations: The image shows irregular shapes and unexpected opacities. There are non-uniform patterns that suggest an anomaly.
Output: 1

**Many outputs for the rest data points ...**

———————— Forward Pass Output ————————
Explanations: The image appears to be a medical X-ray. There are irregular shapes and unexpected opacities present in the lung area, which suggest pathology or anomaly.
Output: 1
———————— Optimization Prompt ————————
You are the optimizer for a model, your goal is to learn the best descriptions for the model. The model used the Current Model Descriptions below predicted the class labels for the given images. You are given the target labels, please optimize the Model Descriptions for better prediction.
** Inputs (a batch of i.i.d. images): **
See the given images.
** Current Model Descriptions: **
1. If the image contains prominent, clearly defined geometric or bodily structures and no unusual opacity or anomalies, output the class label 0.
2. If the image shows abnormal features such as irregular shapes, unexpected opacities, or non-uniform patterns that suggest pathology or anomaly, output the class label 1.
3. When distinguishing medical images specifically, check for fluid accumulation, granular textures, or any signs of abnormality in the expected anatomy; if these are present, output the class label 1.
** The model predictions: **
[[1] [1] [1] [1] [1] [1] [1] [1] [1] [1]]
** The targets: **
[[1] [1] [0] [0] [0] [1] [1] [1] [1]]
If the model is doing well, you can keep using the current descriptions. However, if the model is not performing well, please update the model by improving upon the 'Current Model Descriptions', which should results in lower classification error on both the current and the next batch of i.i.d. data. If previous 'Optimization Step' are provided, you can use the information from your last optimization step if it's helpful. Limit your 'New Model Descriptions' to be less than 100 words! Please think step by step and give your outputs strictly in the following format:
```
Reasoning:
[be explicit and verbose, update the Current Model Descriptions by yourself; please show your work and use the features in the images; note that you don't have access to computers]
New Model Descriptions:
[put your new decision rules here, the rules MUST based on the features in the input image; make sure it is clear when to output which class labels]
```
Please ONLY reply according to this format, don't give me any other words.
———————— Optimization Output ————————
Reasoning:
The current model is too sensitive to features that are indicating anomalies, leading to many false positives. A refined approach would include more granular analysis to distinguish normal variations from actual abnormalities. Specifically, enhancing the detection criteria to differentiate between minor variations and significant pathological markers is necessary. Reviewing more images, I will refine the descriptions to include clearer definitions of significant anomalies versus natural structures.
New Model Descriptions:
1. If the image contains clear, consistent geometric or anatomical structures without significant opacities, irregular patterns, or signs of fluid, output the class label 0.
2. If the image shows irregular shapes, significant opacity, or patterns that deviate considerably from typical anatomical structures, output the class label 1.
3. Examine for fluid accumulation, granular textures, and other significant deviations in expected anatomy. Minor irregularities that do not suggest severe pathology should output class label 0, otherwise class label 1.
———————— Updated Theta ————————
1. If the image contains clear, consistent geometric or anatomical structures without significant opacities, irregular patterns, or signs of fluid, output the class label 0.
2. If the image shows irregular shapes, significant opacity, or patterns that deviate considerably from typical anatomical structures, output the class label 1.
3. Examine for fluid accumulation, granular textures, and other significant deviations in expected anatomy. Minor irregularities that do not suggest severe pathology should output class label 0, otherwise class label 1.

**Training History: Medical Image Classification (without Prior) – Step 3**

**Epoch 0 - Step 3**
——————— **Forward Pass Prompt** ———————
You are the model.
**\*\* Model Descriptions: \*\***
1. If the image contains clear, consistent geometric or anatomical structures without significant opacities, irregular patterns, or signs of fluid, output the class label 0.
2. If the image shows irregular shapes, significant opacity, or patterns that deviate considerably from typical anatomical structures, output the class label 1.
3. Examine for fluid accumulation, granular textures, and other significant deviations in expected anatomy. Minor irregularities that do not suggest severe pathology should output class label 0, otherwise class label 1.
**\*\* Input: \*\***
See the given image.
Please give your output strictly in the following format:
```
Explanations: [Your step-by-step analyses and results]
Output:
[ONLY the integer class label; make necessary assumptions if needed]
```
Please ONLY reply according to this format, don't give me any other words.
——————— **Forward Pass Output** ———————
Explanations: The image shows significant opacity and irregular patterns that deviate considerably from typical anatomical structures.
Output: 1

**Many outputs for the rest data points ...**

——————— **Forward Pass Output** ———————
Explanations: The image shows clear and consistent geometric and anatomical structures without significant opacities, irregular patterns, or signs of fluid. There are no significant deviations from typical anatomy.
Output: 0
——————— **Optimization Prompt** ———————
You are the optimizer for a model, your goal is to learn the best descriptions for the model. The model used the Current Model Descriptions below predicted the class labels for the given images. You are given the target labels, please optimize the Model Descriptions for better prediction.
**\*\* Inputs (a batch of i.i.d. images): \*\***
See the given images.
**\*\* Current Model Descriptions: \*\***
1. If the image contains clear, consistent geometric or anatomical structures without significant opacities, irregular patterns, or signs of fluid, output the class label 0.
2. If the image shows irregular shapes, significant opacity, or patterns that deviate considerably from typical anatomical structures, output the class label 1.
3. Examine for fluid accumulation, granular textures, and other significant deviations in expected anatomy. Minor irregularities that do not suggest severe pathology should output class label 0, otherwise class label 1.
**\*\* The model predictions: \*\***
[[1] [1] [0] [1] [0] [1] [1] [1] [0] [1]]
**\*\* The targets: \*\***
[[1] [1] [0] [0] [0] [0] [1] [0] [1]]
If the model is doing well, you can keep using the current descriptions. However, if the model is not performing well, please update the model by improving upon the 'Current Model Descriptions', which should results in lower classification error on both the current and the next batch of i.i.d. data. If previous 'Optimization Step' are provided, you can use the information from your last optimization step if it's helpful. Limit your 'New Model Descriptions' to be less than 100 words! Please think step by step and give your outputs strictly in the following format:
```
Reasoning:
[be explicit and verbose, update the Current Model Descriptions by yourself; please show your work and use the features in the images; note that you don't have access to computers]
New Model Descriptions:
[put your new decision rules here, the rules MUST based on the features in the input image; make sure it is clear when to output which class labels]
```
Please ONLY reply according to this format, don't give me any other words.
——————— **Optimization Output** ———————
Reasoning:
The model is still producing numerous false positives. The current descriptions are good at detecting anomalies but not at distinguishing minor, non-critical irregularities from significant pathological features. I need to refine the differentiation by including more specific anatomical markers and clearly delineating between slight structures and substantial irregularities.
New Model Descriptions:
1. If the image contains clear, consistent anatomical structures, free from significant opacities, masses, or signs of severe pathology, output the class label 0.
2. If the image shows substantial opacity, masses, or shapes significantly deviating from typical structures indicating severe pathology, output the class label 1.
3. Examine for clear signs of fluid accumulation, large granular textures, or significant abnormalities in anatomical structures. Small, inconsequential variations should output class label 0, otherwise class label 1.
——————— **Updated Theta** ———————
1. If the image contains clear, consistent anatomical structures, free from significant opacities, masses, or signs of severe pathology, output the class label 0.
2. If the image shows substantial opacity, masses, or shapes significantly deviating from typical structures indicating severe pathology, output the class label 1.
3. Examine for clear signs of fluid accumulation, large granular textures, or significant abnormalities in anatomical structures. Small, inconsequential variations should output class label 0, otherwise class label 1.

## Training History: Medical Image Classification (without Prior) – Step 4

**Epoch 0 - Step 4**
—————— **Forward Pass Prompt** ——————
You are the model.
** Model Descriptions: **
1. If the image contains clear, consistent anatomical structures, free from significant opacities, masses, or signs of severe pathology, output the class label 0.
2. If the image shows substantial opacity, masses, or shapes significantly deviating from typical structures indicating severe pathology, output the class label 1.
3. Examine for clear signs of fluid accumulation, large granular textures, or significant abnormalities in anatomical structures. Small, inconsequential variations should output class label 0, otherwise class label 1.
** Input: **
See the given image.
Please give your output strictly in the following format:
"'
Explanations: [Your step-by-step analyses and results]
Output:
[ONLY the integer class label; make necessary assumptions if needed]
"'
Please ONLY reply according to this format, don't give me any other words.
—————— **Forward Pass Output** ——————
Explanations: The image shows substantial opacity and shapes significantly deviating from typical structures, indicating severe pathology.
Output: 1

**Many outputs for the rest data points ...**

—————— **Forward Pass Output** ——————
Explanations: The image shows clear, consistent anatomical structures without significant opacities, masses, or signs of severe pathology. There are no clear signs of fluid accumulation or large granular textures.
Output: 0
—————— **Optimization Prompt** ——————
You are the optimizer for a model, your goal is to learn the best descriptions for the model. The model used the Current Model Descriptions below predicted the class labels for the given images. You are given the target labels, please optimize the Model Descriptions for better prediction.
** Inputs (a batch of i.i.d. images): **
See the given images.
** Current Model Descriptions: **
1. If the image contains clear, consistent anatomical structures, free from significant opacities, masses, or signs of severe pathology, output the class label 0.
2. If the image shows substantial opacity, masses, or shapes significantly deviating from typical structures indicating severe pathology, output the class label 1.
3. Examine for clear signs of fluid accumulation, large granular textures, or significant abnormalities in anatomical structures. Small, inconsequential variations should output class label 0, otherwise class label 1.
** The model predictions: **
[[0] [0] [1] [0] [1] [1] [1] [1] [0] [0]]
** The targets: **
[[0] [0] [0] [0] [1] [1] [1] [0] [1] [0]]
If the model is doing well, you can keep using the current descriptions. However, if the model is not performing well, please update the model by improving upon the 'Current Model Descriptions', which should results in lower classification error on both the current and the next batch of i.i.d. data. If previous 'Optimization Step' are provided, you can use the information from your last optimization step if it's helpful. Limit your 'New Model Descriptions' to be less than 100 words! Please think step by step and give your outputs strictly in the following format:
"'
Reasoning:
[be explicit and verbose, update the Current Model Descriptions by yourself; please show your work and use the features in the images; note that you don't have access to computers]
New Model Descriptions:
[put your new decision rules here, the rules MUST based on the features in the input image; make sure it is clear when to output which class labels]
"'
Please ONLY reply according to this format, don't give me any other words.
—————— **Optimization Output** ——————
Reasoning:
The model is generating several false positives, indicating it may be too conservative and is mistakenly identifying minor irregularities as significant pathology. The descriptions need to further clarify the distinction between mild and serious irregularities. Adding specific attention to the size, shape, and clear delineation of opacities and anatomical deviations might improve accuracy.
New Model Descriptions:
1. If the image displays clear, consistent anatomical structures without significant opacities, masses, or deviations, output the class label 0.
2. If the image highlights substantial, well-defined opacities or masses and shapes deviating significantly from normal structures indicating severe pathology, output the class label 1.
3. Check for large, well-delineated fluid accumulations, substantial granular textures, or major abnormalities in anatomy. Minor, non-critical variations should output class label 0, otherwise class label 1.
—————— **Updated Theta** ——————
1. If the image displays clear, consistent anatomical structures without significant opacities, masses, or deviations, output the class label 0.
2. If the image highlights substantial, well-defined opacities or masses and shapes deviating significantly from normal structures indicating severe pathology, output the class label 1.
3. Check for large, well-delineated fluid accumulations, substantial granular textures, or major abnormalities in anatomy. Minor, non-critical variations should output class label 0, otherwise class label 1.

## Training History: Medical Image Classification (without Prior) – Step 5

**Epoch 0 - Step 5**
——————— **Forward Pass Prompt** ———————
You are the model.
** Model Descriptions: **
1. If the image displays clear, consistent anatomical structures without significant opacities, masses, or deviations, output the class label 0.
2. If the image highlights substantial, well-defined opacities or masses and shapes deviating significantly from normal structures indicating severe pathology, output the class label 1.
3. Check for large, well-delineated fluid accumulations, substantial granular textures, or major abnormalities in anatomy. Minor, non-critical variations should output class label 0, otherwise class label 1.
** Input: **
See the given image.
Please give your output strictly in the following format:
```
Explanations: [Your step-by-step analyses and results]
Output:
[ONLY the integer class label; make necessary assumptions if needed]
```
Please ONLY reply according to this format, don't give me any other words.
——————— **Forward Pass Output** ———————
Explanations: The image displays clear, consistent anatomical structures without significant opacities, masses, or deviations.
Output: 0

**Many outputs for the rest data points ...**

——————— **Forward Pass Output** ———————
Explanations: The image displays clear, consistent anatomical structures without significant opacities, masses, or deviations. There are no large, well-delineated fluid accumulations, substantial granular textures, or major abnormalities in anatomy.
Output: 0
——————— **Optimization Prompt** ———————
You are the optimizer for a model, your goal is to learn the best descriptions for the model. The model used the Current Model Descriptions below predicted the class labels for the given images. You are given the target labels, please optimize the Model Descriptions for better prediction.
** Inputs (a batch of i.i.d. images): **
See the given images.
** Current Model Descriptions: **
1. If the image displays clear, consistent anatomical structures without significant opacities, masses, or deviations, output the class label 0.
2. If the image highlights substantial, well-defined opacities or masses and shapes deviating significantly from normal structures indicating severe pathology, output the class label 1.
3. Check for large, well-delineated fluid accumulations, substantial granular textures, or major abnormalities in anatomy. Minor, non-critical variations should output class label 0, otherwise class label 1.
** The model predictions: **
[[1] [0] [0] [1] [0] [0] [0] [0] [1]]
** The targets: **
[[1] [0] [1] [0] [1] [0] [0] [0] [1] [1]]
If the model is doing well, you can keep using the current descriptions. However, if the model is not performing well, please update the model by improving upon the 'Current Model Descriptions', which should results in lower classification error on both the current and the next batch of i.i.d. data. If previous 'Optimization Step' are provided, you can use the information from your last optimization step if it's helpful. Limit your 'New Model Descriptions' to be less than 100 words! Please think step by step and give your outputs strictly in the following format:
```
Reasoning:
[be explicit and verbose, update the Current Model Descriptions by yourself; please show your work and use the features in the images; note that you don't have access to computers]
New Model Descriptions:
[put your new decision rules here, the rules MUST based on the features in the input image; make sure it is clear when to output which class labels]
```
Please ONLY reply according to this format, don't give me any other words.
——————— **Optimization Output** ———————
Reasoning:
The model still misclassifies some images, both false positives and false negatives. We need to be more precise while identifying anomalies, focusing on substantial deviations such as the density of opacities or masses and their impact on surrounding structures. Including the context of tissue appearance and location could help.
New Model Descriptions:
1. If the image shows clear, consistent anatomical structures without significant opacities, masses, or deviations, output class label 0.
2. If the image highlights substantial, well-defined opacities or masses and shapes with significant density, deviating significantly from normal structures, output class label 1.
3. Check for large, well-delineated fluid accumulations, substantial granular textures, or major abnormalities in anatomy. Minor, non-critical variations should output class label 0, focusing also on the context and location of tissues, otherwise class label 1.
——————— **Updated Theta** ———————
1. If the image shows clear, consistent anatomical structures without significant opacities, masses, or deviations, output class label 0.
2. If the image highlights substantial, well-defined opacities or masses and shapes with significant density, deviating significantly from normal structures, output class label 1.
3. Check for large, well-delineated fluid accumulations, substantial granular textures, or major abnormalities in anatomy. Minor, non-critical variations should output class label 0, focusing also on the context and location of tissues, otherwise class label 1.

## Training History: Medical Image Classification (without Prior) – Step 50

**Epoch 4 - Step 10**
———————— Forward Pass Prompt ————————
You are the model.
** Model Descriptions: **
1. If the image shows clear anatomical structures, uniform texture, symmetry, and normal lung fields without visible opacities, well-defined masses, streaks, irregular granular patterns, or any deviations, output class label 0.
2. If the image shows any distinct opacities or masses, interstitial patterns, visible fluid accumulations, asymmetry, irregular heart shapes, or multiple small opacities, output class label 1.
3. For granular or streaky opacities over any lung region, indistinct or uneven lung fields, subtle masses, minor deviations in symmetry, or any lung irregularities, output class label 1. For minimal irregularities or slight deviations, output class label 0.
** Input: **
See the given image.
Please give your output strictly in the following format:
```
Explanations: [Your step-by-step analyses and results]
Output:
[ONLY the integer class label; make necessary assumptions if needed]
```
Please ONLY reply according to this format, don't give me any other words.
———————— Forward Pass Output ————————
Explanations: The image shows clear anatomical structures with uniform texture and symmetry. The lung fields appear normal without visible opacities, well-defined masses, streaks, or irregular granular patterns. There are no deviations observed.
Output: 0

**Many outputs for the rest data points ...**

———————— Forward Pass Output ————————
Explanations: The image shows clear anatomical structures with uniform texture and symmetry. The lung fields appear normal without visible opacities, well-defined masses, streaks, or irregular granular patterns. There are no deviations observed.
Output: 0
———————— Optimization Prompt ————————
You are the optimizer for a model, your goal is to learn the best descriptions for the model. The model used the Current Model Descriptions below predicted the class labels for the given images. You are given the target labels, please optimize the Model Descriptions for better prediction.
** Inputs (a batch of i.i.d. images): **
See the given images.
** Current Model Descriptions: **
1. If the image shows clear anatomical structures, uniform texture, symmetry, and normal lung fields without visible opacities, well-defined masses, streaks, irregular granular patterns, or any deviations, output class label 0.
2. If the image shows any distinct opacities or masses, interstitial patterns, visible fluid accumulations, asymmetry, irregular heart shapes, or multiple small opacities, output class label 1.
3. For granular or streaky opacities over any lung region, indistinct or uneven lung fields, subtle masses, minor deviations in symmetry, or any lung irregularities, output class label 1. For minimal irregularities or slight deviations, output class label 0.
** The model predictions: **
[[1] [0] [0] [0] [1] [1] [1] [0] [0] [0]]
** The targets: **
[[1] [0] [0] [1] [0] [1] [0] [0] [1] [1]]
If the model is doing well, you can keep using the current descriptions. However, if the model is not performing well, please update the model by improving upon the 'Current Model Descriptions', which should results in lower classification error on both the current and the next batch of i.i.d. data. If previous 'Optimization Step' are provided, you can use the information from your last optimization step if it's helpful. Limit your 'New Model Descriptions' to be less than 100 words! Please think step by step and give your outputs strictly in the following format:
```
Reasoning:
[be explicit and verbose, update the Current Model Descriptions by yourself; please show your work and use the features in the images; note that you don't have access to computers]
New Model Descriptions:
[put your new decision rules here, the rules MUST based on the features in the input image; make sure it is clear when to output which class labels]
```
Please ONLY reply according to this format, don't give me any other words.

**Training History: Medical Image Classification (without Prior) – Step 50 (continued)**

——————— **Optimization Output** ———————

Reasoning:

The model misclassified several images where minor opacities or very light irregular patterns were present. We need to be more precise in detecting these subtle features while examining the overall lung fields more thoroughly to reduce false negatives for label 1. Specifically, minor irregularities or granular patterns that seem inconsequential should still be marked as potential abnormalities if they occur in specific regions, such as the periphery.

New Model Descriptions:

1. If the image shows clear anatomical structures, uniform texture, symmetry, and normal lung fields without any visible opacities, well-defined masses, streaks, or granular patterns, output class label 0.

2. If the image shows any distinct opacities or masses, interstitial patterns, visible fluid accumulations, asymmetry, irregular heart shapes, or multiple small opacities across the lung fields, output class label 1.

3. For granular or streaky opacities over any lung region, indistinct or uneven lung fields, subtle masses, minor deviations in symmetry, or any lung irregularities or granular patterns, including faint ones, in the periphery, output class label 1. For very minimal irregularities or slight deviations in central regions only, output class label 0.

——————— **Updated Theta** ———————

1. If the image shows clear anatomical structures, uniform texture, symmetry, and normal lung fields without any visible opacities, well-defined masses, streaks, or granular patterns, output class label 0.

2. If the image shows any distinct opacities or masses, interstitial patterns, visible fluid accumulations, asymmetry, irregular heart shapes, or multiple small opacities across the lung fields, output class label 1.

3. For granular or streaky opacities over any lung region, indistinct or uneven lung fields, subtle masses, minor deviations in symmetry, or any lung irregularities or granular patterns, including faint ones, in the periphery, output class label 1. For very minimal irregularities or slight deviations in central regions only, output class label 0.

