# OpenReview forum: "Verbalized Machine Learning: Revisiting Machine Learning with Language Models"
_ICLR.cc/2025/Conference — ICLR 2025 Conference Withdrawn Submission_

### Official Review · Reviewer_Yuia · 2024-10-31

**Soundness:** 4
**Presentation:** 4
**Contribution:** 4
**Rating:** 6
**Confidence:** 5

**Summary:**

The paper introduces Verbalized Machine Learning (VML), a framework that leverages large language models (LLMs) to address machine learning problems by constraining the parameter space to human-interpretable natural language. VML treats the LLM's text prompt as model parameters, enabling the model to be optimized over a discrete, sequential, and interpretable space. The framework offers advantages such as easy encoding of inductive bias, automatic model class selection, and interpretable learner updates. The paper empirically evaluates VML's effectiveness on various tasks, including regression, classification, and medical image classification, demonstrating its potential for enhancing interpretability and trustworthiness in machine learning.

**Strengths:**

* Good presentation
* VLM is interesting and provides a new perspective on machine learning.
* This paper investigates two important problems in machine learning: classification and regression.

**Weaknesses:**

* Lack of dicussion and comparison with [1]. [1] introduce agent symbolic learning, a systematic framework that enables language agents to optimize themselves on their own in a data-centric way using symbolic optimizers.
* I found that training loss is not stable in Figure 10. Is common in VML?





------------------------------------
[1] Symbolic Learning Enables Self-Evolving Agents. https://arxiv.org/abs/2406.18532

**Questions:**

Please see weakness

---

> ### Author Response · Authors · 2024-11-18
>
> **Q1: Lack of dicussion and comparison with [1]**
>
> A1: Thanks for the pointer! We will add this into our discussion.
>
> ---
>
> **Q2: I found that training loss is not stable in Figure 10. Is it common in VML?**
>
> A2: Yes, this is due to the fact that we are using mini-batch, just like in deep learning, the training loss over each step is rather noisy, but the trends are consistent (i.e., training loss decreasing).

---

### Official Review · Reviewer_DBUp · 2024-11-03

**Soundness:** 1
**Presentation:** 2
**Contribution:** 2
**Rating:** 3
**Confidence:** 4

**Summary:**

This paper introduces Verbalized Machine Learning (VML), a framework that uses natural language (text prompts) as the parameter space for machine learning models. VLM represents model parameters as text prompts, and uses an optimizer LLM to improve the prompts of the learner LLM. The paper mainly tests the proposed on a series of classical ML problems  such as linear regression, polynomial regression, and shows the proposed framework can enable learning textual prompts for these problems.

**Strengths:**

The paper studies a general and timely problem.

The main experiments cover multiple classical ML problems. The paper also presents some pilot results on image classification problems beyonds language tasks.

**Weaknesses:**

Overall, regarding technical contributions, the paper does not sufficiently differentiate the proposed technique from existing prompt optimization approaches and prior work using LLMs as optimizers. At the same time, the paper does not provide enough comparison against these baseline approaches. Specifically:

The proposed approach is essentially optimizing textual prompts for a learner LLM using an optimizer LLM (as acknowledged in line 160 of the paper). This has been explored in previous prompt optimization work (Pryzant, et al., 2023; Yang et al., 2023; Yuksekgonul et al., 2024; Zhang et al., 2024). The paper states the difference as  "prompt optimization seeks a generic instruction tuning without changing the original meaning." (line 233),
I am not sure if I can buy this claim, modern prompt optimization techniques do have more specific edits in the prompts (e.g., see the examples in appendices of Yang et al., 23 and Zhang et al., 2024).

Moreover, the paper lacks experimental comparisons against these related approaches. The evaluation mainly focuses on regression tasks. The only comparison with prior prompt optimization work is in Section 4.6, which provides a qualitative comparison against APE on text classification. (It's also worth noting that APE is primarily a search-based approach without LLM optimizers, making it a less relevant baseline). In order to clearly show the difference of VLM against prior approaches, it is necessary to compare to up-to-date prompt optimization approaches (Pryzant, et al., 2023; Yang et al., 2023; Yuksekgonul et al., 2024; Zhang et al., 2024) across a broader range of tasks under a controlled computational budget setting.


LARGE LANGUAGE MODELS AS OPTIMIZERS (Yang et al., 23)

Automatic Prompt Optimization with “Gradient Descent” and Beam Search (Pryzant et al., 23)

TextGrad: Automatic "Differentiation" via Text (Pryzant et al., 24)

In-Context Principle Learning from Mistakes (Zhang et al., 24)

**Questions:**

See weakness

---

> ### Author Response · Authors · 2024-11-18
>
> **Q1: The paper does not sufficiently differentiate the proposed technique from existing prompt optimization approaches and prior work using LLMs as optimizers. The proposed approach is essentially optimizing textual prompts for a learner LLM using an optimizer LLM.**
>
> A1: The difference between VML and prompt optimization might seem subtle, but they are very much different concepts. The difference between the two is similar to the difference between ‘classical’ machine learning and optimization. Almost all deep learning algorithms rely on gradient based optimizers, and optimization is happening at each step of learning. But still, we treat them as separate problems.
>
> One distinguishing factor that separates machine learning and optimization is that the goal of machine learning is to learn a model to recognize the pattern in a given training set, while the goal of optimization is to optimize an objective function. We can phrase the goal of a  machine learning problem into an optimization objective, and use the tools from optimization to solve it, but the focus of the two areas are fundamentally different.
>
> Similarly, the goal of verbalized machine learning (VML) is to learn a model that can recognize the pattern in a training set, and the learned pattern descriptions are in natural language space. However, the focus of prompt optimization is not to recognize patterns in a dataset, but to find a prompt that can optimize a given objective.
>
>
> Essentially, VML uses prompt optimization as a tool to reach the goal of pattern recognition, as we indicated in the subsection title of Section 3.4 ‘Iterative training by prompt optimization’.
>
> ---
>
> **Q2: The paper states the difference as "prompt optimization seeks a generic instruction tuning without changing the original meaning." (line 233), I am not sure if I can buy this claim, modern prompt optimization techniques do have more specific edits in the prompts.**
>
> A2: Good point! This statement is indeed problematic. We will update it with the answer to the last question.

---

> > ### Comment · Reviewer_DBUp · 2024-11-24
> > **Thank you for your response**
> >
> > Thank you for your response! I can see there are some conceptual differences here, though it also feels the execution is similar. To this end I would like to see more comparison on the experiment side. I am inclined to keep my score.

---

### Official Review · Reviewer_PJZx · 2024-11-04

**Soundness:** 2
**Presentation:** 3
**Contribution:** 3
**Rating:** 5
**Confidence:** 4

**Summary:**

This paper proposed a new framework of using LLMs as the backbones of machine learning tasks. The authors illustrated and verified the ideas using several simple ML tasks, and demonstrated that the approach is feasible. Furthermore, the authors also conducted ablations to show that the method is more effective than merely prompt optimization.

**Strengths:**

- This paper explores the potential of using LLMs in optimization and learning. The perspective and ideas are novel.
- Overall the paper is well written and conveys the key ideas clearly with a good amount of details.
- The experiments that support the claims are thorough and well designed.

**Weaknesses:**

- To make a strong argument of the generality of this new framework, the paper would need more results to show how practical it is, such as including empirical results measuring the efficiency of this approach. I'd also expect the paper to discuss the scalability of this approach, e.g. how it scales with amount of training data and size of the LLM being used. Given that this approach is not suitable for all or most machine learning tasks, the paper should carve out a concrete area of tasks which the proposed approach shines.
- Since the major differentiator with prior work, e.g. Yang et al. is the function approximation view of LLM, the paper should provide some theoretical analysis of what functions could and could not be approximated, and which properties of the LLMs architecture pose constraints, etc.

**Questions:**

For the training dynamics plots across model scales, i.e. Fig 10 a), it'd be more clear if the authors can change the x-axis to be the FLOPs instead of step size.

---

> ### Author Response · Authors · 2024-11-18
>
> **Q1: How does it scale with the amount of training data and size of the LLM being used?**
>
> A1: Good point. We do provide experiments on various sizes of LLMs (see Figure 10). As for the training data, due to the compute constraint, our experiments are all limited with training dataset of size around 100 to 200, but the training is done in mini-batches of size 10 for each step, and we can see from all the training curves, the loss decreases as the training progress (i.e., more data in terms of mini-batchs), and converged (e.g., Figure 4a).
>
> ---
>
> **Q2: Given that this approach is not suitable for all or most machine learning tasks, the paper should carve out a concrete area of tasks which the proposed approach shines.**
>
> A2: This paper contains quite a few experiments, and they are used as a proof of concept to demonstrate the features of the proposed paradigm. There are still many limitations and possible directions to explore in the framework. We are not trying to solve all the problems, or propose to solve all ML problems in VML at the current stage.
>
> There are many tasks that can be suitable for VML, at the moment it seems like VML is good for ML tasks that require common sense, large knowledge base or semantic reasoning (e.g. the medical image classification task). Of course one can try out VML on any task, and can encounter task specific problems, but we view it as an opportunity rather than a limitation. Just like the early stage of deep learning, many tools (e.g., Adam, CNNs, etc.) have to be invented when NNs are used for new tasks.
>
> ---
>
> **Q3: The paper should provide some theoretical analysis of what functions could and could not be approximated, and which properties of the LLMs architecture pose constraints, etc.**
>
> A3: The proposed VML framework took a function approximation view of using LLMs. We provide a comprehensive empirical study on this framework to show the effectiveness of the framework. Due to the effectiveness of VML, we believe a thorough and in-depth theoretical analysis worth a separate study by itself. Hence, we will leave it to future work.

---

### Official Review · Reviewer_T8tT · 2024-11-04

**Soundness:** 2
**Presentation:** 4
**Contribution:** 2
**Rating:** 5
**Confidence:** 4

**Summary:**

In this paper, the author propose to frame conventional machine learning with LLM as Verbalized Machine Learning: casting the "model parameters" to be the input language to the modeling LLM; and "parameter update" as language feedback from an optimizer LLM. This allows major advantages of VML include (1) easy encoding of inductive bias; (2) automatic model class selection; and (3) interpretable learner updates.

**Strengths:**

* illustration and mathematical formulation are very clear.

**Weaknesses:**

* I do not understand the additional benefit (beyond iterative prompting methods e.g. [3] or retrieval-augmented generator model) provided by this general framework. For example,

1. Using LLM to solve basic numerical computation has been shown to be hard by previous work [1]. Therefore, LLM doesn't seem to be a fit for more advanced calculation like regression. Also, the "classical ML problems" in the paper like regression models are very simple cases, but I do not believe such observation could be extrapolated to more advanced ML setting.

2. Figure 6: couldn't a conventional decision tree achieve the same level of predictive power and interpretability? In VML framework, if the LLM is not good at instruction following anywhere during reading the "model parameter".

3. The author claim VML brings the benefit of encoding human interpretable prior, but the examples given are a bit simple, like "Prior: The input is X-ray image for identifying pneumonia." (Figure 9). And the benefit of such prior seems to be unstable (Figure 9a). Have the author tried to experiment with more complicated prior? or shows more stable benefit over "without prior"?

* Such framework seems to make so strong assumptions --- optimizer LLM are essentially treated as an oracle machine; and the LLM needs to be very strong and faithful (to input) LLM. However, LLM has been shown to be brittle for explanation [2]. If such strong assumption is put on the backbone model, the same level of efforts (to get such a strong LLM) could be spent on alternative framework (a framework to unify various approaches from like a better decision tree, or other interpretable classifier) to achieve the same desiderate claimed by VML; and the "conventional ML" models from such alternative framework could run with much higher efficiency than serving an LLM.



[1]: Faith and Fate: Limits of Transformers on Compositionality
[2]: The Unreliability of Explanations in Few-shot Prompting for Textual Reasoning
[3]: Learning to Retrieve Iteratively for In-Context Learning

**Questions:**

Interpretability is claimed as a benefit of VML framework, but I am uncertain about the actual benefit. For example, Line 515-516 "validated by medical professional". My confusion is: what does it take for a professional to inspect the "model parameter"? If the model parameter is like a document-long, the practitioner might be better off by not using the system right?

---

> ### Author Response · Authors · 2024-11-18
>
> **Q1: Using LLM to solve basic numerical computation has been shown to be hard by previous work. Therefore, LLM doesn't seem to be a fit for more advanced calculation like regression. Also, the "classical ML problems" in the paper like regression models are very simple cases, but I do not believe such observation could be extrapolated to more advanced ML setting.**
>
> A1: Solving numerical problems is a challenge for LLMs at the moment, but they are heavily studied, for example, mathematical problem-solving. The popular MATH dataset (Measuring mathematical problem solving with the math dataset, NeurIPS 2021) requires strong numerical data processing from LLMs, and this dataset is used as a standard evaluation benchmark for LLMs. Moreover, there exist many LLMs (eg, DeepseekMath, WizardMath) that are capable of solving competition-level mathematics problems.
>
> Moreover, LLMs have shown remarkable potential in numerical data tasks for machine learning, and our work is one of the first methods to reveal such a potential. Some concurrent works (eg, TextGrad, LLM processes) also gave empirical evidence that LLMs can be fundamentally suitable for machine learning tasks.
>
> In addition, we are not suggesting to use VML to solve simple regression and classification tasks.  In machine learning, when a new learning algorithm is proposed, we often start with simple classical ML tasks such as regression and classification to gain better insights of the algorithm. Hence, we follow such norms and analyse the effectiveness of VML using these simple tasks.
>
> We do have experiments on more realistic tasks such as medical images classification in the paper, which highlight the unique features (e.g., interpretable model and training; easy encoding of prior knowledge) in VML that differs from the classical deep learning framework.
>
> Verbalized machine learning aims to provide a framework for LLMs to deal with machine learning tasks, with the ability to fully interpret the learned knowledge with natural language. We believe this framework will be increasingly more powerful, as LLMs get more powerful. We have already observed the performance improvement of VML by switching from Llama-3 to GPT-4o.
>
> ---
>
> **Q2: Couldn't a conventional decision tree achieve the same level of predictive power and interpretability?**
>
> A2: Yes, for the specific task in Figure 6, if we choose the decision tree as the model to model it, it will have the same power and interpretability as the current result of VML shown in the figure. However, the important difference here is VML chooses the class of model by itself during training, which does not require manual selection of the model class, unlike in classical ML, we look at the task, and decide that tree is a good model for it, and use it to solve it. Moreover, in VML, we can trace back the reasoning of why the decision tree was used for the task, which provides additional interpretability.
>
> Also, if you look at Figure 9, the learned medical image classifier is a decision tree as well. But this decision tree is on semantic level rather than conditioning on numerical values, which is very difficult for classical ML models. And the semantic level modeling does provide more interpretability.
>
> We would also like to refer the reviewer to Appendix J.1 for more interesting discussion about interpretability for VML!
>
> ---
>
> **Q3:  Have the author tried to experiment with a more complicated prior than in Fig. 9? or shows more stable benefit over "without prior"?**
>
> A3: Providing a more informative prior is like revealing the answer to the model, which makes the task simpler and less comparable to many classical methods. Therefore, we try to restrict the prior to be reasonably informative but not more for a fair study.
>
> However, we would like to highlight the sinusoidal regression task in Figure 5 for a visible benefit and the effectiveness of prior in VML. Using neural networks to model periodic functions like sine without special architectural design is often difficult, as the learned network usually only fits a smooth function in the range of training data, but not able to extrapolate out of the training data range (see Fig 5(b)). In VML, we can simply add the prior in a sentence: “It looks like the data is generated by a periodic function”, which is already effective as we showed in Fig 5.

---

> ### Author Response · Authors · 2024-11-18
>
> ---
>
> **Q4: The framework makes a strong assumption --- optimizer LLM are essentially treated as an oracle machine; and the LLM needs to be very strong and faithful (to input) LLM. However, LLM has been shown to be brittle for explanation. The same level of cost could be spent on other "conventional ML" frameworks than VML to achieve the same benefit of VML.**
>
> A4: We are not treating the optimizer LLM as an oracle, it is just an evaluator based on the reasonable assumption that LLMs can do reasoning, which have been empirically shown possible by the community. The optimizer is not required to know the global optimal solution, it just has to be able to reason on the current configuration and propose a possible change of the model that might improve the performance. We use many experiments in the paper to empirically show that optimizer LLMs do work. Moreover, there are numerous efforts in LLMs research that are currently trying to improve the reliability and reasoning ability of LLMs, which will further improve the performance of VML.
>
> Explanability comes at a cost, but these days compute is not the major issue in most of existing ML tasks. With the excessive amount of compute, we can and should start focusing on other aspect of ML, some important unsolved problems in deep learning includes explananbility of the learned model, and encoding of inductive bias, which VML are good at.
>
> With respect to the comment ‘The same level of cost could be spent on other "conventional ML" frameworks than VML to achieve the same benefit of VML’, we are not aware of any “conventional ML” frameworks that can easily provide benefits such as this level of explananbility in VML at the same level of cost (in terms of manual efforts). We are grateful if the reviewer can point us to the relevant literature.
>
> ---
>
> **Q5: What does it take for a professional to inspect the "model parameter"? If the model parameter is like a document-long, the practitioner might be better off by not using the system right?**
>
> A5: Great question. We believe this is orthogonal to the motivation of VML, which is to provide natural language based explananbility for a ML framework and a model.
>
> On one hand, it is fairly easy to constrain the length of the model parameters (by simply requiring the model parameters to be less than a certain number of words with natural language). On the other hand, reading a document-long explanation is actually very common in many professions. For example, when medical doctors are learning about a new disease, they read text-books or papers to study the explanation why observing certain symptoms is evidence of a disease. Similarly, for machine learning researchers, we write papers around 9 pages to explain a model, a method, etc., and there is rarely a text-based alternative to better explain these ideas in an extremely short format. Explananbility comes at a cost.

---

### Official Review · Reviewer_Cyun · 2024-11-04

**Soundness:** 2
**Presentation:** 3
**Contribution:** 2
**Rating:** 3
**Confidence:** 4

**Summary:**

This paper introduces a framework called Verbalized Machine Learning (VML), which constrains the parameter space to human-interpretable natural language. In this framework, the input prompt of a large language model (LLM) is optimized within the discrete natural language space, and an optimizer LLM iteratively updates the parameters of the learner LLM. VML is more interpretable and adjustable than traditional machine learning as all components are characterized by human-understandable natural language.

**Strengths:**

The idea of parameterizing a machine-learning model with natural language is niche. The new framework, Verbalized Machine Learning (VML), may be used to explain and distinguish between traditional machine learning and LLM-based learning schemas.

**Weaknesses:**

-	The novelty and contribution seem marginal. It is not surprising that LLM-based VML can address classical machine learning problems. Additionally, the core of VML relies on using two LLMs in a role-playing manner as a learner and an optimizer, updating LLMs through prompt engineering, which does not yield significant scientific insights.
-	I find it unclear how the millions and billions of parameters of a language model are represented by natural language. While prompting is an effective way to optimize a model’s output, I do not agree that optimizing model parameters is effectively a prompt optimization problem (Line 161). For example, in Figure 2, the prompt serves more as an additional input to the model rather than representing model parameters. In other words, the prompt adds new contextual information to the input without necessarily changing the model itself.
-	This framework only functions with LLMs that are adept at following instructions, which limits its applicability to smaller or non-instruction-based models.
-	The framework has been tested on regression and classification tasks, which traditional machine learning models can handle quite well. It is unclear to me why it is worth applying this LLM-based VML framework, especially considering the potential costs and computational expense.

**Questions:**

My main question lies in the motivation behind proposing the VLM framework for classical machine learning tasks.

---

> ### Author Response · Authors · 2024-11-18
>
> **Q1: What is the motivation behind VML, what are the contributions?**
>
> A1: Our motivation is not to solve classical ML problems using LLMs. Instead, our motivation is to investigate the two novel questions raised in the paper: can we parameterize functions using natural language instead of numerical matrices? If so, how to find these parameters (i.e., in natural language) to approximate a target function?
>
> We answer these questions in the paper with a comprehensive empirical investigation, and a learning algorithm. In machine learning, when a new learning algorithm is proposed, we often start with simple classical ML tasks such as regression and classification to analyse the generalization behavior of the algorithm. Hence, we follow such norms and show that VML can effectively solve these tasks. We also extend our study beyond classical ML tasks to more realistic tasks such as medical image classification.
>
> In summary, we formulate the framework of verbalized machine learning (VML) and show its effectiveness in many ML tasks. In the perspective of VML, functions are parameterized by natural language. We design an algorithm that iteratively optimizes the natural language based parameters to find an approximation of the target function for a ML task. The advantages of VML includes: encoding prior knowledge by simply writing it down in natural language; automatic model selection; and, interpretability.
>
> ---
>
> **Q2: How the millions and billions of parameters of a language model are represented by natural language?**
>
> A2: There is a misunderstanding. We are not trying to represent the parameters of LLMs as natural language. In the perspective of VML, functions can be described or represented in natural language. For example, say we want to find a function that outputs “1” when an image contains a book, and outputs “0” otherwise. In classical ML, we can try a CNN to learn this function, which will be parameterized by matrices. In VML, the function can simply be a sentence: “outputs ‘1’ when an image contains a book, outputs ‘0’ otherwise.”.
>
> To evaluate a function, we need an inference engine that can understand the parameterization. For example, for CNN, we need a GPU, which is able to do inference with matrices. For natural language parameterized functions, we need something that can understand natural language, which is an LLM with vision ability.
>
> The optimizer in VML has to evaluate the natural language based loss function and update the natural language parameters by rewriting the sentence (i.e., function description).
>
> ---
>
> **Q3: This framework only functions with LLMs that are adept at following instructions, which limits its applicability to smaller or non-instruction-based models.**
>
> A3: Indeed the framework requires instruction-tuned LLMs, but this is not a limitation, it’s just a requirement. As we have elaborated in the response of **Q2**: in order to evaluate functions that are parameterized by natural language, we need an inference engine that can understand such language. Moreover, instruction-tuned LLMs are everywhere and easily accessible, such as open-source models like Llama.
>
> ---
>
> **Q4: Why it is worth applying this LLM-based VML framework to tasks like regression and classification in the paper, especially considering the potential costs and computational expense.**
>
> A4: We are not suggesting to use VML to solve simple regression and classification tasks.  In machine learning, when a new learning algorithm is proposed, we often start with simple classical ML tasks such as regression and classification to analyse the dynamics of the algorithm. Hence, we follow such norms and analyse the effectiveness of VML using these simple tasks.
>
> We do have experiments on more realistic tasks such as medical images classification in the paper, which highlight the unique features (e.g., interpretable model and training; easy encoding of prior knowledge) in VML that differ from the classical deep learning framework. Note that such features come at a reasonable cost, i.e., the compute required for the LLMs. Fortunately, as the LLM inference gets highly optimized in practice (eg, vLLM), this computation overhead is generally not a problem.

---

### Note · Authors · 2024-11-25

**Comment:**

We sincerely thank all the reviewers and ACs for their time and efforts in reviewing our paper. We will take into account the suggestions and criticism to improve our paper. We are withdrawing our submission and will refine the paper for a better version.

**Withdrawal Confirmation:**

I have read and agree with the venue's withdrawal policy on behalf of myself and my co-authors.